# Human degradation of tropical moist forests is greater than previously estimated

C. Bourgoin[1,7 ✉], G. Ceccherini[1,7], M. Girardello[1], C. Vancutsem[1], V. Avitabile[1], P. S. A. Beck[1], R. Beuchle[1], L. Blanc[2,3], G. Duveiller[4], M. Migliavacca[1], G. Vieilledent[5,6], A. Cescatti[1] & F. Achard[1]

Tropical forest degradation from selective logging, fire and edge effects is a major driver of carbon and biodiversity loss[1–3], with annual rates comparable to those of deforestation[4]. However, its actual extent and long-term impacts remain uncertain at global tropical scale[5]. Here we quantify the magnitude and persistence of multiple types of degradation on forest structure by combining satellite remote sensing data on pantropical moist forest cover changes[4] with estimates of canopy height and biomass from spaceborne[6] light detection and ranging (LiDAR). We estimate that forest height decreases owing to selective logging and fire by 15% and 50%, respectively, with low rates of recovery even after 20 years. Agriculture and road expansion trigger a 20% to 30% reduction in canopy height and biomass at the forest edge, with persistent effects being measurable up to 1.5 km inside the forest. Edge effects encroach on 18% (approximately 206 Mha) of the remaining tropical moist forests, an area more than 200% larger than previously estimated[7]. Finally, degraded forests with more than 50% canopy loss are significantly more vulnerable to subsequent deforestation. Collectively, our findings call for greater efforts to prevent degradation and protect already degraded forests to meet the conservation pledges made at recent United Nations Climate Change and Biodiversity conferences.

Tropical moist forests (TMFs) have a major role in the provision of global ecosystem services, including climate and water cycle regulation, carbon sequestration and biodiversity conservation[8]. Despite their importance, TMFs are disappearing at an alarming rate[4]. In addition, degradation from selective logging, fires, edge effects or a combination of these disturbances is affecting forests and their capacity to provide ecosystem services at a rate comparable to—and in some years larger than—deforestation[4,9,10]. Here we define edge effects as changes in forest structure and functionality that occur at forest edges, driven by habitat fragmentation[7]. Furthermore, degraded forests are more vulnerable to additional disturbances such as climate extremes, reducing their potential resilience and threatening their long-term future[11–14]—for instance, Vancutsem et al.[4] showed that nearly half of TMFs are ultimately deforested.

Reducing forest degradation has great potential to reduce carbon emissions and increase carbon sequestration[15]. Yet, large uncertainties remain in quantifying the contribution of forest degradation to the global carbon fluxes (25–69% of overall carbon losses[1,2]). More accurate estimates would directly support Reducing Emissions from Deforestation and Forest Degradation (REDD+) activities under the United Nations Framework Convention on Climate Change (UNFCCC[16]). Despite the advancements in remote sensing capabilities for assessing carbon fluxes associated with each type of disturbance[17–21], a tropical assessment of forest degradation on forest structure is still lacking.

Furthermore, the depth of edge effect penetration within forest interiors is likely to be underestimated, mainly owing to the scarcity of forest structure data across the tropics[22].

The deployment of the Global Ecosystem Dynamics Investigation[6] (GEDI) instrument on the International Space Station in late 2018, which specifically targets forest structure, offers a unique opportunity to shed light on forest degradation at pantropical scale. Here we provide an assessment of the impacts of human-induced degradation on global TMF structure and of the forest's ability to recover to its pre-disturbance condition. Specifically, building on a previous work[23], we quantify in a consistent manner at pantropical scale: (1) the extent of forest degradation in 2022 taking into account edge effects; (2) the impact of different types of disturbance on forest structural characteristics and their persistence over time (over the period 1990 to 2022); (3) the rates of forest structure recovery after each type of degradation (selective logging, fire or edge effect) and forest regrowth after deforestation; and (4) the vulnerability of degraded forests to subsequent deforestation.

Within this scope, we combine a wall-to-wall dataset on forest degradation, deforestation and regrowth dynamics derived from Landsat imagery at 30 m spatial resolution[4] with spatially discontinuous estimates of forest canopy structure from GEDI. We jointly analyse the canopy height (RH98), that is, the top of the canopy or the nearest tallest vegetation in the footprint; the height of median energy (RH50), which describes the vertical distribution of canopy elements and gaps[24];

[1]European Commission, Joint Research Centre, Ispra, Italy. [2]CIRAD, Forêts et Sociétés, Montpellier, France. [3]Forêts et Sociétés, Univ Montpellier, CIRAD, Montpellier, France. [4]Max Planck Institute for Biogeochemistry, Jena, Germany. [5]CIRAD, UMR AMAP, Montpellier, France. [6]AMAP, Univ Montpellier, CIRAD, CNRS, INRAE, IRD, Montpellier, France. [7]These authors contributed equally: C. Bourgoin, G. Ceccherini. ✉e-mail: clement.bourgoin@ec.europa.eu

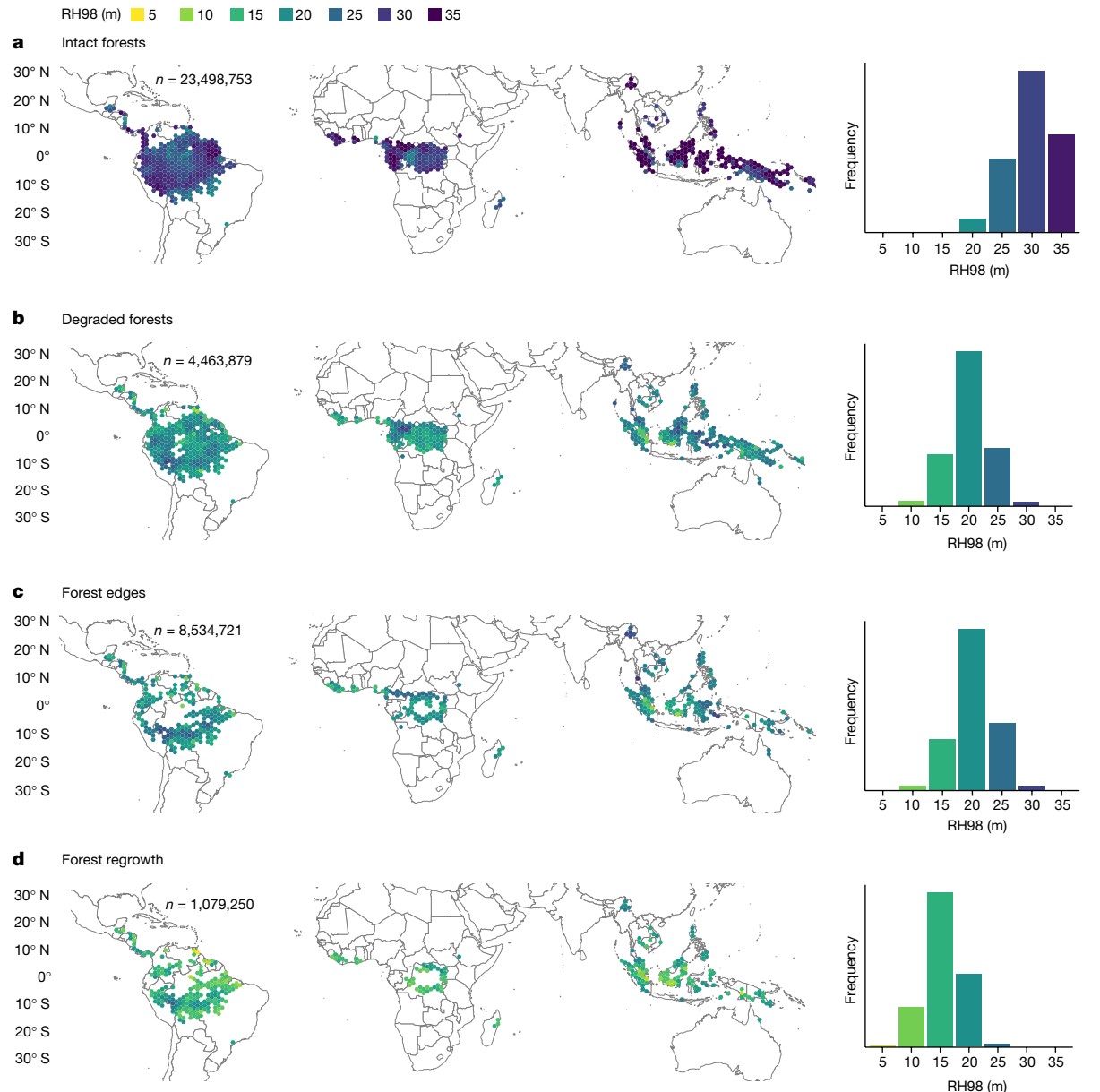

**Fig. 1 | Canopy heights of intact, degraded edge forests and regrowths.**
**a**–**d**, Canopy height (RH98) of intact forests (**a**), degraded forests from logging, fire or natural disturbances (**b**), forest/non-forest edges (120 m width) (**c**) and forest regrowths (**d**) in 1.5° (each side is approximately 167 km) hexagon grid cells between 30° N and 30° S. Canopy heights are mean values calculated over the period 2019–2022. Note that all the forest regrowth areas in **c**, regardless of their age, are shown here (an in-depth analysis by age class is shown in Fig. 3).

Grid cells with fewer than 600 GEDI samples or with no statistically significant differences in canopy height between intact and non-intact forests were masked (Welch two-sided *t*-test. *P* < 0.05). 100%, 99% and 99% of the hexagons for degraded, edge and regrowth forests, respectively, were statistically significant. *n* represents the total number of GEDI sample plots for each forest cover type (Supplementary Fig. 2). Summary statistics of RH98 and AGBD are shown in Extended Data Table 1.

and the aboveground biomass density (AGBD), which represents the aboveground woody biomass per unit area[25].

## Spatial patterns of forest canopy heights

The analysis of more than 23 million sample footprints of GEDI data over intact TMF—that is, areas with no signs of human activity detected during the past three decades and located at least 3 km from forest/non-forest edge (see Methods)—reveals regional and continental variability in forest structure, with the tallest canopies found in Insular Asia, western Africa and western and eastern Amazonia (Fig. 1a and Extended Data Table 1). Overall, canopy heights are higher in Asia (34.4 ± 10.7 m) than in Africa (29.3 ± 8.6 m) and in Americas (28.6± 7.4 m) (mean ± s.d.).

Similar results are found for AGBD (370.8 ± 205.2 Mg ha$^{-1}$ in Asia, 225.5 ± 110.9 Mg ha$^{-1}$ in Africa and 239.5 ± 129.9 Mg ha$^{-1}$ in Americas; Extended Data Fig. 1). These results support previous observations[26] showing that intact tropical forests in Asia, which are typically dominated by hardwood wind-dispersed species, show a higher frequency of large and tall trees (those with RH98 greater than 30 m) compared with Africa and South America (Supplementary Fig. 1).

We found lower RH98 and AGBD values for the three types of disturbed forests considered in our study compared with intact forests. We find that the minimum difference between intact and degraded forests is 10 m for mean RH98 and 122 Mg ha$^{-1}$ for mean AGBD (Fig. 1b). Forests within 120 m to forest/non-forest edge show, on average, a 11 m lower RH98 and 150 Mg ha$^{-1}$ lower AGBD compared to intact forests with an

absence of emergent trees and a canopy height distribution dominated by smaller trees (Fig. 1c and Supplementary Fig. 1). Regrowing forests on deforested land have, on average, a 16 m lower RH98 than intact forests, and an average AGBD of 80.4 ± 87.3 Mg ha[-1] (Fig. 1d).

## Magnitude and scale of edge effects

Potapov et al.[27] showed that fragmentation of intact forest landscapes by agricultural or road expansion initiates an edge effect (here referred to as forest/non-forest edge effect) and a cascade of changes that lead to landscape transformation and loss of conservation values. To a lesser extent, smaller canopy openings following selective logging and fire could also be a source of important and yet unaccounted edge effects[28] (here referred to as forest/degraded forest edge effect). Here we show the impacts of these two anthropogenic edge effects on the vertical structure of the forest, which is caused by microclimatic alterations[22] and leads to large-tree mortality[29].

We first assessed the scale and magnitude of forest/non-forest edge effect on forest structure metrics in the vicinity of deforested lands at varying distances to the forest edge (Extended Data Fig. 2). We used two indicators: the first and more conservative is the distance at which the RH98 of the edge forest reaches 95% of the RH98 of the intact forest; and the second is the distance at which the differences in RH98 between edge and intact forest are not any more significant based on an ANOVA test (Methods).

Forests, classified as undisturbed in the Tropical Moist Forest dataset from the Joint Research Centre (JRC-TMF), showed a decrease in RH98 from the edge up to 350, 400 and 1,500 m into the forest interiors in America, Africa and Asia, respectively (red dotted lines in Fig. 2a corresponding to first criteria). Edges have a significant effect on canopy height for each continent, with maximum distance of effect being largest in the Americas (second criteria). In fact we found significant differences in the RH98 until 7 km distance to the forest edge in the Americas (Tukey's honest significant difference test, $F = 2,141$, $P < 0.0001$) and 1.7 km in Africa ($F = 544$, $P < 0.0001$) and Asia ($F = 582$, $P < 0.0001$). On average, we estimated a 20% reduction in RH98 within the first approximately 200 m of the forest edge relative to the intact forest interior, associated with a change in structure and loss of tallest trees (Fig. 2b). The largest extent of edge effects were detectable along active and consolidated deforestation fronts of the Amazon (Brazilian arc of deforestation, Peru and Colombia active fronts), in Borneo and Sumatra coasts marked by high fragmentation levels, and on the borders of the Congo basin (Fig. 2c). The variability in the magnitude of change and extent of the edge effect within each continent is likely to be linked to forest structure composition and environmental conditions[18,22,30] and to different landscape configurations[29] (for example, forest patch size and connectivity) and composition (for example, proportion and type of agricultural land) that can increase forest sensitivity to edge effects[31].

We detected larger scales of edge effects on AGBD in the Americas (1,000 m) and Africa (750 m) and lower scales in Asia (1,020 m) compared with RH98 (Extended Data Fig. 3). These results show that the edge effect on biomass is far exceeding the previously measured[17,18,32,33] 120 m. In total, the area with edge effects represented 18% (approximately 206 Mha) of the total forest area in 2022. This represents an increase in area of 221% compared with the total area of forest edge zones defined using the 120 m distance to the edge identified in previous studies.

We detect the cumulative impacts of selective logging and forest fire up to around 1.5 km into forest interiors and we quantify an additional 10% decrease in RH98 on average compared with the distribution of forest affected by edge-desiccation effects only (grey distribution of all forests in Fig. 2a). The increased tree mortality due to edge effects triggers positive feedback loops with fires, which can penetrate up to 1 km into forest interiors (red curve in Fig. 2a, inset). At the same time, fragmentation makes the interior of the forest more accessible, which leads to increased hunting and resource extraction[33]. As a result, we found evidence of selective logging within 500 m from the forest edge, except in Asia, where logging operations often occurred deeper within the forest interior (purple curve in Fig. 2a, inset). In addition, the increased frequency of extreme droughts may directly increase tree mortality and fire incidence at the edges[34,35], making the first kilometre of the forest edge highly vulnerable to land use and climate change impacts.

We also assessed the change in forest height in the vicinity of degraded forests (forest-degraded forest edge effects). We observed that canopy heights in undisturbed forests near logged or burned forests (that is, within a 120 m radius) were on average significantly lower than in intact forests by 15% and 22% for RH98, respectively (Extended Data Fig. 4, 22% and 32% lower on average for RH50). These results are due not only to localized edge effects from degradation, but also to the omission in the Landsat-based forest cover change datasets of small-scale (<0.09 ha) and low-intensity disturbances at the interface between undisturbed and degraded forests[36], highlighting the added value of LiDAR-based assessment to compensate for the intrinsic limitations of optical sensors in detecting these phenomena.

Our findings on the spatial extent of the two edge effects (from both deforested land and degraded forests) were used to map the remaining intact TMF landscapes in 2020 (Methods). This resulted in smaller extents (−14%) compared with the 2020 assessment by Potapov et al.[27] (502 Mha versus 426 Mha with our approach). Around 48% of our assessment of intact forest landscape falls within the World Database on Protected Areas (versus 53% estimated by Potapov et al.[27]). Conversely, 57% (approximately 60% in Americas and Africa, and 28% in Asia) of protected TMFs are mapped as intact forest, reinforcing their importance for maintaining forest structure and functioning[37]. The main differences between the two maps (Extended Data Fig. 5) concern mosaics of disturbed forest and deforested land in the JRC-TMF dataset (excluded from our approach but present in Potapov et al.[27]). In Gabon, we have larger estimates of forest cover compared with Potapov et al[27], mainly because undisturbed forest blocks are excluded in their analysis. This discrepancy is attributed to limited Landsat imagery before 2005 in the western part of central Africa, resulting in an underestimation of historical disturbances such as selective logging, particularly in the 2000s[4].

These results demonstrate the relevance of the effects of forest edge creation following tree cover loss at pantropical scale. These effects are especially important in forests with high conservation value and will contribute significantly to carbon emissions from tropical deforestation and degradation through induced edge effects[38].

## Persistence of forest degradation effects

Assessing the persistence of forest/non-forest edge effects over time is critical to understanding the long-term consequences of deforestation and fragmentation on the structure of forest remnants. Here we show that there is no significant recovery in RH98 (Fig. 3a), RH50 and AGBD (Extended Data Fig. 6) during the 30 years following the creation of the forest edge. Undisturbed forests within the first 120 m of the forest edge exhibit on average a 15% lower RH98 compared with intact forests from the first year after edge creation. Even larger decreases of 25% and 30% on average for RH50 and AGBD, respectively, show that the sensitivity of RH50 and AGBD to edge-related desiccation is higher than that of RH98. Degradation of forest edges from logging or fire triggered an additional 30% decrease in RH98 on average (50% and 40% decrease in RH50 and AGBD, respectively), with no evidence of recovery over time, which is corroborated by airborne laser scanning studies at local scales[17,18,31]. This pattern is likely to be due to the long-term persistence of the edge effects driven by changes in the growing conditions and the exposure to additional anthropogenic disturbances[17,39].

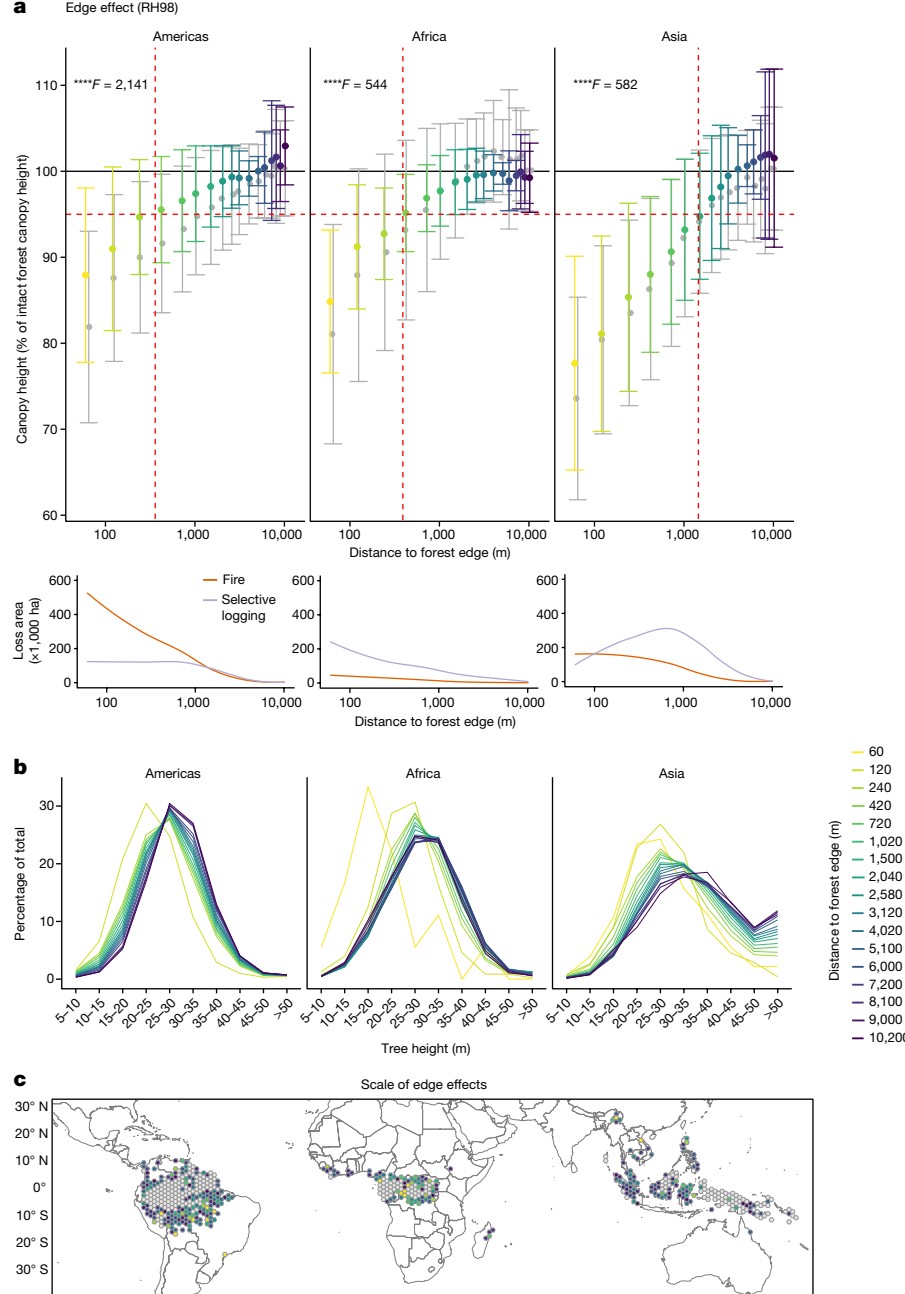

**Fig. 2 | Spatial scale and magnitude of edge effects caused by deforestation.** **a**, Canopy height (RH98) of undisturbed forests (located at more than 120 m from degraded forests) and all forests shown in grey (including undisturbed and degraded forests) at various distances to the forest edge that separates forest cover from agricultural land and other land covers. Inset, degradation area due to fire and selective logging calculated at various distances to the edge. The red dotted vertical line corresponds to the distance between the forest edge and the point at which 95% of intact forest RH98 is reached (red horizontal dotted line): to 350, 400 and 1,500 m for America, Africa and Asia, respectively. Vertical bars indicate the spatial s.d. The number of GEDI sample

footprints for each distance to the forest edge is reported in Supplementary Fig. 3. *F* value from one-sided ANOVA; ****$P \leq 0.0001$; NS, not significant. Tukey post hoc tests are presented in Supplementary Data. **b**, Average distribution of canopy heights of undisturbed forest (located at more than 120 m from degraded forests) at various distances to the forest edge. **c**, Scale of the edge effect, represented as the distance from forest edge at which RH98 reaches 95% of the value of RH98 for intact forest. Colours for the undisturbed forest at indicated distances from the forest edge in **a** correspond to those in **b**,**c**. Grey cells in **c** represent areas where the accumulated deforestation (from 1991 to 2022) is less than 2% of the forest area in 1990.

Beyond 120 m from the forest edge, where effects of edge proximity are reduced (see Fig. 2), we observed immediate effects and post-disturbance recovery dynamics that differ considerably between continents, disturbance types and forest structure metrics (Fig. 3b and Extended Data Fig. 6). Selective logging effects on RH98 and AGBD are higher in Asia (decreases of 20% and 50%, respectively) compared with Americas and Africa (combined decreases of 10% and

30%, respectively), which can be explained by higher selective logging intensity in Asia (30–40 m³ ha⁻¹ for the Amazon, 50 m³ ha⁻¹ for Africa and 270 m³ ha⁻¹ in Asia[40,41]). Within 20 years since the last disturbance, we found that logged forests recovered on average 25%, 15% and 27% of RH50 in America, Africa and Asia, respectively, with slower recovery for AGBD (average of 11% recovery across the three continents). The absence of recovery trends in RH98 can be explained by the slow

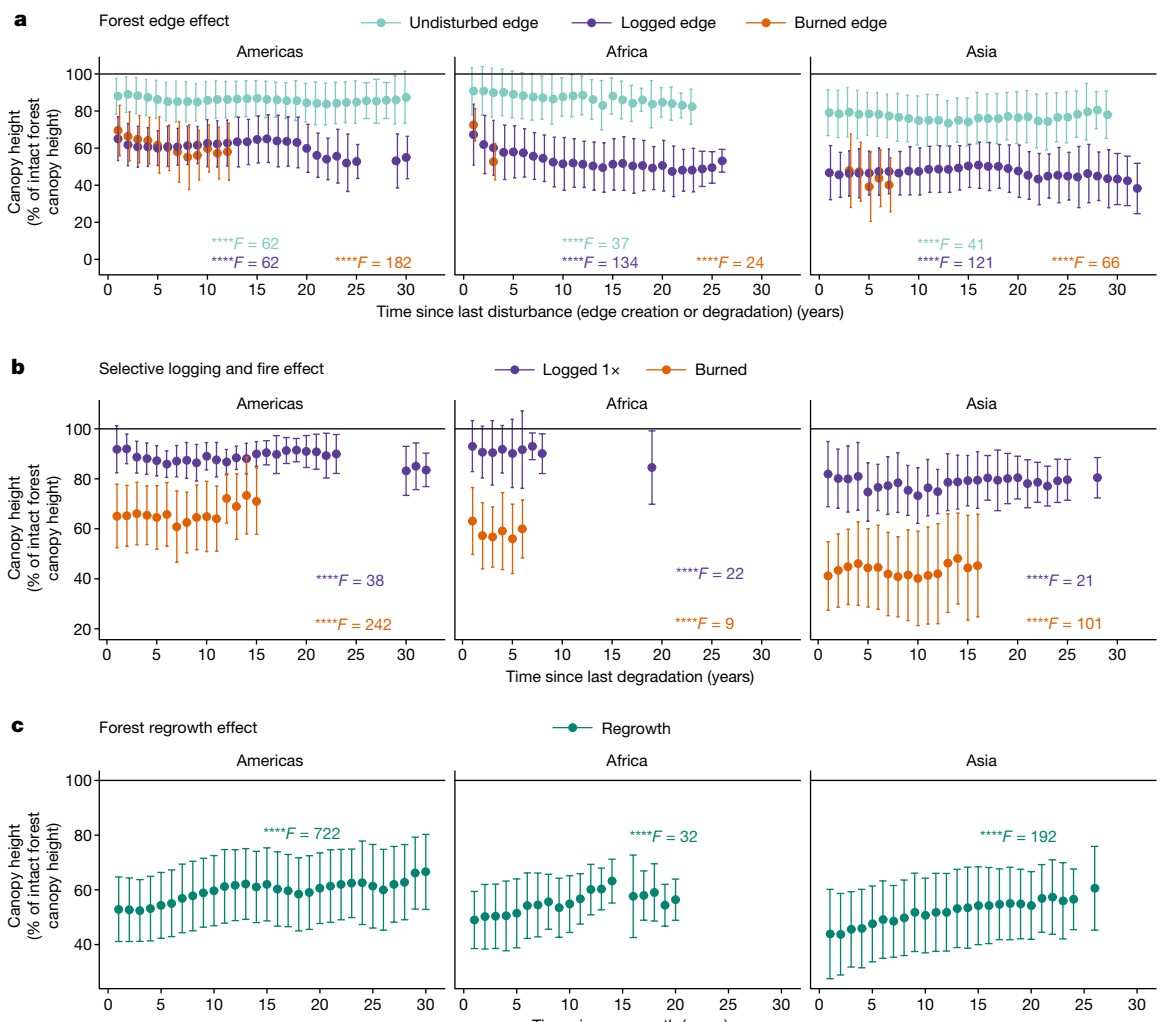

**Fig. 3 | Impacts of forest degradation from selective logging, fire and edge effects. a–c,** Long-term effects on canopy height (RH98) from edge-desiccation effects (**a**), degradation (fire or logging) of edge forest (**a**), selective logging ('logged 1×' indicates an area has been logged once over the past 30 years), fire (**b**) and secondary forests regrowing on abandoned deforested lands (**c**). Results are reported as the percentage of intact forest canopy height (solid line) after normalizing the difference in canopy height (RH98) within each grid cell between intact forest and each forest type (degraded, edge forest and regrowth) and age. Data are mean RH98 ± spatial s.d. GEDI samples for each disturbance type and related time since disturbance are reported in Supplementary Fig. 5. *F* value from one-sided ANOVA; ****$P \le 0.0001$. Tukey post hoc tests are presented in Supplementary Data.

regrowth of late successional, large and emergent trees[19], whereas forest understory dynamics, including tree removals, collateral damages from selective logging (such as dead fallen trees) and the fast regrowth of pioneer and understory species, affect the average vertical distribution of plant material, captured by RH50, making this metric a robust indicator of the long-term effects from degradation and subsequent recovery[42] (Extended Data Fig. 7). Recovery rates for AGBD are lower compared with those reported by Rappaport et al.[21] and Philipson et al.[43] showing an average of 25–32% recovery of AGBD 20 years after logging. The difference in the average annual rates of recovery across continents is likely to be influenced by logging intensity[40], forest composition and climate conditions[44].

The immediate impacts from forest fires are much higher than those from selective logging, with decreases of 35%, 40% and 60% in RH98 for the Americas, Africa and Asia, respectively, and decreases of 60% in AGBD for the Americas and Africa and 80% for Asia. These results are consistent with short-term changes in AGBD from logging and fire reported in the literature[21,45]. No recovery trend in RH98 or AGBD was detected even ten years after the last disturbance, confirming the long-lasting effects of fire on tree mortality and losses of AGBD[46,47]. Manipulative studies of post-fire degradation in the Amazon showed strong understory vegetation regrowth under the remaining dominant and taller trees within 5 years after the disturbance, resulting in partial canopy closure[48] (70–80%). This vegetation dynamic is better captured by changes in RH50 than by changes in RH98. The high variability in recovery rates is probably due to different fire frequencies, intensity, climate and forest-type-specific responses[49].

In comparison with forest degradation, trends of forest regrowing on deforested land could be observed and quantified across continents and forest structure metrics (Fig. 3c and Extended Data Fig. 6). After 10–15 years, the regrowth plateaued at 60% of intact forest RH98 with low growth rates (0.5% yr⁻¹, 0.7% yr⁻¹, and 0.9% yr⁻¹ for the Americas, Africa and Asia, respectively). The regrowth for AGBD was on average half that of RH50, reaching 43% (40%, 33% and 57% Americas, Africa and Asia, respectively) of intact forest AGBD after 20 years of regrowing rates, which are similar to those reported by Poorter et al.[50] (33%) using field inventories, or by Heinrich et al.[15] (36–49%) using remote sensing data. However, the slowdown in regrowth rates of AGBD after ten years of regeneration may indicate that several drivers are affecting forest growth and are not captured by Poorter et al.[50] (Supplementary Data). We found that land use intensity through repeated deforestation events and fire occurrences before forest regrowth may have negative effects

on regrowth after 5–10 years (Supplementary Fig. 4; and corroborated by previous studies[51]), whereby fire legacies could decrease regrowth rates by 20 to 75%, particularly in drier and water-deficient regions.

## The fate of degraded forests

The stage of forest degradation is linked to the type, intensity and recurrence of past disturbances, as well as to the time since the previous disturbance. Here we show that degradation also has a crucial role in predicting future deforestation, whereby the likelihood of total deforestation and land use change increases with the degree of forest degradation. Our results indicate that degraded forests followed by recent deforestation (2020–2022) had significantly lower canopy heights and AGBD compared to those not subjected to deforestation (Extended Data Fig. 8 and Supplementary Fig. 6). On average, degraded forests followed by deforestation experienced severe impacts, with average reductions in RH50, RH98 and AGBD of 60%, 45% and 65%, respectively. These impacts are probably due to unsustainable logging and/or fire, as shown in Fig. 3. Moreover, these structural parameters have a large spatial variability (±12.8%, ±13.3% and ±14.6% for RH50, RH98 and AGBD, respectively), reflecting the complexity of the degradation processes and underlying factors driving deforestation in the tropics[52].

We found that forest relative heights (RH50 and RH98) and distance to the edge were strong predictors of the probability of deforestation (Extended Data Fig. 9 and Supplementary Fig. 7). Degraded forests in America showed, on average, a higher deforestation risk than in Africa or Asia, as 50% of deforestation probability was reached when forests lost 50% of their initial heights (60% in Africa and Asia). Furthermore, proximity to the forest edge, recognized in previous research as a key factor in assessing deforestation risk[53], showed complex interactions with canopy height in degraded forests. This observation highlights the interplay between different factors such as degradation, exposure to human activities and edge-desiccation effects within the first kilometre from the forest edge, contributing to an increased likelihood of subsequent deforestation. However, within 120 m of the forest edge, degradation had a role in enhancing subsequent deforestation only in the Americas, and no statistical differences in RH50, RH98 or AGBD were found for the other continents (Extended Data Fig. 8 and Supplementary Fig. 6).

## Conclusions

Our study demonstrates that the integration of recent and spatially sparse spaceborne LiDAR observations (GEDI), with long-term and spatially continuous spaceborne optical datasets (JRC-TMF) provides a novel approach to assess forest degradation and recovery at the pantropical scale. We show that the magnitude of degradation effects on canopy structure are greater than previously reported, with a 20–80% decrease in canopy height and AGBD. The effects of edges on forest vertical structure were previously assumed[7] to extend no more than about 100 m. Our results show that this is a significant underestimate, and we measure edge effects up to around 1.5 km into the forest interior, implying that the overall spatial impact of fragmentation across the pantropical belt is severely overlooked by at least 200%. We show that the cumulative impacts of selective logging, fires and edge effects have significant long-term effects on the structure of global TMFs, but as the 30-year period of our study is too short to observe a full recovery of the forest structure for most types of forest disturbances and regions, future studies should further address this question. Although the current areas of fast-regrowing forests allow offsetting of around 25% of carbon loss from deforestation[15], we found here that the full recovery of forest structure after deforestation or degradation would take a centennial timescale and may be slowed down by anthropogenic factors. Finally, this study provides new insights for identifying the forests that are most vulnerable to agricultural expansion. Forest canopy structure, combined with disturbance history, is a significant indicator of deforestation risk and should be used to target forest monitoring and prioritize conservation in highly degraded areas. This type of spatially explicit information on tropical forest degradation is crucial for implementing more effective forest-based mitigation policies[54] and conservation activities agreed under the UNFCCC and the UN Convention on Biological Diversity[55] (https://www.cbd.int/meetings/COP-15).

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

# Methods

In this study, we use the spaceborne GEDI[6] from the National Aeronautics and Space Administration (NASA) to analyse the extent of forest degradation on canopy structure at pantropical scale, but its short lifetime limits long-term monitoring. To overcome this limitation, we combine GEDI data with long-term information on forest dynamics from Landsat using a space-time substitution strategy. While this approach has been used in previous studies[15], it assumes that differences in neighbouring land characteristics can be used as a proxy for changes over time and that climate and vegetation remain relatively constant over the 20- to 30-year analysis period. For example, when studying forest recovery, we assume that different height metrics from GEDI represent different ages since the last disturbance.

## Preparation of input datasets

**TMF datasets.** We use JRC-TMF, which provides information on changes in humid forest cover from 1990 to 2022 derived from the Landsat archive collection (more details on the methodology and accuracy assessment in Vancutsem et al.[4]). Mangrove forests were excluded from the analysis as periodical tidal floods affect the consistent estimation of canopy height over time. Bamboo-dominated forests were also excluded, as the dynamics of forest structure are related to seasonal or occasional defoliation rather than anthropogenic disturbances. We used the JRC Transition Map and the Annual Change Collection that capture the TMF extent and the related disturbances on an annual basis to derive the following classes.

**Intact forests.** Undisturbed forest (forest without any disturbance observed over the Landsat time series) located at more than 120 m from degraded forests and more than 3,000 m from the forest/non-forest edge.

**Degraded forests.** Closed evergreen or semi-evergreen forests that have been temporarily disturbed for a period of a maximum of 2.5 years by selective logging, fire, or unusual weather events. We derived the year since the last degradation from the JRC-TMF dataset used as a proxy for forest recovery. To attribute forest degradation to its direct driver, we first used the global forest cover loss due to fire dataset (GFC-Fire) from 2001 to 2021 from The Global Land Analysis and Discovery (GLAD) laboratory[56]. All certainties of forest fires were considered. Regarding degradation due to selective logging, we performed an extensive visual interpretation and delineation of selective logging operations based on their specific spatial features visible on the JRC-TMF Transition Map. The selected degraded forest pixels correspond to temporary logging roads, logging felling gaps, decks, and skid trails. This dataset covers Brazil, French Guiana, Guyana, Cameroon, Central African Republic, Gabon, Congo, the Democratic Republic of Congo, Indonesia, Malaysia and Papua New Guinea. The managed forest concessions dataset from the World Resource Institute was used to guide the collection of polygons in central Africa and southeast Asia while the delineation in the Amazon was generated from previous scientific experience[57,58]. An independent visual interpretation of selective logging was performed in order to analyse how the delineation influenced our results. This sensitivity analysis showed small differences in the magnitude and trends of logging impacts on forest structure without altering the subsequent analysis and conclusions (Supplementary Fig. 8). It also proved to be unbiased and robust when comparing changes in forest height in the vicinity of forest degraded by selective logging (Supplementary Fig. 9). We created a buffer of 300 m radius (10 Landsat pixels) around fire pixels to avoid an overlap between the two causes when analysing impacts from selective logging alone. When looking at forest degradation alone, we excluded pixels within the edge forest defined with a conservative value of 120 m from the edge.

**Forest edge.** To compute forest edges, we considered undisturbed or degraded forest pixels from the JRC-TMF Annual Change Collection dataset for years spanning from 1989 to 2022. We applied a 5 × 5 pixel moving window for all annual forest maps to remove isolated pixels for both forest and non-forest classes using the sieve algorithm and replace them with the value of the most frequent class within the moving window. For the analysis of forest edge effect penetration, we used the extent of forest cover in 2022 to derive undisturbed forest edges using edge widths varying from 60 m to 10,200 m at different intervals (0–60 m, 60–120 m, 120–240 m, 240–420 m, 420–720 m, 720–1,020 m, 1,020–1,500 m, 1,500–2,040 m, 2,040–2,580 m, 2,580–3,120 m, 3,120–4,020 m, 4,020–5,100 m, 5,100–6,000 m, 6,000–7,200 m, 7,200–8,100 m, 8,100–9,000 m, 9,000–10,200 m) using the Euclidean distance calculated from the non-forest class. These distances were selected based on previous studies reporting on the scale at which edge effects operate and affect microclimate[59] (up to 400 m), canopy moisture levels[29] (up to 2.7 km), phenology[60] (up to 5–10km) and forest biomass[22] (up to 1.5 km). The first 6 intervals of distances are centred on the most recent and accurate evaluation of the extent of the edge effect[17,18,32,61] (~100–200 m). To focus on the scale of edge effects due to deforestation, we discarded grid cells of 1.5° showing a value of accumulated deforestation (1991–2022) compared to forest area in 1990 of less or equal than 2% (estimates derived from JRC-TMF). To mitigate the effects of canopy disturbance interactions between degraded and undisturbed forests, we eliminated areas of transition using a buffer of 120 m around degraded forests. This distance corresponds to the area initially affected by the felling of individual trees in selective logging operations[62] where localized edge effects are the highest[28]. To calculate the age of forest edges, we adopted a 120 m edge width, which constitutes the threshold of significant AGBD changes observed in the tropics[7,17,18]. We produced forest edges from 1989 to 2022, masked natural edges (transitions forest/water and forest/savannah), derived the year of forest edge creation and computed the age of all edges classified as forest in 2022. We separated forest edges into undisturbed forest edges, burned forest edges (where a fire from the GLAD dataset occurred after the year of forest edge creation) and logged forest edges (all other types of degradation occurring after the year of forest edge creation).

**Forest regrowth or secondary forest.** Forest regrowth or secondary forest refers to a two-phase transition from moist forests to deforested land to vegetative regrowth. A minimum duration of 3 years (2020–2022) of permanent presence of moist forest cover is needed to classify a pixel as forest regrowth to avoid confusion with other land uses. Using the JRC-TMF Annual Change Collection, we calculated the age of secondary forests (from 1 to 32 years old), which may have an uncertainty of 1 year, depending on whether a deforestation event was detected at the end of a year or at the beginning of the next year. In case of late detection, the area will be classified as regrowing one year later (if it does not show signs of permanent deforestation).

**GEDI dataset.** The GEDI mission uses a LiDAR deployed on the International Space Station from April 2019 until March 2023. One of its primary scientific objectives is to map forest structural properties and understand the effects of vegetation structure on biodiversity. It provides sparse measurements (hereafter sample plots or shots) of vegetation structure, including forest canopy height[6] with a vertical accuracy of about 50 cm, over an area defined by a sampling footprint of ~25 m width. For our analysis, we used GEDI L2A[63] Elevation and Height Metrics (version 2) and GEDI L4A[64] Above Ground Biomass (version 2.1) which represent returned laser energy metrics on canopy height and estimated AGBD for each 25 m diameter GEDI footprint. The footprint data are geolocated and have an expected positional error[6] (that is, horizontal geolocation accuracy) of 11 m. For each footprint, we extracted a set of relative height (RH) metrics, the AGBD and the associated prediction standard error (AGBD_SE). AGBD are reported as weighted averages, using the AGBD_SE as weight. Note that the estimation of AGBD based on RH metrics from GEDI L2A varied considerably in performance across the TMF domain, having a determination coefficient ($R^2$) of 0.66 (mean residual error (MRE) of 10.4 Mg ha$^{-1}$), 0.64 (MRE of 15.32 Mg ha$^{-1}$),

0.36 (MRE of 121.15 Mg ha$^{-1}$) and 0.61 (MRE of 8.17 Mg ha$^{-1}$) for South America, Africa, Asia and Oceania, respectively (further details on the validation of the GEDI L4A are in ref. 25).

RH metrics represent the height (in metres) at which a percentile of the laser energy is returned relative to the ground. RH98 corresponds to the maximum canopy height (hereafter 'canopy height'), which is a more stable height metric than RH100. RH50 (also known as 'height of median energy' (HOME)[24]) is the median height at which the 50th percentile of the cumulative waveform energy returned relative to the ground and has been identified as one of the LiDAR metrics with the greatest potential for estimating structural characteristics in tropical forests[24]. When validated against ground-based data, RH50 generally exhibits a strong correlation with key structural variables, including AGBD, stem diameter, and basal area[65]. Due to its strong dependence on the vertical distribution of canopy elements and gaps within the canopies and canopy cover, RH50 serves as a highly complementary metric to RH98 for characterizing changes in canopy structure from degradation[66] (see also Extended Data Fig. 7).

We selected GEDI data acquired from 1 January 2019 to 31 December 2022. To select the highest quality data, we filtered the GEDI data (both GEDI L2A and L4A) by selecting only the observations collected in power beam mode and labelled them as good quality (quality flag equals 1), thus avoiding risks of having degraded geolocation under suboptimal operating conditions (degrade flag equals 0). Additionally, we filtered GEDI 2A data using only night acquisitions to limit the background noise effects of reflected solar radiation. We used the Shuttle Radar Topography Mission (SRTM) information to exclude GEDI footprints above 20° slopes to avoid errors in vegetation height. Steep slopes might lead to erroneous relative height metrics (especially over sparsely vegetated areas), so applying our threshold of 20° is a conservative approach[67]. Additionally, we filtered out GEDI footprints classified as water in the Global Land Analysis and Discovery Landsat Analysis Ready Data quality layer (ARD; https://glad.umd.edu/ard/home) or when a GEDI footprint was located within an urban area defined by the Global Urban Dataset of Florczyk et al.[68]. Finally, we excluded GEDI footprints with RH98 values below 5 m to be compliant with the Food and Agriculture Organization (FAO) definition of forest.

Further, we used the beam sensitivity information from GEDI L2A as a proxy for signal-to-noise ratio and the ability of GEDI to penetrate the highest canopy cover. For the intact and undisturbed forest classes, we considered only shots with a beam sensitivity greater than 0.98, while for the other classes (for example, degraded, edge and regrowth forests), we used a beam sensitivity greater than 0.95, as previously recommended[67,69].

### Combining datasets

On the temporal scale, we used separately yearly GEDI data to estimate as accurately as possible the year since the last disturbance (that is, degradation, forest edge creation or deforestation). All degraded and edge forests were masked out if the date of disturbance or the year of edge creation occurred during the GEDI acquisition period. A similar step was performed for secondary forests when the year of regrowth overlapped with the GEDI acquisition period. On the spatial scale, to reduce the noise caused by GEDI geolocation errors, we applied a morphological (circular shape) filter of 35 m to the forest cover change class of interest (intact, degraded, edge or regrowth), which resulted in the removal of single-small-patches of pixels. We thus ensured that GEDI samples fell within the class of interest and avoided any partial overlap. The extent of mapped forest change areas in the JRC-TMF dataset was used to target the sampling of GEDI footprints and quantify forest edge effects or canopy disturbance contagiousness between degraded and undisturbed forests on forest structure still classified as 'undisturbed forest'.

To ensure robust and comparable observations of forest structure metrics across the multiple classes of forest cover change, we considered a minimum of 600 GEDI samples for each 1.5° grid cell (~167 km at the Equator; around a given point) and a minimum of 7 grid cells per continent to derive continent-level statistics of forest RHs and AGBD. When analysing the time series (Fig. 3 and Supplementary Fig. 5), a minimum threshold of 30 GEDI samples for each time step of the trajectory—and a minimum of 600 GEDI samples for the sum of all the time steps—within each grid cell was required. Note that the time step does not refer to the GEDI date but to the JRC-TMF dataset where the timing of degradation, regrowth etc. is assessed. Similarly, for edge effect penetration, a minimum of 30 GEDI samples for each distance to the edge within each grid cell—and a minimum of 600 GEDI samples for the sum of all the distances—was required (see Fig. 2 and Supplementary Fig. 3). Metadata on the number of GEDI samples for aggregated classes of forest cover change is provided in Supplementary Fig. 2. Wall-to-wall information of relative heights with high spatial resolution on large scales, such as those produced by Lang[70] for canopy height only, will increase in the future the quantity of data, thus improving the quality and the robustness of the analysis.

The computation of canopy heights for intact, degraded, edge, and regrowing TMFs at the 1.5° grid cell level may vary due to local environmental and anthropogenic factors (for example, soil and forest types), leading to potential high variability in the reported canopy height statistics. In order to reduce sampling bias in the structural variable dataset, we randomly resampled GEDI observations 500 times within each 1.5° × 1.5° grid cell. We then summarized the random samples by calculating the mean and standard deviation of each structural variable, for each grid cell. Using this random sampling procedure based on the iteration (500 times) of sampling 300 GEDI observations for each grid cell, we found that the intra-grid variability of canopy heights was not significant. The results of the random sampling procedure show the low standard deviation for each class of RH98 distribution and forest considered (that is, intact, degraded, edge and regrowth) (Supplementary Fig. 10).

### Intact forest landscape assessment and comparison with Potapov's data product

We selected undisturbed forests in 2020 free from any disturbances located at: (1) a distance higher than the scale of the forest/non-forest edge effect identified at the grid cell level; and (2) more than 120 m distance from degraded forests from the JRC-TMF dataset (identified scale of the forest/degraded forest edge effect). Potapov's map of 2020[27] was constrained to the extent of TMFs (excluding mangroves and bamboo-dominated forests). We resampled our JRC-TMF-derived intact forest landscape (IFL) map from 30 m to 1 km. We computed the number of connected pixels (where each pixel contains the number of 4-connected neighbours) and then restricted them to values greater or equal to 500 to obtain an approximation of forest patch area greater than 500 km$^2$ (to match the definition of IFL of Potapov, with a minimum area of 500 km$^2$). Other criteria in Potapov on minimum IFL patch width (10 km) or minimum corridor width (2 km) were not implemented in our approach.

### Statistical tests

We performed a series of one-way ANOVAs to test for differences in the impacts of edge effects at different distances and times on the long-term recovery of the relative heights and biomass variables. ANOVAs were performed separately for each continent. For the height variables (RH50 and RH98), a series of standard one-way ANOVAs were used. In the analyses involving AGBD, we used a modified approach to propagate the prediction standard error associated with the AGBD dataset values which involved using a Monte Carlo approach ($n = 500$). In brief, we generated random noise that was added to the AGBD data. For each iteration $i$ we generated a noise term, noise$_{ij}$, by drawing a random value from a normal distribution with mean $\mu$ of 0 and s.d. equal to the prediction standard error of the AGBD ($\sigma_j$) for each GEDI footprint. The

noise can be represented as: $\text{noise}_{ij} \sim N(\mu, \sigma_j^2)$. We then perturbed the AGBD values by adding the generated noise to the original dataset ($\text{biomass}_{\text{original},i}$) for the $i$th iteration ($\text{biomass}_{\text{perturbed},i,j}$). We then performed an ANOVA for each iteration using the perturbed dataset and recorded the results. We subsequently examined the distribution density of the $F$ values. The results showed minimal variability suggesting that observed differences are robust to uncertainty associated with the AGBD values (Supplementary Fig. 11). For each ANOVA, we conducted a series of Tukey honest significant difference post hoc tests to assess significant differences between distance classes or time steps. The significance level was set to $P < 0.05$.

## Modelling deforestation risk

We assessed whether changes in RH50, RH98 and AGBD due to the occurrence of forest degradation and the distance to the edge represent an early warning signal of future deforestation. We retrieved GEDI footprints of 2019 sampled in forest degraded before 2018, followed or not by deforestation (2020–2022), together with GEDI footprints of 2020 sampled in forest degraded before 2019, followed or not by deforestation (2021–2022) and, footprints of 2021 sampled in forest degraded before 2020 followed or not by deforestation (2022). We then separated all samples based on their location within the first 120 m to the edge or beyond. The probability of deforestation in degraded forests was modelled using a generalized linear modelling approach. We fitted two models. One included only a single predictor, so that the percentage of intact forest height was the only predictor (Supplementary Fig. 12). The second model included two predictors—that is, the percentage of intact forest height and the distance to the edge. The error structure associated with the models was assumed to be binomial with a logit link function. A given model takes the general form:

$$Y_i \sim B(\pi_i, n_i) \tag{1}$$

$$E(Y_i) \sim n_i \times \pi_i \ \text{ and } \text{var}(Y_i) \sim n_i \times \pi_i \times (1 - \pi_i) \tag{2}$$

$$\text{logit}(\pi_i) = \eta_i \tag{3}$$

$$\eta_i = \alpha + \beta X_i \tag{4}$$

where $Y_i$ is the ith observation corresponding to the occurrence of a deforestation event and $\beta X_i$ is a matrix of regression coefficients.

Models were fitted within a Bayesian framework. We fitted the models using the programming language Stan via the brms package in the R software for statistical computing[71]. Models were run using 4 chains of 4,000 iterations each, with a warm-up of 1,000. We used the brms default priors for our model parameters. Convergence was visually assessed using trace plots (Supplementary Fig. 13) and the Rhat values (that is, the ratio of the effective sample size to the overall number of iterations, with values close to one indicating convergence). Markov chain Monte Carlo diagnostics showed a good convergence of the four chains, while the posterior distributions are centred around one peak value. The discriminatory ability of the models—that is, their ability to successfully predict a deforestation event—was assessed using the receiver operating characteristic (ROC) curve. We calculated the area under the curve (AUC) and compared the values with the guidelines provided by Swets[72].

## Cloud computing platform

All data extraction for this study was performed in Google Earth Engine[73], which provides the ability to compute GEDI footprint statistics and analyse the entire data records with high computational efficiency. The GEE data catalogue contains processed L2A and L4A GEDI data products—that is, the rasterized versions of the original GEDI products, with each GEDI shot footprint represented by a 25 m pixel.

## Reporting summary

Further information on research design is available in the Nature Portfolio Reporting Summary linked to this article.

## Data availability

All data used in this study are from publicly available sources. GEDI data are archived on NASA Distributed Active Archive Centers (DAACs). GEDI's footprint-level Relative Height data were taken from the GEDI02_A height and elevation product, available at LPDAAC: https://doi.org/10.5067/GEDI/GEDI02_A.002. GEDI's biomass data (AGBD) was taken from the GEDI04_A product also available at LPDAAC: https://doi.org/10.3334/ORNLDAAC/2056. The JRC-TMF dataset can be accessed at https://forobs.jrc.ec.europa.eu/TMF/data.php#gee. The slope is processed using SRTM data downloaded from https://developers.google.com/earth-engine/datasets/catalog/SRTMGL1_003. The Intact Forest Landscape dataset for 2020 can be downloaded at https://intactforests.org/data.ifl.html. Managed Forest Concessions dataset (accessed in February 2022) can be downloaded at https://data.globalforestwatch.org/documents/gfw::managed-forest-concessions-downloadable/about. The World Database on Protected Areas (accessed in October 2023) can be downloaded at www.protectedplanet.net. To ensure the full reproducibility and transparency of our research, we provide all of the data analysed during the current study. Pre-processed data, post-processed data, drivers of forest degradation, maps, codes and final figures developed in this study are made publicly available and briefly described to facilitate reproducibility and applicability. These data are permanently and publicly available on a Zenodo repository (https://doi.org/10.5281/zenodo.11235618)[74].

## Code availability

To ensure full reproducibility and transparency of our research, we provide all of the scripts used in our analysis. Codes used for this study (GEE and R scripts) are permanently and publicly available in a Zenodo repository: https://doi.org/10.5281/zenodo.11235618[74].

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

**Acknowledgements** The views expressed are purely those of the writers and may not in any circumstances be regarded as stating an official position of the European Commission. The authors thank S. Carboni for her help in the visual interpretation of selective logging. This study has been partly financed through the European Union's Amazonia+ programme and by the Directorate General for Climate Action of the European Commission (DG-CLIMA) through Lot 2 (TroFoMo (Tropical moist Forest Monitoring)) of the ForMonPol (Forest Monitoring for Policies) Administrative Arrangement.

**Author contributions** C.B., G.C., C.V., M.G., A.C. and F.A. conceived the idea and designed the methodology. C.B., G.C., A.C., M.G. and G.V. analysed the data and wrote the Google Earth Engine and R scripts. R.B. provided data on selective logging in Brazil. C.B. and G.C. wrote the manuscript with contributions from the other authors. All authors contributed critically to the interpretation of the results and gave final approval for publication.

**Competing interests** The authors declare no competing interests.

**Additional information**
**Correspondence and requests for materials** should be addressed to C. Bourgoin.

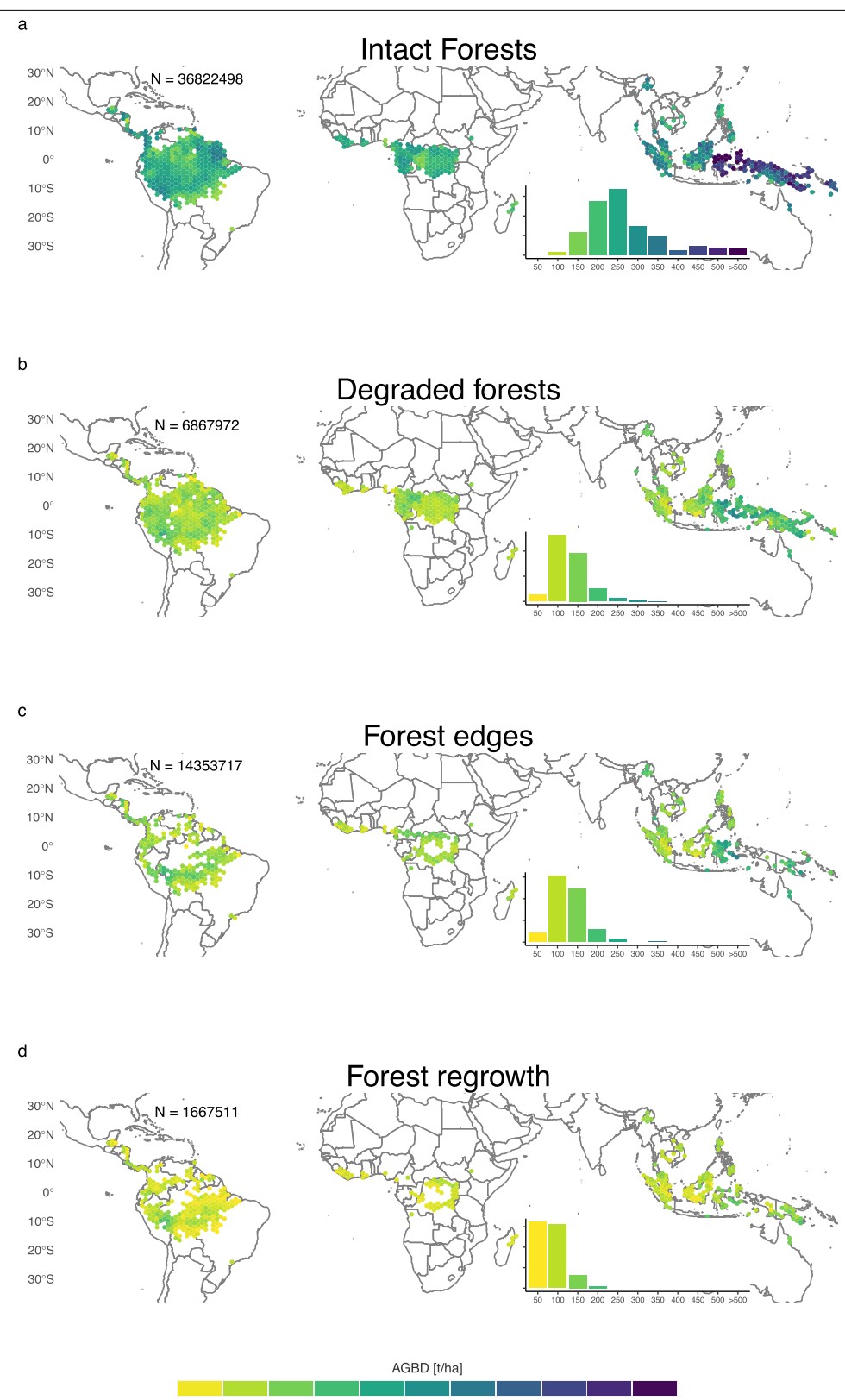

**Extended Data Fig. 1 | AGBD of intact, degraded, edge forests and regrowths.** Above Ground Biomass Density (AGBD) of intact forests (a) degraded forests (b), forest edges (c) and regrowths (d) in 1.5° (~167 km) grid cells between 30°N and 30°S. Grid cells with less than 600 GEDI samples or with no statistically significant differences in canopy height between intact and non-intact forests were masked (Welch two-sided t-test p < 0.05).

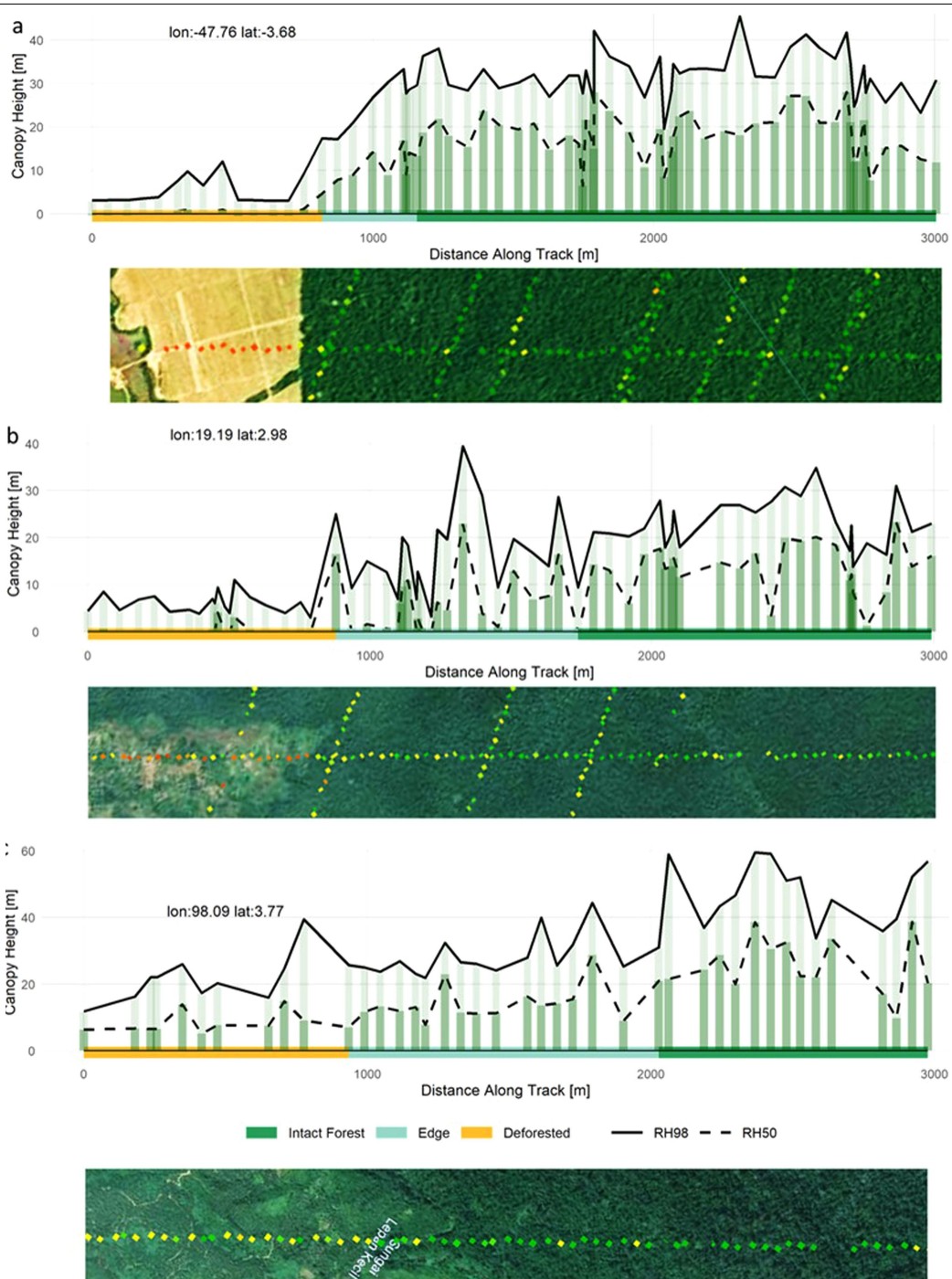

**Extended Data Fig. 2 | Transects of GEDI derived canopy heights.** Transects of GEDI relative heights (RH50 and RH98 from year 2020) in the Brazilian Amazon, Congo basin and West Sumatra crossing deforested land, edge forest (edge width of 350 m, 800 m and 1000 m, respectively) and intact forest. The two black lines represent the height of RH98 and RH50. Background data: Google (08/2019, 12/2023 and 03/2022 for panels a-c, respectively), © 2024 Maxar Technologies.

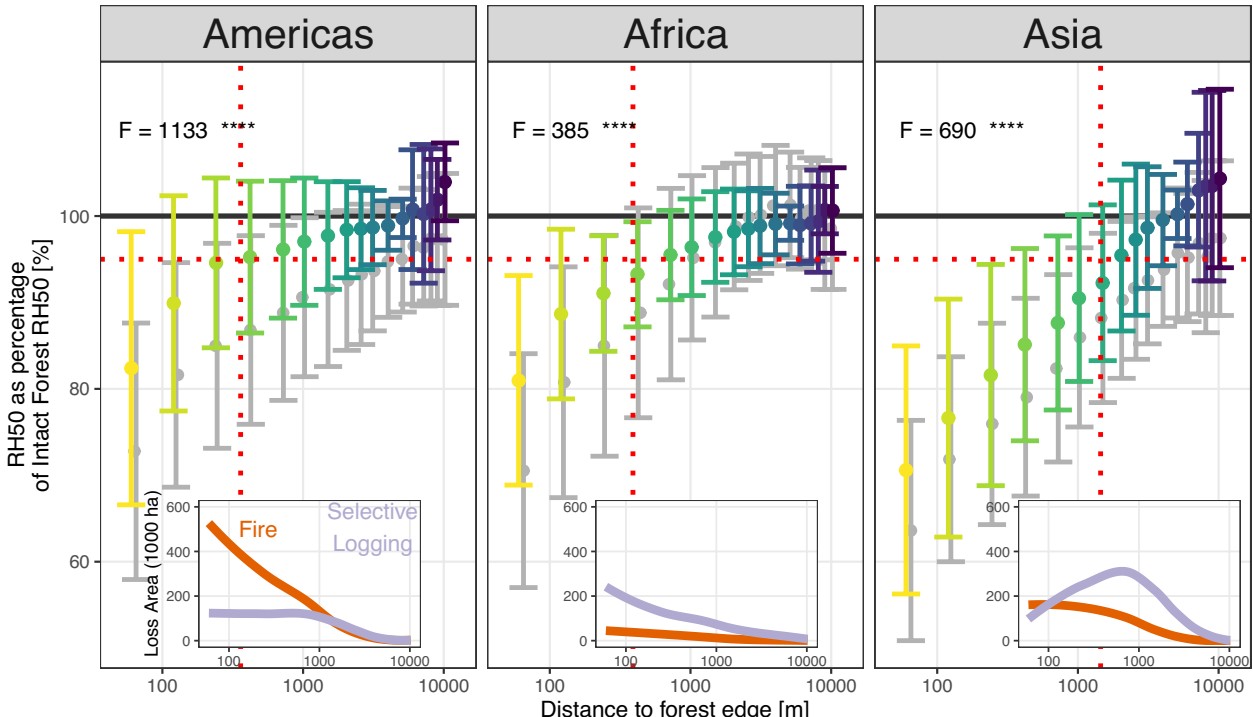

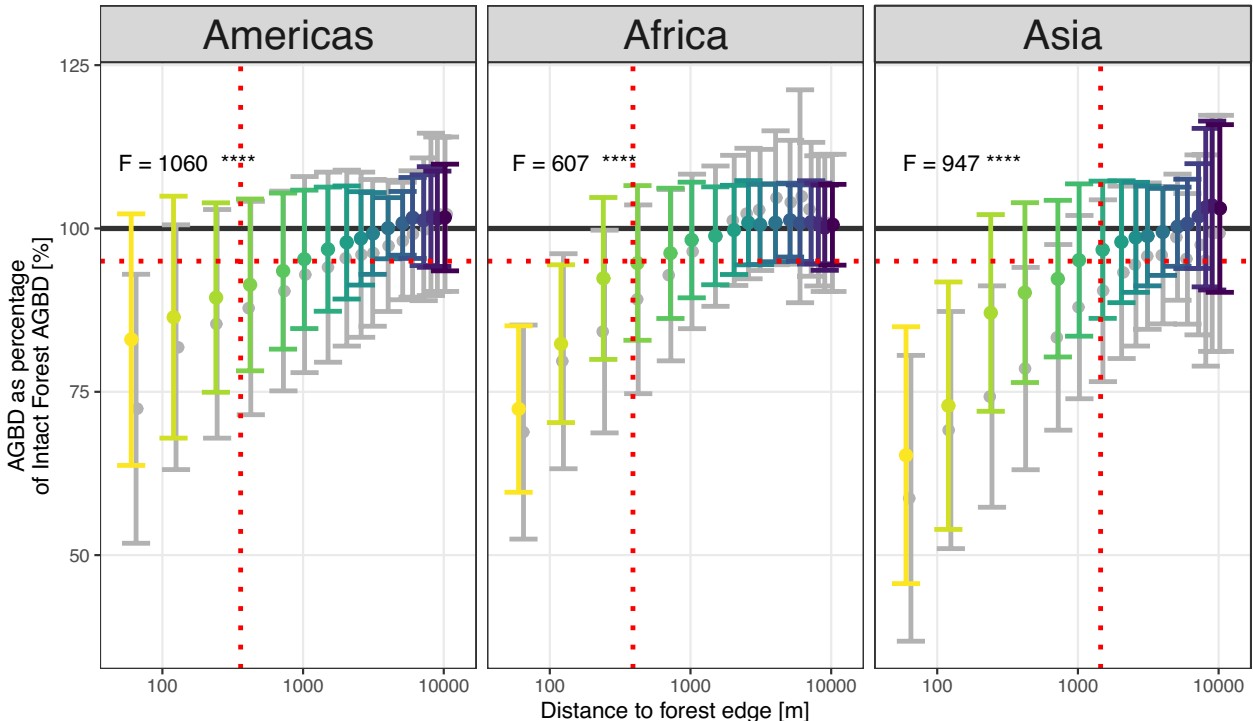

**Extended Data Fig. 3 | Spatial scale and magnitude of edge effects caused by deforestation on AGBD and RH50.** Average distribution of RH50 (panel a) and AGBD (panel b) of undisturbed forests (located at more than 120 metres from degraded forests) and all forests (including undisturbed and degraded forests) at various distances to the forest edge (agricultural and other land covers). The inset caption represents the degradation area due to fire (red curve) and selective logging (purple curve) calculated at various distances to the edge. The red dotted vertical line is placed at a distance equal to 350, 400, and 1500 m for America, Africa, and Asia, respectively, and corresponds to the distance between the forest edge and the point at which 95% of intact forest RH98 is reached (red horizontal dotted line). Vertical bars indicate the spatial standard deviation. F represents the F-Value in one-sided ANOVA and asterisks indicate the level of statistical significance for ANOVA: * $p \leq 0.05$, ** $p \leq 0.01$, *** $p \leq 0.001$, **** $p \leq 0.0001$, ns stands for not significant. Tukey post-hoc tests are available in supplementary data. The number of GEDI sample footprints for each distance to the forest edge is reported in Supplementary Fig. 3.

a

## Undisturbed forest
## Surrounding Degradation (RH98)

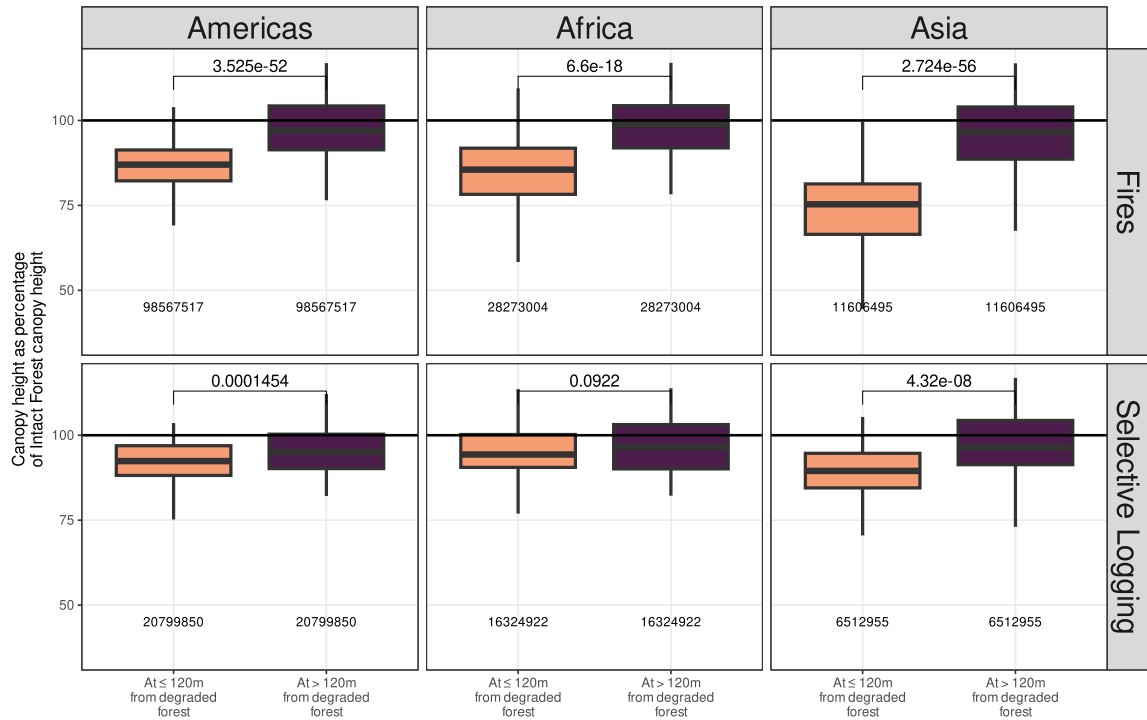

b

## Undisturbed forest
## Surrounding Degradation (RH50)

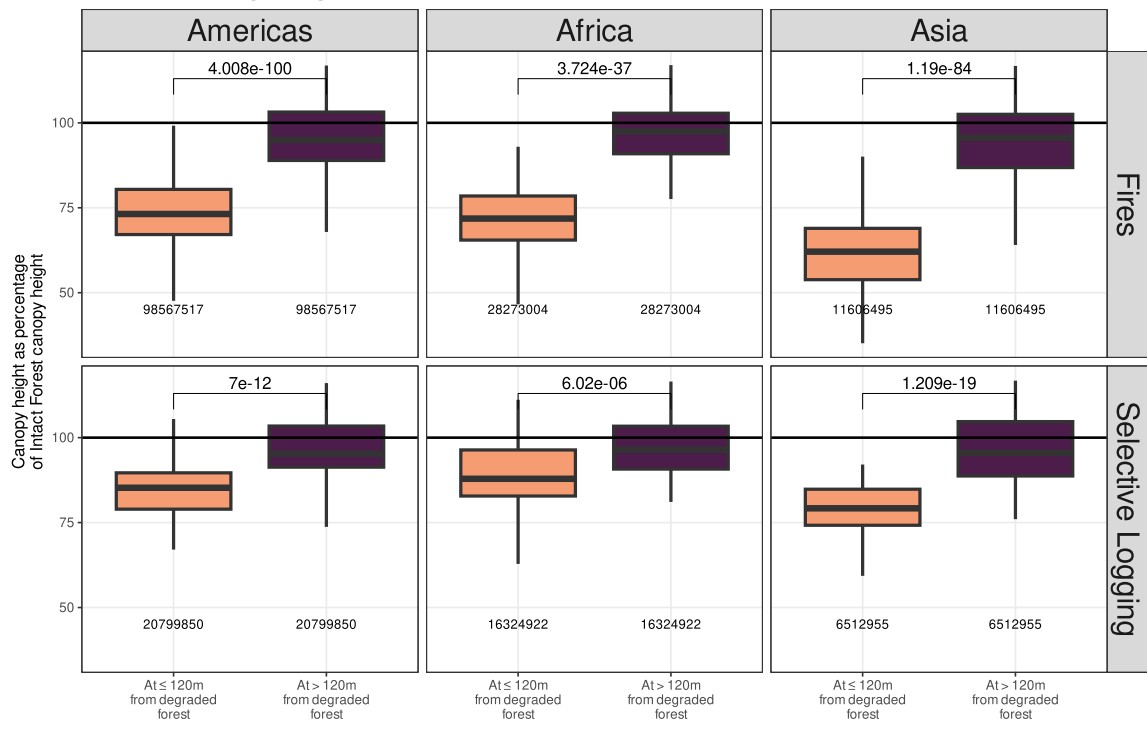

**Extended Data Fig. 4 | Edge effects caused by forest degradation from selective logging and fire.** Difference in RH98 (panel a) and RH50 (panel b) for forest classified as undisturbed in the JRC-TMF dataset located within and outside a buffer area (120 m radius) around logged or burned forest. Asterisks indicate the level of statistical significance of these comparisons: Adjusted p values are determined by two-tailed unpaired T test. The number at the bottom of each boxplot corresponds to the number of GEDI samples. Boxplot shows data from the 25th–75th percentile, the median (line) and whiskers extending to the minimum and maximum within 1.5× interquartile range.

a

## Intact Forest Landscapes – Bourgoin approach

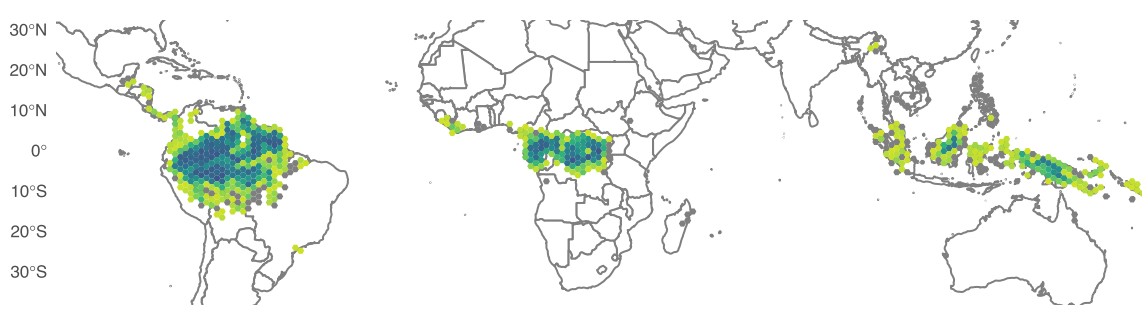

b

## Intact Forest Landscapes – Potapov approach

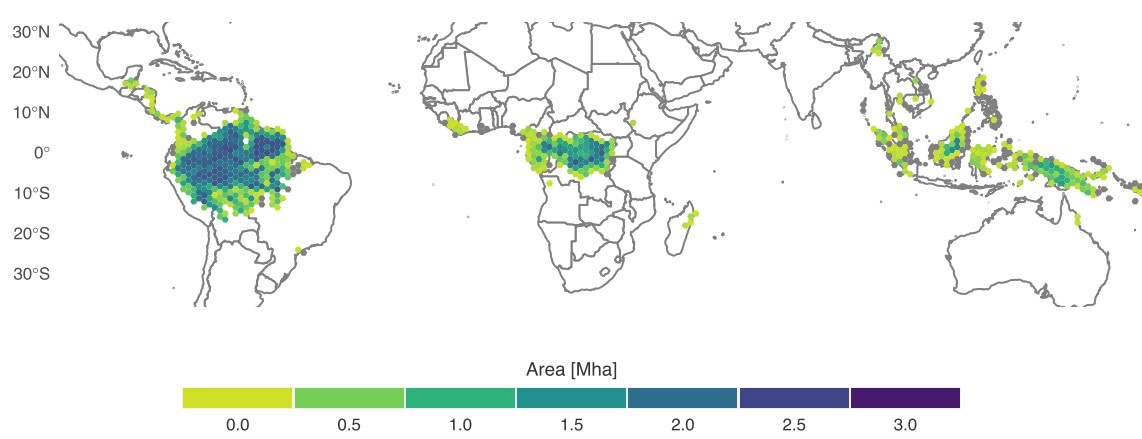

Area [Mha]

0.0    0.5    1.0    1.5    2.0    2.5    3.0

c

## Intact Forest Landscapes
## Bourgoin – Potapov

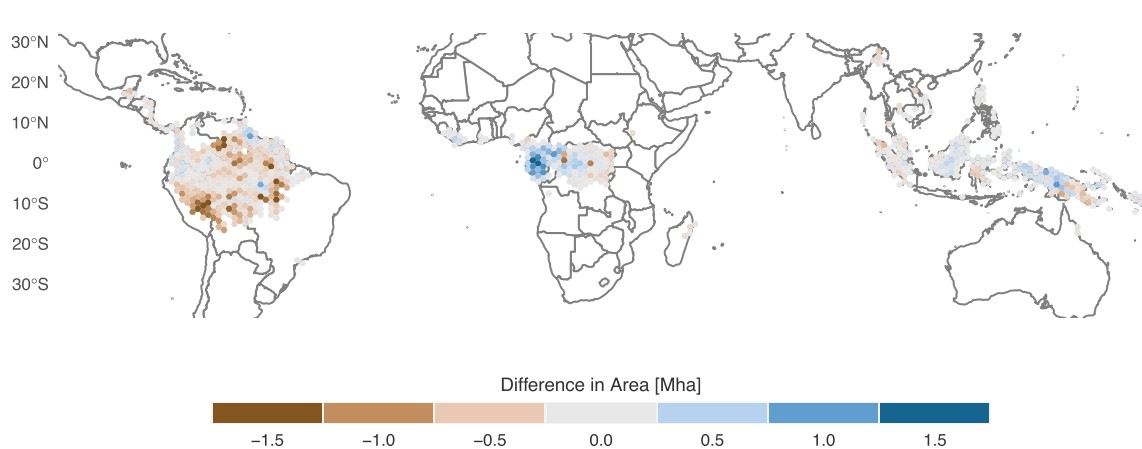

Difference in Area [Mha]

−1.5    −1.0    −0.5    0.0    0.5    1.0    1.5

**Extended Data Fig. 5 | Intact Forest Landscape mapping.** Intact forest landscape (IFL) mapping of the year 2020 in 1.5-degree grid cells between 30°N and 30°S. (a) Area of TMF-based IFL (here referred to as Bourgoin - the main author of this study - approach). Dark grey grid cells present no IFL area. (b) Area of IFL derived from Potapov's 2020. The extent was restricted to the tropical moist forest domain. We further excluded mangrove, forest conversion to water detected in the JRC-TMF dataset and bamboo-dominated forest areas to allow comparison with our approach. (c) Difference in area between our approach (i.e. JRC-derived/Bourgoin) and Potapov's.

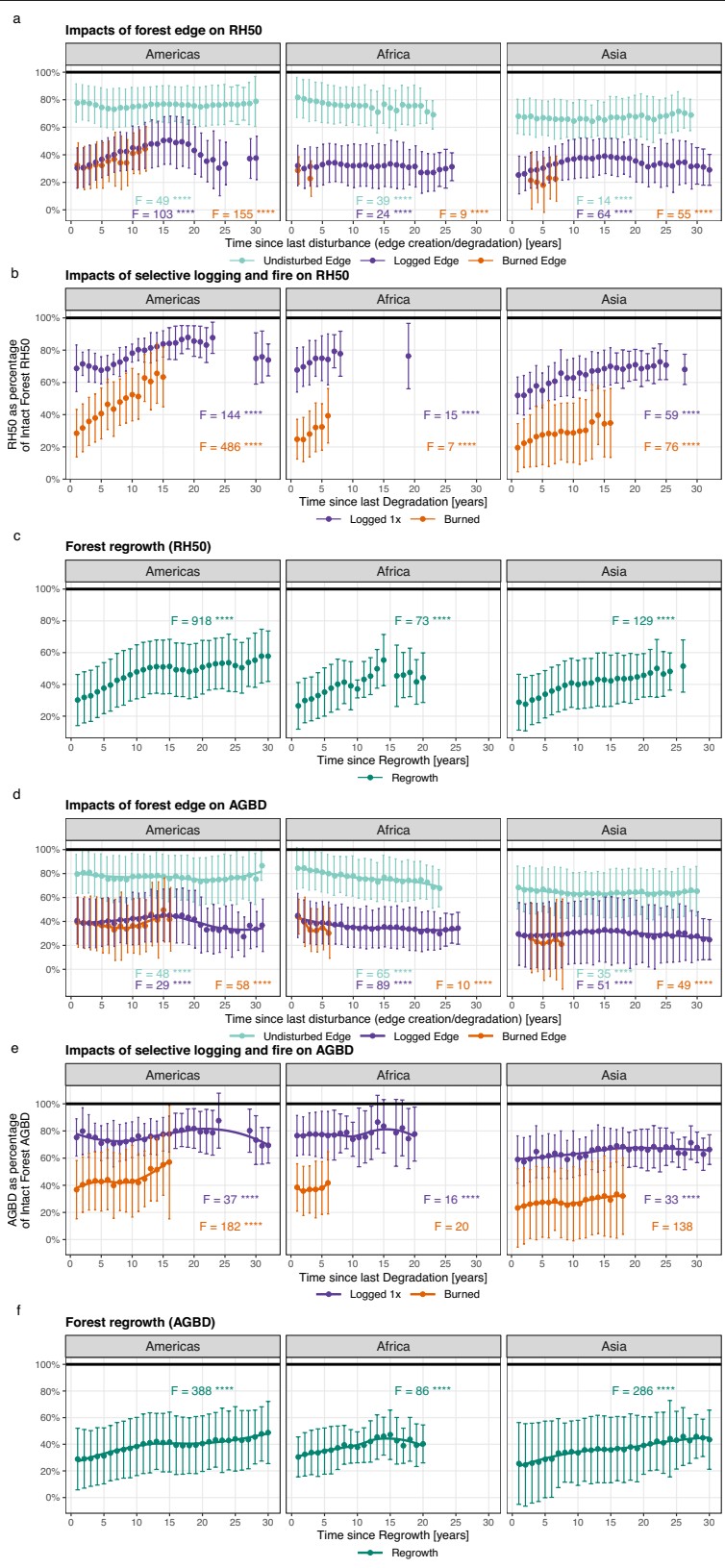

**Extended Data Fig. 6 | Impacts of forest degradation from selective logging, fire and edge effects on AGBD and RH50.** Long-term impacts on RH50 and AGBD from edge-desiccation effect (6a, 6d), degradation (fire or logging) of edge forest (6a, 6d), selective logging (logged 1x corresponds to logged once over the last 3 decades), fire (6b, 6e) and secondary forests regrowing on abandoned deforested lands (6c, 6f). Results are reported as the percentage of intact forest canopy height (solid line) after normalising the difference in RH50 and AGBD within each grid cell between intact forest and each forest type (degraded, edge forest, regrowth) and age. Dots represent the average value of RH50/AGBD and vertical bars indicate the spatial standard deviation. F represents the F-Value in one-sided ANOVA and asterisks indicate the level of statistical significance for ANOVA: * p ≤ 0.05, ** p ≤ 0.01, *** p ≤ 0.001, **** p ≤ 0.0001, ns stands for not significant. Tukey post-hoc tests are available in supplementary data. GEDI samples for each disturbance type and related time since disturbance are reported in Supplementary Fig. 5.

**Cumulative Return Energy**

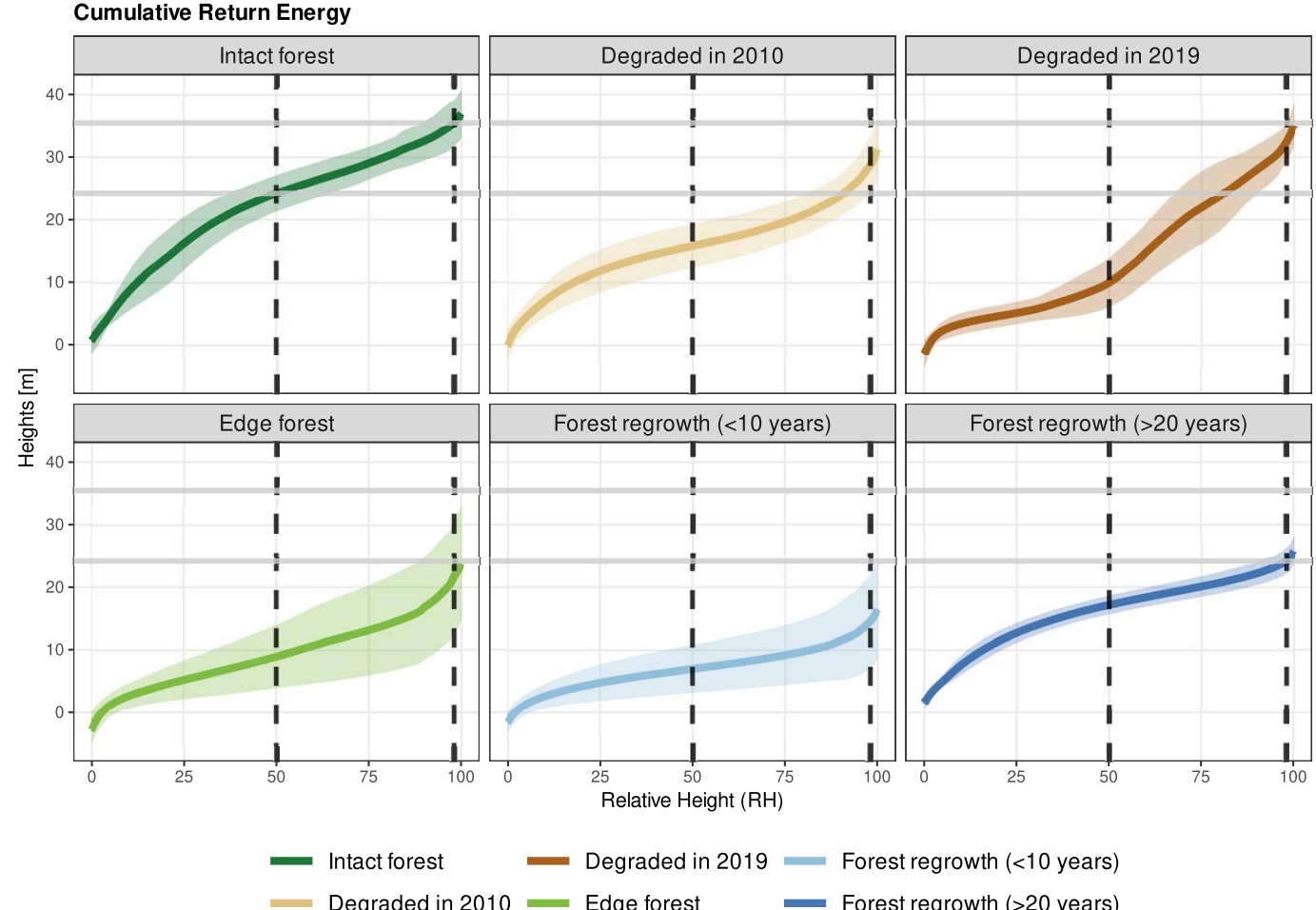

**Extended Data Fig. 7 | Cumulative Return Energy from GEDI for different forest types.** Cumulative Return Energy from GEDI L2A for different forest types (30 GEDI footprints for each forest type were sampled in a grid cell located in the northeast of the Brazilian Amazon). The vertical black lines refer to RH98 and RH50. The horizontal grey lines refer to the height of RH98 and RH50 for Intact Forest. The error band indicates the standard deviation.

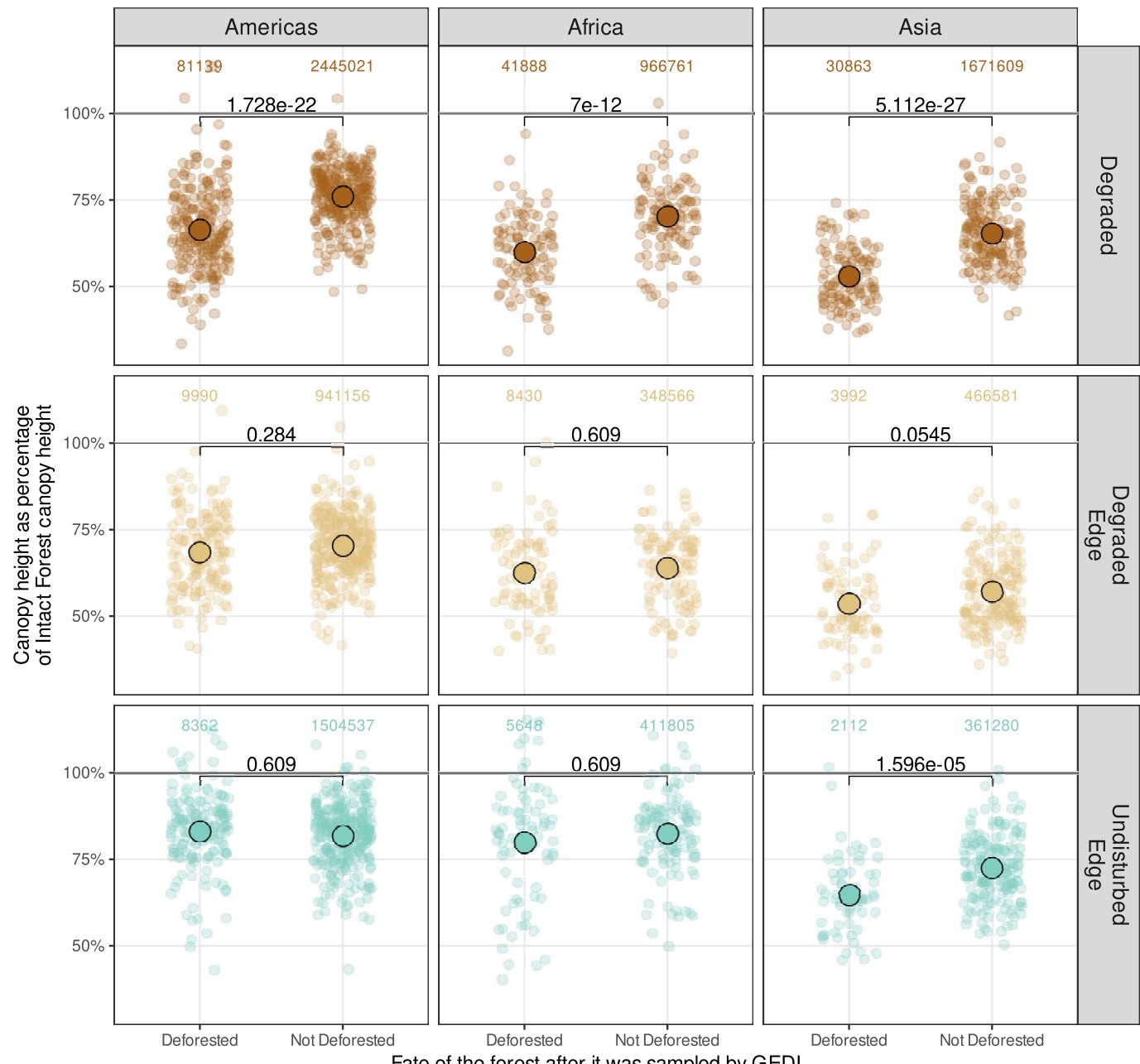

**Vulnerability of degraded and edge forest
to deforestation (RH98)**

**Extended Data Fig. 8 | Canopy heights of degraded forests before being deforested.** Differences in canopy heights of different forest types before deforestation, compared to canopy heights of similar and contemporary forests that are not deforested. The canopy heights (RH98) are all retrieved during the period 2019-2021, while the deforestation events occurred 1-3 years after GEDI measurements (2020-2022). The different forest types are degraded forest (located beyond the edge), degraded edge forest (width 120 m), and undisturbed edge forests. For degraded forests, degradation occurred before 2019, and no disturbance was observed during the year of GEDI data acquisition. Big circles represent the averages, and the small dots are individual GEDI samples. Adjusted p values are determined by two-tailed unpaired T test. The number at the top of each distribution corresponds to the number of GEDI samples.

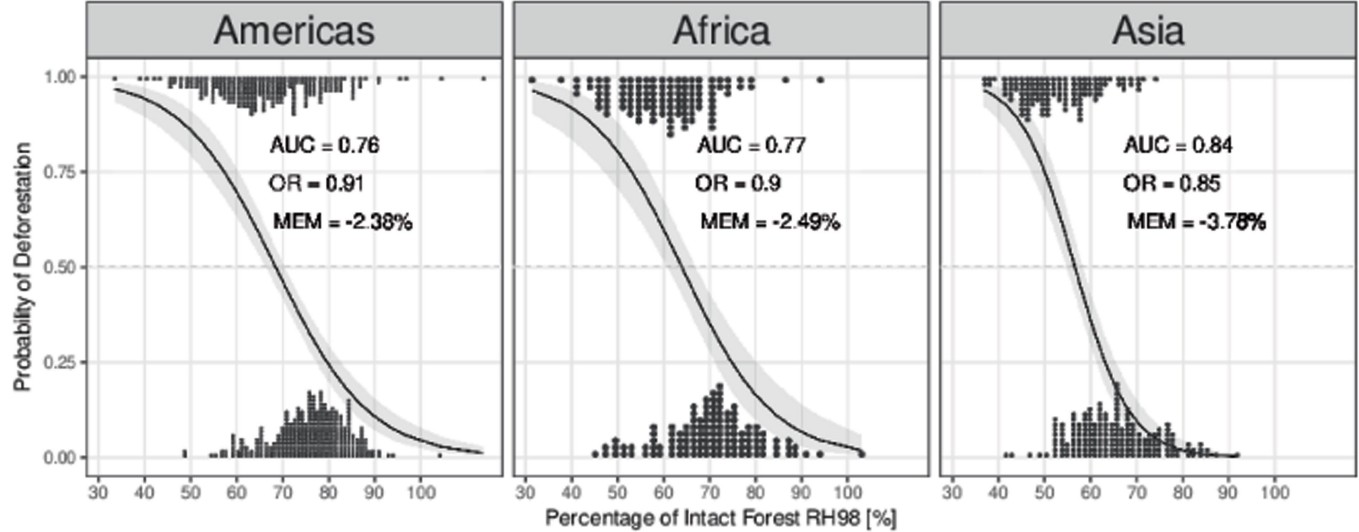

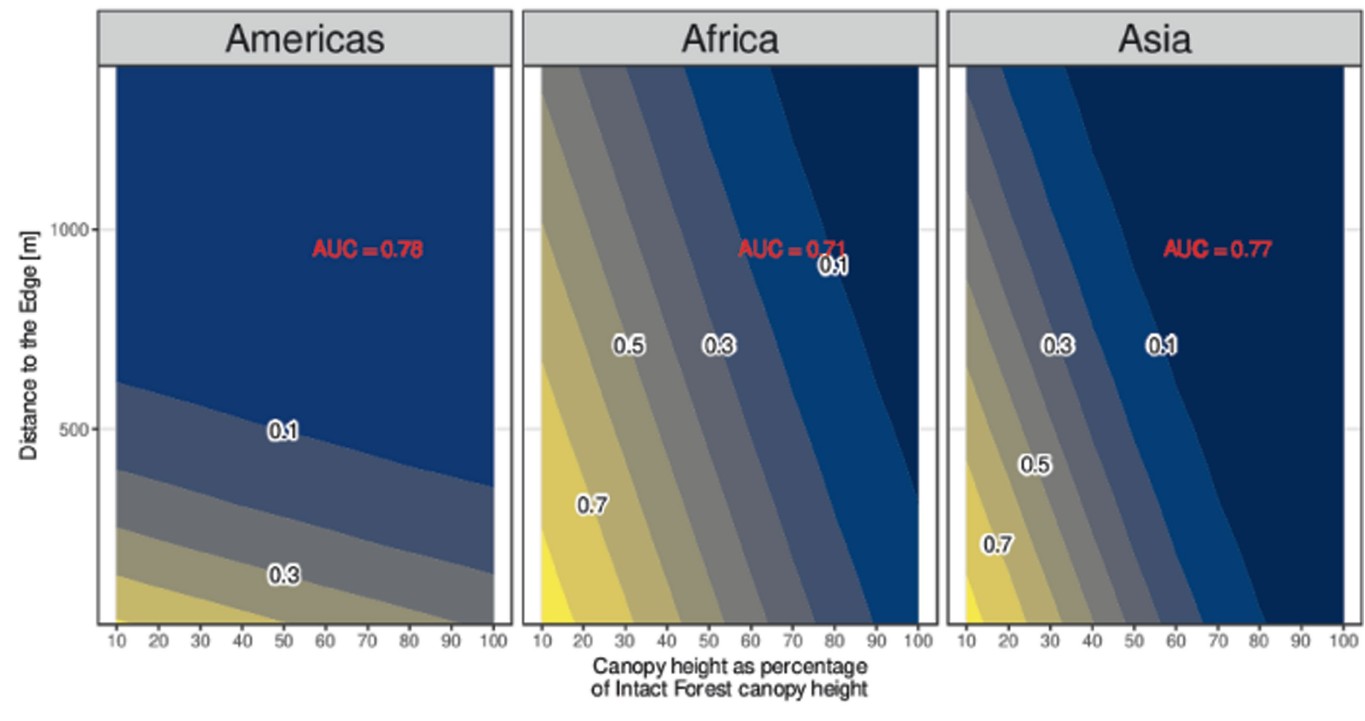

**Extended Data Fig. 9 | Modelling of deforestation risk based on canopy heights and distance to forest edge.** a) Marginal model plot of the binary logistic regression model of RH98 predicting whether deforestation was reported. The plot shows the expected influence of degraded forest structure (canopy height, expressed as a percentage of intact forest RH98) on the probability of deforestation. The grey shaded areas indicate the 95% credibility intervals of the predicted values. OR is the odds ratio. Stacked dots represent the GEDI samples deforested/not deforested. The marginal effect at the mean (MEM) quantifies for a one-point increase in RH98 (i.e. the x-axis) the associated percentage point variation in the probability of deforestation. AUC is the area under the ROC curve101. Marginal effects are partial derivatives of the regression equation for each variable in the model for each unit in the data. b) Plot of the marginal effects showing the probability of deforestation based on degraded forest structure (canopy height, expressed as a percentage of intact forest height) and the distance to the forest edge.

**Extended Data Table 1 | Summary statistics of canopy heights and AGBD for intact, degraded, edge forests and regrowths**

| Forest type | Continent | Number of GEDI shots | | Mean | | Standard Deviation | | Precision of the Mean | |
|---|---|---|---|---|---|---|---|---|---|
| | | RH98 | AGBD | RH98 [m] | AGBD [Mg/ha] | RH98 [m] | AGBD [Mg/ha] | RH98 [m] | AGBD [Mg/ha] |
| Intact forests | Americas | 16087545 | 25303980 | 28.563 | 239.512 | 7.440 | 129.876 | 0.002 | 0.026 |
| | Africa | 5880126 | 8986956 | 29.259 | 225.661 | 8.572 | 110.906 | 0.004 | 0.037 |
| | Asia | 1523450 | 2519005 | 34.397 | 370.812 | 10.750 | 205.194 | 0.009 | 0.129 |
| Degraded forests | Americas | 1728729 | 2791301 | 18.885 | 111.762 | 8.661 | 115.661 | 0.007 | 0.069 |
| | Africa | 704074 | 1099960 | 17.965 | 103.272 | 9.821 | 87.832 | 0.012 | 0.084 |
| | Asia | 1749092 | 2766798 | 20.173 | 142.991 | 10.718 | 128.692 | 0.008 | 0.077 |
| Forest edges (120m width) | Americas | 4234908 | 6903997 | 19.253 | 116.365 | 9.037 | 117.970 | 0.004 | 0.045 |
| | Africa | 2654101 | 4494123 | 19.799 | 121.980 | 10.052 | 96.790 | 0.006 | 0.046 |
| | Asia | 1585174 | 2809352 | 21.139 | 147.523 | 10.971 | 133.289 | 0.008 | 0.080 |
| Forest regrowth | Americas | 594291 | 936495 | 13.944 | 68.972 | 7.765 | 94.260 | 0.010 | 0.097 |
| | Africa | 67313 | 120529 | 14.299 | 73.730 | 9.234 | 73.817 | 0.036 | 0.213 |
| | Asia | 177179 | 306946 | 15.721 | 98.413 | 9.348 | 93.695 | 0.022 | 0.169 |

Number of GEDI shots and statistics (mean, standard deviation and precision) of grid cells by continent of RH98 and AGBD for intact forest, degraded forest, forest edge and forest regrowth. The precision of the RH98 was obtained by computing the standard error of the RH98 values at footprint levels for each continent. The precision of the AGBD was obtained by computing the standard error of the AGBD values at footprint levels for each continent. The mean, standard deviation and precision of the AGBD predictive standard error (AGBD_SE) are shown in Supplementary Table 1. Note that issues linked to the non-randomness and spatial autocorrelation of GEDI samples and the propagation of the regression error associated to each AGBD estimate are not integrated in this computation of summary statistics[67,75].

# Reporting Summary

## Statistics

For all statistical analyses, confirm that the following items are present in the figure legend, table legend, main text, or Methods section.

| n/a | Confirmed | |
|---|---|---|
| ☐ | ☒ | The exact sample size (*n*) for each experimental group/condition, given as a discrete number and unit of measurement |
| ☐ | ☒ | A statement on whether measurements were taken from distinct samples or whether the same sample was measured repeatedly |
| ☐ | ☒ | The statistical test(s) used AND whether they are one- or two-sided<br>*Only common tests should be described solely by name; describe more complex techniques in the Methods section.* |
| ☒ | ☐ | A description of all covariates tested |
| ☒ | ☐ | A description of any assumptions or corrections, such as tests of normality and adjustment for multiple comparisons |
| ☐ | ☒ | A full description of the statistical parameters including central tendency (e.g. means) or other basic estimates (e.g. regression coefficient) AND variation (e.g. standard deviation) or associated estimates of uncertainty (e.g. confidence intervals) |
| ☐ | ☒ | For null hypothesis testing, the test statistic (e.g. *F*, *t*, *r*) with confidence intervals, effect sizes, degrees of freedom and *P* value noted<br>*Give P values as exact values whenever suitable.* |
| ☐ | ☒ | For Bayesian analysis, information on the choice of priors and Markov chain Monte Carlo settings |
| ☒ | ☐ | For hierarchical and complex designs, identification of the appropriate level for tests and full reporting of outcomes |
| ☒ | ☐ | Estimates of effect sizes (e.g. Cohen's *d*, Pearson's *r*), indicating how they were calculated |

*Our web collection on statistics for biologists contains articles on many of the points above.*

## Software and code

Policy information about availability of computer code

| Data collection | All data extraction for this study was performed in Google Earth Engine, which provides the ability to compute LiDAR (GEDI) statistics and analyze the entire data records with high computational efficiency. Google Earth Engine is a cloud-based infrastructure that enables "access to high-performance computing resources for processing very large geospatial datasets". It consists of "a multi-petabyte analysis-ready data catalogue co-located with a high-performance, intrinsically parallel computation service". |
|---|---|
| Data analysis | Data analysis was performed on R (version 4.2.2). |

For manuscripts utilizing custom algorithms or software that are central to the research but not yet described in published literature, software must be made available to editors and reviewers. We strongly encourage code deposition in a community repository (e.g. GitHub). See the Nature Portfolio guidelines for submitting code & software for further information.

# Data

Policy information about availability of data

All manuscripts must include a data availability statement. This statement should provide the following information, where applicable:

- Accession codes, unique identifiers, or web links for publicly available datasets
- A description of any restrictions on data availability
- For clinical datasets or third party data, please ensure that the statement adheres to our policy

> To ensure full reproducibility and transparency of our research, we provide all of the data analysed during the current study. The data are permanently and publicly available on a Github and Zenodo repository (immediately upon publication or for reviewers).

# Research involving human participants, their data, or biological material

Policy information about studies with human participants or human data. See also policy information about sex, gender (identity/presentation), and sexual orientation and race, ethnicity and racism.

| | |
|---|---|
| Reporting on sex and gender | *Use the terms sex (biological attribute) and gender (shaped by social and cultural circumstances) carefully in order to avoid confusing both terms. Indicate if findings apply to only one sex or gender; describe whether sex and gender were considered in study design; whether sex and/or gender was determined based on self-reporting or assigned and methods used. Provide in the source data disaggregated sex and gender data, where this information has been collected, and if consent has been obtained for sharing of individual-level data; provide overall numbers in this Reporting Summary. Please state if this information has not been collected. Report sex- and gender-based analyses where performed, justify reasons for lack of sex- and gender-based analysis.* |
| Reporting on race, ethnicity, or other socially relevant groupings | *Please specify the socially constructed or socially relevant categorization variable(s) used in your manuscript and explain why they were used. Please note that such variables should not be used as proxies for other socially constructed/relevant variables (for example, race or ethnicity should not be used as a proxy for socioeconomic status). Provide clear definitions of the relevant terms used, how they were provided (by the participants/respondents, the researchers, or third parties), and the method(s) used to classify people into the different categories (e.g. self-report, census or administrative data, social media data, etc.) Please provide details about how you controlled for confounding variables in your analyses.* |
| Population characteristics | *Describe the covariate-relevant population characteristics of the human research participants (e.g. age, genotypic information, past and current diagnosis and treatment categories). If you filled out the behavioural & social sciences study design questions and have nothing to add here, write "See above."* |
| Recruitment | *Describe how participants were recruited. Outline any potential self-selection bias or other biases that may be present and how these are likely to impact results.* |
| Ethics oversight | *Identify the organization(s) that approved the study protocol.* |

Note that full information on the approval of the study protocol must also be provided in the manuscript.

# Field-specific reporting

Please select the one below that is the best fit for your research. If you are not sure, read the appropriate sections before making your selection.

☐ Life sciences  ☐ Behavioural & social sciences  ☒ Ecological, evolutionary & environmental sciences

For a reference copy of the document with all sections, see nature.com/documents/nr-reporting-summary-flat.pdf

# Ecological, evolutionary & environmental sciences study design

All studies must disclose on these points even when the disclosure is negative.

| | |
|---|---|
| Study description | This study consists in analyzing the spatial and temporal effects of human-driven forest disturbances (including forest degradation due to selective logging, fire and edge effect and forest regrowth) on forest vertical canopy structure using around 39 million GEDI measurements across the pan-tropical belt. |
| Research sample | The research sample consists of LiDAR (GEDI) samples over different forest types from the JRC-TMF dataset (https://forobs.jrc.ec.europa.eu/TMF/). It covers intact (~23 million sample plots), degraded (~4.5 million), edge (~8.5 million) and regrowing (~1.1 million) tropical moist forests, representing large gradients of environmental, climatic and anthropogenic conditions of the humid tropics. Regarding edge effects, the sample covers forest at various distances to non-forest edge (i.e. urban or agricultural areas), varying from 60m to 10km. On the temporal aspects, the sample covers the last 33 years of forest disturbances and recovery (1990-2022) across various combination of disturbance types representative of the main direct drivers of forest degradation in the tropics (selective logging, fire and edge effect). |

| | |
|---|---|
| Sampling strategy | The sampling size depended on the availability of GEDI data (filtered to its highest quality). The resulting sampling sizes for each forest dynamics and for each types of forest disturbances are reported in the manuscript as numbers and maps. |
| Data collection | Data collection was based on the extraction of GEDI-derived canopy structure metrics (canopy height - RH98, height of median energy - RH50 and aboveground biomass density - AGBD) from 2019-2022 for different forest types captured by the JRC-TMF dataset. We created a fishnet grid of 1.5*1.5 degrees (~167km) covering the pantropical spatial domain. GEDI-derived metrics were extracted within each grid cell of this fishnet system. Ancillary datasets on selective logging (visual delineation of selective logging areas covering Brazil, French Guiana, Guyana, Cameroon, Central African Republic, Gabon, Congo, the Democratic Republic of Congo, Indonesia, Malaysia and Papua New Guinea) and forest fire (Global forest loss due to fire from the Global Land Analysis and Discovery) were used to refine the attribution of forest disturbance type. |
| Timing and spatial scale | The GEDI data was acquired from 2019 to end of 2022. It was analyzed on a yearly basis to ensure good precision in the assessment of the years since the last disturbance, the years since edge creation, or the age of forest regrowth. Forest cover dynamics of degradation, recovery and regrowth covered the period 1990-2022 coinciding with the monitoring period of forest cover change of the JRC-TMF dataset. Deforestation risk analysis was done on tropical moist forest cover change between 2020-2022 (1-3 years after the acquisition of GEDI canopy structure measurement). The spatial scale covers the tropical humid forest cover as defined by the Food and Agriculture Organization of the United Nations. |
| Data exclusions | GEDI data not passing the different quality filters (e.g. power beam mode, night acquisition, quality flags, slope less than 20 degrees, not classified as urban or water, beam sensitivity higher than 0.95) or forest definition (canopy height more than 5m) were excluded from the analysis. Mangrove forest were excluded from the analysis as periodical tidal floods will impact the consistent estimation of canopy height over time. Afforestation was excluded from the analysis due to the limited number of GEDI samples. |
| Reproducibility | All attempts to repeat the experiment were successful. |
| Randomization | Not relevant, as we used all the data to perform our analysis. |
| Blinding | Not relevant for this field (geoscience and forestry). |

Did the study involve field work? ☐ Yes ☒ No

# Reporting for specific materials, systems and methods

We require information from authors about some types of materials, experimental systems and methods used in many studies. Here, indicate whether each material, system or method listed is relevant to your study. If you are not sure if a list item applies to your research, read the appropriate section before selecting a response.

## Materials & experimental systems

| n/a | Involved in the study |
|---|---|
| ☒ | Antibodies |
| ☒ | Eukaryotic cell lines |
| ☒ | Palaeontology and archaeology |
| ☒ | Animals and other organisms |
| ☒ | Clinical data |
| ☒ | Dual use research of concern |
| ☒ | Plants |

## Methods

| n/a | Involved in the study |
|---|---|
| ☒ | ChIP-seq |
| ☒ | Flow cytometry |
| ☒ | MRI-based neuroimaging |

## Plants

| | |
|---|---|
| Seed stocks | *Report on the source of all seed stocks or other plant material used. If applicable, state the seed stock centre and catalogue number. If plant specimens were collected from the field, describe the collection location, date and sampling procedures.* |
| Novel plant genotypes | *Describe the methods by which all novel plant genotypes were produced. This includes those generated by transgenic approaches, gene editing, chemical/radiation-based mutagenesis and hybridization. For transgenic lines, describe the transformation method, the number of independent lines analyzed and the generation upon which experiments were performed. For gene-edited lines, describe the editor used, the endogenous sequence targeted for editing, the targeting guide RNA sequence (if applicable) and how the editor was applied.* |
| Authentication | *Describe any authentication procedures for each seed stock used or novel genotype generated. Describe any experiments used to assess the effect of a mutation and, where applicable, how potential secondary effects (e.g. second site T-DNA insertions, mosiacism, off-target gene editing) were examined.* |

