## [Peer Review file · Nature]

Manuscript Title: Human degradation of tropical moist forests is greater than previously estimated

Reviewer Comments & Author Rebuttals

Reviewer Reports on the Initial Version:

Referees' comments:

Referee #1 (Remarks to the Author):

This paper presents an analysis of the impact of human disturbance (fire, logging, edge effects) on tropical moist forests, on structure and resilience. The paper combines GEDI (for the structural element, particularly the height metrics RH50 and RH98, along with estimates of disturbance/degradation state from Landsat time series.

There is some very interesting work here – in particular the assessment of the distance from edges that structural change can persist & also the potential impact of structural change on future recovery. These are important topics in our understanding of forest resilience, C stocks and also how we go about managing and mitigating change in TMFs. There is new insight here, which is potentially really important in changing how we think about forest disturbance more widely in the tropics.

The approach is broadly sound in general, but there are a lot of unanswered questions, rather vague analysis and assertions, and some important areas of uncertainty about how the data are used throughout. This means that in many places, there are rather vague and discursive assertions made, efforts to explain cause and effect, and generalities that are not really supported by the analysis. There is also a lack of clarity in what we are meant to be looking at i.e. a failure to guide the reader to the most important aspects. I have provided specific comments on the annotated manuscript, but I note some more general issues here. I realise that Nature discourages the use of marked-up manuscripts, but this is by far the best way to provide specific feedback. I provide this file to be used as is seen fit.

There is no way to validate the assessment of disturbance – unless there were an independent dataset eg from locally-assessed v specific impacts. So the term 'accuracy' is used in many places, but there is no reference measurement against which to assess this. Eg in the Conclusions the authors say "... provides an incontrovertible assessment of forest structural attributes and enables the large-scale monitoring of forest degradation and recovery. This type of accurate and transparent assessment ...". But this study is neither incontrovertible, nor accurate in a specific, quantifiable way. The best that can be said is 'precise' and then to compare with other studies. Similarly in the same passage the authors say "... previously undervalued and considered de facto of ~ 100 m - demonstrates that the overall impact of fragmentation across the pantropical belt is severely underestimated ..." But underestimated compared to what? What they mean is that their estimate is different (larger) from previous work. Which is potentially important of course, but without a direct causal explanation and/or independent validation, there is no 'underestimation' per se.

Different height metrics are used in different ways in different places, without any clear reasoning as to why RH50 or RH98 or both. This is v important as these are the key structural indicators. So unless there is a clear reason why each is used and where, this is somewhat hard to follow and confusing. On a related note, FHD (the foliage height diversity) is mentioned early on, but then never used except in the Extended Data – it is not presented in the main results at all. So why is it even mentioned?

A general question arises in terms of the use of Landsat for one aspect of the analysis (defining the degradation status and change), and GEDI for the other – the analysis of the impact on canopy height. As the authors acknowledge in a couple of places, passive optical data like Landsat are relatively insensitive to structural variation that isn't high intensity deforestation – which is why GEDI is so useful. BUT you then have a data source (Landsat) that is insensitive to lower levels of disturbance being used to define the regions, where gedi is then being used to assess canopy height. This seems somewhat of a mismatch in that we know Landsat will underestimate lower levels of degradation, but those data are then used to assess the spatial impact (of disturbance) on height. There is also a lack of mention of Radar here – there is quite a lot of work done to assess structural variation in tropical forests from radar, but that is not mentioned at all which seems like an oversight (particularly given the v long list of references).

Overall, I would emphasise that there is definitely some very interesting work here, particularly on the distance and temporal persistence of disturbance. I think with a bit of work this could be a really interesting paper of wide interest to Nature readership. But I do think it needs some work – streamlining, and a much clearer focus on the key message(s). In particular (see my comment on lin 264) the observation that “No statistical differences in CH were found for undisturbed edges in the three continents, confirming the role of unsustainable logging and fire in enhancing the risk of deforestation.” – I think this is really important, but it is actually more understated than some of the other less convincing results.

Referee #2 (Remarks to the Author):

General Comments

This paper provides a nice example of the application of structural measurements of forests from NASA's GEDI mission towards understanding changes that occur after degradation and deforestation with regards to intact forests. It also leverages these data to examine the impact of disturbance edges on interior intact forests over a vast area of the tropics. In both cases, there is an attempt to analyze impact of the type and timing of disturbances. The research underscores the new avenues of exploration global estimates of structure from GEDI are enabling.

In reviewing this submission, several points emerged. First, there may be potential issues in the paper with respect to statistical methods and GEDI data that should be addressed (and expanded on below). Secondly, while the paper in any section is well written, its overall narrative structure could be improved. There needs to be a cleaner story that the reader can more easily grasp and not be

confused by the very many statistics and results that are presented. Some type of overall table of results might help with proper positioning relative to the context of and motivations of the paper. What specific hypotheses are being tested? How are these reflective of major unanswered questions regarding the role of deforestation and degradation? What was discovered relative to these?

The concept of GEDI “unveiling” information about forest structure that has been hidden (from the title) is a very good one and could be emphasized more. Even so, the reader may ask what exactly has been hidden from view? That structure changes after disturbance and will take time to recover is self-evident. Rather is not the discovery element the quantification of these changes that GEDI enables? Approaching the paper and its organization of results from that perspective of discovery grounded in major conclusions within a framework of inquiry that is clearly linked to the important ecological issues would make it more readable, accessible, and ultimately exciting.

The paper could benefit from a conceptual model that relates GEDI metrics (RH98, RH50, FHD) to biophysical elements on the landscape. An introductory paragraph about the metrics and their ecological significance would go a long way towards getting the reader to stop thinking in terms of metrics. It may be that these kinds of lidar metrics are so new as applied to this problem over vast areas that an unambiguous connection between “degradation” and changes in the metrics is not possible yet. Rather, it is the exploratory nature of this submission that is pointing the way to a new paradigm – these metrics provide new ways of identifying and understanding impacts.

It appears the paper is lacking a definition of what “degradation” is. How are the use of these GEDI metrics consistent with established definitions of degradation? How would recovery from degradation appear in the metrics through time? Again, loss of canopy height usually means loss of trees. But “loss” and “recovery” of metrics are not so well-defined.

The paper asserts that “intact” forests show signs of degradation 2 km in from an edge – so how can they be considered intact? This is one of the critical points of the paper, that passive optical remote sensing is misrepresenting the notion of intact. If this result is validated in other studies, it would be an exciting finding indeed. The authors should make more of this point from the very outset – perhaps this is the central finding? Can comparisons with other maps of intact forests be done formally? Under what conditions and where geographically do we see this divergence from optical estimates? Do the specifics lead to summary conclusions that are geospatially comprehensible?

It was disappointing to find that aboveground biomass was not also included in the analysis. Those data are readily available from the mission. Because biomass incorporates multiple aspects of the complex factors that could affect forests after disturbance and degradation (soil compaction, loss of fertility, microclimate changes) it would be a logical inclusion. Yes, it makes the analysis more difficult because there is an added source of uncertainty which comes from calibration models that turn GEDI metrics into biomass estimates. But we should see concomitant losses in biomass that are quantified. Furthermore, the accumulation of biomass through time after a disturbance is of great interest (the net impact of deforestation and degradation and subsequent regrowth). The logical inference is that the carbon content of intact forests is impacted by edges, but what was the magnitude of those losses? This paper could quantify that and seems within scope.

The statistical methods assume random sampling. GEDI shots are not random, they are cluster samples. Some justification is required here and discussion of impacts if the assumption is violated in the formal testing. Some of the minimum shot criteria (e.g., 10 shots) seem quite low and highly susceptible to non-random sampling issues and should be tested. There also needs to be a quantitative justification for why a cell with only 300 shots over 200 km is sufficient. Was this tested? With regards to the above, it would be helpful if the paper could show examples of actual GEDI transects, not aggregates, but actual track lines, that illustrate interior forest structures are impacted near edges.

The comments on the impact of slope are not quite correct. Slopes over bare areas can lead to spurious canopy height but its more complicated than that. Slopes on vegetated areas do not have a consistent lowering or raising of RH98 and RH50, i.e., a consistent bias. This depends on the actual configuration of stems in the footprint, and whether the slope is facing towards or away from the sensor (so a function of the slope aspect and the sensor off-nadir angle). The restriction to low slopes likely will bias results (e.g., steep slopes may be much less susceptible to degradation from logging as its far more difficult to access these). What can we say about this potential bias?

I did not find the explanation of why the analysis was limited to only year of GEDI data readily justified. It that what was on GEE? There are at 3+ years of data that are there. Why can we not filter out shots in subsequent years that were disturbed in those years? That would have greatly increased available data, especially given some of the low shot counts.

The analysis is not entirely novel given that four of the co-authors have just published a paper (29 March 2023) in Nature Communications Earth & Environment: "Spaceborne LiDAR reveals the effectiveness of European Protected Areas in conserving forest height and vertical structure" (2nd author Ceccherini is 1st author of the Europe paper). That paper shares many elements to this submission. Referencing this paper and including some discussion of the difference between "intact" and "protected" areas should be done. If this paper were repeated only for protected areas in the tropics, would we see the same results?

Lastly, the conclusions are too emphatic. Saying the results are "incontrovertible" is fraught here given uncertainties in data and methods. The paper would be improved with a more meaningful discussion of limitations and the potential impact of assumptions on the conclusions. We have seen many examples of how reanalysis of remote sensing data can lead to different outcomes. Hence, circumspection in the conclusions and acknowledgement that subsequent processing and analysis of the GEDI data might lead to some changes would be advised. This is especially true given that the final data sets from the mission are not published and only one year of data was used.

Specific comments:

Line 48-49: Check sentence construction

Line 51: Reference UNFCC

Line 70: Reference GEDI

Line 88-89: Need more detail on what RH98/RH50 are to understand going forward.

Line 92: "plots" should be "footprints."

Line 93: No definition what “range” means here. Is the standard deviation of the bootstrap heights? Also, remember to define somewhere that “canopy height” refers to the tallest thing in a GEDI footprint.

Line 97-98: What are these canopy heights refereeing too? What does 29.4 +/- 2.94 m refer to? That is the mean height of something, and the SD is only 2.94m? Is that the precision of mean? What exactly? How did you statistically prove the differences are significant? If the uncertainties are precision of the mean, they don’t look significant.

Line 122: Can you do a t-test with non-random samples that GEDI produces? Do we believe 300 samples is sufficient in a 200 km grid cell and why? What if all those samples are from 1 single track going through the area? Should probably also show a map of the number tracks used in each 200 km cell.

Line 132: Do you mean Americas (vs Amazonia) here?

Line 157: It’s difficult to have the single assertion in the sentence requiring the reader to go to three different places to understand it. You need to rewrite that sentence and then have clauses that separate out Fig 2b, from EDT 1 and EDF 6 instead of lumping them all together.

Line 162: Are the rates per year?

Line 169: This is where the lack of good ecological definition of RH50 creates issues. RH50 does not “recover” – it is a radiative transfer metric (50% of the cumulative energy is below that height relative to the ground). So, explain in clear terms what you mean when you say that. The intact RH50 may be dominated by the tallest trees, whereas when the forest recovers its dominated by young trees growing? What are you asserting here ecologically?

Line 171: Most ecosystem models show that height saturates much more quickly than biomass and should probably cite some of those. Your results are consistent with those previous modeling studies.

Line 176-179: This sentence is difficult to understand, would you please rewrite to clarify, especially the first part of it. But the whole paragraph is missing that conceptual model mentioned earlier.

Line 180: Fig2c. Can you statistically assert which time since regrowth periods are statistically different (i.e., what segments of the slope of those lines are different from 0)? For example, in Asia is any of the change significant after 10 years?

Line 196: Do you mean 120 m here?

Line 210-212: I cannot follow the logic of this statement.

Line 227&following: Where is the map of IFL given. You say you have an “approach to mapping IFL” but I could not find the result of that map. Since you are also comparing to Potapov, why would you not include a comparison of maps in the Supplement? That would be exceptionally interesting if you are asserting that GEDI data provide a very different view of what it means to be “intact”.

Line 243: Fig 3 (Africa). Something doesn’t seem right here. How can the recovery of “All Forests” (undisturbed and disturbed) exceed undisturbed after 1000 m?. Can you explain? Also, there is this 210 m reference again. Where is that coming from? Do you mean 120 m? How did you pick, 1000 , 1500 and 2000 m? What quantitative rules got you to those numbers?

Line 283: Same question with statistical test for non-random data, especially if for some areas most of the GEDI footprints come from a single line.

Line 315: provide a reference for GEDI since one did not appear earlier.

Line 447: GEDI L2A and L2B have citations on the DAAC so you do not need to have web reference (i.e. they have a DOI as with any other publication).

Line 456: Define “HOME”

Line 464: You should reference the paper some of the co-authors just released on European protected areas as well.

Line 465&: It's not clear from your writing if you processed waveforms, simply used metrics from GEE, or just used filtering criterion on L2A and L2B data. Please be more precise here.

Line 476: When you filtered at 10 deg you did two things that you do not address. (1) you biased your sample against forests on slopes (which may have different degradation regimes) and (2) you added noise because GEDI geolocation is only 10 m at one standard deviation. This means that many shots have geolocation error more than 20 m. Slopes change very rapidly in space and since the shots cannot be precisely matched to SRTM you included some shots that should have been filtered and you removed some shots that should have remained. This is not acknowledged nor discussed.

Line 471: Topography only results in an overestimation of height as a bias if its unvegetated. On vegetated surfaces it can make the heights taller or smaller.

Line 483-485: This is not a true statement as given. I think you could probably just delete that first sentence and add a qualifier to the next one.

Line 502: 300 seems quite small for such a large area. Again, how many tracks are used? This also shows why using multiple years of data, properly filtered would perhaps help.

Line 505: That seems like very, very few shots, especially if they all came from one line. Same thing with the 13 minimum in next sentence. Again, in this paragraph you clump together too many figures and data sets. Guide the reader here.

Line 512: Why do you assert the intragrid variability is not large? It looks like its 10 to 15 m from the bootstrap. What were the SDs of the bootstrap? Also, it's unclear what you are doing. Suppose a grid cell has 1000 shots you are still doing a 300 bootstrap? What if it has 300 shots? The first sentence does make much sense to me. Many factors will make canopy heights variable. If you have non-random sample or too few samples, you may either get variability that is either too big (not enough samples) or too small (non-random sampling). So, what exactly are you doing here and why? Why is high variability if its real a bad thing?

Line 520: Once again, can you do a t-test with non-random data?

Line 857: Those figures are highly ineffective. Why are they needed?

Line 863: EDT 1. Is there an equivalent table for RH100? What does the shading mean?

Referee #3 (Remarks to the Author):

This is a valuable paper that characterises the spatial variation in height of tropical moist forest (TMF) as a function of distance to disturbance – the key learning being that TMF are shorter much further (up to 2000m) from observable disturbance than previously thought. This result changes our understanding of TMF states and has broad implications.

The authors use new satellite lidar data (GEDI) from a single year alongside a three decade map of forest disturbance histories to build a space-for-time analysis of height versus time since disturbance. The analysis is used to estimate forest height regrowth post-disturbance.

Some of the results found are to be expected – forests that have been disturbed are found to be shorter than undisturbed forests in the same region. The study here confirms that selective logging has less impact on height than does burning.

Some other results are more controversial and unexpected. The authors state that “In the 30 years

of the analysis, the complete recovery of the canopy height cannot be achieved for most types of forest disturbances and regions, documenting indisputably the persistent marks of cumulative impacts from logging, fires and edge effects on the structure of global TMFs". This global conclusion seems to be counter to other records generated through much higher resolution airborne lidar at specific TMF locations. It is true that forests in Borneo show slow canopy recovery (Pfeifer et al. 2015 Riutta et al., 2018). But these forests have very tall canopies reaching up to 70 m, which are not captured in this analysis by GEDI. Logging disturbance in Borneo reduces forest height by 30-40 m, so with typical height regrowth of ~0.8 m per annum, multi-decadal recovery in height is expected. But there is rapid recovery in vertical forest structure in La Selva, Costa Rica (Tang et al., 2012). Here height profiles twenty years after clear-felling are similar to those found in undisturbed forests, which are shorter stature than those in Borneo. Chave et al (2020) report lower regrowth rates in French Guyana than Costa Rica and link this to poor soils on the Guiana shield. What we conclude from these studies is that regrowth rate varies significantly across TMF.

The authors raise some interesting theories to explain their observation of slow recovery across all TMF – suggesting that logged and burned forests are highly vulnerable to further degradation and deforestation. This may be true, but the global consistency is unexpected. Land use, land use change, fire resilience, climate, biodiversity and soils all vary spatially at sub-continental scale. The lack of nuance in the results here raises concerns that there may be intrinsic biases in the data and the analysis. These should be resolved before proceeding.

A challenge for the rigour of the results may be linked to the difficult concept of 'canopy height', a scale dependent property which will vary according to the resolution of the laser. GEDI has a footprint of 25 m, whereas airborne lidar resolve down to 0.1 m, up to two orders of magnitude difference. GEDI samples part of the landscape, not its entirety, whereas airborne lidar can give complete returns albeit for a smaller domain. Comparison of GEDI with airborne lidar in Amazonia reveals an RMSE of 4.4 m (Milenković et al, 2022). In this published study the airborne data were used to correct inherent bias in GEDI, and then to estimate forest regrowth rates. Milenković et al note that without airborne lidar for calibration, GEDI estimates of regrowth rate will be under-estimates.

We could hypothesise that biases arise in remote measurement of height in undisturbed forests, due to rougher canopy surfaces linked to presence of canopy emergent trees and gap dynamics. This heterogeneity might have length scales <25m resolution of GEDI. It would be helpful to reflect on what a GEDI measure of height represents and how can it be related to ecological information. Does GEDI have different biases depending on forest structure, including the size, number and arrangement of stems and canopies? A detailed comparison of the GEDI approach used here against the same analysis using airborne lidar and in situ estimates of disturbance history would provide enhanced credibility for global GEDI outputs.

Another challenge for the study lies with potential errors in the mapping of forest disturbance. It is not clear if the degradation mapping has been validated independently, and therefore what its potential biases are. I found extended data figure 1 confusing – the colours used are difficult to match between legend and map, and the shading is difficult to interpret.

Most of the figures present height changes as a percentage relative to an intact forest baseline – this allows comparison, but limits sense checking against direct field data of e.g. forest height growth rates.

In conclusion, the most important conclusion of this paper is that degradation extends far from observed disturbance in TMF, up to 2 km. The challenge for the authors is that this conclusion relies

on a space for time substitution approach that generates a global consistent result at odds with the nuanced learnings from airborne lidar in regional studies. To have confidence in the GEDI analyses it is important to link these to airborne lidar and confirm results are consistent.

P 329 – what is 2 σ ?

P 475 – is the 10 σ slope threshold robust and valid?

Ext Data fig 7 – is there any ground check on these rates of canopy height during regrowth?

References:

- Chave, J., C. Piponiot, I. Maréchaux, H. De Foresta, D. Larpin, F. J. Fischer, G. Derroire, G. Vincent, and B. Hérault. 2020. Slow rate of secondary forest carbon accumulation in the Guianas compared with the rest of the Neotropics. *Ecological Applications* 30:e02004.
- Milenković, M., J. Reiche, J. Armston, A. Neuenschwander, W. De Keersmaecker, M. Herold, and J. Verbesselt. 2022. Assessing Amazon rainforest regrowth with GEDI and ICESat-2 data. *Science of Remote Sensing* 5:100051.
- Pfeifer, M., Kor, L., Nilus, R., Turner, E., Cusack, J., Lysenko, I., Khoo, M., Chey, V. K., Chung, A. C., & Ewers, R. M. (2016). Mapping the structure of Borneo's tropical forests across a degradation gradient. *Remote Sensing of Environment*, 176, 84– 97. <https://doi.org/10.1016/j.rse.2016.01.014>
- Riutta, T., Malhi, Y., Kho, L. K., Marthews, T. R., Huaraca Huasco, W., Khoo, M. S., Tan, S., Turner, E., Reynolds, G., Both, S., Burslem, D. F. R. P., Teh, Y. A., Vairappan, C. S., Majalap, N., & Ewers, R. M. (2018). Logging disturbance shifts net primary productivity and its allocation in Bornean tropical forests. *Global Change Biology*, 24(7), 2913– 2928. <https://doi.org/10.1111/gcb.14068>
- Tang, H., Dubayah, R., Swatantran, A., Hofton, M., Sheldon, S., Clark, D. B., & Blair, B. (2012). Retrieval of vertical LAI profiles over tropical rain forests using waveform lidar at La Selva, Costa Rica. *Remote Sensing of Environment*, 124, 242– 250. <https://doi.org/10.1016/j.rse.2012.05.005>

Author Rebuttals to Initial Comments:

Action taken:

-> Following the recommendations, in the revised manuscript we have provided additional observational evidence and analysis to reinforce our findings, corroborate the statistics of canopy heights, and deepen the understanding of the factors affecting canopy height dynamics. We provide below a detailed answer to each comment.

Referee #1

This paper presents an analysis of the impact of human disturbance (fire, logging, edge effects) on tropical moist forests, on structure and resilience. The paper combines GEDI (for

the structural element, particularly the height metrics RH50 and RH98, along with estimates of disturbance/degradation state from Landsat time series.

There is some very interesting work here – in particular the assessment of the distance from edges that structural change can persist & also the potential impact of structural change on future recovery. These are important topics in our understanding of forest resilience, C stocks and also how we go about managing and mitigating change in TMFs. There is new insight here, which is potentially really important in changing how we think about forest disturbance more widely in the tropics.

The approach is broadly sound in general, but there are a lot of unanswered questions, rather vague analysis and assertions, and some important areas of uncertainty about how the data are used throughout. This means that in many places, there are rather vague and discursive assertions made, efforts to explain cause and effect, and generalities that are not really supported by the analysis. There is also a lack of clarity in what we are meant to be looking at i.e. a failure to guide the reader to the most important aspects. I have provided specific comments on the annotated manuscript, but I note some more general issues here. I realise that Nature discourages the use of marked-up manuscripts, but this is by far the best way to provide specific feedback. I provide this file to be used as is seen fit.

We thank the reviewer for the appreciation of our work and for stressing the relevance of the topic in the current debate about forest disturbances in the tropics. Following the reviewer's recommendation, in the updated version of the manuscript, we have provided additional observational evidence that reinforces our findings and re-arranged the storyline. Note that we have also integrated GEDI measurements from 2019-2022 (instead of 2019-2020) and the JRC-TMF dataset up to the year 2022. We provide below a detailed answer to each of the reviewer's comments. We have also improved how we report our results, emphasizing the key messages and being transparent on the underlying variability behind the data supported by the most recent and relevant references.

1) There is no way to validate the assessment of disturbance – unless there were an independent dataset eg from locally-assessed v specific impacts. So the term 'accuracy' is used in many places, but there is no reference measurement against which to assess this. Eg in the Conclusions the authors say "... provides an incontrovertible assessment of forest structural attributes and enables the large-scale monitoring of forest degradation and recovery.

This type of accurate and transparent assessment ...” . But this study is neither incontrovertible, nor accurate in a specific, quantifiable way. The best that can be said is ‘precise’ and then to compare with other studies.

We thank the reviewer for pointing out the issue of the use of the terms ‘accurate’, ‘indisputably’, or ‘incontrovertible’ when presenting our estimates. Even though we have acknowledged in the method section that both JRC-TMF and GEDI datasets have already been validated separately (ref Vancutsem, Duncanson etc), we agree with the reviewer that the validation of the combination of these products requires an independent dataset that goes beyond the scope of this paper.

Consequently, we have now rearranged the text to tone down the use of phrases describing what has not been directly assessed. In the Conclusion section, we deleted the term ‘indisputably’, and we modified the sentence on the incontrovertible assessment of forest structural attributes. Now the sentence reads (Lines 334 -337 of the revised manuscript):

“Our study demonstrates that the integration of recent and spatially sparse observations from spaceborne LiDAR (this corresponds to over 39 million individual sample plots) with long-term and spatially continuous spaceborne optical datasets (i.e. JRC-TMF) provides a novel approach to assess the forest structural attributes and enables the large-scale monitoring of forest degradation and recovery.”

We replaced “accurate” with “precise” in this sentence of the conclusion (Line 351 of the revised manuscript):

” This type of precise, spatially detailed, recent and transparent information on tropical forest degradation ...”

We also deleted the term ‘high-accuracy’ in the method section (Lines 358-359 in the revised manuscript):

“The spaceborne Global Ecosystem Dynamics Investigation (GEDI10) from NASA revolutionises this field by consistently acquiring forest canopy heights and structure measurements at scales that have been unattainable until now across these vast areas”.

Finally, we added these sentences on the validation of the JRC-TMF dataset knowing that more details can be found in the respective papers (lines 401-406 of the revised manuscript):

” The accuracy of the disturbance mapping of every single Landsat image is 91.4%, with omission and commission errors for non-forest cover detection of 9.4% and 7.9%, respectively. The accuracy matrix for the transition map shows an overall accuracy of 92.8% for the classes of the moist forest domain. The omission and commission errors for the forest change areas are 19% and 8.4%, respectively. Details on the accuracy assessment at the continental level are presented in the supplementary material of Vancutsem et al. 2021”.

2) Similarly in the same passage the authors say “... previously undervalued and considered de facto of ~ 100 m - demonstrates that the overall impact of fragmentation across the pantropical belt is severely underestimated ...” But underestimated compared to what? What they mean is that their estimate is different (larger) from previous work. Which is potentially important of course, but without a direct causal explanation and/or independent validation, there is no ‘underestimation’ per se.

We thank the reviewer for this comment and we completely agree. We changed the word “underestimated” to “overlooked” and we calculated the edge area with a distance of 120m compared with an edge with a distance varying from 350, 600 to 1700m (for Americas, Africa, and Asia, respectively, see new figure 2 for more details). We showed now in the text that there are large differences and that the edge effects were previously overlooked by a factor of 221%. This sentence in the Conclusion section now reads as such (lines 340-343 of the revised manuscript):

“The forest edge effect was previously considered de facto of ~ 100 m²⁴. Our results show a far-reaching effect of forest edge implying that the overall impact of fragmentation across the pantropical belt is severely overlooked by a factor of over 200%, with relevant potential consequences on the forest carbon dynamics and other ecosystem services”.

3) Different height metrics are used in different ways in different places, without any clear reasoning as to why RH50 or RH98 or both. This is v important as these are the key structural indicators. So unless there is a clear reason why each is used and where, this is somewhat hard to follow and confusing.

Following the recommendations, we only present the main results using the RH50 metric in the updated version of the manuscript, keeping RH98 and AGB figures in the supplementary materials. We also provide a more explicit and ecologically grounded definition of relative height metrics in the introduction and in the method section and demonstrate why RH50 is the most relevant metric to quantify forest structure changes following degradation and recovery. We finally added an extended data figure (Extended Data Fig. 1) on the cumulative return energy of GEDI relative height metrics for the main classes of interest in this study i.e. intact, degraded, edge, and regrowth showing the sensitivity of RH50 to capture vertical variations in canopy structure compared to RH98.

The end of the introduction reads as such:

“ We selected two structural metrics (Relative Height RH98 and RH50) that capture various aspects of forest structure, encompassing canopy height (RH98, hereafter referred to as top canopy height) and the distribution of canopy elements, as well as canopy cover and the fractions of gaps within the canopies (RH50, hereafter referred as relative height). We used RH50, i.e., the average height of median energy⁵⁰, as the main indicator in our study as it is strongly correlated to key structural variables including AGB, stem diameter, and basal area^{51,52}, and exhibits a high sensitivity to the variation in forest structure along degradation gradients compared to RH98^{53,54} (see also Extended Data Fig 1)“.

4) On a related note, FHD (the foliage height diversity) is mentioned early on, but then never used except in the Extended Data – it is not presented in the main results at all. So why is it even mentioned?

We agree with the reviewer that FHD is not discussed enough in the manuscript. In our initial draft, we often presented results in terms of FHD. Still, in the end, we decided to move them

to the Supplementary given their similarity with RH98 and RH50 (see scatterplots in the SI, Fig. S1 in the original version).

Action taken:

-> We removed FHD and analysed instead Above Ground Biomass (AGB) as requested by reviewer #2. For further details see Point#6 Reviewer#2.

5) A general question arises in terms of the use of Landsat for one aspect of the analysis (defining the degradation status and change), and GEDI for the other – the analysis of the impact on canopy height. As the authors acknowledge in a couple of places, passive optical data like Landsat are relatively insensitive to structural variation that isn't high intensity deforestation – which is why GEDI is so useful. BUT you then have a data source (Landsat) that is insensitive to lower levels of disturbance being used to define the regions, where gedi is then being used to assess canopy height. This seems somewhat of a mismatch in that we know Landsat will underestimate lower levels of degradation, but those data are then used to assess the spatial impact (of disturbance) on height.

Landsat is sensitive to high-intensity deforestation but also to short-term disturbances due to selective logging and fires. A long list of recognized studies has used Landsat to map forest degradation across the tropics¹⁻⁴. However, we agree with the reviewer that small-scale (i.e. less than 0.09 hectares) and very short disturbances will not be detectable by Landsat. Despite its partial shortcomings in detecting structural variation, Landsat's strength lies in its spatial coverage (that is complete, unlike GEDI) and its temporal coverage (its archive is unprecedented). These assets give it the proper credentials to make the first guess of where to sample GEDI footprints to analyze structural variation.

We know that Landsat will underestimate lower-intensity degradation, particularly when canopy disturbances fall below its spatial resolution (0.09ha). Other types of degradation (with stronger severity) may also be missed if visible only during a short time due to the temporal resolution of satellite coverage (data gaps in time series + high cloud coverage). To tackle this, we include buffer areas around pixels that are depicted as degradation by the TMF approach (using a buffer distance of 120m from logging and fire). Including such buffered areas decreases the omission of low-intensity degradation⁵. Regarding the edge effect that can be also considered as low-intensity degradation (~20% decrease in CH), we also used a buffer around the undisturbed forest with tests of various distances to the edge and disturbance to retrieve information on impacts due to this type of low-intensity degradation.

Action taken:

-> We have modified the introduction and presented the limitations of Landsat in mapping different types and intensities of forest degradation. How we now introduce the use of GEDI data suggests that the combination of Landsat-based maps of forest cover change and GEDI would allow to: 1) quantify forest structure changes within mapped areas of forest disturbances and 2) target the sampling of GEDI footprint at proximity to main forest cover changes.

-> We have modified the combination of datasets paragraph of the Method section (lines 572-575) to be more explicit about how we sampled GEDI footprints according to the analysis of each degradation type.

“ The extent of mapped forest change areas (short and long-term disturbances) in the JRC-TMF dataset was used to target the sampling of GEDI footprints and quantify forest edge effects or canopy disturbance contagiousness between degraded and undisturbed forests on forest structure still classified as ‘undisturbed forest’”.

6) There is also a lack of mention of Radar here – there is quite a lot of work done to assess structural variation in tropical forests from radar, but that is not mentioned at all which seems like an oversight (particularly given the v long list of references).

Thank you for this comment. We have accordingly restructured the introduction in order to show that Radar, along with airborne LiDAR data, has already been widely used in quantifying forest degradation impacts at a local scale. We have inserted the most recent and relevant references to support this.

7) Overall, I would emphasise that there is definitely some very interesting work here, particularly on the distance and temporal persistence of disturbance. I think with a bit of work this could be a really interesting paper of wide interest to Nature readership. But I do think it needs some work – streamlining, and a much clearer focus on the key message(s). In particular (see my comment on lin 264) the observation that “No statistical differences in CH were found for undisturbed edges in the three continents, confirming the role of unsustainable

logging and fire in enhancing the risk of deforestation.” – I think this is really important, but it is actually more understated than some of the other less convincing results.

Thank you for your encouraging comment. We have accordingly restructured the results and improved the overall narrative around the quantification of forest degradation impacts at both spatial (result number 1 and 2) and temporal scales (result number 3) which then feed into the last part on the fate of degraded forests (result number 4). We consequently modified the abstract, introduction and conclusion.

Specific comments (annotated manuscript):

8) Title: Unveiling [MOU1] the magnitude and persistence of human-driven degradation in Tropical moist forests

[MOU1]This word doesn't really add anything here. Quantifying? Assessing?

Thank you, we used the word “Quantifying” instead to refer to a broader and more complete evaluation of the degradation.

9) Line 19: Anthropogenic and natural [MOU1] disturbances leading to the degradation of tropical moist forests are increasing in intensity and

[MOU1]You only mention human-driven in the title

Thank you for this comment, we have rephrased it accordingly: **“Direct and indirect anthropogenic and natural disturbances such as land use and climate change are widely considered the main drivers of degradation7”**.

10) Line 22: They may eventually exceed forests' capacity to withstand or recover [MOU1] from these impacts

[MOU1]Mean v similar things here?

We agree with the reviewer and we removed "withstand".

11) Lines 23-24: Here, we combine forest cover change maps [MOU1] with 20 million canopy height measurements[MOU2] from spaceborne Light Detection And Ranging (LiDAR) to

[MOU1]No comment on how these are derived?

The reviewer's comment is well-taken. In fact, initially, we cited two papers, but in a second moment, given the word limitations, we removed them.

We changed the sentence as such (Lines 23-25):

" Here, we combine satellite remote sensing data on forest cover change^{1,9} with canopy height and biomass measurements from a spaceborne¹⁰ Light Detection And Ranging (LiDAR)..."

and added the references to the two main data sources.

[MOU2]I never like this idea of using 'big' numbers to imply strength of evidence - it's reductive and doesn't really add anything. I could claim there are 50 million measurements of reflectance in a single pic from my cell phone :-)

We agree with the reviewer's comment about avoiding the misuse of big numbers. These numbers allow us to reach "strength of evidence" quoting the reviewer. The numbers have been removed from the abstract. However, we chose to keep them in the introductory section:

" The analysis of more than 23 million sample footprints of GEDI data (Americas: 16,092,827; Africa: 5,882,035; Asia: 1,523,891) shows that intact forests"

and in the Conclusion (lines 334 - 337 of the revised manuscript):

"Our study demonstrates that the integration of recent and spatially sparse observations from spaceborne LiDAR (this corresponds to over 39 million individual

sample plots) with long-term and spatially continuous spaceborne optical datasets (i.e. JRC-TMF) provides a novel approach to assess the forest structural attributes and enables the large-scale monitoring of forest degradation and recovery.”

12) Line 33: the necessity to protect the recovering forests to meet the pledges on forest enhancement and conservation made at the recent United Nations Climate Change and Biodiversity Conferences[MOU1] .

[MOU1]This latter part is a bit vague - not saying it's not important, but can you say more concisely what the new findings imply re doing something about this? Do we need to change the way we assess & manage forest disturbance impacts? What are the likely impacts on how we deal with this now?

Thank you for this comment, we changed this sentence accordingly into (lines 31-33 of the revised manuscript):

“Our findings on the cumulative impacts of degradation and fragmentation together representing 18% of the remaining forest area demonstrate the increasing vulnerability of tropical moist forests to deforestation and call for their protection.”

13) Line 40 : In 2020, 54% of the TMFs area (~593 million ha) was impacted by anthropogenic[MOU1] disturbances,

[MOU1]There's a conflation here of loss with disturbance. Clearly loss encompasses disturbance, but not the other way around - needs to stick with disturbance

Thank you for this comment, we changed this sentence accordingly into (lines 59-61 of the revised manuscript):

“ While only 46% of the TMFs (~505 Mha) can be considered intact forest landscapes³⁷, free from human influence and disturbances, little is known about the state and extent of forest degradation across the pantropical belt³⁸”.

14) Line 49: If not further disturbed, the recovery of degraded forests, together with the regrowth of secondary forests on abandoned agricultural land, may bring a relevant contribution to addressing the climate and ecological[MOU1] emergencies

[MOU1]This is again too vague

We agree with the reviewer. We improved this sentence by quoting two very recent studies that quantify the contribution of recovering forests in carbon absorption, counterbalancing part of carbon emissions due to deforestation and degradation (lines 53-58):

“ Presently the recovery of degraded forests and the regrowth of secondary forests on abandoned agricultural land are offsetting ~ ¼ of deforestation and degradation emissions¹¹. However, even if all disturbances ceased now, current forests only have a limited additional carbon storage potential³⁵. Furthermore, forest or ecosystem restoration is included in results-based payment frameworks such as Reducing Emissions from Deforestation and Degradation (REDD+) activities under the United Nations Framework Convention on Climate Change (UNFCCC³⁶) ”.

15) Lines 57-58 Improving the understanding and knowledge [MOU1] of the impact of disturbances on the structure of TMFs, along with the recovery speed and rate [MOU2] of disturbed forests, is thus of paramount importance to inform forest-based mitigation strategies and policies

[MOU1]?

[MOU2]This whole sentence doesn't really make sense

Thank you for this comment. We rephrased the sentence and made a more explicit connection between the quantification of degradation, the ecological importance of recovering forests and policies as such (lines 61-63 of the revised manuscript):

“ Disentangling the effects of the multiple drivers of degradation and quantifying their impacts on forest structure over a long time period is crucial for understanding forest recovery dynamics and defining appropriate forest-based mitigation strategies”.

16) Lines 59-60: Forest recovery pathways are linked to the nature of the disturbance, potentially making them quite different in extent and speed[MOU1] .

[MOU1]Again, this isn't really clear

We agree with the reviewer that this sentence was bringing too much confusion. We deleted this sentence and the message that forest recovery depends on forest degradation intensity which greatly varies depending on the type and co-occurrence of logging, fire, and edge effects and is now presented in two separate sentences in the introduction (lines 45-49):

” Forest degradation is mainly caused by selective logging, fires and edge effects⁷, i.e. increased mortality of trees at forest edges²⁴, and it is often exacerbated by a combination of these disturbances occurring together, repeated over time and driven by continued deforestation, habitat fragmentation and human-induced climate change^{6,25,26}”

and (lines 61-63) :

“ Disentangling the effects of the multiple drivers of degradation and quantifying their impacts on forest structure over a long time period is crucial for understanding forest recovery dynamics and defining appropriate forest-based mitigation strategies”.

17) Line 63: However, documentation of the extent and types of forest degradation mostly covers low-intensity [MOU1] disturbances, mainly focusing on timber harvesting in logging concessions, whereas information on degradation from high-intensity logging remains scarce

[MOU1]What do low and high mean in this context. In themselves they don't seem to mean anything, so this needs to be defined really eg C loss / ha.

Action taken:

We agree on the need to evaluate the degradation intensity. For this reason, in the revised version of the manuscript, we modified this part of the introduction, emphasizing the overall need to assess the extent of forest degradation and, more specifically, to characterize the effects of the multiple drivers of degradation and quantify their impacts on forest structure. As the paper is not directly aimed at assessing carbon loss after degradation, we decided to only provide the range of quantified carbon loss due to each main driver of degradation (i.e. selective logging, fire and edge effect), which implies that great variability exists within each type of disturbance. See the following point to see how the sentence was rephrased.

18) Lines 65-67: Airborne Light Detection and Ranging (LiDAR) has been used to study higher-intensity degradation, such as understory fires and edge effects, but only for single-cause forest disturbances and at regional scales[MOU1] .

[MOU1]I don't think this is true, and also this seems out of place here. Why mention airborne lidar but not eg radar & other types of observation. See for eg

Joshi, N., Mitchard, E.T., Woo, N., Torres, J., Moll-Rocek, J., Ehammer, A., Collins, M., Jepsen, M.R. and Fensholt, R., 2015. Mapping dynamics of deforestation and forest degradation in tropical forests using radar satellite data. *Environmental Research Letters*, 10(3), p.034014.

Milodowski ref 60 here.

Carstairs, H., Mitchard, E.T., McNicol, I., Aquino, C., Burt, A., Ebanega, M.O., Dikongo, A.M., Bueso-Bello, J.L. and Disney, M., 2022. An effective method for InSAR mapping of tropical forest degradation in hilly areas. *Remote Sensing*, 14(3), p.452.

Thanks for this comment. Following the suggestion of the reviewer, we evaluated the literature carefully and we modified the text by adding these references and by presenting the added value of Radar together with ALS on the quantification of degradation impacts on the vertical forest structure. The corrected sentence reads as such (lines 66-72):

“ Despite the novelty of these products in mapping disturbances beyond deforestation, small-scale disturbances or edge effects affecting forest structure remain undetected, leading to an underestimation of degradation⁴¹. Active remote sensing using Synthetic Aperture Radar (SAR) or Airborne LiDAR Scanning (ALS) is a promising source for detecting subtle disturbances^{42–46} and assessing biomass loss associated with edge effects (20-30% loss within the first 120m of forest edge^{20,47}), selective logging (10-50% loss^{21,48,49}), and fire (40-90% loss^{21,49}) at local scale.”

19) Line 69: Altogether, potential feedbacks and synergies between different disturbance types, intensities, and frequencies and their long-term impacts on forest structure remain largely unknown and most likely underestimated[MOU1] .

[MOU1]This to me is the crux of it - this is not well-established here, but really ought to be pretty much the first statement of the paper and then real well-justified. Ref 5 is projections, and 47 is from 8 years ago. This doesn't provide the justification for the paper that is needed.

Thank you for pointing this out. We agree and have moved this sentence further up in the introduction and reads now as the main need for the study. It is then justified by a literature review on the main advancements in remote sensing in reducing uncertainties related to quantifying impacts in forest structure and biomass losses attributable to degradation processes.

20) Line 70: The 2018 deployment of the Global Ecosystem Dynamics Investigation (GEDI) LiDAR

change recent into 2018

The reviewer is right. The text has been updated accordingly.

21) Line 72: which specifically targets forest structure, is now opening new opportunities to address these questions [MOU1] with proper quantitative tools [MOU2] at the scale

[MOU1]Which questions? Do you mean issues above?

[MOU2]? This is a bit meaningless

Following the recommendations, in the updated version of the manuscript we modified this sentence as such (lines 72-75):

“Yet another opportunity arises at near-global scale from the deployment of the Global Ecosystem Dynamics Investigation (GEDI10) LiDAR instrument on the International Space Station (ISS) in late 2018. GEDI specifically targets forest structure and is opening new opportunities to quantify the long-term impacts of forest degradation over the tropical belt”.

22) Lines 74-77: this study has the main objectives to ~~shed new light on the impact of human pressures on global TMFs~~ by 1) quantify the impact of different types of disturbances on forest structural characteristics across the entire tropical moist domain; 2) assess the capacity of remaining forests to buffer future disturbances, and 3) quantify the recovery rates of TMFs after various type of anthropogenic disturbances.

Thank you for this suggestion, it now read as such (lines 76-79): **“ Using GEDI, we aim to assess across the entire tropical moist domain: 1) the impact of different types of disturbances on forest structural characteristics; 2) the real extent of forest degradation when considering edge effects; 3) the forest recovery rates after various types of anthropogenic disturbances; and 4) the vulnerability of degraded forests to subsequent deforestation.”**

23) Line 80: We evaluate the range of degradation severity by analysing the effects of single disturbance processes (e.g., [MOU1] selective logging, fire, edge effect)

[MOU1]Eg or ie? Need to be clear

Thanks, it is i.e. The text has been changed accordingly:“(i.e., selective logging, fire, edge effect)”

24) Line 83: We thereby provide an overall assessment of the recent [MOU1] impacts of anthropogenic disturbances

[MOU1]Meaning?

Following the recommendations, in the updated version of the manuscript, we modified this sentence as such (lines 88-89):

“ We thereby provide an overall assessment of the impacts of anthropogenic disturbances on tropical moist forests and an assessment of the forest’s ability to recover to its pre-disturbance state”.

25) Line 85; To[MOU1] this scope, we document and quantify 30 years of pantropical forest degradation and forest regrowth by

[MOU1]Why is ref 48 relevant here?

The reviewer is right. Ref 48 is the Lang et al. paper, *de facto* the paper on RH98 extrapolation with Sentinel 1 and 2 at global scale. However, this reference is off-topic here and has been removed.

26) Line 87: from a wall-to-wall map of tropical moist forest cover changes at fine spatial resolution and accurate [MOU1] samples from GEDI

[MOU1]?

-> In the updated version of the manuscript we removed the word “accurate”. Also, the term wall-to-wall was removed because it is not necessary and not needed for a general reader. The modified sentence reads as such (lines 90-92):

“ Within this scope, we combine a wall-to-wall dataset on tropical moist forest cover changes derived from Landsat imagery at 30 m spatial resolution^{1,9} with a sample dataset of estimates on forest structure and Above Ground Biomass (AGB) derived from the GEDI LiDAR sensor. ”.

27) Line 89: heterogeneity of the foliage profile (Foliage Height Diversity - FHD[MOU1]) (details in Methods, Extended Data Fig.1).

[MOU1]You mention this here and then in the Extended Data, but you do no analysis of it at all in the main part of the paper. So why is it here? It feels like it's been left in, but only in a very half-hearted way and doesn't feed into any of the main conclusions at all.

Thanks for this comment. We agree with the reviewer and have removed the FHD from the manuscript as it showed redundant trends compared to the two RH metrics. Instead, we added the Above Ground Biomass (AGB) from GEDI Level 4A as suggested by reviewer#2. The text has been updated accordingly (main and methods) and new figures can be found in the supplementary.

28) Line 94: Intact forests are [MOU1] areas with no signs of human activity detected

[MOU1]Finally we get a definition

As suggested, we have moved the definition of intact forest to the introduction as it is the baseline of our methodological approach. It now reads as such (lines 84-88):

” We also assess the speed of forest regrowth after full canopy cover removal using forest structure metrics to gauge the evolution over time of the level of recovery compared to the reference structure of intact forests, i.e., areas with no signs of human activity detected during the last three decades and located at least 3 km from other land covers (see Preparation of input datasets section in Methods)”.

29) Line 96: The spatial patterns of RH98[MOU1] , RH50 and FHD of intact forests

[MOU1]I know this is discussed in the methods, but I think there needs to be some clarity here on what RH means and the implication of using this metric and the difference of RH98 and RH50. These are likely to be somewhat opaque to a more general reader

Thank you for this comment that was also raised by the other reviewers. We have now a dedicated paragraph at the end of the introduction presenting the ecological meaning of RH98 and RH50 and its sensitivity to forest structure changes along degradation gradients. It now reads as such (lines 92-100):

” We selected two structural metrics (Relative Height RH98 and RH50) that capture various aspects of forest structure, encompassing canopy height (RH98, hereafter referred to as top canopy height) and the distribution of canopy elements, as well as canopy cover and the fractions of gaps within the canopies (RH50, hereafter referred as relative height). We used RH50, i.e., the average height of median energy50, as the main indicator in our study as it is strongly correlated to key structural variables including AGB, stem diameter, and basal area^{51,52}, and exhibits a high sensitivity to the variation in forest structure along degradation gradients compared to RH98^{53,54} (see also Extended Data Fig 1). Results based on the RH98 and AGB measurements are presented in the supplementary materials (details in Methods).

More details can also be found in the Method section.

We have also computed the cumulative energy return (new extended data fig 1.) using all relative height metrics over the main forest types of interest (e.g. intact, degraded, edge forest and regrowth) and showed how RH50 and RH98 metrics change along degradation and recovery gradients.

Finally, we have extracted all relative heights values along a transect of deforested land - edge forest - intact forest (new extended data fig 1) to illustrate the edge effect and how the main canopy height metrics (RH50, RH98) are changing:

Extended Data Fig.1 (a) Cumulative Return Energy from GEDI L2A for different forest types (30 GEDI footprints for each forest type were sampled in a grid cell located in the northeast of the Brazilian Amazon). The vertical black lines refer to RH98 and RH50. The horizontal grey

lines refer to the height of RH98 and RH50 for Intact Forest. (b) Transect (-47.76, -3.68) of GEDI relative heights across deforested land, edge forest (width of 350m) and intact forest collected in the Brazilian Amazon (left panel). The vertical lines represent the GEDI Relative Heights (RH), which give the height at which a certain quantile returned energy is reached relative to the ground. The three black lines represent the height of RH98, RH50 and RH0. Zoom (right panel) on the vertical line, representing the quantile levels of the GEDI Relative Height (RH). The grey curves represent its density distribution, while the dots are the individual RH. RH98, RH50 and RH0 are shown in green.

These two new figures have been requested by reviewer#2, see point#8.

30) Line 98: Top canopy heights (RH98) were higher in Africa ($29.4 \text{ m} \pm 2.94 \text{ m}$) and Asia ($32.3 \text{ m} \pm 4[\text{MOU1}] .62 \text{ m}$) than in South America ($28.9 \text{ m} \pm 2.43 \text{ m}$).

[MOU1]These numbers all seem on the low side, even for RH98. A big question is how well the GEDI footprints are distributed relative to density and height. There is a strong impact of cloud cover, so there will be variable sampling across the tropical belt - how does that impact these numbers ie is it more of an impact in SE Asia than S Am for eg?

Thanks for this comment. In this response to the editor and reviewer, we performed a sensitivity analysis to explore the distribution of GEDI shots. Histograms and maps are presented at point#7 reviewer#2.

With regard to the mentioned statistics, the reviewer is right, in the original version of the manuscript is not clear what we computed. As also stressed by reviewer#2 (see point#19), we rearranged the text and presented the stats in the form of tables in Extended Data Tables 1,2 and 3.

In relation to the effects of cloud cover filtering on observation density, we concur with the reviewer that such filtering can introduce bias in the distribution and density of observations within the spatial domain of interest. As LiDAR wavelength can interact with clouds and produce low-quality shots, we followed standard practice and diligently eliminated all potential

cloud contamination by employing various combinations of Quality flags (refer to the 'GEDl dataset' section in Methods). We took great care in mitigating any spatial biases that could impact subsequent calculations. To achieve this, we utilized data spanning a 4-year period, covering the entire duration of the GEDl mission up to the present. Furthermore, to ensure a balanced representation of observations for subsequent calculations (e.g., Fig. 1), we implemented a subsampling procedure across grid cells of approximately 1.5 degrees, covering the study area. This involved selecting a sufficiently large random sample that accurately represents all forest disturbance classes. Grid cells with an insufficient number of observations were excluded from the subsampling procedure, with a minimum threshold of 600 observations set. By applying this threshold, only a negligible fraction of forested areas across the pantropics were excluded (for Intact forest only 46 gridcell - out of a total of 658 - were excluded).

31) Line 98: We found similar [MOU1] differences for RH50 and Foliage Height Diversity (FHD)(Supplementary Fig.1).

[MOU1]What does 'similar' mean though? This is v vague

We have now clarified the original sentence by being more specific and by inserting statistics directly from RH98 and AGB metrics. We invite the reviewer to read this first result section that has been rewritten. Here is below an extract of the revised text (lines 120-124):

“ Our results show that in 2019-2022, the mean height (RH50) of forests that have previously been logged, burned or disturbed due to natural events was at least 7.5 m lower than intact old-growth forests (Fig.1b) with AGB densities of 50-150 Mg/ha, which corresponds to a relative difference of more than 100 Mg/ha compared to intact forests (Extended data Fig 4-6). ”.

32) Line 101: These results are consistent with previous observations, showing that old-growth tropical forests in Asia and Africa, which are dominated by wind-dispersed species, show a

higher frequency of large trees [MOU1] compared to South America. Still, the relative importance of environmental

[MOU1]But this is different to an overall change of canopy height - the two things aren't the same

Thank you, we rephrased this sentence accordingly (lines 114-117):

” These results support previous observations^{55–59}, showing that old-growth tropical forests in Asia and Africa, which are typically dominated by wind-dispersed species, show a higher frequency of large trees compared to South America.”.

33) Line 114: Forest regrowths have, on average, a 10 m to 15 m lower canopy compared to the intact forest (Fig.1d[MOU1]).

[MOU1]But this depends on the time from disturbance doesn't it? Is there any way to unpick this, as this seems key to understanding the rates of regrowth

The reviewer is right. The regrowth depends on the time from disturbance as shown in Fig. 3. However, for the sake of visualization, we have shown in Fig.1d the average values of all the areas undergoing regrowth. The sentence has been framed as (lines 127-130):

“ Forests showing regrowth (Fig.1d) have, on average, a 10 m to 15 m lower RH50 than intact forests, and an AGB varying within a range of 50-100 Mg/ha. Note that all the regrowth areas - regardless of their age - are shown here (an in-depth analysis by age classes is shown in Fig. 3). ”.

Note that we also added extended data tables 8-11 to report on the various rates of regrowth.

34) Line 115: The other forest structural variables (i.e., RH50 and FHD) show similar [MOU1] spatial patterns

[MOU1]?

See point#31 above.

35) Line 118: Supplementary Fig.5[MOU1]).

[MOU1]'m not convinced Fig 1 is that useful in that the grid cells are v large - why 200 km - this is v coarse? And the color scheme means it is v hard to see the differences, partic where there are so few points eg d in central Africa. Dots are almost impossible to see as so small and also against dark blue - this figure is really hard to interpret.

The reviewer is right. In the new version of the manuscript, we changed the spatial resolution from 2 degrees to 1.5 degrees. In the new version of the study, we used all the GEDI archive (2019 - 2022 instead of 2019 - 2020 as in the original version) - for further details see point#10 reviewer#2. By doubling the GEDI shots, we could change the size of the grid cells, passing from 2 degrees to 1.5 degrees, which corresponds to ~ 167 km at the Equator. Also, we took the liberty to use regular hexagons instead of square grids as in the original version of the manuscript. Regular hexagons are the closest shape to a circle that can be used for the regular tessellation of a plane and the distance between centroids is the same for all neighbours.

In order to support this choice, we performed first a sensitivity analysis of the gridcell size. Specifically, for intact forest and degraded forests, we computed for different grid sizes (i.e. 0.5, 1, 1.5 and 2 degrees):

- 1) the mean and the 10th percentile of the number of GEDI **shots** per cell (Fig. A1 and 2 for intact and degraded forests, respectively),
- 2) the mean and the 10th percentile of the number of GEDI **orbits** per cell (Fig. A3 and 4 for intact and degraded forests, respectively),
- 3) the mean of Koppen Geiger⁶ climate classes per cell (Fig. A5 and 6 for intact and degraded forests, respectively)

Results show that by using a 1.5-degree cell (the dotted vertical line in Fig. A1 to 6), there is a trade-off between the cell size and the number of GEDI shots.

Sensitivity Analysis Intact Forest

Fig. A1 Mean and 10th percentile number of GEDI shots per gridcell for Intact forest

Sensitivity Analysis Degraded Forest

Fig. A2 Mean and 10th percentile number of GEDI shots per gridcell for Degraded forest

Sensitivity Analysis Intact Forest

Fig. A3 Mean number of Koppen Geiger climate classes per gridcell for Intact forest

Sensitivity Analysis Degraded Forest

Fig. A4 Mean number of Koppen Geiger climate classes per gridcell for Degraded forest

Sensitivity Analysis Intact Forest

Fig. A5 Mean and 10th percentile number of GEDI orbits per gridcell for Intact forest

Sensitivity Analysis Degraded Forest

Fig. A6 Mean and 10th percentile number of GEDI orbits per gridcell for Degraded forest

Finally, to demonstrate that grid size is not heavily affecting our results, we also performed an additional sensitivity analysis. Maps of Figure 1 have been computed using a grid size of 1

degree, and results (see Fig. A7 to 12) show that spatial patterns of RH are the same as for a gridcell of 1.5 (new version of the manuscript) and 2 degrees (original version of the manuscript).

Fig. A7 Same as Fig 1 in the original version of the manuscript with a gridcell of 1 degree.

Fig. A8 Same as Fig 1 in the original version with a gridcell of 1 degree.

a

Fig. A9 Same as Fig 1 in the original version with a gridcell of 1 degree.

a

Fig. A10 Same as Fig 1 in the original version with a gridcell of 1 degree.

a

Fig. A11 Same as Fig 1 in the original version with a gridcell of 1 degree.

a

Fig. A12 Same as Fig 1 in the original version with a gridcell of 1 degree.

Regarding the colour scheme and the dots, in the new version of the manuscript, in Fig.1 we show only RH50 height and not the differences between intact forest and the forest classes that are now in the Extended Data section to avoid a colour clash. Furthermore, dots have been removed, we took into consideration only cells where a Welch t-test is passed. An example of the new map of differences is shown in Fig. A13.

a

b

c

d

Fig. 1 showing RH50 instead of RH98

d

Fig A13 Example of map of differences between intact forest and edge for RH50, now in Extended Data Figure section. Note that the dots have been removed.

36) Line 120: Fig.1 Canopy height (CH)[MOU1] of intact forests

[MOU1]This is confusing - is. The CH or RH98? I know there's more detail in the methods but you switch between the 2 here.

The reviewer's comment is well-taken. We have significantly revised the text to add more information on RH98 and RH50. We now only present RH50 in the main results and keep RH98 in the supplementary to avoid confusion.

See point#31 above for more details on this issue.

37) Line 129: Selective logging triggers an immediate decrease of 40-70% in RH50 [MOU1] compared to a 70-90% decrease following fires

[MOU1]Why switching between RH98 and RH50? This is not clear

Following the recommendations, in the updated version of the manuscript, we show only results in terms of RH50 or relative height in the main. Figures with canopy heights or RH98 have been moved to the Extended Data and Supplementary Information sections.

See point#31 above.

38) Line 138: African and Asian logged forests remain structurally different from intact forest areas for up to 30 years after disturbance, presenting a top canopy height [MOU1] 10 m lower compared to intact forest and an absence of the tallest trees (>25 m) (Extended Data Fig.6).

[MOU1]Different metric again

See point#31 above.

39) Line 141: , with rates 25-65% (rewording)

We changed accordingly.

40) Line 148: Logged and burned forests showed highly degraded forest structures (CH 70 to 85% lower than intact forest on average), slow CH recovery in Africa (0.77% per year), and continuous degradation (-1.55% and -1.42% of CH loss per year[MOU1]) in Asia and America

[MOU1]Respectively for logged and burned? Not clear

Thank you , we have now clarified this sentence and it reads as such (lines 256-259):

” Immediate impacts on RH50, RH98 and AGB from compound disturbances (e.g. selective logging followed by fire) were on average twice stronger than logging alone.

However, longer time series would be needed to fully assess the long-term dynamic and the legacy of cumulative selective logging and fire impacts on canopy heights and AGB recovery.”.

41) Lines 162-164 The average rates of regrowth in forest height after deforestation (2.33%, 2.38%, 2.03% of canopy height expressed as a percentage of intact forest CH for the Americas, Africa and Asia, respectively) were 18-33% lower than after logging (Fig.2c, Extended Data Table 1[MOU1]).

[MOU1]This could be clearer - hard to follow sentence

We simplified and rephrased the sentence as such (lines 260-263):

“ The average annual rates of forest regrowing on deforested land were ~2 to 3 times higher than after selective logging (Fig.3c, Extended Data Fig.16, Extended Data Table 8). After 10-15 years, the recovery plateaued at 40-50% of intact old-growth forest RH50 with low growth rates (0.77%/yr, 0.72%, and 1.31%/yr for the Americas, Africa, and Asia, respectively). ”.

42) Line 171: As the tree's maximum height recovers more quickly than the tree's maximum diameter[MOU1] , A

[MOU1]I don't understand how a tree's maximum diameter can 'recover'. Not clear whether this refers to crown diameter or DBH either. Do you mean canopy grows taller for the same mean canopy diameter?

Action taken:

-> As suggested, we modified the text such as (lines 263-274):

" The recovery for AGB was on average twice slower than RH50, reaching 40% (38%, 39% and 41% Americas, Africa, and Asia, respectively) of intact forest AGB after 20 years of regrowing, rates which are similar to those reported by Poorter et al.88 (33%)

using field inventories, or by Heinrich et al.¹¹ (36-49%) using remote sensing data. However, the slowdown in recovery rates of AGB after 10 years of regeneration (Extended Data Tables 9-11) may indicate that several drivers are impacting forest growth and are not captured by Poorter et al. due to the limited representativeness of the field inventories data across climatic, environmental and anthropogenic gradients. Despite a complete time series, we found that land use intensity through repeated deforestation events and fire occurrences prior to forest regrowth may have negative effects on regrowth after 5-10 years (Supplementary Fig.6), which was corroborated by previous studies^{89–91} where fire legacies could decrease regrowth rates by 20 to 75%, particularly in drier and water-deficient regions."

43) Line 172: This overestimation [MOU1] of secondary forest resiliency

[MOU1]Compared to what? These are both estimates, not measurements - so you are just talking about a difference between your results and those of others, unless you have real validation data.

Many thanks for this relevant comment. We have updated the text accordingly (see point #42 above).

44) Line 176: The other forest structural variables (RH98 and FHD) show similar [MOU1] patterns

[MOU1]?

See point#31 above. The whole section on forest regrowth has been rewritten integrating directly in the main text results from the other structural metrics.

45) Line 179: recovery after a disturbance was slower than RH50 (Extended Data Fig.8-9), which is likely because emergent trees captured by RH98 will take decades to recover⁵⁵[MOU1].

[MOU1]There is a lot of rather vague speculation about causality to explain the observed patterns.

Following the recommendations, in the updated version of the manuscript, we re-wrote this entire paragraph on forest regrowth knowing that AGB recovery from GEDI is now allowing us to compare directly with previous studies based on forest inventories or airborne LiDAR analysis or a combination of both. We showed that our results are aligned with field data estimates and that global remote sensing products allow capturing changes occurring spatially across the three continents, covering 30 years of disturbance history, something which is difficult with field data. See point #42 above for more details.

46) Line 185: Results are reported as the percentage of intact forest canopy height (solid line[MOU1]) after normalising the difference in canopy height within each grid cell between intact forest and each forest type

[MOU1]Without looking at the ED (I will) but is this a spline fit, or a model? If the former, then it is misleading to join up the points here, particularly in a where there are 5 & 10 year gaps with no data in the America, Africa plots.

Thanks for this comment. A loess smoother was used to visualise trends within the data. Loess smooths are constructed by making a large number of quadratic (or possibly linear) regression lines as a window moves along the x-axis. The predictions from these many local regressions are then computed and plotted as a “smooth” of the data. The actual results from a smoothing spline or loess are going to be pretty similar (see point #52 reviewer#1). We did use the loess regression to perform any statistical inference. Our aim was to facilitate the identification of patterns, as detecting trends visually can be challenging. As the reviewer noticed there are indeed missing data gaps, but our intention was to highlight broad trends. If a parametric method like GAM (Generalized Additive Models) had been employed, it is likely that confidence intervals for regions with no data would have been quite large.

With regard to the existing gaps, the only way to have a “denser” time series is to relax quality checks (i.e using different thresholds of GEDI shots per year), but there is a delicate tradeoff between samples size and uncertainties as noted by reviewer#2 point#7.

Action taken:

-> Following the recommendations, in the updated version of the manuscript we added the following sentence to explain the smoothing method used:

“The solid lines depict the local polynomial regression (loess) fit to the data.”

47) Line 213: These results are likely due to a combination of localised edge effects from degradation but also to omissions in the forest [MOU1] cover change datasets to detect changes in forest structure from small-scale and low-intensity disturbances at the interface between ‘undisturbed’ and degraded forests.

[MOU1]Can you not test this explicitly with more detailed info at site level say? Again, a bit vague

The reviewer is right, we can be more clear. The text has been rephrased such as (lines 178-180):

” These results are not only due to localised edge effects from degradation but also to the omission in the Landsat-based forest cover change datasets of small-scale and low-intensity disturbances at the interface between ‘undisturbed’ and degraded forests^{1,69}. ”.

Omissions in the detection of short-duration forest disturbances at proximity to selective logging roads and gaps detected in the Landsat products (e.g. JRC-TMF dataset) have been quantified by Lima et al.⁷:

“Logging infrastructure was better detected from Sentinel-2 (43.2%) than Landsat 8 (35.5%) data, confirming its potential for mapping small-scale logging.”

and it was also clearly identified by Vancutsem et al. in the accuracy assessment of the JRC-TMF dataset² (see Vancutsem et al. 2021, supplementary materials):

”most of the omissions of the single-date classification algorithm were ‘temporary’ omissions, as most of these disturbances were then confirmed from the full temporal Landsat time series. These probably correspond to omissions at the beginning of disturbance events. It is also

important to mention that 87% of these omissions were plots with minor non-forest extent (fewer than five pixels interpreted as non-forest within the sample plot)."

Separating the impact of omitted disturbances at the edge of logged or burned forest with localized effects that impact vertical structure at low scales is for the moment impossible with the current data, however, large-scale applications using Sentinel 1 and 2 imagery would improve forest cover change mapping by reducing omissions of small scale disturbances and thus refine the analysis of localized edge effects which remain understudied.

48) Line 228: Accounting for forest structure degradation related to the penetration of edge effects that remain undetected by optical [MOU1] satellite imagery has major consequences for pantropical intact forest landscape mapping

[MOU1]Passive optical? And what about radar?

We agree about the need to cite Radar literature on tropical moist forest (see point#18 above). However, optical remote sensing - such as Landsat - provides the longest observation on forest cover change in intact forest which is critical when monitoring forest degradation and regeneration.

Action taken:

-> Following the reviewer's suggestion, we have added an appropriate reference list in the text and included a paragraph about the radar EO. See also point#18 above.

49) Lines 234-236: At the continental level, our approach mapped smaller intact forests in America (-9%, 378.1 Mha Potapov vs 347.1 Mha), while in Africa and Asia, intact forest landscape was more strictly defined in Potapov than in our approach [MOU1] (79.2 Mha Potapov vs 102.1 Mha in Africa and Mha Potapov vs 65.2 Mha in Asia). We expect that the inclusion of additional criteria on minimal forest patch size and connectivity[MOU2] would significantly reduce our estimation of IFL extent

[MOU1]How? And what happens if you tighten your definition to match theirs?

[MOU2] I think you need to do this if you want to generalise in terms of explanation.

Thank you for this comment, we have now added an additional criteria on the minimum patch size (500 km²) to better match the Intact Forest Landscape (IFL) extent as defined by Potapov et al.⁸. We resampled our TMF-derived IFL map from 30m to 1km in order to compute - using Google Earth Engine - the number of connected pixels (where each pixel contains the number of 4-connected neighbours including itself) and then restrict it to 500 to get an approximation of forest patches greater than 500 km² (i.e. the minimum area of intact forest defined by Potapov). We updated the text accordingly and provided maps of comparison between our approach and Potapov's (see below Fig A10-A12, now in the extended data fig 12) and added a paragraph in the Method section.

The final text reads as such (lines 197-199): “ **However, by including additional criteria on the minimal patch size of 500 km² (37see Methods for more details), IFL was further reduced by 23%, and it resulted in smaller extents compared to Potapov's (-14%, 427 Mha with this approach).** ”.

a

Intact Forest Landscapes - Bourgoin approach

Fig A10: Area of TMF-derived IFL (here referred to as Bourgoin - the main author of this study - approach). IFL was defined following these criteria: undisturbed forests in 2020 located at 1) more than 350, 600 and 1700m from the edge for America, Africa, and Asia, respectively, 2) at more than 120 m distance from selective logging or fire and 3) with minimal patch size of 500km². Dark grey grid cells present no IFL area.

b

Intact Forest Landscapes - Potapov approach

Fig A11: Area of IFL extent, IFL was extracted from Potapov's 2020 and the extent was restricted to the tropical moist forest domain. We further excluded mangrove, forest conversion to water detected in the JRC-TMF dataset and bamboo-dominated forest areas to allow comparison with our approach. Dark grey grid cells present no IFL area.

c

Intact Forest Landscapes Bourgoin - Potapov

Fig A12: Difference in area between our approach and Potapov's.

The resampling of our IFL map from 30m to 1km was necessary as counting connected pixels using the Google Earth Engine algorithm is restricted to 1024. We retained 1km cells only

when presenting more than 1000 IFL pixels from our 30m original map and evaluated the sensitivity of this threshold on the final comparison with Potapov, varying it from 375, 750 and 1125 pixels which represent one quarter, half and three-quarters of the maximum number of pixels of 30m within ~1km grid cells (Table A1). We selected the threshold 1000 as a good compromise to tackle overlapping and forest edges issues. You can find below that the spatial trends in the differences between our approach and Potapov are similar and we also report areas of IFL when varying the threshold:

Table A1: Area estimates (Mha) of Intact Forest Landscape in 2020 in the tropical moist forest domain from Potapov and our approach (30m resampled to 1km). Sensitivity analyses on the estimates from our approach when varying the number of IFL pixels of 30m within 1km grid cells after resampling.

thresholds	IFL Potapov (Mha)	IFL Bourgoin (Mha)	% difference Bourgoin - Potapov
375	501.71	620.20	23.62
750	501.71	518.76	3.40
1000	501.71	427.13	-14.87
1125	501.71	364.74	-27.30

Fig A13: Sensitivity analysis when resampling our map of IFL 30m to 1km

50) Line 243: Fig.3 Forest edge effects: RH50 [MOU1] of undisturbed forests

[MOU1]Again, why Rh50 this time, not RH98? There's no consistency or seeming logic to why some one metric used in one instance, and another elsewhere.

The reviewer's comment is well-taken. In the revised version of the manuscript, we always presented results in the main in terms of RH50 (or relative heights), while canopy heights - RH98 (and the added Above Ground Biomass) are presented in the Extended Data and Supplementary Information sections.

Action taken:

-> See point #3 reviewer#1.

51) Line 243: located at more than 210 [MOU1] metres from degraded forests

[MOU1]Why 210

Thank you for pointing out this issue, we meant 120m. The 120-m buffer was selected following previous studies and it is likely a conservative buffer distance for a global analysis.

Action taken:

-> We have included more information on the choice of this distance of 120m in the Methods section (lines 478-480):

“ This distance of 120 m corresponds to the area initially affected by the felling of individual trees in selective logging operations^{106,107} where localised edge effects (i.e. increased tree mortality and altered forest structure) are the highest²⁷.”

52) Line 246:). The inset caption represents the cumulative area (1000 ha) of degradation due to fire and selective logging calculated at various distances to the edge[MOU1] . The

[MOU1]And is the solid line a spline fit again, or a model?

-> The solid line is a loess smoothing (see point #46 above).

-> We added Fig.A14 (below) to show that between spline and loess smoothing there are minimal differences.

Fig. A14 Example of LOESS (top) and spline (bottom) smoothing for Americas (first inset on the left of Fig.2). Note that the x-axis is not in log scale as in the main to allow comparison between the two methods.

53) Line 249: Vertical bars indicate the spatial standard deviation. GEDI samples for each distance to the forest edge are reported in Extended Data Fig.13. Red dotted line indicates 95% of intact forest CH[MOU1] .

[MOU1]So this seems to me a more useful figure than fig 1 particularly. But there are also a number of issues in trying to interpret this. Why 95% specifically? What are the solid lines (are above)? And the solid and dotted blue lines are not explained. I presume this is undisturbed (and black bars) and degraded (gray bars) but this is only my interpretation. And why not

express both in terms of the 95% height level? For Africa they coincide but for the other two the latter is much further from the forest edge (as noted in the text)

We fully agree that Figure 3 is far more important than Figure 1. For this reason, in the new version of the manuscript Figure 3 is now Figure 2. However, we have decided to retain Figure 1, as it is important to understand the methods used in this study, i.e. dividing the spatial domain into hexagonal cells, and to summarize the main patterns which are then further developed in the next sections.

Regarding the 95% threshold, it was primarily chosen to aid in the interpretation of edge effects depicted on the graph. Specifically, the red dotted line highlights the reaching of an asymptote for changes in RH50 in relation to distance from the edge in undisturbed forests. Similar approach was adopted by Chaplin-Kramer et al.⁹ at the tropical scale.

The solid line refers to undisturbed forest at various distances to the edge and we used this to assess the extent of edge effects that are uniquely due to canopy desiccation. We then present the cumulative effects of edge effects with impacts from logging and fire that co-occur and are exacerbated by drier conditions at the edge and by increased accessibility to forest resources. The result of the vertical forest structure is represented by the dotted curve where we sampled both undisturbed and degraded forest at various distances to the edge.

Action taken:

-> Following the recommendations, in the updated version of the manuscript we have specified solid curve, dotted curve when presenting edge effects and cumulative impacts from edge effects and from logging and fire disturbances. It now reads like this (lines 210-218):

” Fig.2 Forest edge effects: RH50 of undisturbed forests (located at more than 120 metres from degraded forests) and all forests (including undisturbed and degraded forests) at various distances to the forest edge (agricultural and other land covers). The inset caption represents the degradation area due to fire (red curve) and selective logging (purple curve) calculated at various distances to the edge. The turquoise vertical line is placed at a distance equal to 350, 600, and 1700m for America, Africa, and Asia, respectively, and corresponds to the distance between the forest edge and the point at which 95% of intact forest RH50 is reached (red dotted line). Vertical bars indicate the spatial standard deviation. The number of GEDI sample footprints for each

distance to the forest edge is reported in Extended Data Fig.13. The blue lines depict the local polynomial regression (loess) fit to the data. ”.

54) Lines 254-256: Here we show that it is also a determinant of future deforestation, where the risk of full removal of forest cover increases with the degradation stage of forests.[MOU1]

[MOU1]Arguably this is the most important conclusion of the paper isn't it? Seems to be rather understated, particularly compared to some of the other parts

We agree with the reviewer that it is important to put this result forward. In the updated version of the manuscript, we have corroborated this figure by using 3 years of GEDI data instead of 1 (as in the original version of the study). Furthermore, we presented results in terms of RH98 and Above Ground Biomass. We have finally made the link clearer between the long-term changes in forest structure after degradation (e.g. following fire and unsustainable logging) in section 3 (located in section 2 in the first version of the manuscript) and this final section on the fate of degraded and edge forests.

Note that the x labels in Fig.4 (and in the equivalent ones Fig for RH98 and AGB) have been changed because now we are using 3 years of GEDI data instead of 1.

55) Line 259: (32-41% difference in RH50 - Fig.4, 18-23% difference in RH98 in Extended Data Fig.14[MOU1]).

[MOU1]This is hard to follow. But if I follow it right isn't this because as outlined above, degraded forests lead to greater logging and fire? So same reasons as above

Thank you for this comment, we rephrased this sentence and in the updated version it reads as such (lines 293-299):

” We show that relative heights and AGB of degraded forests followed subsequently by deforestation (occurring after the year of GEDI measurements) are significantly lower

than those not followed by deforestation (Fig.4, Extended Data Fig.18-19). Degraded forests followed subsequently by deforestation were on average severely impacted by the degradation processes with reductions in mean RH50, RH98 and AGB by 60%, 45% and 65%, respectively, which corresponds to severe impacts from unsustainable logging and/or fire (Fig.3, Extended Data Fig.14-15). ”.

This part means that deforestation occurs most probably in forests that have been heavily degraded (lowest canopy heights and biomass) compared to low degraded forests now recovering.

Action taken:

-> Following the recommendations, in the updated version of the manuscript we

56) Line 265: No statistical differences in CH were found for undisturbed edges in the three continents, confirming the role of unsustainable logging and fire in enhancing the risk of deforestation[MOU1] .

[MOU1]Near to the existing disturbance

Thank you for this comment. We invite you to read again this last result section as it was re-written due to the fact that the increase in GEDI samples - to characterize the structure of degraded forests followed by deforestation or not - slightly modified the reported statistics.

57) Line 266: Similar [MOU1] results were found using Generalised Linear Models (GLM)

[MOU1]?

We have now clarified the sentence. The amended text reads as follows (lines 302-311):

“Using Generalised Linear Models (GLM) (see the section on statistical analyses in the Methods) we confirmed that canopy height variables (RH50, RH98) and distance to the edge were strong predictors of the probability of deforestation (Extended Data Fig. 20a and 21a). Degraded forests in America showed, on average, a higher deforestation risk than in Africa or Asia, as 50% of deforestation probability was reached when degraded

forests lost 50% (60% in Africa and Asia) of their initial heights. Also, distance to the forest edge, identified in other studies as the most critical indicator in determining the risk of deforestation^{94,95}, showed confounding effects with the relative heights of degraded forests (Extended Data 20b and 21b), confirming the synergies between various effects (namely, degradation, exposure to human activities and edge-desiccation effects within the first kilometre from the forest edge) in increasing the probability to subsequent deforestation”

Action taken:

-> We added new text to clarify the sentence.

-> In the updated version of the manuscript we took the liberty to add a figure (in the Supplementary Information) to illustrate the logistic regression probability results with RH50 data over Africa of whether a forest is deforested or not based on the canopy height (Fig. S9).

Fig. S9 Logistic regression probability results with RH50 data over Africa of whether a forest is deforested or not based on the canopy height (expressed as a percentage of Intact forest canopy height).

58) Fig4 (line 283): Different vertical scales don't help comparison

Thanks for this comment. In the new version of the manuscript, we used fixed vertical scales to facilitate the comparison.

59) Lines 293-297: Our study unveils the destructive impacts of degradation on the canopy structure of tropical moist forests (TMFs), varying from a 40% to a 90% decrease in canopy height and extending up to ~2 km inside the forest interiors. [MOU1] This far-reaching effect of forest edge - previously undervalued and [MOU2] considered *de facto* of ~ 100 m⁸⁰ - demonstrates that the overall impact of fragmentation across the pantropical belt is severely underestimated [MOU3] with relevant potential impacts on the forest carbon dynamics and other ecosystem services.

[MOU1] So, overall it seems to me that the height aspect is not surprising, but that the distance of the impact is more so

[MOU2] Well, there is no absolute reference so it's not undervalued, it's just smaller than you have found here.

[MOU3] See above - no it doesn't

Action taken:

-> Following the recommendations, in the updated version of the manuscript we re-structured the conclusion with an emphasis on the extent of forest degradation and corrected the terms "undervalued" or "underestimated". We invite the reviewer to have another look at this section.

60) Line 305: Our study demonstrates that the integration of recent and spatially sparse observations from spaceborne LiDAR (~ 20 million [MOU1] GEDI sample plots)

[MOU1]So what? As above, why is this number relevant? It's still a tiny fraction of the total area covered by TMFs and is likely biased in terms of coverage (as I note above). So be careful about making a point of highlighting this unless there is some importance attached to that number.

We would like to mention it, but only once, and not with the intent to try to highlight the number. Note that we already removed the number reference in the abstract and we prefer to keep it somewhere in the main to give a general idea for the reader. Before the advent of GEDI, analyses at this scale with this amount of data (a tiny but representative fraction) were not possible or would have at least been less robust. We rephrased the text to put it more verbose such as (line 335):

“This corresponds to over 39 million individual sample plots”

61) Line 306: provides an incontrovertible [MOU1] assessment of forest structural attributes and enables the large-scale monitoring

[MOU1]You can't say this without some independent validation. So no it isn't.

We removed the word “incontrovertible”.

The text has been rephrased such as (line 336) :

“novel approach to assess the forest structural...”

62) Line 308: This type of accurate [MOU1] and transparent assessment of degradation is key for implementing more effective forest-based mitigation policies

[MOU1]I think you are eliding accurate with precise. The former implies some absolute reference, which is not possible here. Transparent yes, and potentially important because anyone can do and it can be refined eg to do increased local sampling w airborne lidar, but accuracy is a separate issue.

Thanks. We agree with the reviewer on the need to remove “accurate” from this paragraph.

Action taken:

-> Following the recommendations, we used (line 351):

“precise, spatially detailed, recent and transparent”

63) Line 317: The spaceborne Global Ecosystem Dynamics Investigation (GEDI) from NASA revolutionises this field by consistently acquiring high-accuracy forest canopy heights and structure measurements at scales that have been unattainable until now[MOU1] .

[MOU1]Make it clear you mean pan-tropical. This can be done on M of ha scale with airborne, national and regional even, and be careful as you ignore radar here as well.

Thanks for this comment, in the new version we made explicit that this study was done at pantropical scale (lines 357-363):

” In this study, we infer the long-term effect of degradation and regrowth on forest structure at the pantropical scale. The spaceborne Global Ecosystem Dynamics Investigation (GEDI10) from NASA revolutionises this field by consistently acquiring forest canopy heights and structure measurements at scales that have been unattainable until now across these vast areas. ”

64) Line 318: Fortunately, the development reviewer deleted the word recent)

Action taken:

-> We removed the word “recent” accordingly.

65) Line 348: The JRC-TMF product is based on the entire Landsat archive where every single image [MOU1] was classified allowing the precise detection of forest disturbances

[MOU1]Within the TMF region, not every single image?

We modified the sentence as such (lines 393-396):

“The JRC-TMF product is based on the entire Landsat archive collection (1982-2022) where every single image covering the TMF region was classified, allowing the precise detection of forest disturbances (i.e. absence of tree foliage cover within a 0.09 ha size Landsat pixel due to e.g. selective logging or fire) that are visible only over a short period from space.”

66) Line 390: The resulting pixels correspond to temporary logging roads, logging gaps, decks, and skid trails. The WRI dataset covers the Central Africa region (Cameroon, Central African Republic, Gabon, Congo and the Democratic Republic of Congo), Indonesia, and Malaysia and was completed by an extensive [MOU1] visual interpretation and delineation of selective logging operations in Brazil, French Guyana and Guiana based on Google Earth and Landsat images

[MOU1]By what metric?

Thank you for this comment, we rephrased this sentence accordingly (lines 442-445) :

“... was completed by an extensive visual interpretation and delineation of selective logging operations in Brazil, French Guyana and Guiana based on its spatial patterns (e.g. logging roads, skids trails and logging gaps) visible on the JRC-TMF Transition Map1. ”

67) Line 438: ● In this case, it will be classified as regrowing the year after (when no deforestation will be detected all over the year[MOU1]).

[MOU1]There is a potential problem here in that the analysis of all degradation and edge is done using Landsat ie optical data that as noted in the text, is insensitive to many aspects of change, particularly structural. So you start from a definition of these areas that is relatively insensitive to the very thing you want to look at - is this an issue?

This issue has already been tackled in point#5 above.

68) Line 450: For each footprint, we extracted a set of relative height (RH) [MOU1] metrics and the Foliage Height Diversity (FHD).

[MOU1]These are provided, so there is no additional processing here right?

Yes, these metrics are provided, but we prefer to keep the word “extract” because with Google Earth Engine we literally extracted/pulled out statistics of RHs for each gridcell, as stated by Gorelick¹⁰.

69) Lines 461-464: . A recent analysis of intact forests with GEDI assessed the human impact on the structural density of forests globally. However, this was only focused on intact forest and protected areas and did not consider degradation and forest regrowth⁹⁴. [MOU1]

[MOU1]You need to address this in the main text as it is v relevant

Thanks for this comment. In fact, the reference was used in the Method section and not in the main because, until June 1st this article was available only in preprint format. Now that the article is out in Nature Communications the article has been cited (see point#11 reviewer#2). There are some similarities between the papers, but all in all, they are distinct in scope and target (e.g. forest conservation vs forest degradation). But it is important to note that the targets are very different, and that intact does not mean protected, and degraded does not mean not protected or managed. For this reason, we prefer to cite this article in Methods.

70) Line 466: We selected GEDI data acquired from the 1st of January 2019 to the 31 of December 2020 to ensure having a large number of samples without creating too much bias [MOU1] in the assessment of the years since the last disturbance,

[MOU1]What is large and what is too much?

Thanks for this comment. We agree with the reviewer that the terms “large” and “too much” are arbitrary. Now we selected the entire GEDI collection from 2019 to 2022 (see point #10 reviewer#2) and there we removed this part of the sentence.

Referee #2

General Comments

This paper provides a nice example of the application of structural measurements of forests from NASA's GEDI mission towards understanding changes that occur after degradation and deforestation with regards to intact forests. It also leverages these data to examine the impact of disturbance edges on interior intact forests over a vast area of the tropics. In both cases, there is an attempt to analyze impact of the type and timing of disturbances. The research underscores the new avenues of exploration global estimates of structure from GEDI are enabling.

We thank the reviewer for the appreciation of our work.

1) In reviewing this submission, several points emerged. First, there may be potential issues in the paper with respect to statistical methods and GEDI data that should be addressed (and expanded on below). Secondly, while the paper in any section is well written, its overall narrative structure could be improved. There needs to be a cleaner story that the reader can more easily grasp and not be confused by the very many statistics and results that are presented. Some type of overall table of results might help with proper positioning relative to the context of and motivations of the paper. What specific hypotheses are being tested? How are these reflective of major unanswered questions regarding the role of deforestation and degradation? What was discovered relative to these?

Thank you, we tried to circumvent all the issues mentioned in the specific comments below with respect to statistical methods (using for instance sensitivity analyses) and to the GEDI data (by integrating the full 2019-2022 GEDI archive). Regarding the narrative, we rearranged the text and emphasized on the need to quantify the impacts of forest degradation across time and space. We changed the order of the figures, first presenting the general spatial patterns of the different forest types of interest, then presenting the magnitude and scale of edge effect (most important and novel result), then the temporal aspects and particularly the persistence of degradation at the edge, the plateauing of forest regrowth rates, finishing the paper on the fate of degraded forest and the call for their protection. For each result, we have made more explicit links with local studies based on airborne lidar or field inventories.

Following your recommendations, we have also added tables on the immediate impacts of forest degradation and on the recovery rates following disturbances (expressed as average annual rates and by calculating the slope of recovery rates for forest regrowth and forest edge).

Regarding the major unanswered questions on the role of deforestation and degradation, we made them explicit from the introduction to the final result section.

2) The concept of GEDI “unveiling” information about forest structure that has been hidden (from the title) is a very good one and could be emphasized more. Even so, the reader may ask what exactly has been hidden from view? That structure changes after disturbance and will take time to recover is self-evident. Rather is not the discovery element the quantification of these changes that GEDI enables? Approaching the paper and its organization of results from that perspective of discovery grounded in major conclusions within a framework of inquiry that is clearly linked to the important ecological issues would make it more readable, accessible, and ultimately exciting.

Thanks for this comment. In fact, the enormous amount of optical or radar images that have been available for the Tropical Moist Forest so far, (almost) always lost any three-dimensional information. We do agree with the reviewer’s comment and we have modified the manuscript and the organization of results. However, as asked by reviewer#1 (see point#8), we used the word “Quantifying” instead of “Unveiling”, to refer to a broader and more complete evaluation of the degradation.

Action taken:

- > Following the recommendations, in the updated version of the manuscript we changed the order of Fig 2 and 3.
- > We highlighted the ecological role of RH50 - see point#24 below.
- > We modified the introduction trying our best to use the perspective of discovery.

3) The paper could benefit from a conceptual model that relates GEDI metrics (RH98, RH50, FHD) to biophysical elements on the landscape. An introductory paragraph about the metrics and their ecological significance would go a long way towards getting the reader to stop thinking in terms of metrics. It may be that these kinds of lidar metrics are so new as applied to this problem over vast areas that an unambiguous connection between “degradation” and changes in the metrics is not possible yet. Rather, it is the exploratory nature of this submission that is pointing the way to a new paradigm – these metrics provide new ways of identifying and understanding impacts.

Thanks for this comment, we added an introductory paragraph as indicated. See point#24 below.

4) It appears the paper is lacking a definition of what “degradation” is. How are the use of these GEDI metrics consistent with established definitions of degradation? How would recovery from degradation appear in the metrics through time? Again, loss of canopy height usually means loss of trees. But “loss” and “recovery” of metrics are not so well-defined.

Thank you for this comment, we have added a more general definition of forest degradation at the very beginning of the introduction in order to stress that deforestation and degradation are two different processes and explain why quantifying its impacts across time and space is challenging. Here is below an extract from the updated introduction (lines 42-51):

“ Anthropogenic disturbances also cause the degradation of remaining TMFs, that is, the reduction of forest ecosystem services with no change in land use^{18–21}, at a global rate similar to deforestation^{1,22}. It leads to forest cover and biomass changes, accounting for 25-69% of overall carbon losses from tropical forests^{3,23}. Forest degradation is mainly caused by selective logging, fires and edge effects⁷, i.e. increased mortality of trees at forest edges²⁴, and it is often exacerbated by a combination of these disturbances occurring together, repeated over time and driven by continued deforestation, habitat fragmentation and human-induced climate change^{6,25,26}. The synergies between agricultural expansion, selective logging, edge effects, drought, and fire occurrences^{27–34} have made forests vulnerable to further

disturbances⁷, which may hamper their potential resilience and threaten their long-term future¹. ”.

At the end of the introduction we have made a clearer link between forest degradation/recovery and forest structure metrics and have produced a figure on the cumulative return energy of GEDI relative height metrics for the main classes of interest in this study i.e. intact, degraded, edge and regrowth to show the sensitivity of RH50 to capture vertical variations in canopy structure compared to RH98. See point #29 from Reviewer 1 for more details.

5) The paper asserts that “intact” forests show signs of degradation 2 km in from an edge – so how can they be considered intact? This is one of the critical points of the paper, that passive optical remote sensing is misrepresenting the notion of intact. If this result is validated in other studies, it would be an exciting finding indeed. The authors should make more of this point from the very outset – perhaps this is the central finding? Can comparisons with other maps of intact forests be done formally? Under what conditions and where geographically do we see this divergence from optical estimates? Do the specifics lead to summary conclusions that are geospatially comprehensible?

The reviewer is right. This notion is now better explained in the introduction where we stress the importance of using forest cover change maps from optical remote sensing (short and long term disturbances) to target the sampling of GEDI footprints to unveil edge effects patterns. Despite the capacity to directly validate our results, we added to the Method section some specifications on the accuracy of the JRC-TMF dataset (see point 1 from Reviewer 1) and we have compared more explicitly our results with other recent studies using airborne Lidar data covering localized studies across the TMF domain. Our results are aligned with some previous studies^{9,11–15} and bring more refinement on the scale and magnitude of edge effects across the pantropical scale. By having 4 years of GEDI we have also increased by 47% the number of footprints falling within the forest edge class which help to consolidate our results.

We agree with the reviewer that this result is central and for this reason we have switched figure 2 and 3 presenting edge effects first. We have also changed the order of figure 3 panels

to present the persistence of edge effects over time which creates a more logical transition with the previous result (on the scale and magnitude of edge effects).

Furthermore, in the new version of the manuscript, we performed a comparison with Potapov's map of intact forest valid for year 2020 and identified similar extent although our mapping of intact forest landscapes is reduced when adding additional criterias (i.e. minimal forest patch size). We mapped areas of agreement and disagreement (see extended data fig 12) and discussed it in the main. For further details, please refer to point 49 from reviewer 1.

6) It was disappointing to find that aboveground biomass was not also included in the analysis. Those data are readily available from the mission. Because biomass incorporates multiple aspects of the complex factors that could affect forests after disturbance and degradation (soil compaction, loss of fertility, microclimate changes) it would be a logical inclusion. Yes, it makes the analysis more difficult because there is an added source of uncertainty which comes from calibration models that turn GEDI metrics into biomass estimates. But we should see concomitant losses in biomass that are quantified. Furthermore, the accumulation of biomass through time after a disturbance is of great interest (the net impact of deforestation and degradation and subsequent regrowth). The logical inference is that the carbon content of intact forests is impacted by edges, but what was the magnitude of those losses? This paper could quantify that and seems within scope.

Thanks for this comment. The reviewer is right. In the original version, we excluded AGB because the GEDI L4A product (version 2.1, with footprints of ~25 m in diameter) was only released in Google Earth Engine when our manuscript had been finalised and was ready for submission. Also, we didn't want to include another variable with many uncertainties. Following the recommendation from the reviewer, we now included AGB in the analysis. However, we keep the analysis in the Extended Data and Supplementary Information sections, because the focus of the paper is on forest structure rather than the AGB. We do however discuss AGB results in the main which allow us to compare with airborne lidar-based studies on the edge effects or on other impacts from disturbances and regrowth.

Action taken:

-> Following the recommendations, in the updated version of the manuscript we added the GEDI Above Ground Biomass (AGB) product to the analysis (Fig. A15). Further, for the sake of readability, we excluded FHD that was not explained in the text and its interpretation is more difficult for a generalist reader.

a

Forest Edge (AGB)

b

Forest degradation from selective logging and fire (AGB)

c

Forest regrowth (AGB)

Edge Effect (AGB)

— Undisturbed ··· All Forests (Undisturbed or Degraded)

Vulnerability of degraded and edge forest to deforestation (AGB)

RH50 vs AGB

Rh98 vs AGB

Fig A15, as Figures in the manuscript with Above Ground Biomass (AGB) instead of RH98 or RH50.

Interestingly, for AGB we found differences in Asia (Scatterplot in Fig. A15), probably due to the stratification and how AGB is computed. To further investigate the bifurcation of the blue points in the scatterplot of Fig.A15, we split the Asia continent into two subclasses: Asia and Oceania. This was done because we aggregated Asia and Oceania in our analysis, mainly because we did not have hexagons enough for robust statistics in Oceania.

Fig.A16 show the scatterplot (with r^2) for the four continents. Again, over Asia, we can observe two different clusters. For this reason, we let the data speak and we performed an unsupervised classification, using a K-means algorithm with two groups. The K-means method is able to separate the two distinct clusters (see Fig. A17) that roughly correspond to Continental and Insular Asia (see map in Fig.A18). This is confirmed by what was found in this study from Duncanson et al.¹⁶ where it is stated that AGB is computed fitting models stratified

by continent and PFT and that over Continental Asia there is poor model performance mainly due to relatively sparse data.

Fig A16 RH50 versus AGB for the four continents.

Fig A17 K-means clustering (cluster 1 - left, cluster 2 - right) with two groups for RH50 versus AGB for Asia.

K-Means

Fig A18 Maps of K-Means cluster over asia.

7) The statistical methods assume random sampling. GEDI shots are not random, they are cluster samples. Some justification is required here and discussion of impacts if the assumption is violated in the formal testing. Some of the minimum shot criteria (e.g., 10 shots) seem quite low and highly susceptible to non-random sampling issues and should be tested. There also needs to be a quantitative justification for why a cell with only 300 shots over 200 km is sufficient. Was this tested?

We do agree that additional analysis on the statistical methods would be helpful to strengthen the robustness of our findings, and, for this reason, we performed a sensitivity analysis with different gridcell sizes.

“300 shots over 200 km is sufficient”?

-> Summary: We are now using 600 shots over 1.5 degrees (~167 km) - see point#35 reviewer#1.

Regarding the 300 GEDI minimum sample threshold, we performed a sensitivity analysis and we raised this threshold to 600 samples per cell. Also, the issue of the non-randomness and the GEDI orbit/tracks per gridcell has been addressed in point#32 below.

Fig. A19 to 22 show the distribution of the number of GEDI shots per gridcell (now of 1.5 degrees instead of 2) for Intact Forest, Forest Edge, Degraded Forest and Forest Regrowth. While it is generally possible to raise this threshold to 1200 samples for Intact Forest, Forest Edge and Degraded Forest, for Forest Regrowth we might encounter some issues due to the scarcity of samples especially in Africa. In other words, higher thresholds are reducing the number of acceptable gridcells. Fig. A23 show the maps of RH98 for Forest Regrowth with 300, 600 and 1200 samples per cell. Results of this loss of information are also reported in Table A1, where we reported the percent decrease in the study area as a function of the minimum sample threshold adopted. Cell in green in Table A1 is showing the high value of decrease in study area covered by selected grids when a selection criterion of a minimum of 1200 samples is adopted over Asia.

Fig. A19 Histogram of the number of GEDI samples per gridcell for Intact Forest. Vertical dotted lines are at 300, 600 and 1200 samples per cell.

Fig. A20 Histogram of number of GEDI samples per gridcell for Forest Edge

Fig. A21 Histogram of the number of GEDI samples per gridcell for Degraded Forest

Fig. A22 Histogram of the number of GEDI samples per gridcell for Forest Regrowth

Table A1 Percent decrease in study area covered by selected grids when a selection criterion of a minimum of 600 and 1200 samples is adopted

	minimum of 600 samples	minimum of 1200 samples
Intact	7%	14%
Edge	3%	13%
Degraded	4%	15%
Regrowth	20%	44%

Regrowth - 300 samples RH98

Regrowth RH98

Regrowth - 1200 samples RH98

Fig. A23. As Fig. 1 in the original version of the manuscript, when a selection criterion of a minimum of 300 (top), 600 (centre) and 1200 (bottom) samples is adopted.

Regarding the 10 GEDI shots, the reviewer is right and refers to this sentence:

"The threshold of GEDI samples for each grid cell was lowered to 10 for recovering forests after selective logging and fire and to 100 for other degradation and regrowing classes for each time step of the trajectory as those shown in Fig.2." .

In the new version of the manuscript, we always used a threshold of 600 GEDI shots per cell, and when analysing Figure 2 and 3, we used the following scheme:

For each forest class (i.e. logged forest, forest regrowth etc) we considered only cells where there are at least 600 GEDI shots.

Then, in the new Fig.3 (Fig.2 in the original version) where we are analyzing forest height with an annual time step, we selected only years when we had at least 30 GEDI shots per year. In other words, for logged forest class (i.e. top panel Fig. 3a) we selected only cells with a minimum of 600 samples (total) and 30 samples per year. Please note that the year does not refer to the GEDI date (where we only have 4 years of data) but to the JRC-TMF dataset where the timing of degradation/regrowth etc. is assessed.

We performed a sensitivity analysis to explore how changing this annual threshold to 50 or 100 samples per year is changing the results. Fig.A24 to 26 show the effect of this annual threshold.

For the new Fig 3, results show that already raising the threshold to 50 samples per year, we start losing annual information while raising it to 100 samples per year, and we completely lose forest classes (e.g. logged forest 1x in panel b).

Regarding the new Fig.2 (Fig.3 in the original version), instead of annual classes, we had distance classes, and, as for Fig.3, we used the same 30 sample thresholds per distance class. Results of the sensitivity analysis are presented in Fig. A27 to 29. Figures show that raising the threshold for distance classes to 50 or 100 has a limited impact on the results. However, to be coherent throughout the manuscript we always used the 30-sample threshold.

The text has been rephrased such as (Method section, GEDI minimum thresholds sub-section, lines 592-603):

" To ensure robust and comparable observations of forest structure metrics across the multiple classes of forest cover change, we considered a minimum of 600 GEDI samples for each 1.5° grid cell and a minimum of 7 grid cells per continent to derive continent-level statistics of forest RHs and AGB. When analysing the time series (Fig.3

and Extended Data Fig.14-15), a minimum threshold of 30 GEDI samples for each time step of the trajectory - and a minimum of 600 GEDI samples for the sum of all the time steps - within each grid cell was required. Note that the time step does not refer to the GEDI date but to the JRC-TMF dataset where the timing of degradation/regrowth etc. is assessed. Similarly, for edge effect penetration, a minimum of 30 GEDI samples for each distance to edge within each grid cell - and a minimum of 600 GEDI samples for the sum of all the distances - was required (see Fig.2 and Extended Data Fig.8-10). Metadata on the number of GEDI samples for aggregated classes of forest cover change and for time series of forest recovery after disturbances or edge effect penetration are provided in Extended Data Fig.7, 13, and 17. “

Fig.A24. Fig. 3 with at least 30 samples per year (and total sample per category equal or greater than 600 samples).

Fig.A25. Fig. 3 with at least 50 samples per year (and total sample per category equal or greater than 600 samples)

Fig.A26. Fig. 3 with at least 100 samples per year (and total sample per category equal or greater than 600 samples)

Edge Effect (RH50)

Fig.A27. Fig.2 with at least 30 samples per distance (and total sample per category equal or greater than 600 samples)

Edge Effect (RH50)

Fig.A28. Fig.2 with at least 50 samples per distance (and total sample per category equal to or greater than 600 samples)

Edge Effect (RH50)

Fig.A29. Fig.2 with at least 100 samples per distance (and total sample per category equal to or greater than 600 samples)

8) With regards to the above, it would be helpful if the paper could show examples of actual GEDI transects, not aggregates, but actual track lines, that illustrate interior forest structures are impacted near edges.

Thanks for this suggestion. We agree with the reviewer on the need to show examples of actual GEDI transects. Our initial idea was to show a transect similar to the one in Fig.1 in this study¹³. However, it is difficult to show at the same time a long transect (i.e. in the order of km) and the impact of edge/degradation/regrowth on forest structure (i.e. an abrupt drop mainly visible for RH50 rather than RH98). For this reason, we prefer to show: 1) a ~3km transect of GEDI shots and 2) as an additional figure (see point#24 reviewer#2) the actual impacts of edge/degradation/regrowth on different RHs.

Action taken:

-> We added Extended Data Fig. 1a and b (see point 29 from Reviewer 1), where we show a GEDI transect in Amazonia (extended data Fig. 1b) and plot the vertical distribution of RHs, expressed as quantile levels.

-> text has been modified accordingly such as (lines 981-990):

“ Transect (-47.76, -3.68) of GEDI relative heights across deforested land, edge forest (width of 350m) and intact forest collected in the Brazilian Amazon (left panel). The vertical lines represent the GEDI Relative Heights (RH), which give the height at which a certain quantile returned energy is reached relative to the ground. The three black lines represent the height of RH98, RH50 and RH0. Zoom (right panel) on the vertical line, representing the quantile levels of the GEDI Relative Height (RH). The grey curves represent its density distribution, while the dots are the individual RH. RH98, RH50 and RH0 are shown in green. “.

“Cumulative Return Energy from GEDI L2A for different forest types (30 GEDI footprints for each forest type were sampled in a grid cell located in the northeast of the Brazilian Amazon). The vertical black lines refer to RH98 and RH50. The horizontal grey lines refer to the height of RH98 and RH50 for Intact Forest.”

9) The comments on the impact of slope are not quite correct. Slopes over bare areas can lead to spurious canopy height but its more complicated than that. Slopes on vegetated areas do not have a consistent lowering or raising of RH98 and RH50, i.e., a consistent bias. This depends on the actual configuration of stems in the footprint, and whether the slope is facing towards or away from the sensor (so a function of the slope aspect and the sensor off-nadir angle). The restriction to low slopes likely will bias results (e.g., steep slopes may be much less susceptible to degradation from logging as its far more difficult to access these). What can we say about this potential bias?

Thanks to this inspiring comment, we had the opportunity to learn something new and to improve the study by removing the restriction to low slopes. In the new version of the manuscript, we reprocessed the GEDI time series using a 20-degree threshold (in the original we used a 10-degree threshold) and we performed a sensitivity analysis.

We used these two main references ^{17,18} where the main messages are that

- 1) steep topography (generally exceeding about 15°–20°), especially in bare soils - might lead to incorrect data interpretation. Over mostly bare-soil terrain with high slopes,
- 2) GEDI sensitivity seems to be the most important parameter of all in tropical areas (also on slope daytime nighttime etc.)

By using a 20-degree threshold, we show that we generally removed few percent of GEDI samples for areas affected by logging or fires (more details in the Method on how logged and burned forest were defined), as shown in Fig. A30. Only areas in Asia are more affected by this threshold (in any case less than 20%, while for Americas and Africa, we are well below 10%).

Fig A30 can answer the direct request of the reviewer: "What can we say about this potential bias?"

We changed the text such as (lines 545-549):

“ We used the Shuttle Radar Topography Mission (SRTM) information to exclude GEDI footprints above 20° slopes to avoid errors in vegetation heights. Steep slopes might lead to spurious relative height metrics (especially over sparsely vegetated areas), so applying our threshold of 20° is a conservative approach¹¹⁹”

Fig A30 Distribution of GEDI shots for different slope classes for classes of intact forest, logging and fires (more details in the Method section on how logged and burned forest areas were derived). The colour of the points shows the value of the mean RH98 from each slope class.

10) I did not find the explanation of why the analysis was limited to only year of GEDI data readily justified. It that what was on GEE? There are at 3+ years of data that are there. Why can we not filter out shots in subsequent years that were disturbed in those years? That would have greatly increased available data, especially given some of the low shot counts.

The reviewer's comment is well-taken. In the original version, we used the JRC-TMF product which provides information on changes in forest cover from 1990 up to the year 2021. For this very reason, we excluded the GEDI shots for the years 2021 and 2022 because we want to be conservative, excluding thus GEDI observations over areas with potential changes in forest structure not yet detected by the JRC-TMF product. In April 2023, the updated version of JRC-TMF was released, and this concern was gone away. We added GEDI 2021 and 2022, almost doubling the available number of GEDI shots for the analysis (23 million sample footprints in the first manuscript vs more than 39 million now).

11) The analysis is not entirely novel given that four of the co-authors have just published a paper (29 March 2023) in Nature Communications Earth & Environment: "Spaceborne LiDAR reveals the effectiveness of European Protected Areas in conserving forest height and vertical structure" (2nd author Ceccherini is 1st author of the Europe paper). That paper shares many elements to this submission. Referencing this paper and including some discussion of the difference between "intact" and "protected" areas should be done. If this paper were repeated only for protected areas in the tropics, would we see the same results?

We agree with the reviewer that the analysis has some connections with this other study and that we can cite this paper. In fact, the PA article, needed for European forest assessment, was also a "playground" to get familiar with GEDI data structure over a well-known spatial domain. At the time of submission, the other paper was not yet published.

There are some similarities between the papers, but all in all, they are distinct in scope and target (e.g. forest conservation vs forest degradation). Also, we don't know if this paper provides the same results if the focus was only on PAs in the tropics. But it is important to note that the targets are very different, and that intact does not mean protected, and degraded does not mean not protected or managed.

The paper in Communication Earth and Environment fits into a line of research fully devoted to forest conservation rather than degradation and analysis of drivers of forest disturbances. Further, the paper was a proof of concept designed for small areas, not as the (very) recent Nat Comms paper by Duncanson et al. on PAs¹⁹.

Action taken:

-> Following the recommendations, in the updated version of the manuscript we cited the paper "Spaceborne LiDAR reveals the effectiveness of European Protected Areas in conserving forest height and vertical structure" (lines 537-540):

" A recent analysis of intact forests with GEDI assessed the human impact on the structural density of forests globally¹¹⁵. Other studies^{116,117} used GEDI to assess the effectiveness of Protected Areas in conserving forest ecosystems and mitigate climate change. However, these studies were only focused on intact forest and protected areas and did not consider degradation and forest regrowth."

12) Lastly, the conclusions are too emphatic. Saying the results are "incontrovertible" is fraught here given uncertainties in data and methods. The paper would be improved with a more meaningful discussion of limitations and the potential impact of assumptions on the conclusions. We have seen many examples of how reanalysis of remote sensing data can lead to different outcomes. Hence, circumspection in the conclusions and acknowledgement that subsequent processing and analysis of the GEDI data might lead to some changes would be advised. This is especially true given that the final data sets from the mission are not published and only one year of data was used.

We agree with the reviewer.

Action taken:

-> Too much emphasis: see point #1 reviewer #1. We have changed incontrovertible taking into account the potential uncertainty in TMF and GEDI data. We rephrase into "precise".

-> We used four years of data in the revised version of the manuscript since the JRC-TFM was released for 2022 during this round of review (see point#10 above). Adding 2021 and 2022 (in total 4 years of the GEDI mission) is consolidating and reinforcing the results.

-> We acknowledge that subsequent processing and analysis of the GEDI data might lead to some changes would be advised (lines 535-536).

“ We acknowledge that future reprocessing of the GEDI archive (now version 1) might lead to some change and substantial accuracy improvements as it was in the transition from version 1 to version 2. “

We invite the reviewer to have another read of the conclusions as it was fully re-written and re-structured to address this comment.

Specific comments:

13) Line 48-49: Check sentence construction

Action taken:

-> We modified the sentence such as (lines 53-56):

“ Presently the recovery of degraded forests and the regrowth of secondary forests on abandoned agricultural land are offsetting ~ ¼ of deforestation and degradation emissions¹¹. However, even if all disturbances ceased now, current forests only have a limited additional carbon storage potential³⁵. ”

14) Line 51: Reference UNFCC

We added a reference.

15) Line 70: Reference GEDI

We agree with the reviewer. We added a reference.

16) Line 88-89: Need more detail on what RH98/RH50 are to understand going forward.

We agree with the reviewer on the need to add more details on RH98 and RH50. See point 3 reviewer#1.

17) Line 92: “plots” should be “footprints.”

Thanks for this comment. We replaced “plots” with “footprints”.

18) Line 93: No definition what “range” means here. Is the standard deviation of the bootstrap heights? Also, remember to define somewhere that “canopy height” refers to the tallest thing in a GEDI footprint.

We agree with the reviewer that the “range” is not clear. We rephrased the text such as (lines 104-108):

“ The analysis of more than 23 million sample footprints of GEDI data (Americas: 16,092,827; Africa: 5,882,035; Asia: 1,523,891) shows that intact forests derived from the JRC-TMF dataset (for further details see section "Tropical Moist Forest datasets" in Methods) present RH50 values above 17m on average (Fig.1a), top canopy heights (RH98) varying from 25 to more than 35m (Extended data Fig 2) and an average Above Ground Biomass (AGB) of 250 Mg/ha (Extended data Fig 3). ”

19) Line 97-98: What are these canopy heights refereeing too? What does 29.4 +/- 2.94 m refer to? That is the mean height of something, and the SD is only 2.94m? Is that the precision of mean? What exactly? How did you statistically prove the differences are significant? If the uncertainties are precision of the mean, they don't look significant.

The reviewers' comment is well-taken. We agree on the need to better explain these stats. Within each gridcell, we computed first the mean values of RH98 and RH50 for intact forest. Then, we grouped by continent and we computed the average and the precision (i.e. the standard deviation) of the mean.

However, we agree with the reviewer that these statistics are not sufficient to get a complete picture. In the new version of the manuscript, where the grid system is different (we reduced the gricell size) and with more GEDI shots (we added GEDI data up to 2022), for each gridcell we computed first the mean and the **standard deviation** of RH98 and RH50 for intact forest. Then we computed the mean value and the standard deviation of these statistics per continent.

Extended Data Table.1 Number of GEDI shots and statistics of gridcells by continent of RH98 for Intact Forest

Continent	Number of GEDI shots	Mean Height [m]	Mean Standard Deviation [m]	Precision of the Mean [m]
Africa	5882035	30.1	8.13	3.49
Americas	16092827	28.3	7.03	3.26
Asia	1523891	33.9	9.13	4.73

Extended Data Table.2 Number of GEDI shots and statistics of gridcells by continent of RH50 for Intact Forest

Continent	Tot	Mean	Mean Standard Deviation	Precision of the Mean
Africa	5882035	17.6	6.71	2.34
Americas	16092827	15.8	5.39	2.23
Asia	1523891	19.0	6.69	3.11

Extended Data Table.3 Number of GEDI shots and statistics of gridcells by continent of AGB for Intact Forest

Continent	Number of GEDI shots	Mean AGB [t/ha]	Mean Standard Deviation [t/ha]	Precision of the AGB [t/ha]
Africa	5818569	227	104	39

Americas	15818644	229	121	53
Asia	1502346	366	176	113

Also, to avoid confusion we rephrased the text (lines 108-117) such as:

“ The spatial patterns of RH50, RH98, and AGB of intact forests reveal regional and continental variability (Extended Data Table 1 to 3) with the highest canopy heights and AGB located in Insular Asia and Western Amazonia. Overall, relative heights (RH50) were higher in Africa (17.49 m on average) and Asia (18.91 m) than in South America (15.77 m). We found comparable among-continent variations for RH98 and AGB, emphasizing consistent patterns across these metrics (Supplementary Fig.1). Note that the variability of AGB (“precision” column in Extended Data Table 3) is quite high over Asia. These results support previous observations^{55–59}, showing that old-growth tropical forests in Asia and Africa, which are typically dominated by wind-dispersed species, show a higher frequency of large trees compared to South America. ”.

20) Line 122: Can you do a t-test with non-random samples that GEDI produces? Do we believe 300 samples is sufficient in a 200 km grid cell and why? What if all those samples are from 1 single track going through the area? Should probably also show a map of the number tracks used in each 200 km cell.

Thanks for this comment. We agree with the reviewer to clarify this point. Regarding the statistical test, we used a Welch t-test to deal with unequal sample sizes. Regarding the non-random samples, the reviewer is correctly concerned about the spatial autocorrelation of the samples. Since we analyzed the number of orbits, as shown in point#32 below (i.e. a map of the number of tracks used in each 1.5 deg cell), we tried our best to circumvent this issue.

21) Line 132: Do you mean Americas (vs Amazonia) here?

Thanks for this comment, we meant Americas. The text has been updated accordingly.

22) Line 157: It’s difficult to have the single assertion in the sentence requiring the reader to go to three different places to understand it. You need to rewrite that sentence and then

have clauses that separate out Fig 2b, from EDT 1 and EDF 6 instead of lumping them all together.

Following the recommendations, in the updated version of the manuscript we rephrased this into (lines 224-231):

“ Undisturbed forests within the first 120 m of the forest edge exhibit on average a 20-30% lower RH50 compared to intact forests from the first year after edge creation (decrease of 10-20% and 20-40% on average for RH98 and AGB resp.). Fire or logging impacting undisturbed edge forests triggered an additional 40-50% decrease in RH50, RH98 and AGB on average with no evidence of recovery over time (Fig 3a, Extended Data Tables 7-8), which is corroborated by Airborne Laser Scanning (ALS) studies at local scales^{20,47,67}. This pattern is likely due to the long-term persistence of the edge effects driven by changes in the growing conditions and the exposure to additional anthropogenic disturbances^{20,47,74,75}”

23) Line 162: Are the rates per year?

These are 5 years average annual rates of forest regrowth. We have updated the extended data tables 8-11 to present recovery rates for all forest types considered here. This sentence reads like this (lines 260-263):

“ The average annual rates of forest regrowing on deforested land were ~2 to 3 times higher than after selective logging (Fig.3c, Extended Data Fig.16, Extended Data Table 8). After 10-15 years, the recovery plateaued at 40-50% of intact old-growth forest RH50 with low growth rates (0.77%/yr, 0.72%, and 1.31%/yr for the Americas, Africa, and Asia, respectively). ”

24) Line 169: This is where the lack of good ecological definition of RH50 creates issues. RH50 does not “recover” – it is a radiative transfer metric (50% of the cumulative energy is below that height relative to the ground). So, explain in clear terms what you mean when you say that. The intact RH50 may be dominated by the tallest trees, whereas when the forest recovers its dominated by young trees growing? What are you asserting here ecologically?

We would like to thank the reviewer for the useful comment. We have now amended the text written between Lines 169 and 171 to explain more clearly what we mean by 'recovery of RH50' from an ecological point of view. The text (without final references in this extract) now reads as (lines 260-274):

“ The average annual rates of forest regrowing on deforested land were ~2 to 3 times higher than after selective logging (Fig.3c, Extended Data Fig.16, Extended Data Table 8). After 10-15 years, the recovery plateaued at 40-50% of intact old-growth forest RH50 with low growth rates (0.77%/yr, 0.72%, and 1.31%/yr for the Americas, Africa, and Asia, respectively). The recovery for AGB was on average twice slower than RH50, reaching 40% (38%, 39% and 41% Americas, Africa, and Asia, respectively) of intact forest AGB after 20 years of regrowing, rates which are similar to those reported by Poorter et al.⁸⁸ (33%) using field inventories, or by Heinrich et al.¹¹ (36-49%) using remote sensing data. However, the slowdown in recovery rates of AGB after 10 years of regeneration (Extended Data Tables 9-11) may indicate that several drivers are impacting forest growth and are not captured by Poorter et al. due to the limited representativeness of the field inventories data across climatic, environmental and anthropogenic gradients. Despite a complete time series, we found that land use intensity through repeated deforestation events and fire occurrences prior to forest regrowth may have negative effects on regrowth after 5-10 years (Supplementary Fig.6), which was corroborated by previous studies^{89–91} where fire legacies could decrease regrowth rates by 20 to 75%, particularly in drier and water-deficient regions. ”

We have also provided a clearer explanation of the ecological meaning of RH50 in the introduction (lines 92-100):

“ We selected two structural metrics (Relative Height RH98 and RH50) that capture various aspects of forest structure, encompassing canopy height (RH98, hereafter referred to as top canopy height) and the distribution of canopy elements, as well as canopy cover and the fractions of gaps within the canopies (RH50, hereafter referred as relative height). We used RH50, i.e., the average height of median energy⁵⁰, as the main indicator in our study as it is strongly correlated to key structural variables including AGB, stem diameter, and basal area^{51,52}, and exhibits a high sensitivity to the variation in forest structure along degradation gradients compared to RH98^{53,54} (see also Extended Data Fig 1). Results based on the RH98 and AGB measurements are presented in the supplementary materials (details in Methods). ”

and in the Method section (GEDI dataset sub-section, lines 516-530):

“ RH metrics represent the height (in metres) at which a percentile of the laser's energy is returned relative to the ground. For example, an RH50 of 20 m means that 50% of the laser's energy was returned by objects up to 20 m above the ground. RH metrics are saved at 1% intervals, so each shot contains 101 values representing RH at 0%–100%. RH98 corresponds to the maximum canopy height (hereafter “canopy height”), which is a more stable height metric than RH100. RH50 specifically indicates the height at which half of the vegetation falls below and half falls above. It represents the midpoint of the vertical structure of the vegetation (also known as “Height Of Median Energy”-HOME50) and has been identified as one of the LiDAR metrics with the greatest potential for estimating structural characteristics in tropical forests⁵⁰. When validated against ground-based data, RH50 generally exhibits a strong correlation with key structural variables, including AGB, stem diameter, and basal area^{51,52}. Due to its strong dependence on the vertical distribution of canopy elements⁵⁰ and gaps within the canopies and canopy cover, RH50 serves as a highly complementary metric to RH98. Notably, RH50 exhibits a high sensitivity to the variation in forest structure along degradation gradients, as shown in previous studies^{53,54} (see also extended data Fig 1) making it a strong indicator for characterizing changes in canopy structure^{21,112,113}.“

Finally, as explained previously, we added extended data figure 1 to show the actual values of RHs for different forest classes in order to be more transparent on the underlying ecological processes and what RHs capture.

25) Line 171: Most ecosystem models show that height saturates much more quickly than biomass and should probably cite some of those. Your results are consistent with those previous modeling studies.

Thank you for this comment. Now that we have introduced AGB from GEDI, we have decided to keep the comparison of our results with two relevant papers (Poorter et al.²⁰ and Heinrich et al.²¹ who already provide an in-depth comparison with Cook-Patton et al.²², Philipson et al.²³ and Heinrich et al.²⁴). We showed that we are in the same order of regrowth although we

detect a plateauing in regrowth rates that differ from Poorter et al. The revised texts (lines 267-274) reads as such:

“ However, the slowdown in recovery rates of AGB after 10 years of regeneration (Extended Data Tables 9-11) may indicate that several drivers are impacting forest growth and are not captured by Poorter et al. due to the limited representativeness of the field inventories data across climatic, environmental and anthropogenic gradients. Despite a complete time series, we found that land use intensity through repeated deforestation events and fire occurrences prior to forest regrowth may have negative effects on regrowth after 5-10 years (Supplementary Fig.6), which was corroborated by previous studies^{89–91} where fire legacies could decrease regrowth rates by 20 to 75%, particularly in drier and water-deficient regions. ”

26) Line 176-179: This sentence is difficult to understand, would you please rewrite to clarify, especially the first part of it. But the whole paragraph is missing that conceptual model mentioned earlier.

Thank you, we have deleted this sentence accordingly and have reported results in RH98 and AGB directly in the text along with RH50 (main metric). This issue was also reported by reviewer 1.

27) Line 180: Fig2c. Can you statistically assert which time since regrowth periods are statistically different (i.e., what segments of the slope of those lines are different from 0)? For example, in Asia is any of the change significant after 10 years?

Thanks for this comment. We performed a statistical analysis and we grouped our dataset for regrowth and forest edge by continent and by decade (i.e. three groups, each with 10 years) and we pulled out the p-values of the slope from simple linear models (see extended data table 9-11). In Table 4 to 9 we reported the statistics (for Regrowth and Forest Edge, and RH50, RH98 and AGB). We selected 10 years to have more robust statistics (an alternative could have been a “breakpoint” algorithm, but it was suboptimal for reporting results in tables). As suggested by the reviewer, over Asia for forest regrowth, the p-value of the slope is well below 0.001. Also, for forest edge the p-value of the slope is not statistically significant (i.e. always above 0.2).

Table 4. Significance of the slope of the regeneration of forest regrowth (RH50) by continent and for timespans of 10 years. The stars are intended to flag levels of significance for 3 of the most commonly used levels. If a p-value is less than 0.05, it is flagged with one star (*). If a p-value is less than 0.01, it is flagged with 2 stars (**). If a p-value is less than 0.001, it is flagged with three stars (***). The last column refers to the corresponding percentage of intact forest.

Continent	TimeSinceDisturbance	p.value	signif	PercentageIntactForest
Africa	0-10	0.0002731	***	36 %
Africa	10-20	0.8339832		48 %
Americas	0-10	0.0000000	***	38 %
Americas	10-20	0.2603213		50 %
Americas	20-30	0.0281074	*	54 %
Asia	0-10	0.0000006	***	34 %
Asia	10-20	0.0003674	***	44 %
Asia	20-30	0.2563087		50 %

Table 5. Significance of the slope of the regeneration of forest regrowth (RH98) by continent and for timespans of 10 years. The stars are intended to flag levels of significance for 3 of the

most commonly used levels. If a p-value is less than 0.05, it is flagged with one star (*). If a p-value is less than 0.01, it is flagged with 2 stars (**). If a p-value is less than 0.001, it is flagged with three stars (***). The last column refers to the corresponding percentage of intact forest.

Continent	TimeSinceDisturbance	p.value	signif	PercentageIntactForest
Africa	0-10	0.0001274	***	52 %
Africa	10-20	0.4300495		59 %
Americas	0-10	0.0000003	***	54 %
Americas	10-20	0.0252918	*	60 %
Americas	20-30	0.0396967	*	62 %
Asia	0-10	0.0000012	***	46 %
Asia	10-20	0.0000500	***	54 %
Asia	20-30	0.1801195		58 %

Table 6. Significance of the slope of the regeneration of forest regrowth (AGB) by continent and for timespans of 10 years. The stars are intended to flag levels of significance for 3 of the most commonly used levels. If a p-value is less than 0.05, it is flagged with one star (*). If a p-value is less than 0.01, it is flagged with 2 stars (**). If a p-value is less than 0.001, it is flagged with three stars (***). The last column refers to the corresponding percentage of intact forest.

Continent	TimeSinceDisturbance	p.value	signif	PercentageIntactForest
Africa	0-10	0.0003976	***	34 %
Africa	10-20	0.3263024		42 %
Americas	0-10	0.0000003	***	31 %
Americas	10-20	0.0579074	.	38 %
Americas	20-30	0.0547736	.	41 %
Asia	0-10	0.0000062	***	28 %
Asia	10-20	0.0009677	***	36 %
Asia	20-30	0.4697922		41 %

Table 7 Significance of the slope of undisturbed edge forest recovery following edge creation (RH50) by continent and for timespans of 10 years. The stars are intended to flag levels of significance for 3 of the most commonly used levels. If a p-value is less than 0.05, it is flagged with one star (*). If a p-value is less than 0.01, it is flagged with 2 stars (**). If a p-value is less than 0.001, it is flagged with three stars (***). The last column refers to the corresponding percentage of intact forest.

Continent	TimeSinceDisturbance	p.value	signif	PercentageIntactForest
-----------	----------------------	---------	--------	------------------------

Africa	0-10	0.4254926		50 %
Africa	10-20	0.9931230		55 %
Africa	20-30	0.2268761		42 %
Americas	0-10	0.5540300		50 %
Americas	10-20	0.5380417		62 %
Americas	20-30	0.6836947		60 %
Asia	0-10	0.5588607		43 %
Asia	10-20	0.9621818		53 %
Asia	20-30	0.2970753		49 %

Table 8 Significance of the slope of undisturbed edge forest recovery following edge creation (RH98) by continent and for timespans of 10 years. The stars are intended to flag levels of significance for 3 of the most commonly used levels. If a p-value is less than 0.05, it is flagged with one star (*). If a p-value is less than 0.01, it is flagged with 2 stars (**). If a p-value is less than 0.001, it is flagged with three stars (***). The last column refers to the corresponding percentage of intact forest.

Continent	TimeSinceDisturbance	p.value	signif	PercentageIntactForest
Africa	0-10	0.8674811		69 %
Africa	10-20	0.8617110		68 %

Africa	20-30	0.2611471		59 %
Americas	0-10	0.6898782		69 %
Americas	10-20	0.5816447		73 %
Americas	20-30	0.6979281		71 %
Asia	0-10	0.8498510		57 %
Asia	10-20	0.9126037		63 %
Asia	20-30	0.2725013		58 %

Table 9 Significance of the slope of undisturbed edge forest recovery following edge creation (AGB) by continent and for timespans of 10 years. The stars are intended to flag levels of significance for 3 of the most commonly used levels. If a p-value is less than 0.05, it is flagged with one star (*). If a p-value is less than 0.01, it is flagged with 2 stars (**). If a p-value is less than 0.001, it is flagged with three stars (***). The last column refers to the corresponding percentage of intact forest.

Continent	TimeSinceDisturbance	p.value	signif	PercentageIntactForest
Africa	0-10	0.6648768		53 %
Africa	10-20	0.9134773		54 %
Africa	20-30	0.2517759		42 %

Americas	0-10	0.9405949		49 %
Americas	10-20	0.5724160		57 %
Americas	20-30	0.7456494		55 %
Asia	0-10	0.8511684		41 %
Asia	10-20	0.9719744		47 %
Asia	20-30	0.3288108		43 %

Note that we have decided to remove the section on modelling forest regrowth and extrapolation as the 30-year period of our study (20 year period for Africa) is too short to extrapolate regrowth rates and assess the time it would take to fully recover old-growth forest structure. We preferred in this revised version of the manuscript to focus on better interpreting our results using the three metrics of forest structure within the timespan of our observations.

28) Line 196: Do you mean 120 m here?

Thank you, we meant 120m and corrected this mistake. Note that we added a paragraph in the Method section on the rationale behind this distance. See point #51 from reviewer 1.

29) Line 210-212: I cannot follow the logic of this statement.

In the revised manuscript, this sentence has been deleted and final text reads as such (lines 174-180):

“ To a lesser extent, smaller canopy openings following selective logging and fire could also be a source of important and yet unaccounted edge effects^{27,68}. Here we quantified that relative heights (RH50) in the undisturbed forests near logged or burned forests (i.e. within a 120 m radius) were on average lower than in intact forests by 22%

(20-25%) and 32% (25-38%), respectively (Extended Data Fig.11, 15% and 22% lower on average for RH98). These results are not only due to localised edge effects from degradation but also to the omission in the Landsat-based forest cover change datasets of small-scale and low-intensity disturbances at the interface between ‘undisturbed’ and degraded forests^{1,69}. ”.

30) Line 227&following: Where is the map of IFL given. You say you have an “approach to mapping IFL” but I could not find the result of that map. Since you are also comparing to Potapov, why would you not include a comparison of maps in the Supplement? That would be exceptionally interesting if you are asserting that GEDI data provide a very different view of what it means to be “intact”.

Thank you, we have now included a comparison of maps in extended data figures and performed additional computations to allow a better comparison with Potapov dataset (forest patch area needs to be greater than 500 km², according to Potapov definition of IFL). We have in this sense added a dedicated section to the Method (see sub-section “Intact Forest Landscape assessment and comparison with Potapov”). Please refer to point #49 from Reviewer 1 to visualize the different maps (which can be also found in Extended data figures 12) and other sensitivity analyses. We have updated the main text and it reads as such (196-199):

” It resulted in comparable extents to Potapov et al.³⁷, assessment in 2020 (502 Mha vs 535 Mha with this approach). However, by including additional criteria on the minimal patch size of 500 km² (37see Methods for more details), IFL was further reduced by 23%, and it resulted in smaller extents compared to Potapov’s (-14%, 427 Mha with this approach). ”

31) Line 243: Fig 3 (Africa). Something doesn’t seem right here. How can the recovery of “All Forests” (undisturbed and disturbed) exceed undisturbed after 1000 m?. Can you explain? Also, there is this 210 m reference again. Where is that coming from? Do you mean 120 m? How did you pick, 1000 , 1500 and 2000 m? What quantitative rules got you to those numbers?

Thank you for this comment. In the updated figure (see below), we can see that undisturbed forests heights measured at various distances to the edge (height mainly driven by edge-related desiccation) always exhibit on average a higher relative height compared to all forest (mainly driven by edge-related desiccation and logging/fire impacts) except for Africa where beyond 1500 - 2040m, no clear differences are visible between the two types of forests with clear overlapping standard deviations. This may be explained by the fact that the majority of degradation due to selective logging and fire (minor impacts in this region) are located at short distances to the forest edge.

Regarding the 210m reference, we actually meant 120m and corrected it accordingly. To explain the rationale behind this distance, we added a paragraph in the Method section (lines 476-480):

“ To mitigate the effects of canopy disturbance contagiousness between degraded and undisturbed forests, we eliminated areas of transition using a buffer of 120 m around degraded forests. This distance of 120 m corresponds to the area initially affected by the felling of individual trees in selective logging operations^{106,107} where localised

edge effects (i.e. increased tree mortality and altered forest structure) are the highest²⁷.”

Regarding how we picked the scale of the edge effect, we added information in the main and reads like this (lines 159-162):

” Undisturbed forests showed a decrease in RH50 from the edge up to 350, 600, and 1700m into forest interiors in America, Africa, and Asia, respectively (solid curve in Fig.2, defined as the distance between the forest edge and the point at which 95% of intact forest RH50 is reached).”

A similar evaluation has been made by Chaplin-Kramer et al.⁹. Note that we also added some explanation in the Method on the rationale behind the distances to forest edge and how we decided to split them into various widths (lines 471-475):

“ These distances were selected on the basis of previous studies reporting on the scale at which edge effects operate and affect microclimate (up to 400m⁶²), canopy-moisture levels (up to 2.7 km²⁸), phenology (up to 5-10km¹⁰⁵) and forest biomass (up to 1.5 km⁶³). The first six intervals of distances are centered on the most recent and accurate evaluation of the scale of edge effect (~100-200m^{20,47,64,67}).”

32) Line 283: Same question with statistical test for non-random data, especially if for some areas most of the GEDI footprints come from a single line.

Thanks for this comment. The issue of the tracks/orbits has already been addressed in point#35 reviewer#1. Also, by extending the GEDI time series to 2021 and 2022 (and removing the 10-degree threshold) we can exclude that many GEDI shots are coming from a single line. Fig. A31 and A32 show the maps of GEDI orbits per cell for intact and degraded forest. Further, the most important thing to say is that our spatial scale (1.5 degrees, ~ 167 km) is mitigating this bias. Since the swath width of GEDI's eight beams is 4.8 km, literature¹⁷ reported that 6 km grid cells would adequately mitigate varying sample intensity caused both by orbital resonance and broader latitude- and cloud-based factors while minimizing the number of tiles with fewer than two tracks.

Fig.A31. Map of GEDI orbits per cell for intact forest.

Fig. A32. Map of GEDI orbits per cell for degraded forest.

33) Line 315: provide a reference for GEDI since one did not appear earlier.

We added a reference for GEDI.

34) Line 447: GEDI L2A and L2B have citations on the DAAC so you do not need to have web reference (i.e. they have a DOI as with any other publication).

We agree with the reviewer. We removed the web references and added the references to the doi.

35) Line 456: Define "HOME"

Following the recommendations, in the updated version of the manuscript we now specified the definition of the acronym HOME, which is the Height of Median Energy. HOME is the same as the RH50, so the definition is already provided in the sentence before the text "(also known as Height of Median Energy or HOME)" in line 522.

36) Line 464: You should reference the paper some of the co-authors just released on European protected areas as well.

We added the reference to the Ceccherini and Girardello paper on European Protected Areas with GEDI. See point#11 above.

37) Line 465&: It's not clear from your writing if you processed waveforms, simply used metrics from GEE, or just used filtering criterion on L2A and L2B data. Please be more precise here.

The reviewer's comment is well-taken. We have now clarified the type of data, available, on GEE that were used and how these were filtered.

Action taken:

-> We have now better specified the type of data which were used in “Cloud-computing platform: Google Earth Engine” section in Methods (lines 666-671):

“ The GEE data catalogue contains processed Level-2A and Level-4A GEDI data products, i.e. the rasterized versions of the original GEDI products, with each GEDI shot footprint represented by a 25 m pixel¹²⁶. The Level-2A product contains information derived from waveforms, including metrics such as energy quantile heights and relative height. The Level-4A product provides predictions of aboveground biomass density, along with estimates of the prediction standard error within each geolocated laser footprint. ”

As described in the methods section (see GEDI dataset sub-section), we filtered the data based on several quality criteria.

38) Line 476: When you filtered at 10 deg you did two things that you do not address. (1) you biased your sample against forests on slopes (which may have different degradation regimes) and (2) you added noise because GEDI geolocation is only 10 m at one standard deviation. This means that many shots have geolocation error more than 20 m. Slopes change very rapidly in space and since the shots cannot be precisely matched to SRTM you included some shots that should have been filtered and you removed some shots that should have remained. This is not acknowledged nor discussed.

We do thank the reviewer for this specific comment. In fact, we were not aware of this bias introduced with a 10 deg threshold. We used a 20 deg threshold instead. See point #9 above.

Regarding the geolocation error for the slope, the reviewer is right. However, we filtered GEDI shots over an area that is larger than the GEDI footprint. In other words, we took a larger area than the GEDI footprint and we declared the degradation/forest (i.e. the JRC-TMF) class based on that. This is acknowledged in the text (Dataset combination in the Method section).

39) Line 471: Topography only results in an overestimation of height as a bias if its unvegetated. On vegetated surfaces it can make the heights taller or smaller.

See point #9 above. The text has been changed accordingly.

40) Line 483-485: This is not a true statement as given. I think you could probably just delete that first sentence and add a qualifier to the next one.

We agree with the reviewer and have deleted the sentence accordingly.

41) Line 502: 300 seems quite small for such a large area. Again, how many tracks are used? This also shows why using multiple years of data, properly filtered would perhaps help.

Reviewer's comment is well-taken. In fact, we used multiple years of data (see point #10 reviewer#1).

Action taken:

-> We used multiple years of data

-> We have provided the reviewer with a figure of the number of GEDI orbits per gridcell, thus indicating the spatial density of GEDI samples (see point#32 above).

42) Line 505: That seems like very, very few shots, especially if they all came from one line.

Reviewer's comment is right. Now we changed this threshold and the new methods and text are shown in point#7 above.

43) Same thing with the 13 minimum in next sentence. Again, in this paragraph you clump together too many figures and data sets. Guide the reader here.

Reviewer's comment is right. Now we changed this threshold and the new methods and text are shown in point#7 above.

Regarding this paragraph, formerly grouping different aspects under the title "Robustness of forest structure statistics", we have now subdivided it into GEDI minimum thresholds and statistical tests.

44) Line 512: Why do you assert the intragrid variability is not large? It looks like its 10 to 15 m from the bootstrap. What were the SDs of the bootstrap? Also, it's unclear what you are doing. Suppose a grid cell has 1000 shots you are still doing a 300 bootstrap? What if it has 300 shots? The first sentence does make much sense to me. Many factors will make canopy heights variable. If you have non-random sample or too few samples, you may either get variability that is either too big (not enough samples) or too small (non-random sampling). So, what exactly are you doing here and why? Why is high variability if its real a bad thing?

While we acknowledge that there is a certain range of variability in the results from the random sampling procedure, we have now clarified how the procedure works and the advantages of random sampling over using the "full" GEDI sample.

Note that we have replaced the term "bootstrap" with "Random sampling procedure".

We have added the following text to the methods section:

Action taken:

In order to reduce sampling bias in the structural variable dataset, we randomly resampled GEDI observations 500 times within each 1.5 degrees x 1.5 degrees gridcell. We then summarised the random samples by calculating the mean and standard deviation of each structural variable, for each grid cell. The use of random sub-samples has several advantages over the full GEDI sample. These include:

- i) ensuring equal representation of observations across all grid cells. The density of GEDI observations varies across different regions or areas because of differences in orbital coverage by the International Space Station. This non-uniform distribution of data points can introduce a bias in the estimation of summary statistics within a given grid cell.
- ii) removing redundancy. Data thinning achieves a more equal representation of information on forest structural parameters across the spatial domain of interest, within each grid cell.

Working with a reduced dataset that covers different grid cells proportionately can help mitigate the aforementioned bias arising from uneven data density and redundant observations.

Following the reviewer's suggestions, we also modified the random sampling procedure in such a way only cells with a minimum number of observations are used. Each grid should contain at least double the number of points (600) compared to the size of the random sample (300). By setting a fixed threshold for the number of observations, we ensured that only grid

cells with a large number of observations are sampled. While this approach led to the exclusion of a small proportion of 1.5 deg x 1.5 deg cells, the geographical distribution of the remaining cells remains highly representative and does not affect the results. Additionally, we conducted a sensitivity analysis using random samples of different sizes (i.e. 300 and 450), and the results are presented in Figure A33.

Fig. A33 Results of random sampling with 300 (top) and 450 (bottom) samples per gridcell.

We have also reworded the first sentence following the reviewer's suggestion explaining the drawbacks of using the "full" GEDI sample.

A comparison of canopy heights for intact, degraded, edge, and regrowing tropical moist forests at the 1.5 deg grid cell level may be impacted by the high density of GEDI samples. This high density of observations can lead to an excess of variability within certain grid cells. While a larger set of observations may capture structural different local environmental and

anthropogenic factors (e.g., soil and forest types), it is important to ensure that comparisons among cells are not affected by bias, such as density of observations and redundancy.

45) Line 520: Once again, can you do a t-test with non-random data?

See point#20 above.

46) Line 857: Those figures are highly ineffective. Why are they needed?

We do agree with the reviewer that these figures are not explained enough in the manuscript. For this reason, we have decided to place them in supplementary instead of in the extended data figures.

47) Line 863: EDT 1. Is there an equivalent table for RH100? What does the shading mean?

In line of principle, with Google Earth Engine we can extract all the RHs of GEDI (and RH100 too). However, we decided to show in the manuscript only results in terms of RH50 (in the main) and RH98 (in the other sections). Generally, among RH metrics RH98 and RH50 were most frequently selected¹⁶.

Also, one of the first GEDI papers by Potapov⁸ stated that RH100 is overestimating canopy height. In any case, adding RH100 might create confusion in the reader.

We removed the shades in blue and in red to avoid confusion. The blue and red were negative and positive recovery rates.

Referee #3

This is a valuable paper that characterises the spatial variation in height of tropical moist forest (TMF) as a function of distance to disturbance – the key learning being that TMF are shorter much further (up to 2000m) from observable disturbance than previously thought. This result changes our understanding of TMF states and has broad implications.

We thank the reviewer for the appreciation of our work.

The authors use new satellite lidar data (GEDI) from a single year alongside a three decade map of forest disturbance histories to build a space-for-time analysis of height versus time since disturbance. The analysis is used to estimate forest height regrowth post-disturbance. Some of the results found are to be expected – forests that have been disturbed are found to be shorter than undisturbed forests in the same region. The study here confirms that selective logging has less impact on height than does burning.

We appreciate the positive comments of the reviewer. We have also done the following improvements in this version of the manuscript (following the request of all reviewers), which further strengthens our conclusions.

- We have significantly revised the analysis using 4 years of GEDI observations (instead of 2) to corroborate our findings. This was possible because in April 2023 the updated version of the JRC-TMF dataset was released, thus allowing us to expand our analysis up to the end of 2022.
- Further, we have included aboveground biomass (AGB) derived from GEDI along with the initial canopy height metrics.
- We have updated the figures to address the reviewer's concerns and inform the readers on the post-disturbance regrowth.

These new findings clearly show that, despite the high spatial variability (i.e. vertical bars in Fig.3, for further details see point#2 below), selective logging has less impact on height than burning. And the same can be said for single versus multiple degradations (Fig.3).

However, note that these results were expected but never quantified (to the best of the authors' knowledge) using these instruments nor at that scale in a consistent way. Studies on the

impacts of degradation and long-term recovery of tropical forest structure remain scarce especially when disturbance type spans beyond timber extraction done within a forest management framework (i.e. Illegal logging, fire and fire following logging).

1) Some other results are more controversial and unexpected. The authors state that “In the 30 years of the analysis, the complete recovery of the canopy height cannot be achieved for most types of forest disturbances and regions, documenting indisputably the persistent marks of cumulative impacts from logging, fires and edge effects on the structure of global TMFs”. This global conclusion seems to be counter to other records generated through much higher-resolution airborne lidar at specific TMF locations. It is true that forests in Borneo show slow canopy recovery (Pfeifer et al. 2015 Riutta et al., 2018). But these forests have very tall canopies reaching up to 70 m, which are not captured in this analysis by GEDI. Logging disturbance in Borneo reduces forest height by 30-40 m, so with typical height regrowth of ~0.8 m per annum, multi-decadal recovery in height is expected. But there is rapid recovery in vertical forest structure in La Selva, Costa Rica (Tang et al., 2012). Here height profiles twenty years after clear-felling are similar to those found in undisturbed forests, which are shorter stature than those in Borneo. Chave et al (2020) report lower regrowth rates in French Guyana than Costa Rica and link this to poor soils on the Guiana shield. What we conclude from these studies is that regrowth rate varies significantly across TMF.

Thanks for this comment: the reviewer rightly points out how the recovery of some forests varies, and may take longer than the 30 years considered in this study. Therefore we agree that considering the impacts as "persistent marks.. on the structure of global TMFs" is a bit of a stretch. We invite the reviewer to look at point #1 from reviewer 1 who mentions a similar issue and see how we addressed it. We modified the conclusion accordingly, the revised text reads as such (lines 334-354):

“ Our study demonstrates that the integration of recent and spatially sparse observations from spaceborne LiDAR (this corresponds to over 39 million individual sample plots) with long-term and spatially continuous spaceborne optical datasets (i.e. JRC-TMF) provides a novel approach to assess the forest structural attributes and enables the large-scale monitoring of forest degradation and recovery. We unveil the spatial extent and quantification of the impacts of degradation on the canopy structure

of tropical moist forests (TMFs), varying from a 40% to a 90% decrease in canopy height and aboveground biomass and extending up to ~2 km inside the forest interiors. The forest edge effect was previously considered de facto of ~ 100 m²⁴. Our results show a far-reaching effect of forest edge implying that the overall impact of fragmentation across the pantropical belt is severely overlooked by a factor of over 200%, with relevant potential consequences on the forest carbon dynamics and other ecosystem services. The 30-year period of our study is too short to observe a full recovery of the forest structure for most types of forest disturbances and regions. We show that cumulative impacts from selective logging, fires and edge effects have significant long-term effects on the structure of global TMFs. Although the present areas of fast-regrowing forests allow counterbalancing ~25% of carbon loss from deforestation¹¹, we found here that the full recovery of forest structure after deforestation or degradation would take a centennial timescale and may be slowed down by anthropogenic factors. Moreover, our study demonstrates for the first time that logged and burned forests are highly vulnerable to further degradation and deforestation. This type of precise, spatially detailed, recent and transparent information on tropical forest degradation is key for implementing more effective forest-based mitigation policies⁹⁸ and conservation activities that have been agreed under the UN Conventions on Climate Change and Biodiversity^{99,100} (UNFCCC COP and CBD). ”

Following this recommendation, we indicated that the recovery was, on average, not complete within the 30-year study period and that there is some local variability documented by local studies (based on field inventories or high-resolution data such as airborne LiDAR).

We also wanted to thank the reviewer for pointing to these references. They reflect low-intensity logging that may not be well captured by Landsat (due to Landsat's spatiotemporal limitations). Nonetheless, we agree with the reviewer that logging has the lowest impact over time among the other disturbances analysed. So in that sense, our results are not controversial.

Note that we present the average canopy height of intact forest within a 1.5-degree grid but the GEDI data behind this does capture tall canopies reaching more than 70m in height, you can see below the distribution of RH98 from GEDI for intact forest for each continent (Fig. A34):

Fig.A34 Histogram of all GEDI RH98 values per continent. The dotted vertical line is at 70 m.

Action taken:

-> In the updated version of the manuscript we have integrated the references of Pfeifer et al. 2015, Riutta et al. 2018, Tang et al. 2012 and Chave et al. 2020 in the discussion of our results on the recovery rates of recovering forests from degradation and deforestation.

-> We have throughout the manuscript made comparisons with ALS data and other relevant studies to corroborate our results on the edge effects and on immediate and long-term changes in forest structure after forest disturbances.

-> We have put more emphasis on the variability in the reported results and what may drive this variability (e.g. climate, environmental and anthropogenic gradients).

2) The authors raise some interesting theories to explain their observation of slow recovery across all TMF – suggesting that logged and burned forests are highly vulnerable to further degradation and deforestation. This may be true, but the global consistency is unexpected. Land use, land use change, fire resilience, climate, biodiversity and soils all vary spatially at sub-continental scale. The lack of nuance in the results here raises concerns that there may be intrinsic biases in the data and the analysis. These should be resolved before proceeding.

We agree with the reviewer on the need to give more relevance to the variability in the results. Following this comment, we added an in-depth analysis to indicate and describe that there is indeed some variability in the recovery curves depending on the environmental conditions and land use history.

Regarding edge effects variability, we improved this already existing sentence (lines 169-173):
“ The variability in the magnitude of change and scale of edge effect within each continent is likely to be linked to forest structure composition and environmental conditions^{47,63,65} but also to different landscape configurations^{28,66} (e.g, forest patch size and connectivity) that can increase forest sensitivity to edge effects⁶⁷. ”

Regarding the persistence of edge effects over time, we improved this already existing sentence (lines 229-231):
“ This pattern is likely due to the long-term persistence of the edge effects driven by changes in the growing conditions and the exposure to additional anthropogenic disturbances^{20,47,74,75}. ”

Regarding the recovery of logged forests, we added this sentence (lines 246-247):
“ Continental variability of average annual rates of recovery is likely influenced by logging intensity⁷⁸, forest composition and climate conditions⁸². ”

Regarding the recovery of burned forests, we improved this already existing sentence (lines 255-256):
“ The high variability in recovery rates is likely due to different fire frequencies, intensity, climate and forest-type specific responses⁸⁷. ”

We added this sentence in the result section “fate of degraded forest” to highlight the complex driving forces in tropical deforestation (lines 296-301):

“ Degraded forests followed subsequently by deforestation were on average severely impacted by the degradation processes with reductions in mean RH50, RH98 and AGB by 60%, 45% and 65%, respectively, which corresponds to severe impacts from unsustainable logging and/or fire (Fig.3, Extended Data Fig.14-15). Moreover, these structural parameters have a large variability ($\pm 12.8\%$, $\pm 13.3\%$ and $\pm 14.6\%$ for RH50, RH98 and AGB respectively) that reflects the complexity of degradation processes and underlying factors driving deforestation in the tropics^{15,93}.”

Further, we tried our best to explore nuances in results, for example showing how the recovery rate of regrowth varies in different cases. Fig. S6 show for RH50 and RH98 the recovery in case of single/multiple deforestation events (panel a) or fire occurrence (panel b). These two factors have been identified by Heinrich et al.²⁴ as key anthropogenic drivers of forest regrowing rates.

Fig.S6 Long-term recovery of relative heights (RH50 and RH98) of secondary forests regrowing on abandoned deforested lands. Previous land use type and intensity are assessed on the basis of two proxies as previously identified by Heinrich et al.2: (a) Forest regrowing after single or multiple deforestation events (repeated cycles events of minimum 3 years duration) and (b) forest regrowing after forest fires or not (identified in the GFC loss3 from forest product from UMD GLAD 2001-2021 Tyukavina et al.4). Results are reported as the percentage of intact forest canopy height (solid line) after normalising the difference in canopy height within each grid cell between intact forest and each forest regrowth type and age. The solid lines depict the local polynomial regression (loess) fit to the data. Vertical bars indicate the spatial standard deviation.

The revised text on forest regrowth variability reads as such (lines 263-274):

“ The recovery for AGB was on average twice slower than RH50, reaching 40% (38%, 39% and 41% Americas, Africa, and Asia, respectively) of intact forest AGB after 20 years of regrowing, rates which are similar to those reported by Poorter et al.⁸⁸ (33%) using field inventories, or by Heinrich et al.¹¹ (36-49%) using remote sensing data. However, the slowdown in recovery rates of AGB after 10 years of regeneration (Extended Data Tables 9-11) may indicate that several drivers are impacting forest growth and are not captured by Poorter et al. due to the limited representativeness of the field inventories data across climatic, environmental and anthropogenic gradients. Despite a complete time series, we found that land use intensity through repeated deforestation events and fire occurrences prior to forest regrowth may have negative effects on regrowth after 5-10 years (Supplementary Fig.6), which was corroborated by previous studies^{89–91} where fire legacies could decrease regrowth rates by 20 to 75%, particularly in drier and water-deficient regions. ”

3) A challenge for the rigour of the results may be linked to the difficult concept of ‘canopy height’, a scale dependent property which will vary according to the resolution of the laser. GEDI has a footprint of 25 m, whereas airborne lidar resolve down to 0.1 m, up to two orders of magnitude difference. GEDI samples part of the landscape, not its entirety, whereas airborne lidar can give complete returns albeit for a smaller domain. Comparison of GEDI with airborne lidar in Amazonia reveals an RMSE of 4.4 m (Milenković et al, 2022). In this published study the airborne data were used to correct inherent bias in GEDI, and then to estimate forest regrowth rates. Milenković et al note that without airborne lidar for calibration, GEDI estimates of regrowth rate will be under-estimates.

We could hypothesise that biases arise in remote measurement of height in undisturbed forests, due to rougher canopy surfaces linked to presence of canopy emergent trees and gap dynamics. This heterogeneity might have length scales <25m resolution of GEDI. It would be helpful to reflect on what a GEDI measure of height represents and how can it be related to ecological information. Does GEDI have different biases depending on forest structure, including the size, number and arrangement of stems and canopies? A detailed comparison of the GEDI approach used here against the same analysis using airborne lidar and in situ estimates of disturbance history would provide enhanced credibility for global GEDI outputs.

This comment is interesting and relevant but it is also very complex and cannot be fully addressed within this study as it calls for a calibration of GEDI estimates at global scale, which is out of the scope of this study (even though it is certainly important to enhance the credibility of the GEDI results). To circumvent this, we have for each result section and each spatio-temporal pattern found, made clear links with ALS results. We showed in the revised text that our results corroborate with ALS studies over multiple types of forests and along complex gradients of degradation. For instance, similar changes in aboveground biomass (~20-30% loss) at the forest edge (width of 120m) were measured by Silva Junior et al.¹³. Asner et al.²⁵ reported similar forest structure (height and AGB density) for old-growth intact forest than us in Borneo. Finally, for burned forests that are structurally very different from intact forests, we found a similar percentage of canopy heights and estimated AGB to Rappaport et al.²⁶.

The presence of biases in the GEDI recovery rates is most relevant here and the results of the cited paper has been considered in the Method section, but also keeping in mind that the Milenkovic study refers to the Amazon and it may not apply equally to other regions. The revised text reads as such (lines 531-534):

“However, a recent study²⁷ showed the importance of calibrating canopy heights from GEDI with Airborne Laser Scanning (ALS) to prevent an underestimation bias. Given the limited availability of ALS data at tropical scale, it is not possible, at the moment, to calibrate the GEDI archive”.

In addition, the fact that GEDI and ALS see different canopies (which may have some effect also on the changes in their height) because of their different resolutions and characteristics is to be acknowledged when comparing the results. In other words, tree height measurements from ALS and GEDI are not always directly comparable. Differences can be expected comparing ALS height to GEDI RHs where height is based on energy accumulation while for ALS, height is based on the elevation of ALS echoes not classified as ground²⁸.

In the new version of the manuscript, we added a discussion on local studies comparing ALS with GEDI. Please note that there are few ALS dataset available, mainly on Gabon (Fig. A35) and Democratic Republic of Congo (DRC)²⁹.

Figure A.2: Evaluation w.r.t. canopy top height (RH98) derived from independent LVIS airborne LIDAR data (Blair, 2018). a) Locations of LVIS LIDAR campaigns. a-c) Confusion plots showing the relationship between LVIS reference data and predictions from Sentinel-2 for a) All available LVIS areas, b) Only regions within the GEDI range, and c) Only regions north of GEDI.

Fig. A35. From ³⁰ <https://lvis.gsfc.nasa.gov/Home/index.html> The map shows the LVIS LiDAR data that are available for calibration. In our case, there are only few points over Gabon.

Regarding the ecological meaning of what GEDI relative heights capture, we have (also requested by reviewers 1 and 2) added a dedicated paragraph at the end of the introduction and a more lengthy one in the method section along with the extraction of GEDI relative heights over the main classes of interest in this study (i.e. intact forest, edge, degraded and regrowth). We have also created a new plot in Extended Data Fig. 1 showing Cumulative Return Energy from GEDI L2A for different forest types and a transect of GEDI relative heights across deforested land, edge forest and intact forest.

4) Another challenge for the study lies with potential errors in the mapping of forest disturbance. It is not clear if the degradation mapping has been validated independently, and

therefore what its potential biases are. I found extended data figure 1 confusing – the colours used are difficult to match between legend and map, and the shading is difficult to interpret.

We thank the reviewer for pointing out this issue. The degradation mapping in the JRC-TMF dataset has been validated in Vancutsem et al. 2021. We added a few sentences related to this in the Method section (lines 401-406):

” The accuracy of the disturbance mapping of every single Landsat image is 91.4%, with omission and commission errors for non-forest cover detection of 9.4% and 7.9%, respectively. The accuracy matrix for the transition map shows an overall accuracy of 92.8% for the classes of the moist forest domain. The omission and commission errors for the forest change areas are 19% and 8.4%, respectively. Details on the accuracy assessment at the continental level are presented in the supplementary material of Vancutsem et al. 2021. ”

The validation of the combination of GEDI and JRC-TMF products requires an independent dataset which goes beyond the scope of this paper. We tried our best to circumvent the potential biases linked to commission and omission errors in the degradation mapping. Regarding commission errors, we applied a morphological (circular shape) filter of 35 m to the degradation class before intersecting with GEDI footprints. This method which originally was applied to tackle GEDI geolocation errors (see in Dataset combination section in Method) resulted in the removal of single/small-patches of degraded forest pixels which may likely be commission errors. Regarding omission errors i.e. non detected disturbed forest areas which are still classified as undisturbed forest, we applied a 120m buffer around the mapped degraded forest to mask out GEDI footprints within this area when analyzing edge effects (Fig 1 and 2) which is where omissions are most likely located (Vancutsem). We performed an analysis looking at forest canopy heights (using RH50 and RH98 from GEDI footprints) within undisturbed forest at 120m from mapped degraded forests (logged or burned forests). We could see that canopy heights show in average marks of degradation which may be linked to either localized edge effects but also to omissions in the detection of forest disturbances (see the result section on the Cumulative impacts from edge effects in undisturbed and degraded forests for more information).

Regarding the extended data Fig 1, we agree with the reviewer and we deleted this figure. We replaced it with two figures showing GEDI relative height metrics sampled along a transect crossing deforested area, edge and intact forest and within the main classes of interest in this study i.e. intact forest, edge forest (120m width), degraded forests and forest regrowth at various ages or time since disturbances. These two additional figures were requested by

reviewer #2 and bring much more information on the focus of this study compared to previous extended data Fig 1.

5) Most of the figures present height changes as a percentage relative to an intact forest baseline – this allows comparison, but limits sense checking against direct field data of e.g. forest height growth rates.

We agree with the reviewer that evidence of actual canopy heights is needed. However, in doing so there is a delicate yet important trade-off between the need to show actual canopy height values (easy to grasp) and the fact that heights (and their difference thereof) might be determined by environmental properties. In other words, canopy heights might be determined by environmental conditions, and it is possible that in areas with different elevation, fertility, temperature and precipitation regimes, the heights - and their differences - are different. For this reason, we prefer to keep the figures with normalized values - that are circumventing this issue - in the revised version of the manuscript and show Figure 2 with actual values only in the Supplementary Information section.

Action taken:

-> We have inserted in Extended data section multifacet plot with actual canopy height values (see Extended data fig.16) and not with heights as a percentage of intact forest height.

Forest regrowth (RH50)

Extended Data Fig.16 Long-term recovery of RH98 and RH50 of secondary forests regrowing on abandoned deforested lands compared to intact forest. Vertical bars indicate the spatial standard deviation.

6) In conclusion, the most important conclusion of this paper is that degradation extends far from observed disturbance in TMF, up to 2 km. The challenge for the authors is that this conclusion relies on a space for time substitution approach that generates a global consistent result at odds with the nuanced learnings from airborne lidar in regional studies. To have confidence in the GEDI analyses it is be important to link these to airborne lidar and confirm results are consistent.

While plots in Fig.2 (now Fig.3 in the updated version of the manuscript) fully rely on a space-for-time substitution, the analysis of the extent of edge effects where degradation extends far from the observed disturbance in TMF, up to 2 km was obtained through a spatial approach where we measured forest structures within different distances to the forest edge. The temporal component is integrated into this result as the sampling of GEDI footprints was limited to undisturbed forest area (i.e. where no disturbance has been detected using Landsat time series) but not through a time-for-space substitution approach per se. Besides, we also report the substantial spatial variability (as shown by the standard deviation, and vertical bars in Fig2) which likely include the local/regional studies based on airborne LiDAR. Following the recommendations, in the updated version of the manuscript, we have better introduced the most relevant and recent studies on edge effects using ALS, and have better linked our results on the persistence and extent of edge effects with ALS-based studies^{13,15,31} which shows our results are aligned with these studies.

7) P 329 – what is 2o?

The reviewer is right, we meant 2 degrees of spatial resolution. The text has been rephrased.

Further, we performed a sensitivity analysis of the grid size (see point #35 reviewer#1) and now we considered gridcell of 1.5 degrees (~ 167 km at the Equator).

8) P 475 – is the 10o slope threshold robust and valid?

Thanks for this comment. As suggested by reviewer #2 (and corroborated by recent literature), we modified this threshold (now it is 20 degrees) backed with a sensitivity analysis: See point #9 reviewer#2.

9) Ext Data fig 7 – is there any ground check on these rates of canopy height during regrowth?

With the integration of AGB in our study, we have now been able to compare our results of regrowth rates with the most recent and relevant studies. See point #24 and #25 from

Reviewer 2, as well as the updated text in the result section “Forest recovery dynamics after degradation and deforestation” on forest regrowth.

References

1. Bullock, E. L., Woodcock, C. E., Souza Jr., C. & Olofsson, P. Satellite-based estimates reveal widespread forest degradation in the Amazon. *Global Change Biology* **26**, 2956–2969 (2020).
2. Vancutsem, C. *et al.* Long-term (1990–2019) monitoring of forest cover changes in the humid tropics. *Sci. Adv.* **7**, eabe1603 (2021).
3. Langner, A. *et al.* Towards Operational Monitoring of Forest Canopy Disturbance in Evergreen Rain Forests: A Test Case in Continental Southeast Asia. *Remote Sensing* **10**, 544 (2018).
4. Souza, C. Mapping forest degradation in the Eastern Amazon from SPOT 4 through spectral mixture models. *Remote Sensing of Environment* **87**, 494–506 (2003).
5. Broadbent, E. *et al.* Forest fragmentation and edge effects from deforestation and selective logging in the Brazilian Amazon. *Biological Conservation* **141**, 1745–1757 (2008).
6. Beck, H. E. *et al.* Present and future Köppen-Geiger climate classification maps at 1-km resolution. *Sci Data* **5**, 180214 (2018).
7. Lima, T. A. *et al.* Comparing Sentinel-2 MSI and Landsat 8 OLI Imagery for Monitoring Selective Logging in the Brazilian Amazon. *Remote Sensing* **11**, 961 (2019).

8. Potapov, P. *et al.* The last frontiers of wilderness: Tracking loss of intact forest landscapes from 2000 to 2013. *Science Advances* **3**, e1600821 (2017).
9. Chaplin-Kramer, R. *et al.* Degradation in carbon stocks near tropical forest edges. *Nat Commun* **6**, 10158 (2015).
10. Gorelick, N. *et al.* Google Earth Engine: Planetary-scale geospatial analysis for everyone. *Remote Sensing of Environment* **202**, 18–27 (2017).
11. Briant, G., Gond, V. & Laurance, S. G. W. Habitat fragmentation and the desiccation of forest canopies: A case study from eastern Amazonia. *Biological Conservation* **143**, 2763–2769 (2010).
12. Ordway, E. M. & Asner, G. P. Carbon declines along tropical forest edges correspond to heterogeneous effects on canopy structure and function. *Proc Natl Acad Sci USA* **117**, 7863–7870 (2020).
13. Silva Junior, C. H. L. *et al.* Persistent collapse of biomass in Amazonian forest edges following deforestation leads to unaccounted carbon losses. *Sci. Adv.* **6**, eaaz8360 (2020).
14. Almeida, D. R. A. *et al.* Persistent effects of fragmentation on tropical rainforest canopy structure after 20 yr of isolation. *Ecol Appl* (2019) doi:10.1002/eap.1952.
15. Zhao, Z. *et al.* Fire enhances forest degradation within forest edge zones in Africa. *Nat. Geosci.* **14**, 479–483 (2021).
16. Duncanson, L. *et al.* Aboveground biomass density models for NASA's Global Ecosystem Dynamics Investigation (GEDI) lidar mission. *Remote Sensing of Environment* **270**, 112845 (2022).
17. Dubayah, R. *et al.* GEDI launches a new era of biomass inference from space. *Environ. Res. Lett.* **17**, 095001 (2022).
18. V.C. Oliveira, P., Zhang, X., Peterson, B. & Ometto, J. P. Using simulated GEDI waveforms to evaluate the effects of beam sensitivity and terrain slope on GEDI L2A relative height metrics over the Brazilian Amazon Forest. *Science of Remote Sensing* **7**, 100083 (2023).

19. Duncanson, L. *et al.* The effectiveness of global protected areas for climate change mitigation. *Nat Commun* **14**, 2908 (2023).
20. Poorter, L. *et al.* Multidimensional tropical forest recovery. *Science* **374**, 1370–1376 (2021).
21. Heinrich, V. H. A. *et al.* The carbon sink of secondary and degraded humid tropical forests. *Nature* **615**, 436–442 (2023).
22. Cook-Patton, S. C. *et al.* Mapping carbon accumulation potential from global natural forest regrowth. *Nature* **585**, 545–550 (2020).
23. Philipson, C. D. *et al.* Active restoration accelerates the carbon recovery of human-modified tropical forests. *Science* **369**, 838–841 (2020).
24. Heinrich, V. H. A. *et al.* Large carbon sink potential of secondary forests in the Brazilian Amazon to mitigate climate change. *Nat Commun* **12**, 1785 (2021).
25. Asner, G. P. *et al.* Mapped aboveground carbon stocks to advance forest conservation and recovery in Malaysian Borneo. *Biological Conservation* **217**, 289–310 (2018).
26. Rappaport, D. I. *et al.* Quantifying long-term changes in carbon stocks and forest structure from Amazon forest degradation. *Environmental Research Letters* (2018) doi:10.1088/1748-9326/aac331.
27. Milenković, M. *et al.* Assessing Amazon rainforest regrowth with GEDI and ICESat-2 data. *Science of Remote Sensing* **5**, 100051 (2022).
28. Pascual, A. *et al.* Assessing the performance of NASA's GEDI L4A footprint aboveground biomass density models using National Forest Inventory and airborne laser scanning data in Mediterranean forest ecosystems. *Forest Ecology and Management* **538**, 120975 (2023).
29. Xu, L. *et al.* Spatial Distribution of Carbon Stored in Forests of the Democratic Republic of Congo. *Scientific Reports* **7**, (2017).

30. Lang, N. *et al.* Global canopy height regression and uncertainty estimation from GEDI LIDAR waveforms with deep ensembles. *Remote Sensing of Environment* **268**, 112760 (2022).
31. Shapiro, A. C., Aguilar-Amuchastegui, N., Hostert, P. & Bastin, J.-F. Using fragmentation to assess degradation of forest edges in Democratic Republic of Congo. *Carbon Balance and Management* **11**, (2016).

Reviewer Reports on the First Revision:

Referees' comments:

Referee #1 (Remarks to the Author):

This manuscript presents an analysis of the impact of anthropogenic disturbance on tropical moist forests over about 30 years, using space for time substitution and recent GEDI lidar data as a proxy for height (& structure change). It's an interesting topic and there is some interesting result here, particularly on the persistence in space and time of disturbance (and particularly compounding effects).

While there is some good and interesting analysis here, there is a quite a lot of rather vague assertion, which is largely a function of the lack of clarity in definitions: which height metrics and what they mean very specifically for the ensuing analysis; why each metric is being used in each case; and then choice of crucial distances (edge size etc). In particular, there is a lack of clarity of what is meant by structure right from the outset - it is used in a vertical and horizontal sense at various times, as well as an areal measure which is then interpreted in terms of individual tree size distributions. There is also a difficulty in assessing the results because key figures are rather poorly presented. The use of the 200 km cells as a base spatial scale with then different underlying amounts of the type of forest being analysed, make comparisons hard (not helped by choices of colour, scales etc).

As a result, the conclusions end up being a bit muddled and similarly lacking in clarity. There is also so much material in the ED and SI, much of which feels unnecessary rather than adding to the scope or providing clarifying detail.

I think there could be a much more focused piece of work here, with clearer definitions and better figures, that would present a far more convincing story.

Referee #2 (Remarks to the Author):

General Comments:

The revised paper has been modified significantly from its original form. The scope of these changes reflects corrections and improvements of the original manuscript suggested from the reviewers. The authors are to be commended for making a strong effort to address the concerns raised and this is greatly appreciated. This is perhaps the most complete assessment of quantifying changes in structure for degraded and regrowth areas that I have seen using remote sensing data, and as I said previously, an excellent application of the analyses new remote sensing data will contribute to key issues in ecology.

My first review identified several areas of concern. These were: (1) Confusing and ineffective narrative and presentation that downplayed a significant result; (2) Questions about the robustness and appropriateness of statistical methods as well as justification for choosing important thresholds;

(3) Failure to use the complete observational archive of the GEDI mission; (4) Not including aboveground biomass (AGB) in the analyses.

The revised manuscript addressed each of these with but with varying effectiveness. The overall outline of the paper is improved but I found even an informed reader would still find the presentation somewhat hard to get through. I was hoping for a clearer, simpler story: Here are the major uncertainties and why they are important, this is what we tested, this is what we discovered. The presentation of the results is now broadly put into categories which helps, but it stills feels like one must adventure through a thicket of results and figures by continent and I had to read the paper and extensions again several times to really grasp the research. I still question whether the single result of structural changes far from an edge should be even more isolated and highlighted and saving the attribution aspect and its detangling for another paper where these can be presented in much more detail and with more validation.

The inclusion of AGB is much appreciated. As I cautioned in the first review, this would then lead to complications with regards to statistics and conclusions (which it does), but AGB should not logically be excluded from the research. I am not sure the authors made the correct decision in including the AGB results in the extended portion. Their argument is that paper is about structure, but it is widely accepted that structure includes biomass. Secondly, the motivation is dominated by carbon concerns, not biodiversity, water/energy cycles, nor conservation. Why do we care about canopy height and RH50? For this manuscript, it is mostly for carbon. If it were for biodiversity, for example, then biodiversity metrics derived from GEDI would be assessed (e.g., number of canopy layers, canopy cover profiles, FHD, etc.). I do not consider this a major flaw, but it does not lend clarity.

I am not convinced on certain aspects of the statistics. While a kind of bootstrapping (the authors call random sampling) is done, I still do not see how it enables a reliable derivation of measures of variability that is sufficient to make unambiguous assertions. For RH98 and RH50, I can accept their method, though it should state that it assumes there are no systematic errors in the GEDI observations of these. AGB is not so simple. Any statistical test involving AGB, or the precision of its mean will be far too optimistic, especially with regards to distinguishing between areas or time intervals. AGB is a product of regression equations. Failure to include regression error, along with sample error, means the variance is too small, and therefore we will accept, say a t-test for difference of means, when we should have concluded there was no difference. The implementation of methods to account for this exist. Each GEDI estimate of footprint AGB has an associated prediction error, and these will not “average out” when considering the precision of areal means. Note that the protected areas paper cited by the authors (Duncanson et al., 2023), used a statistical method (hybrid estimation) that propagated the footprint AGB regression model errors. Short of including model errors, the authors risk incorrect conclusions regarding AGB and that a subsequent study may invalidate some of their results and conclusions. I fear the authors need to go back redo the analyses for AGB accounting for this error or present a convincing rebuttal of why the error may be ignored.

I continue to miss the overall ecological model that links RH50 to edge effects. RH98 is clear as is AGB. But RH50 is not so. It is correlated to things we care about but unlike RH98, it is not unambiguous with regards to structure. The tallest thing in a 25 m footprint is easy to grasp and

unique. Not so for RH50. There are many different canopy configurations that could lead to same value of RH50. There is no discussion of this. For example, is RH50 lower because some tall trees were removed? Is the “recovery” of RH50 due to small trees growing or infilling of canopy from remaining canopy forming trees? I do not think it is reasonable to expect the authors to anticipate every potential set of canopy configurations, nor for us to focus on rare, pathological exceptions that would not impact the conclusions. But it is reasonable to expect that the metric be placed in proper context ecologically and its limitations noted. Furthermore, the paper loses the distinction between cumulative waveform metrics and quantities derived from the actual waveform (not the cumulative waveform). As a simple example, the number of layers in a canopy, that is the vertical organization, surely changes with disturbance, but the cumulative metrics would not be used to assess this. Can the authors provide a bit more framework on how they expect RH50 to vary and why. An alternative approach is to acknowledge the ambiguity of RH50 from an interpretation standpoint, do their data-driven analyses, and then to speculate how the exploratory data analyses seem to make sense with a bit more clarity. We know that Rh98 and Rh50 are correlated, but Rh50 captures other elements of canopy organization (as an assumption) in the portion that is not correlated.

Comments on Response to Referees:

(Numbers refer to authors’ reference list in response letter)

3) Conceptual model -> I do not think the intro paragraph is adequate for reasons listed above.

4) AGB -> Authors decision to not include AGB analysis in main body was discussed above. AGB is structure and because the motivations of the paper are almost entirely about carbon, and because non-cumulative metrics that are indicative of structure are not evaluated, this might need revisiting (but recognizing length limitations).

5) Authors correctly point out poor model performance of AGB in continental Asia, but not how this impacts their results or conclusions.

11) Novelty of analysis given previous work by authors and Nat Comms protected area paper -> Perhaps we could highlight these papers a bit more, especially since one is by the authors? Even a sentence such as “Building off of our previous work that established the efficacy of methods that utilize lidar to do ...” would be sufficient and helpful. In terms of the PA paper, perhaps more than a passing reference would be good. I fully agree the present manuscript covers more ground, but in terms of final conclusions and the impact on carbon, there is commonality. How many protected areas are also “intact” –would be good to know. Again, a sentence or two takes care of this.

19) Extended Data Tables -> I do not understand how we find the precision of the mean for any of the quantities. Are we doing that by bootstrapping the samples, then looking at the distribution curve, and picking the first standard deviation quantile? Are we dividing the mean standard deviation by some quantity, the \sqrt{n} ? As discussed above, the precision of the mean for AGB for an entire continent should include the regression error.

24) The definition of RH50 “...height at which half of the vegetation falls below and half falls above...” is technically not correct. If we have a very open canopy, say with a 20 m tree in the

middle, RH50 may be close to the ground because so much of the return energy in is in the ground return, far away from any notion that half of the vegetation is above this height. It might 100% of it.

27) Statistical significance of slopes -> Thank you very much for including these. What would be good is to now have some kind of discussion about the many examples that are not statistically significant. How does that impact our conclusions?

Comments on the revised manuscript:

44: What does "It" refer to here?

106-108: Inconsistent statistics here. "RH values above 17m on average", "... (RH98) varying from 25 to more than 25 m" (and what do you mean by varying here?), "...average AGB of 250 Mg/ha" -> one is above, another is a range, another is a mean.

113-118: You state elsewhere that the Asia models of biomass are not so good, yet in this paragraph you speculate on AGB variability based on large tree distributions. The variability may simply be poor AGB models.

126-127: How do you know this suggests "drastic changes in composition"? Even "suggests" is too strong here.

129: Do you mean true statistical "range" here. That is the true min/max?

156: Probably should not use the word "measured" for two reasons. First, the space instrument took the measurements, not the authors, so maybe retrieved is better. Secondly, the instrument does not measure biomass, it estimates biomass using models.

169: "associated with a change in forest composition". Again, how do you know this? How did the composition change?

177: Why are some given as ranges in parentheses? What does that mean?

210-215: Fig.2 I still am not clear on these figures. I believe I asked last time which of these distance intervals are significantly different given the wide range of the SD. For example, the SD bar at 100 m looks like it overlaps with every other distance. Should you do a similar set of analyses on the significance of the heights vs. distance as slopes that was performed in the time domain>

270-274: Is the time for space substitution valid under changing climate regimes or is the assumption that the climate is stationary over the record?

350: "for first time", it's always hard to prove statements like this and no loss of credit or significance of the research is lost by simply deleting that phrase.

502: Think this should be March 2023.

521: See notes above on interpretation of RH50.

531-534: These sentences are misleading. The authors seem to assert that the GEDI data need be calibrated (with ALS), with is not true. The paper cited does not properly geolocate GEDI data, so errors reported in that article are incorrect with respect to true height accuracy, and that paper acknowledges this fact.

535: Change “might” to “will”.

537-540: See my comments above. Probably a good idea to expand this a bit but it is not clear why this paragraph is at this spot in the paper.

543-546: Removing daytime AGB estimates was overly conservative. GEDI data include a sensitivity flag that can be used to filter. You also state you “label” shots as good quality. Is that not what the GEDI quality flags already do?

555: I am not aware of any study that justifies using GEDI data down to 2 m because the pulse width is too large. Can you cite one? On flat areas 3-5 m is probably good, but on slopes, 3 m is still too low because of pulse spreading therefore your results are likely unreliable there. That said if ALS data validate this (where you have it) that would be great to know.

608 and following: With regards to AGB, this was discussed above. The errors from regression that are ignored could greatly hamper ability to detect differences. The implied assumption that is being made is that the AGB estimates are perfect at the plot level, and for areas, the resampling procedure done by the authors generates truly random samples, so that the only effect is sample size. These assumptions are not tenable.

978: Ext. Fig 1A. Thank you for including the figure. I do not understand what the right-hand plot means, nor the grey circles. It would have been more effective if you had removed the elevation from the plot. The top of the canopy is going up in part because the elevation is going up. I really like the idea of this figure, perhaps try again but remove the ground elevation first?

1019: Ext. Data Tbl 1: See my comments above on interpreting the precision of the mean (how was it found) and the issue with AGB.

Referee #3 (Remarks to the Author):

The authors have addressed the comments of my review effectively. The paper is now more focused, clear, and balanced.

I do note that in abstract l. 29 – the theorised control ‘due to canopy desiccation and exposure to human activities’ should be moderated as 'likely' or 'probably', as there are a range of potential causes.

Author Rebuttals to First Revision:

Referee #1 :

This manuscript presents an analysis of the impact of anthropogenic disturbance on tropical moist forests over about 30 years, using space for time substitution and recent GEDI lidar data as a proxy for height (& structure change). It's an interesting topic and there is some interesting result here, particularly on the persistence in space and time of disturbance (and particularly compounding effects).

While there is some good and interesting analysis here, there is a quite a lot of rather vague assertion, which is largely a function of the lack of clarity in definitions: which height metrics and what they mean very specifically for the ensuing analysis; why each metric is being used in each case; and then choice of crucial distances (edge size etc). In particular, there is a lack of clarity of what is meant by structure right from the outset - it is used in a vertical and horizontal sense at various times, as well as an areal measure which is then interpreted in terms of individual tree size distributions. There is also a difficulty in assessing the results because key figures are rather poorly presented. The use of the 200 km cells as a base spatial scale with then different underlying amounts of the type of forest being analysed, make comparisons hard (not helped by choices of colour, scales etc).

As a result, the conclusions end up being a bit muddled and similarly lacking in clarity. There is also so much material in the ED and SI, much of which feels unnecessary rather than adding to the scope or providing clarifying detail.

I think there could be a much more focused piece of work here, with clearer definitions and better figures, that would present a far more convincing story.

We thank the reviewer for the appreciation of our work and for the comments. Reading the comments, though, it seems to us that this review was made on the first version of the manuscript rather than the revised version. For example, the reviewer mentions the 200km cells in his response above. Following the reviewer's recommendations reported in the first revision, we have changed this spatial scale at which we aggregate our results from 200 km to ~150 km in the resubmitted version. Moreover, the annotated manuscript that came along with this referee's revision was the first initial version and not the one resubmitted.

The many points raised here have already been addressed in the previous round of reviews and we invite the referee to review our responses. As suggested by the reviewer in the first review, we have taken great care in homogenizing the height metric we present in the main text. In particular: we have linked the interpretation of the forest structural metrics with relevant literature and more explicit definitions of forest degradation ('Preparation of input datasets' sub-section of the Methods), we have added in the method section justification of the choice of distances to edge effects that we selected to sample GEDI footprints and we have improved the figures and made them clearer.

Overall we have improved the narrative and better organized the presentation of the results to present a more convincing story, and this in fact was also recognized by the other reviewers. In this new round of reviews, we have consolidated our main results with robust statistics and assessed the influence of error propagation. Finally, we have simplified the ED

and SI as much as possible. Each main result is accompanied by an analysis with additional structural indicators in the ED and SI along with key tables of statistics in order to provide the most complete assessment of the robustness of the conclusions on forest degradation and regrowth at the pantropical scale.

Referee #2:

General Comments:

The revised paper has been modified significantly from its original form. The scope of these changes reflects corrections and improvements of the original manuscript suggested from the reviewers. The authors are to be commended for making a strong effort to address the concerns raised and this is greatly appreciated. This is perhaps the most complete assessment of quantifying changes in structure for degraded and regrowth areas that I have seen using remote sensing data, and as I said previously, an excellent application of the analyses new remote sensing data will contribute to key issues in ecology.

We thank the reviewer for the appreciation of our work.

My first review identified several areas of concern. These were: (1) Confusing and ineffective narrative and presentation that downplayed a significant result; (2) Questions about the robustness and appropriateness of statistical methods as well as justification for choosing important thresholds; (3) Failure to use the complete observational archive of the GEDI mission; (4) Not including aboveground biomass (AGB) in the analyses.

We agree with the reviewer and have addressed these four areas of concern in this revised manuscript:

- (1) We have improved the narrative by emphasizing the research gaps and presenting clearer objectives
- (2) We have used state-of-the-art statistical tests to assess the robustness of our results and to support the observed trends (in both spatial and temporal dimensions for all forest types/disturbances analyzed)
- (3) We have used the entire GEDI archive and followed best practices on quality check, filtered the GEDI dataset and retain good quality data (i.e. night filter has been removed for AGB as suggested by reviewer or by this study from the GEDI team¹)
- (4) In the revised version the aboveground biomass (AGB) has been included as well as the associated predictive error. We now report on this in the tables (Extended data Table 3 in the revised version). Moreover, we propagated the uncertainty associated with the AGB maps to the results to assess the potential impact on the spatial trends previously quantified.

Please find below our detailed response to each specific point.

1) The revised manuscript addressed each of these with but with varying effectiveness. The overall outline of the paper is improved but I found even an informed reader would still find the presentation somewhat hard to get through. I was hoping for a clearer, simpler story: Here are the major uncertainties and why they are important, this is what we tested, this is what we discovered. The presentation of the results is now broadly put into categories which helps, but it stills feels like one must adventure through a thicket of results and figures by

continent and I had to read the paper and extensions again several times to really grasp the research. I still question whether the single result of structural changes far from an edge should be even more isolated and highlighted and saving the attribution aspect and its detangling for another paper where these can be presented in much more detail and with more validation.

Thank you for this relevant comment. We have now improved the narrative to help better frame our work with the current research gaps and the major importance of degradation and forest recovery for carbon and biodiversity. We have restructured and re-written several sections of the (e.g. introduction and discussion).

Here we report few examples of the changes introduced:

The research gap is more explicitly presented and positioned at the beginning of the introduction (lines 45-47 of the revised manuscript):

“Yet, little is known about the state and extent of forest degradation along the pantropical belt³⁸ and large uncertainties remain in quantifying its contribution to the global carbon fluxes (25-69% of overall carbon losses^{3,4}).

The main objectives now reads (lines 77-79 of the revised manuscript):

“ Here we provide an overall assessment of the impacts of anthropogenic degradation on global tropical moist forest structure and an assessment of the forest's ability to recover to its pre-disturbance condition.”

and between lines 80-85 of the revised manuscript:

“ ...we quantify in a consistent manner and across the tropical humid belt: 1) the extent of forest degradation at year 2022 taking into account edge effects; 2) the impact of different types of disturbance on forest structural characteristics and their persistence over time (over the period 1990 to 2022); 3) the rates of forest structure recovery after each type of degradation (selective logging, fire, edge effect) and forest regrowth after deforestation; and 4) the vulnerability of degraded forests to subsequent deforestation.”

We have reduced the number of extended data figures to the most essential ones and moved the non essential figures/tables to supplementary information. This way we hope that it will make the understanding of our research easier. We also deleted redundant figures from the supplementary figures list and reduced the number of references.

We kindly disagree on the idea of focusing on one single topic as we are convinced that proposing a spatial and temporal analysis of forest degradation processes and dynamics is the added value of our work and would eventually have a broader impact. Those two dimensions are obviously intertwined, and now they are more explicitly presented in the introduction in terms of ecological importance, research gaps and objectives. The abstract now better captures these two main results.

2) The inclusion of AGB is much appreciated. As I cautioned in the first review, this would then lead to complications with regards to statistics and conclusions (which it does), but AGB should not logically be excluded from the research. I am not sure the authors made the correct decision in including the AGB results in the extended portion. Their argument is that paper is about structure, but it is widely accepted that structure includes biomass. Secondly, the motivation is dominated by carbon concerns, not biodiversity, water/energy cycles, nor conservation. Why do we care about canopy height and RH50? For this manuscript, it is mostly for carbon. If it were for biodiversity, for example, then biodiversity metrics derived from GEDI would be assessed (e.g., number of canopy layers, canopy cover profiles, FHD, etc.). I do not consider this a major flaw, but it does not lend clarity.

Thank you for your comment. The AGB results are reported in the text. We agree with the reviewer that the paper is more oriented towards carbon issues, and we worked in the revised version to make this point more clear, although we believe that canopy heights and forest structure are also key determinants for biodiversity (i.e. abundance of tall trees and ecological niches). These aspects (AGB, height, structure  Carbon, biodiversity etc) are so interconnected that we made an effort to look at the whole set of metrics together, and we believe this is the added value of the paper. On the other hand, not all metrics can fit the main text for reasons of limited space and we made choices.

In the manuscript, we seek to address forest conservation issues by mapping intact forest landscapes and emphasizing the critical need to manage highly degraded forests, which are most vulnerable to deforestation. The use of canopy heights (e.g. RH98) in the main text allows to broaden the potential audience and future application. The figures with AGB data are placed in the extended data figures section because of space limitation but mostly because we wanted to emphasize results based on direct measurements of forest structure (i.e., RH98 and RH50). AGB estimates derived from GEDI are still a preliminary product with larger uncertainties than height retrieval, mainly due to the scarcity of calibration data for the allometric models in some regions². We believe that our work opens the door towards the quantification of emission and removal factors from various forest degradation pathways at the pantropical scale, but this would require further fine-tuning of AGB modeling and validation.

3) I am not convinced on certain aspects of the statistics. While a kind of bootstrapping (the authors call random sampling) is done, I still do not see how it enables a reliable derivation of measures of variability that is sufficient to make unambiguous assertions. For RH98 and RH50, I can accept their method, though it should state that it assumes there are no systematic errors in the GEDI observations of these. AGB is not so simple. Any statistical test involving AGB, or the precision of its mean will be far too optimistic, especially with regards to distinguishing between areas or time intervals. AGB is a product of regression equations. Failure to include regression error, along with sample error, means the variance is too small, and therefore we will accept, say a t-test for difference of means, when we should have concluded there was no difference. The implementation of methods to account for this exist. Each GEDI estimate of footprint AGB has an associated prediction error, and these will not “average out” when considering the precision of areal means. Note that the

protected areas paper cited by the authors (Duncanson et al., 2023), used a statistical method (hybrid estimation) that propagated the footprint AGB regression model errors. Short of including model errors, the authors risk incorrect conclusions regarding AGB and that a subsequent study may invalidate some of their results and conclusions. I fear the authors need to go back redo the analyses for AGB accounting for this error or present a convincing rebuttal of why the error may be ignored.

Thank you for this relevant comment and constructive criticism. We thank also for pointing out few possibilities to improve the characterization of the error that can better include systematic errors. We have redone all analyses involving AGB (also removing the night filter as suggested) and have now taken into account the associated prediction error (i.e. Aboveground biomass density prediction standard error) into our results. To do this we have first decided to compute and report the weighted arithmetic mean (equivalent of Fig 1-4), in other words, AGB values will contribute more when their associated predictive error is low. Then, Analysis of Variance (ANOVA) and post-hoc Tukey HSD tests were computed to evaluate differences in RH98, RH50 and predicted AGB between the different distances to forest edge classes (Fig 2 and Extended data Fig.7) and time since last disturbance intervals (Fig 3 and Extended data Fig.11-12). For AGB, we used 500 Monte Carlo simulations to perturb AGB distribution using the associated predictive error and extract ANOVA outputs (p-values and F value) for each continent. We then analysed the frequency distribution of the ANOVA F test to evaluate if differences in forest classes vary when propagating the error. The results shown in the figure below (also in Supplementary Figure 8 in the revised manuscript) report the distribution of the ANOVA F-test. The results that observed differences in the different AGB of undisturbed forest measured at various distances to forest edge are robust to the uncertainty associated with the AGB values as no significant variability in the F values were observed:

**Distribution ANOVA F-test
Montecarlo**

Figure S8: Distribution of Anova f-test after Monte Carlo simulation to propagate AGB prediction standard error in the analysis of edge effects (see Extended data fig 7)

We have updated the Method section (GEDl dataset sub-section) accordingly and it reads as such (lines 528-531 of the revised manuscript):

“For each footprint, we extracted a set of Relative Height (RH) metrics, the Above Ground Biomass (AGB) and associated prediction standard error. AGB results are reported in terms of weighted arithmetic mean AGB where AGB values will contribute more when their associated predictive error is low”.

We have updated the Method section (statistical test sub-section) accordingly and it reads as such (lines 631-642 of the revised manuscript):

“We performed a series of one-way Analysis of Variance (ANOVAs) to test for differences in the impacts of edge effects at different distances and times on the long-term recovery of the relative heights and biomass variables. ANOVAs were performed separately for each continent. For the height variables (RH50 and RH98), a series of standard one-way ANOVAs

were used. In the analyses involving AGB, we used a modified approach to propagate the prediction standard error associated with the AGB dataset values: with a Monte Carlo approach ($n=500$), we generated random noise (mean of 0, σ = standard error of the AGB product) that was added to the AGB data. Then for each $n=500$ realization of the dataset, we calculated the F -values. We subsequently examined the distribution density of the F -values. The results showed minimal variability suggesting that observed differences are robust to uncertainty associated with the AGB values (Supplementary figure 8). For each ANOVA, we conducted a series of Tukey HSD post-hoc tests to assess significant differences between distance classes or time-steps. The significance level was set to $p < 0.05$.”

Finally, we have extracted the average AGB prediction standard error in 1.5° (~ 167 km) grid cells for intact forest areas. The map below (also found in Supplementary figure 4) shows that the average AGB prediction standard error varies from 13 Mg/ha (on average) for Americas, 17 Mg/ha for Africa, 18 Mg/ha for Borneo and Sumatra and 6 Mg/ha for Sulawesi, West Papua and Papua New Guinea.

Figure S4: Average AGB prediction error for intact forest in 1.5° grid cells between 30°N and 30°S .

4) I continue to miss the overall ecological model that links RH50 to edge effects. RH98 is clear as is AGB. But RH50 is not so. It is correlated to things we care about but unlike RH98, it is not unambiguous with regards to structure. The tallest thing in a 25 m footprint is easy to grasp and unique. Not so for RH50. There are many different canopy configurations that could lead to same value of RH50. There is no discussion of this. For example, is RH50 lower because some tall trees were removed? Is the “recovery” of RH50 due to small trees

growing or infilling of canopy from remaining canopy forming trees? I do not think it is reasonable to expect the authors to anticipate every potential set of canopy configurations, nor for us to focus on rare, pathological exceptions that would not impact the conclusions. But it is reasonable to expect that the metric be placed in proper context ecologically and its limitations noted.

Furthermore, the paper loses the distinction between cumulative waveform metrics and quantities derived from the actual waveform (not the cumulative waveform). As a simple example, the number of layers in a canopy, that is the vertical organization, surely changes with disturbance, but the cumulative metrics would not be used to assess this. Can the authors provide a bit more framework on how they expect RH50 to vary and why. An alternative approach is to acknowledge the ambiguity of RH50 from an interpretation standpoint, do their data-driven analyses, and then to speculate how the exploratory data analyses seem to make sense with a bit more clarity. We know that Rh98 and Rh50 are correlated, but Rh50 captures other elements of canopy organization (as an assumption) in the portion that is not correlated.

We agree with the reviewer's comment about the difficulty of relying on a widely recognized ecological framework to fully explain all RH50 variations along degradation and recovery gradients. We thus decided to present the RH98 figures in the main as suggested by the reviewer and to keep some explanation on RH50 when relevant. Generally, the reviewer can appreciate that we found co-variations between RH98 and RH50 values (e.g. on the scale of the edge effect in Fig.2). However, we believe that it is also important to show when RH98 and RH50 have different trends, particularly when looking at temporal profiles of forest recovery after degradation where we report clear trends of recovery after selective logging in RH50 versus persistent low values in RH98. As quantified by another recent study in the tropical moist forest of Malaysian Borneo (Ghizoni Santos et al. 2022), RH50 revealed to be a strong predictor of disturbance intensity as it measures the height at which 50% of plant material is (i.e. median of the vertical distribution of the canopy). This corresponds to forest layers where the logging impacts (tree removals, collateral damage of logging operations) and the fast regrowth of pioneer species affect the distribution of plant material. This dynamic can be effectively quantified and monitored using RH50, while RH98 is sensitive only to the height of the tallest trees and therefore less indicative to quantify the recovery dynamics.

In order to make this clearer for degradation from selective logging we added this paragraph at lines 253-257 of the revised manuscript:

“ The low recovery trends in RH98 can be explained by the slow regrowth of late successional, large and emergent trees³⁴, whereas forest understory dynamics including the removal of trees, collateral damages of selective logging (e.g. dead fallen trees) and the fast regrowth of pioneer and understory species affect the average vertical distribution of plant material,

captured by RH50, making this metric a robust indicator of long-term impacts from degradation and regrowth58 (Extended data fig 13).“

In order to make this clearer for degradation from fire we added this paragraph at lines 268-271 of the revised manuscript:

“ Manipulative studies of post-fire degradation in the Amazon showed strong understory vegetation regrowth under the remaining dominant and taller trees within five years after the disturbance, triggering partial canopy closure (70-80%64). This vegetation dynamic is more likely to be captured by changes in RH50 than by changes in RH98”

In order to make this clearer for degradation from edge effects we added this paragraph at lines 232-235 of the revised manuscript:

“ An even larger decrease of 20-30% and 20-40% on average for RH50 and AGB respectively, shows that the sensitivity of the median height and the biomass stock to single tree mortality events at the edge, is understandably higher than that of canopy height.”

Comments on Response to Referees:

(Numbers refer to authors' reference list in response letter)

5) Conceptual model -> I do not think the intro paragraph is adequate for reasons listed above.

This paragraph has been removed, all figures in the main show RH98. We added a paragraph of explanation dedicated to RH50 to stress its relevance when looking at direct impacts and recovery (see our response above).

6) AGB -> Authors decision to not include AGB analysis in main body was discussed above. AGB is structure and because the motivations of the paper are almost entirely about carbon, and because non-cumulative metrics that are indicative of structure are not evaluated, this might need revisiting (but recognizing length limitations).

We agree with the reviewer on the need to re-organise the storyline and incorporate the AGB material prepared in the review process. However, in doing so there is a delicate yet important trade-off between the need to convey all the relevant information and the journal's limitation in terms of text length and number of figures (i.e. maximum of ten, A4 size, multi-paneled display items). Therefore, as mentioned above, in the revised version we found the compromise to keep AGB figures in the extended data figures section but keep reporting the key AGB results in the main text to inform the reader about these results.

7) Authors correctly point out poor model performance of AGB in continental Asia, but not how this impacts their results or conclusions.

We have indeed acknowledged that AGB models are less performant in Asia using literature, through the computation of the precision of the mean AGB and other statistics and more recently in this review through the integration of the average AGB prediction standard error (Extended data table 3). Specifically, we show that propagating the error in AGB values does not impact the conclusions (Supplementary Fig 8). Figure S8 presented above shows spatial variability in the AGB prediction standard error in Asia and Oceania, varying from 18 Mg/ha for continental Asia, Borneo and Sumatra and 6 Mg/ha for Sulawesi, West Papua and Papua New Guinea. In order to more explicitly mention the potential impacts of poor model performance of AGB in Asia onto our results and conclusion, we added this sentence at lines 550-553 of the revised manuscript:

“ We acknowledge that future reprocessing of the GEDI archive (now version 1) will lead to some change and substantial accuracy improvements as it was in the transition from version 1 to version 2. We can for instance expect higher performance in future AGB models particularly in Asia where GEDI’s estimates of biomass have higher uncertainty in areas where calibration data are sparse. ”

It is difficult to quantify at this stage of development of the GEDI L4A product the potential impacts on our results. This is the reason why the paper emphasizes direct measurement of canopy height (RH98) and the median of the vertical distribution of canopy elements (RH50).

8) Novelty of analysis given previous work by authors and Nat Comms protected area paper -> Perhaps we could highlight these papers a bit more, especially since one is by the authors? Even a sentence such as “Building off of our previous work that established the efficacy of methods that utilize lidar to do ...” would be sufficient and helpful. In terms of the PA paper, perhaps more than a passing reference would be good. I fully agree the present manuscript covers more ground, but in terms of final conclusions and the impact on carbon, there is commonality. How many protected areas are also “intact” –would be good to know. Again, a sentence or two takes care of this.

Thank you for this relevant comment.

We have added a sentence in the introduction to make the link with previous work by some of the coauthors on utilizing GEDI Lidar to assess the effectiveness of European protected areas in conserving forest vertical structure. Now lines 79 and 85 of the revised manuscript reads as:

“ Building on a previous work to assess the effectiveness of Protected Areas in conserving forest structure with GEDI LiDAR38, we quantify in a consistent manner and across the tropical humid belt: 1) the extent of forest degradation at the year 2022 taking into account edge effects; 2) the impact of different types of disturbance on forest structural characteristics and their persistence over time (over the period 1990 to 2022); 3) the rates of forest structure recovery after each type of degradation (selective logging, fire, edge effect)

and forest regrowth after deforestation; and 4) the vulnerability of degraded forests to subsequent deforestation.”

We used World Database on Protected Areas (WDPA) database (version October/2023) to calculate the area of tropical moist forest (including undisturbed and degraded) and intact tropical moist forest (our approach and Potapov's) that fall within PA polygons in 2020. We added a sentence in the section **Cumulative impacts from edge effects in undisturbed and degraded forests** and made the link with Duncanson et al. 2023 paper³. It reads as such (lines 194-197 of the revised manuscript):

“ Around 48% of our IFLs fall within the World Database on Protected Areas (vs 53% with Potapov's) and conversely 57% (~60% in Americas and Africa, 28% in Asia) of protected tropical moist forest are mapped as intact forest which reinforces their importance for safeguarding forest structure and functioning⁵²”

We also updated the **Data availability** section with adequate reference to the WDPA database (line 707 of the revised manuscript):

“The World Database on Protected Areas (accessed in October 2023) can be downloaded at www.protectedplanet.net.”

9) Extended Data Tables -> I do not understand how we find the precision of the mean for any of the quantities. Are we doing that by bootstrapping the samples, then looking at the distribution curve, and picking the first standard deviation quantile? Are we dividing the mean standard deviation by some quantity, the \sqrt{n} ? As discussed above, the precision of the mean for AGB for an entire continent should include the regression error.

Thank you for this comment. We have revised the Updated Extended Data Table 3 and included the average of the AGB prediction standard error for each continent for Intact Forest:

Extended Data Table.3 Number of GEDI shots and statistics (mean, standard deviation and precision) of grid cells by continent of AGB and its associated prediction standard error for Intact Forest.

Continent	Number of GEDI shots	Mean AGB [Mg/ha]	Mean standard error [Mg/ha]	Standard Deviation of the AGB [Mg/ha]	Standard Deviation of the standard error [Mg/ha]	Precision of the AGB [Mg/ha]	Precision of the standard error [Mg/ha]
Africa	8989847	225	17.0	106	0.000829	37.9	0.0000589
Americas	25312312	228	13.0	123	0.00527	52.9	0.167
Asia	2520339	362	12.4	175	0.178	114	4.47

The precision of the AGB was obtained by computing the standard deviation of the AGB mean (*agbd_mean* layer in GEDI data). The precision of the standard error was obtained by computing the standard deviation of the AGB standard error mean (*agbd_se_mean* layer in GEDI data). To make things clearer, we added this information in the title of the extended data table and it reads as such (lines 925-929 of the revised manuscript):

"Extended Data Table.3 Number of GEDI shots and statistics (mean, standard deviation and precision) of grid cells by continent of AGB and its associated prediction standard error for Intact Forest. The precision of the AGB was obtained by computing the standard deviation of the AGB mean. The precision of the standard error was obtained by computing the standard deviation of the AGB standard error mean."

10) The definition of RH50 "...height at which half of the vegetation falls below and half falls above..." is technically not correct. If we have a very open canopy, say with a 20 m tree in the middle, RH50 may be close to the ground because so much of the return energy is in the ground return, far away from any notion that half of the vegetation is above this height. It might 100% of it.

We agree with the reviewer and rephrased. It now reads as such (lines 541-544 of the revised manuscript):

"RH50 (also known as "Height Of Median Energy"- HOME80) is the median height at which the 50th percentile of the cumulative waveform energy returned relative to the ground and has been identified as one of the LiDAR metrics with the greatest potential for estimating structural characteristics in tropical forests80".

We also renamed throughout the manuscript the term associated to RH50 into median height (instead of relative height) to improve the clarity and avoid confusion with the relative heights retrieved from GEDI L2A.

11) Statistical significance of slopes -> Thank you very much for including these. What would be good is to now have some kind of discussion about the many examples that are not statistically significant. How does that impact our conclusions?

We have explained in the results using relevant literature why some recovery trends are not statistically significant, in other words when we have either a persistence of degradation impacts at the edge of the forest for instance or when forest growth reaches a plateau. Regarding differences between RH50 and RH98 trends, we have now included a paragraph explaining what the two metrics capture (see response above).

Comments on the revised manuscript:

12) Line 44: What does “It” refer to here?

Thank you, we modified this sentence (and generally the whole introduction) to make it clearer. “It” referred to forest degradation. This first part of introduction now reads as such (lines 36-47 of the revised manuscript):

“ Tropical moist forests (TMFs) play a major role in the provision of global ecosystem services, including climate and water cycle regulation, carbon sequestration, and biodiversity conservation¹¹. Despite their importance, TMFs are disappearing at an alarming rate, mainly in landscapes where agricultural expansion is the main driver of forest conversion^{12,13}. Besides deforestation, anthropogenic disturbances also cause the degradation of remaining TMFs at a similar rate¹, resulting in the reduction of forest ecosystem services^{14–16}. Forest degradation is mainly caused by selective logging, fires and edge effects⁷, (edge effects being characterised by increased mortality of trees at forest edges¹⁰), and is often exacerbated by a combination of these disturbances occurring together, repeatedly over time. Moreover, forest degradation is driven by ongoing deforestation, habitat fragmentation and human-induced climate change^{17–19}. Yet, little is known about the state and extent of forest degradation along the pantropical belt³⁸ and large uncertainties remain in quantifying its contribution to the global carbon fluxes (25-69% of overall carbon losses^{3,4}). ”

13) Line 106-108: Inconsistent statistics here. “RH values above 17m on average”, “...(RH98) varying from 25 to more than 25 m” (and what do you mean by varying here?), “...average AGB of 250 Mg/ha” -> one is above, another is a range, another is a mean.

We agree that the reported statistics were inconsistent. We simplified the beginning part of this paragraph and decided to only present continental average with the associated spatial standard deviation. It now reads as such (lines 100-103 of the revised manuscript):

“ The analysis of more than 23 million sample footprints of GEDI data over intact forest area from the JRC-TMF dataset (Americas: 16,092,827; Africa: 5,882,035; Asia: 1,523,891) reveal regional and continental variability in forest structure (Extended Data Table 1 to 3) where the highest canopy heights are located in Insular Asia, Western Africa and Western and Eastern Amazonia (Fig. 1a).“

14) Line 113-118: You state elsewhere that the Asia models of biomass are not so good, yet in this paragraph you speculate on AGB variability based on large tree distributions. The variability may simply be poor AGB models.

Thank you for this comment. When we refer to large tree distributions, it was when using the RH98 results (see Supplementary Fig 2), not the AGB results. To be more clear, we have removed this sentence from the result section. In the methods, we already mention poor AGB model performance in Asia compared to the other regions. See point #7.

15) Line 126-127: How do you know this suggests “drastic changes in composition”? Even “suggests” is too strong here.

We have deleted this part accordingly. It now reads as such (lines 116-119 of the revised manuscript):

“ Forests within 120 m from the edge with another land cover class show a lower RH98 and AGB compared to intact forests (Fig.1c) with an absence of emergent trees and a canopy height distribution dominated by smaller trees (Supplementary Fig.2). ”

16) Line 129: Do you mean true statistical “range” here. That is the true min/max?

We decided to report the average and spatial standard deviation at pantropical scale in order to be more clear. It now reads as such (lines 119-120 of the revised manuscript):

“ Regrowth forests (Fig.1d) have, on average, a 10 m to 15 m lower RH98 than intact forests, and an average AGB of 87 Mg/ha \pm 38.”

17) Line 156: Probably should not use the word “measured” for two reasons. First, the space instrument took the measurements, not the authors, so maybe retrieved is better. Secondly, the instrument does not measure biomass, it estimates biomass using models.

Thank you for your suggestion, we replaced ‘measuring’ by ‘retrieving’ throughout the whole manuscript.

18) Line 169: “associated with a change in forest composition”. Again, how do you know this? How did the composition change?

Besides calculating average values of RH98 metrics, we also computed the frequency histogram for each grid cell and distance to forest edge showing the distribution of canopy heights in 5m bins. We agree with the reviewer that a change in forest composition is a far

stretch interpretation of the frequency histogram and rephrased this sentence into (lines 162-164 of the revised manuscript):

“ We estimated a 10-20% reduction in RH98 (20-30% in RH50 and AGB) within the first ~ 200 m of the forest edge relative to intact forest interior, associated with a change in structure and loss of tallest trees (Fig.2b) ”.

Note that Fig 2 has been modified and we now present these frequency histograms at continental scale for each distance to forest edge as they clearly show the loss of the tallest trees for forest close to the edge. This is now presented in a second panel:

Fig 2b: Average distribution of undisturbed forest (located at more than 120 meters from degraded forests) canopy height at various distances to the forest edge (b). The colours for the undisturbed forest (solid line) at various distances from the forest edge in panel a correspond to those in panel b

19) Line 177: Why are some given as ranges in parentheses? What does that mean?

These ranges correspond to the first and third quartile of the boxplot distribution. We have simplified the sentence and now only report the average. It reads as such (lines 168-171 of the revised manuscript):

“ In addition to edge effects induced by deforestation, we quantified that canopy heights in undisturbed forests near logged or burned forests (i.e. within a 120 m radius) were on average significantly lower than in intact forests by 15% and 22%, respectively (Extended Data Fig.8, 22% and 32% lower on average for RH50). ”

20) Line 210-215: Fig.2 I still am not clear on these figures. I believe I asked last time which of these distance intervals are significantly different given the wide range of the SD. For example, the SD bar at 100 m looks like it overlaps with every other distance. Should you do a similar set of analyses on the significance of the heights vs. distance as slopes that was performed in the time domain>

Thank you for this suggestion. The SD bars represent spatial standard deviation at the continental scale while standard error bars (not shown in the figure) do not overlap due to the high number of observations. Nonetheless, we performed one-way ANOVAs for each continent followed by a post hoc Tukey test to evaluate differences in RH98 (resp. RH50 and AGB) between the undisturbed forest located at various distances to the forest edge. The results of the ANOVA (F value) are pasted on the figure while the results of the post hoc Tukey tests are provided as supplementary files due to visibility reasons. You can see below that pasting the results of the ANOVA and Tukey on the figures blurs the message and makes the figures unreadable:

a

Edge Effect (RH98)

Figure A1: Canopy height (RH98) of undisturbed forests (located at more than 120 metres from degraded forests) and all forests (including undisturbed and degraded forests) at various distances to the forest edge separating forest cover from agricultural land and other land covers.

a

Forest Edge (RH98)

Figure A2: Long-term recovery of top canopy height (RH98) following degradation due to forest edge creation and degradation (fire or logging) of edge forest.

We preferred to present the results of the Tukey tests in supplementary files. Twelve files in total are attached to the submission of the revised manuscript and complement Fig2, Fig3, Extended data Fig 7, 11 and 12.

We report on the results of these statistical test in the text, see lines 153-156 of the revised manuscript :

“ Distance to edge had a significant effect on canopy height for each continent (Supplementary file 1). We found significant differences up to 7 km in the Americas (Tukey's HSD Test,

$F=4449$, $p<0.0001$) and 1.7 km in Africa ($F=647$, $p<0.0001$) and Asia ($F=909$, $p<0.0001$), suggesting that the impact of the edge effect may be greater in the Americas. ”

21) Line 270-274: Is the time for space substitution valid under changing climate regimes or is the assumption that the climate is stationary over the record?

Thanks for this comment. In fact, this assumption holds if the background climate and vegetation acclimation (including species shifts) do not change substantially, and could further be used to explore impacts in the near-future, but would require special attention to projecting them in a changing climate. However time series analyzed in this manuscript are too short to explore this aspect.

We added this sentence in lines 388-390 of the revised manuscript:

“ The central assumption is that differences in properties of neighbouring patches of land can serve as a surrogate for changes in time and that the background climate and vegetation acclimation do not change substantially over the 20-30 years of analysis. ”

22) Line 350: “for first time”, it’s always hard to prove statements like this and no loss of credit or significance of the research is lost by simply deleting that phrase.

Thanks for this comment, we deleted this accordingly.

23) Line 502: Think this should be March 2023.

We believe the mission was extended until January 2023:
<https://gedi.umd.edu/mission/timeline/>

24) Line 521: See notes above on interpretation of RH50.

Thank you, this was already answered above.

25) Line 531-534: These sentences are misleading. The authors seem to assert that the GEDI data need be calibrated (with ALS), with is not true. The paper cited does not properly geolocate GEDI data, so errors reported in that article are incorrect with respect to true height accuracy, and that paper acknowledges this fact.

Thank you, we agreed with the reviewer and decided to remove these sentences.

26) Line 535: Change “might” to “will”.

Thank you, we changed it accordingly.

27) Line 537-540: See my comments above. Probably a good idea to expand this a bit but it is not clear why this paragraph is at this spot in the paper.

Thank you, this was already answered above.

28) Line 543-546: Removing daytime AGB estimates was overly conservative. GEDI data include a sensitivity flag that can be used to filter. You also state you “label” shots as good quality. Is that not what the GEDI quality flags already do?

We completely agree with the reviewer and we relaunched all analysis (data extraction, data aggregation, statistics etc) involving AGB without the night filter as the quality flags (l4_quality_flag - Flag simplifying selection of most useful biomass prediction) already does that. We modified the method section accordingly. It resulted in an increase of 56% in the number of GEDI shots used (compared to level 2A). We updated the method accordingly and it reads as such (lines 554-559 of the revised manuscript):

“ To select the highest quality data, we filtered the GEDI data (both GEDI L2A and L4A) by selecting only the observations collected in power beam mode and labelled them as good quality (i.e. quality flag equals 1), thus avoiding risks of having degraded geolocation under suboptimal operating conditions (i.e. degrade flag equals 0). Additionally, we filtered GEDI 2A data using only night acquisitions to limit the background noise effects of reflected solar radiation. ”

We also updated extended data table 3.

29) Line 555: I am not aware of any study that justifies using GEDI data down to 2 m because the pulse width is too large. Can you cite one? On flat areas 3-5 m is probably good, but on slopes, 3 m is still too low because of pulse spreading therefore your results are likely unreliable there. That said if ALS data validate this (where you have it) that would be great to know.

The recent paper from Turubanova et al. 2023⁴ used GEDI data down to 3m (in RH83 value):” We assigned zero tree canopy height values if the RH83 value was <3 m”. Moreover, recently degraded forest from selective logging or understory fire can present heterogeneous canopy heights^{5,6} varying from 2m (corresponding to canopy openings) to 15m (remaining standing trees) reinforcing the importance of lowering the threshold at which we filter GEDI data for degraded forests.

We assessed that the large majority of GEDI footprints (>75%) with canopy heights from 2 and 5m fall within 0-5 degree slope, meaning that the potential risk of pulse spreading on high slopes and thus yield unreliable heights is minimal (Figure A3).

Figure A3: Distribution of GEDI shots with canopy heights (RH98) ranging from 2-5m for different slope classes and classes of intact, logged, burned and regrowth forests.

30) Line 608 and following: With regards to AGB, this was discussed above. The errors from regression that are ignored could greatly hamper ability to detect differences. The implied assumption that is being made is that the AGB estimates are perfect at the plot level, and for areas, the resampling procedure done by the authors generates truly random samples, so that the only effect is sample size. These assumptions are not tenable.

This has now been addressed in the previous comments, see point #3.

31) Line 978: Ext. Fig 1A. Thank you for including the figure. I do not understand what the right-hand plot means, nor the grey circles. It would have been more effective if you had removed the elevation from the plot. The top of the canopy is going up in part because the elevation is going up. I really like the idea of this figure, perhaps try again but remove the ground elevation first?

Thank you for this feedback, we have accordingly simplified the figure and removed the ground elevation. We updated the Extended data figure 6 in the manuscript.

Extended Data Fig.6 Transect (-47.76, -3.68) of GEDI relative heights (RH50 and RH98) across deforested land, edge forest (width of 350 m) and intact forest collected in the Brazilian Amazon. The two black lines represent the height of RH98 and RH50.

32) Line 1019: Ext. Data Tbl 1: See my comments above on interpreting the precision of the mean (how was it found) and the issue with AGB.

This has now been addressed in the previous comments, see points #3 and #9.

References :

1. Liang, M., Duncanson, L., Silva, J. A. & Sedano, F. Quantifying aboveground biomass dynamics from charcoal degradation in Mozambique using GEDI Lidar and Landsat. *Remote Sensing of Environment* **284**, 113367 (2023).
2. Dubayah, R. *et al.* The Global Ecosystem Dynamics Investigation: High-resolution laser ranging of the Earth's forests and topography. *Science of Remote Sensing* **1**, 100002 (2020).
3. Duncanson, L. *et al.* The effectiveness of global protected areas for climate change mitigation. *Nat Commun* **14**, 2908 (2023).
4. Turubanova, S. *et al.* Tree canopy extent and height change in Europe, 2001–2021, quantified using Landsat data archive. *Remote Sensing of Environment* **298**, 113797 (2023).
5. Bourgoin, C. *et al.* UAV-based canopy textures assess changes in forest structure from long-term degradation. *Ecological Indicators* **115**, 106386 (2020).
6. Silva, C. *et al.* Impacts of Airborne Lidar Pulse Density on Estimating Biomass Stocks and Changes in a Selectively Logged Tropical Forest. *Remote Sensing* **9**, 1068 (2017).

Referee #3:

The authors have addressed the comments of my review effectively. The paper is now more focused, clear, and balanced.

We thank the reviewer for the appreciation of our work.

- 1) I do note that in abstract l. 29 – the theorised control ‘due to canopy desiccation and exposure to human activities’ should be moderated as ‘likely’ or ‘probably’, as there are a range of potential causes.

We agree with the reviewer. However, we decided to remove this sentence in order to ensure that the abstract length complies with the journal guidelines. The causes of the edge effect are explained and documented in the main. The revised abstract reads as such:

“ Tropical forest degradation from selective logging, fire, and edge effects impacts a larger area than deforestation^{1,2} and threatens carbon stocks, habitat, and biodiversity^{3–6}. Yet, little is known about structural changes in degraded forests along the pantropical belt, and large uncertainties remain in quantifying their long-term consequences⁷. Here, we combine satellite remote sensing data (1990-2022) on forest cover change^{1,8} with canopy height and biomass measurements from a spaceborne⁹ Light Detection And Ranging (LiDAR) to assess the magnitude and persistence of degradation impacts on the vertical structure of global moist forests. Anthropogenic disturbances decrease forest heights by 20-80%, showing understory regrowth but low rates of canopy height recovery even after 20 years. Also, the creation of forest edges as a result of agricultural or road expansion triggers a 20-30% reduction of canopy height and biomass, with persistent impacts measurable up to 1.5 km inside the forest. Our findings show that 18% (~206 Mha) of the remaining tropical moist forests are affected by edge effects, an area more than 200% larger than previous estimates¹⁰. We demonstrate that the cumulative impacts from logging and fire on forest structure significantly increase their vulnerability to subsequent deforestation. This calls for greater efforts to prevent degradation and protect already degraded forests to meet the pledges on conservation made at the recent United Nations Climate Change and Biodiversity Conferences ”

Reviewer Reports on the Second Revision:

Referees' comments:

Referee #1 (Remarks to the Author):

General comments

Overall I think the paper is an improved version and a number of the previous issues have been dealt with. As I noted before I think the results are definitely interesting – the range of degradation impact into the intact forest is particularly interesting and far exceeds what is generally assumed. Same for the persistence, and also the covariance of the impact types. So I think this would definitely be of interest to a wide readership and has some important implications.

That said, I still think those key results could be explained much more clearly and simply – it takes quite a bit of time and effort to get to these. This is also reflected in the way results are presented in the abstract and even the title, which could definitely be punchier given their results (eg something like Human-driven degradation of tropical forests is much greater than previously estimated in both space and time). The overall narrative is disjointed by technical detail where it is not needed, as well as the many references to extended data. That needs a rethink – see my comments below. If the ED is so crucial to the point being made, then it should either be in the main text, or the main text needs to change to reflect those data.

I have a question about the nature of the manually delineated data they used for classifying selective logging activities from S America, where there were no WRI concessions data. If this is not the same as the WRI data, then it is possible (perhaps likely) that the results for S America will be different to those for Africa and SE Asia; that is what is found in the analysis. So those manually delineated logging activities have to be shown to be comparable, otherwise any subsequent attributions of difference are not likely to be valid as they are based on different assumptions (hence systematic bias is quite possible).

Specific comments

Line 38-39: Little is known about structural changes in degraded forests Is this true though? The authors cite some of this work, but I think it's fair to say we do know quite a bit about this but it is usually regional rather than pan-tropical / comparative.

Longo et al 2020, water energy and c cycling, Amazon

Milodowski et al. 2021, vertical canopy structure, Borneo

Silverio et al 2019 forest. Fire, fragmentation and wind, Amazon

Stark et al 2020, structure and energy balance, Amazon

Dalagnol et al 2023, C emissions v degradation from NICFI Planet data, Amazon

Jayanthilake et al. 2021 Drivers of this (social science approach), global

Line 41 Not biomass measurements, estimates

Still slightly confusing degradation from ... edge effects. This isn't clear how this happens – it's not a

single thing, but a complex mix of logging, development, fires etc.

Line 44 – showing understory regrowth ... the implication is that it is anthrop disturbance that is showing this. Is this per area, or where impact on H is highest, or ? Not clear.

From the abstract and lower down the key results seems to be the persistence of these impacts away from the edges (but need to define persistent impacts eg > 20 years) and the underestimate of prev edge impacts. If so, the title and abstract could be better framed eg something like: Human degradation of tropical moist forests is far more extensive and persistent than previously estimated

Line 121/2 – can degradation occur at the same rate as deforestation if they are essentially quite different things?

Edge effects are a tricky thing to wrap up into the general discussion of degradation as they are co-varying (driven by, exacerbated by) with the other effects. This part has never quite been clear in the different versions. It is explained well in the Methods, but ideally needs really carefully laying out here in the main text ie Here we define edge effects as these will increase in response to and so are closely-dependent on

Line 125 – moreover ... is this needed? Can you have degradation without deforestation or vice versa? If so, then prob need to state this either way ie are they inextricably causally linked and which way(s).

Line 131 – not a clear sentence.

Line 150 – remains localized and limited in access – what does that mean? If you mean coverage, then these are just the same thing. I'd argue that thousands of ha that can be covered by ALS is not so local. It's not pan-tropical but can be regional eg see the CAO work in Peru. If you mean cost and availability (access) then that is surely changing – nearly everywhere has some ALS systems available either from state, local gov or commercial or some combination.

151 – what is small-scale?

Lines 158 on - Use of pan-tropical scale, global, near-global-scale in different places and in different ways. These don't really mean anything – it's either pan-tropical or it's not. Neither of these are global or even near-global. Then it switches to global tropical moist forest. Just say tropical MF, or pantropical. There are many different uses of the word scale throughout without clear definition or context, which is confusing.

Line 159 – overall? Meaning what?

Line 162 – now it's tropical humid belt (see above). Needs to be consistent terminology throughout.

Line 170 – sample dataset. I know what is meant but this suggests a kind of test dataset. Better to phrase as a spatially discontinuous or point-based.

Line 175 – you need to explain what RH98 and RH50 are here as these are your critical metrics and non-experts will have no idea what they actually represent. They are defined in the methods, but this is critical info and so should be summarised here. Also you then refer to RH98 as the proxy for canopy height – fine, but then don't use top height and other terms elsewhere.

Line 177 – I don't think you can argue this is comprehensive. It doesn't for eg give leaf area density, n layers, gap fraction etc. It's definitely a good set of metrics, but

Line 208 – I think intact needs defining given all that is to come, particularly as you are defining disturbed/degraded relative to this (eg impact distance from an edge). Are forest edges 120 m width, then how far does the other side of this extend before this becomes eg a patch? And is there some linear extent?

Line 210 – highest canopy heights – can you just say tallest canopies? Then, overall canopies are taller

Our results show that the mean canopy height and AGB densities of degraded forests are at least 10 m and 100 Mg/ha lower, respectively, than intact old-growth forests (Fig.1b, Extended Data Fig 3-4).

This is of course what we'd expect – by definition almost - but the interesting part is quantifying this.

In figure 1, why mask out the hexagons where there is no difference between intact and other forest types? This is a different category from < 600 footprints, but both are masked out. Also, that sentence needs to be rephrased as Statistically significant differences in canopy height

Line 226 Regrowth forests 10m to 15m lower RH98 ... ? Which is it? Or if it's different based on location, state this. Same for lots of the other statements like this – the range is confusing as it implies uncertainty, rather than comparing specific regional differences.

Line 315 Distance to edge impact in the Americas up to 7 km – this is a major result. I'd mention this in the abstract. Also, say where this is (not just the Americas, but specifically). And it IS greater isn't it (not may)? The impact on AGB is not similar scale though (line 317) – it's a lot smaller, and a different pattern (much more similar between continents).

Line 333 – small-scale again. Meaning?

Line 338 – and this resulted in

And these numbers 15-25%, 30-40% RH50, 25-40% AGB – see my comment above re line 226 - are these variations those between continents? If so, you should probably say so as otherwise it doesn't make sense given you have discussed differences between continents elsewhere.

Fig 2: A lot of info here and not all of it clear. The colorscale of the vertical bars in 2a just aren't clear enough – lines would need to be much thicker. Solid lines – I think this is the loess fit (seeing the caption) but there is no blue line – previous version? Why is loess fit necessary? If the fig was clearer you could just use the points & don't need some arbitrary line fit, which seems to have been added

just to help navigate the dots. It's not a model prediction / interpolation so don't use. Same for the inset – what do these lines represent? Also the red dotted horiz line referred to later is almost invisible, but this is important information.

Some similar issues on fig 2b. Can't see the intact line. Also these are histograms so why have lines (again)? they should be bars with errors or even box-whisker would be better. Overall this figure is important but it needs to be clearer. If the 2b histos could be overlaid on a single plot that would help compare one continent to the other also.

474: you start using top canopy height here when you haven't defined or used it before.

Line 486: twice as strong as logging alone. And how long time series would you need? I'd leave this sentence out, or you need to justify what you mean here w some evidence on what 'longer' means.

Line 493: half that of?

Lines 498-9: do you know they are not representative? If not, leave this out.

Line 499: a complete time series? Not sure what that means (annual?) – implies measurements all the time which is definitely not the case here, so leave this out.

Fig 3 – also a bit unclear. You are referring to top canopy height again but not defined. Also the panels say one thing eg Forest Edge then (RH98) – so do you mean the impact of edge on RH98? If so, just say that. Same for others. And the line fits again. Not needed, and not even consistent eg missing on Americas panel in line b, and the line fit for 1x logged in Asia has a huge jump of 8 years! The logged + burned points in these plots are impossible to distinguish, except for 2 in the Americas panel. They are either too similar in color to the logged 1+ or not there.

Line 541 – this is a much clearer statement of the problem, which should come right at the top.

Line 543

Here we show using three years of GEDI data (more details in the Modelling deforestation risk section of the Method) that degradation is also a determinant of future deforestation, where the probability of full removal of forest cover and change in land use increases with the degradation stage of forests.

You can just say, here we show that degradation is (and then refer to Methods at the end).

This is an issue throughout. There is a lot of reference to Extended Data, Methods etc that is quite distracting from the flow and the overall message. If they are important enough that they need to be referred to throughout then they are not ED. I think some more consideration is needed in terms of what is left in/out in terms of what the key results are and what needs to be in ED or left out altogether. See my comments on that part below also.

Line 547: recently degraded? Same w line 549 – this whole bit is hard to follow and needs rephrasing eg Forests that are recently (within X years) degraded and are then deforested within X years,

showed reductions in mean ... This corresponds to 'severe' impacts from unsustainable logging and/or fire (where severe is defined as)

Line 555: see comments above. Using Generalised Linear Models (GLM) (see the section on statistical analyses in the Methods) we confirmed that ... why do you need that qualifier? Just state Parameters related to ... were shown to be strong predictors ...

Line 560: again, this section is not clear and needs clarifying – what confounding effects, and is this different to previous studies?

Fig 4 – if you were looking to leave something out, I would suggest this – it is interesting but it doesn't really need a whole figure as you've already stated the main result in the text. A more interesting figure would be the map of where this puts the most vulnerable to deforestation places & even a statement on that eg from our results, the top X most vulnerable locations are X, Y and Z.

Line 591 You mean GEDI samples not plots. Also, what is the intention of using seemingly large numbers like this (see above also) – are they meaningful? If you're using a Sentinel-2 image it has 125 M samples (or pixels as they are usually called). Nobody ever says 'we processed X trillion' samples of Sentinel or Landsat data, but it's effectively the same thing. So unless there's a specific reason for using this number, leave it out. A more meaningful number might be eg the total ha of TMF covered by GEDI samples out of the total in each continent, but that wouldn't sound so impressive.

Line 594 – this could be clearer. We show that the spatial extent and magnitude of degradation impacts on canopy structure are greater than aboveground biomass doesn't need all caps. Then – the impact of edges on forest vertical structure was previously assumed to extend no more than ~ 100 m into intact forest. Our results show this is a significant underestimate, and that

Line 610: these terms are a bit vague. Can you either define them or just leave as 'spatially explicit, at scale of Xm, over Y ha'?

Line 620: might want to revise in light of the re-deployment plan. Also, is this really all that relevant – seems more like context. Could just start from Line 624. I think this could also be shortened quite a lot – stick to the actual methods used here.

Don't really need any of the detail lines 669 to 680 as that is described elsewhere.

Line 717; Do you mean complemented / augmented? And what does extensive visual interpretation mean? How accurate are these data and have they been validated and if so how? This is quite important as it suggests you are using a different scheme for S America than you are for Africa and SE Asia. If so, could that explain any of the differences you observe for SA v Afr v SEA? Is there a way to show that if you did the same for eg in Congo, Gabon etc that you get the same / similar results as in the WRI data? And that if different people did the analysis they would get the same results?

Lines 850-857 I think this is what needs to go into the main section to define what RH98 and RH50

actually are. This is where you also define RH98 as canopy height, but then go on to refer to top canopy height in a number of places still.

Lines 928-32: why these numbers (7, 600, 30)?

Line 1026 on: don't need a description of GEE really.

Extended data

There is a LOT of material here, some of which is interesting, but a lot of which seems almost like draft material – possibly nice to have, but not essential. I think unless the material is vital to evidence a specific result or point in the main text, it should probably be cut. It would help if it were also organised so that all results here are organised under sub-headings that relate specifically to which result/point in the main text they are related to.

You could show here for eg comparisons of the visual assessment of the S American selective logging impacts flagged above.

ED Fig 6 – this is interesting and useful to show. Might be good to see other examples of these from eg locations where you find differences for S America, Africa and SE Asia. It's a good way for reader to visualise what you're talking about – refer to this v early on in the main text to show what it is you're actually deriving. The RH98 suggests the canopy at the deforested end is around 3m, RH50 says it's flat - & it looks like bare soil (although I know that might be misleading). If it's bare wouldn't RH98 and RH50 both be zero? Or is this because GEDI transect was at a different time to the image data when crops were present? Can you give the dates of the GEDI and image data?

Referee #2 (Remarks to the Author):

Summary

Thank you for the effort in responding to my second round of comments. I applaud your patience and work in this effort.

At this point I think we have cleared up many of the major issues, including things that were not quite right, adding clarity, and most importantly using nearly the full archive of GEDI data. This last point, however, does make review difficult because each subsequent version thus far has had significant changes, especially with respect to the kinds and quantity of GEDI data used. And each review of a revision starts to find different things that were missed in previous reviews because the entire paper needs to be evaluated again. I apologize for bringing in a few new concerns, but this essentially what has happened and is unavoidable.

My main issue still concerns the statistical methods and the meaning of some of the descriptive statistics and uncertainties that are being reported. These are described below but center on the monte carlo approach to propagating regression uncertainty and the results presented in Extended Data Tables 1-3. For example, the precisions shown for height are greater than what GEDI reports as

RMSE against airborne data globally, so its not clear why aggregates of these data and the precision of those means should be nearly as large as reported. So either there is a methodolglcal issue or the description of what is being reported lacks clarity and needs to be refined.

Comments on Responses

1. "AGB estimates derived from GEDI are still a preliminary product with larger uncertainties than height retrieval, mainly due to the scarcity of calibration data for the allometric models in some regions². We believe that our work opens the door towards the quantification of emission and removal factors from various forest degradation pathways at the pantropical scale, but this would require further fine-tuning of AGB modeling and validation."

=> This is not so important but what you assert here is not actually true. GEDI V2 biomass data was released, and these are no more "preliminary" than any other global biomass map; indeed, GEDI has at least as much and maybe more calibration data than any other product.

2. "We have redone all analyses involving AGB (also removing the night filter as suggested) and have now taken into account the associated prediction error (i.e. Aboveground biomass density prediction standard error) into our results."

=> Thank you for doing these analyses. I am still not sure about how you propagated the prediction error. The process is described under "Statistical Tests" in the manuscript. In the above description this seems to suggest that all the prediction errors at 25m from GEDI are used to find the standard deviation of this distribution? The manuscript says the sigma of the distribution is set to "the standard error of the AGB product". That product does not have a single standard error, it only has prediction standard errors for each footprint and at 1 km cells. So, it's not clear what the parameters of the distribution are being sampled from are exactly. It seems as if it's the standard deviation of the standard error distribution? Secondly, randomly permuted the AGB data set 500 times and calculated an F statistic. I do not see why this would work as the prediction errors are highly correlated with the AGB value. They tend to get larger as the value gets larger, so as you sample from that noised distribution you should take that into account. Third, even if we randomly sample from the AGB, I do not see how we overcome the issue that the residuals are spatially correlated unless we specifically draw samples that are not spatially correlated. All the above, and the actual F values make me think something is not quite right. Would you please provide more exact description of the methods here, for which data (in 1.5 deg cells, in polygons, etc) so that it can be recreated (that is step by step)?

Note that the Committee on Earth Observation Satellites (CEOS) has published a good practices protocol that includes methodology on using monte carlo methods for estimating biomass uncertainty and at least one of the manuscript authors (Avitabile) is a coauthor on this report. https://lpvs.gsfc.nasa.gov/PDF/CEOS_WGCV_LPV_Biomass_Protocol_2021_V1.0.pdf

3. Figure S8.

=> I cannot say I understand Figure S8. Looks like for each continent there is no distribution of the F-stat or the way its plotted is masking it. Can we explain this figure more? Why does it look like a

single vertical line?

5. Extended Table 1 and 2

=> If we look at Extended Table 1, we report the spatial S.D. and the precision of the mean. I understand these column headings. But how did we find that precision of the mean? If the samples are random and uncorrelated, that precision should be the $\sqrt{(\text{Merr}^2 + \text{var})/n}$, where Merr is the measurement error for GEDI heights and var is the sample height variance of the GEDI samples in the area – this is not what you did (nor should you for GEDI). So, how was this found for height? Unless there is a bias, can explain how you can get such a large error about the mean when the S.D. are (a) on the order of 5-6 m, (b) you have millions of shots, and (c) you are ignoring measurement error?

Alternately, do we mean this is the precision of grid cell means? If so, how did we propagate the uncertainty about each grid cell mean to get a precision for the entire continent? If we could see the formula and the inputs used to recreate one of the rows, that would help greatly.

4. Extended Data Table 3.

=> I apologize but I still do not understand the SE's in this table. I'm also confused about what is being presented. Each GEDI footprint has a prediction error from its associated regression calibration. The table reports something called "Mean Standard Error", say for Africa, with 17.0 Mg/ha. What is that number and how do you find it? We then see "precision of the AGB" (37.9), but again how do we find it? That is just the standard deviation of all the GEDI shots in the area? Is it the precision of mean biomass which we can use to form a confidence interval?

=> The usual metric associated with uncertainty where sampling is involved is the "precision of the mean", or the or "standard error the mean". If GEDI shots were actual observations of AGB and if they were random, the precision of the mean would be the common formula of $\sqrt{\text{variance}/n}$. But this is entirely incorrect for GEDI. The observations have a regression error and they are not random, hence the montecarlo approach. It does not seem to me that we have calculated a "precision" as normally thought of in this table, where the precision is a function of both the regression error and the sampling error and would tell us the limits of the mean for some confidence level. Or am I missing something here? I also point that even if we take the sampled data and make a grid, we still must propagate the grid level sample uncertainty when reporting values that combine grid cells, say over a continent.

=> It might help if we note everywhere where we mean standard deviation and where we mean "standard error". I could not always tell.

6. "We assessed that the large majority of GEDI footprints (>75%) with canopy heights from 2 and 5m fall within 0-5 degree slope, meaning that the potential risk of pulse spreading on high slopes and thus yield unreliable heights is minimal (Figure A3)."

=> This is not true, even on a flat surface. GEDI cannot reliably get down to 2 m. It's not just pulse spreading. The GEDI pulse width is 15 ns which is 4.5 m (i.e. the full-width half max is that). You can see from this value that it will be rare that you would believe you can consistently capture heights <

3m with a pulse-width that large.

Notes on the Revised Manuscript

60. As with last revision, "It" is ambiguous. Suggest you substitute "such estimates" for it.

74. its Global Dynamics Ecosystem Investigation, no LiDAR at the end of that which appears in multiple places

94. I find there is ambiguity throughout the manuscript with "AGB". AGB usually refers to a stock and AGBD refers to a density. You seem to use these interchangeably in the text and figures. Perhaps just define up front and let the reader know what is meant where?

104. I don't think it's clear what the number after +/- refers to. When we mean the standard deviation of a sampled distribution, we should specify somewhere S.D. so there is no ambiguity. When we mean the standard error of the sample mean of that distribution, say S.E. (i.e. the precision of the mean).

115. Can we make this more precise? To say "at least 10 m" means that the minimum value your observed was 10 m. Is that correct?

119. Same thing when we say "on average 10 m to 15 m lower" -> this range covers some set of averages?

265. We defined RH50 early in the paper as "hereafter median height", however RH50 continues to be used throughout the manuscript. Secondly, I still think there is confusion over what RH50 is. It is most definitely not the "median height of forests" if what you are referencing here is RH50. The median height of forests that were burned would be the median of all the RH98 values. Is this what is meant or do we mean RH50 recovered faster in this sentence? Remember that RH50 is an energy metric, it's the height at which the cumulative return is at 50%.

520. GEDI is *not* an instrument. GEDI refers to an investigation that uses an instrument. So, this sentence should really be something like "NASA's GEDI mission uses a lidar deployed on the ISS." I know this seems somewhat trivial but is more accurate.

521: The website date is wrong (so I understand why you report it!). GEDI took its last observations on March 16, 2023.

528: The current GEDI geolocation is 11 m (1 sigma) and has been for some time now.

539: Just delete "was returned by objects 20m above the ground" as this is not true. There may be nothing at all at 20m, it's just the cumulative return energy as you say to start the sentence.

550. Suggest we substitute for "(now version 1)" with "(we used version 1 in this paper)".

Referee #3 (Remarks to the Author):

I thank the authors for their revisions.

Author Rebuttals to Second Revision:

Referee #1 (Remarks to the Author):

General comments

Overall I think the paper is an improved version and a number of the previous issues have been dealt with. As I noted before I think the results are definitely interesting – the range of degradation impact into the intact forest is particularly interesting and far exceeds what is generally assumed. Same for the persistence, and also the covariance of the impact types. So I think this would definitely be of interest to a wide readership and has some important implications.

That said, I still think those key results could be explained much more clearly and simply – it takes quite a bit of time and effort to get to these. This is also reflected in the way results are presented in the abstract and even the title, which could definitely be punchier given their results (eg something like Human-driven degradation of tropical forests is much greater than previously estimated in both space and time). The overall narrative is disjointed by technical detail where it is not needed, as well as the many references to extended data. That needs a rethink – see my comments below. If the ED is so crucial to the point being made, then it should either be in the main text, or the main text needs to change to reflect those data.

We would like to thank the reviewer for stressing the need to improve the narrative and better present the results in the main and the abstract. We have addressed all specific comments that go in that direction (see below). In summary, we have followed the advice and have improved the structure and flow of the sentences, especially the ones with too many references to the extended data or supplementary information. We changed the title of the paper following the reviewer's suggestion. We have also removed sentences that were redundant with the Method section in order to get more directly to the key results. We simplified the titles of the results section emphasizing on the two new aspects of this paper, namely the extensive and persistence aspects of forest degradation. We bring forward the ecological explanation of the key forest structure metrics we are using at the very beginning of the paper. Following Reviewer #2 comments, we changed AGB into AGBD acronym throughout the manuscript. We have simplified the ED and moved non-essential figures and tables to supplementary information.

I have a question about the nature of the manually delineated data they used for classifying selective logging activities from S America, where there were no WRI concessions data. If this is not the same as the WRI data, then it is possible (perhaps likely) that the results for S America will be different to those for Africa and SE Asia; that is what is found in the analysis. So those manually delineated logging activities have to be shown to be comparable, otherwise any subsequent attributions of difference are not likely to be valid as they are based on different assumptions (hence systematic bias is quite possible).

Thank you for this relevant comment. We conducted a sensitivity analysis over Africa and SouthEast Asia by comparing the potential impact of using the WRI concession data against using a manually delineated approach onto our degradation/recovery results. We delineated a total of 697 polygons corresponding to selective logging areas in Central/West Africa and SouthEast Asia (Figure A1). We followed the same interpretation and delineation scheme of selective logging activities as we did previously for the Americas (i.e. Amazon basin and Guiana shield). The interpretation is based on the visual assessment of degraded forest pixels from the JRC TMF Transition maps that are depicting sequential trajectories of forest cover change from 1990 to 2022. The network of logging roads and felling gaps are usually classified as degraded forests in the JRC TMF dataset (i.e. as short-duration disturbances where the opening of the canopy after logging is only visible for a short time) and are key spatial features that allow to characterize selective logging activities¹⁻⁶. When delineating the polygons of selective logging we excluded areas corresponding to other land uses (e.g. shifting cultivation agriculture) that may appear temporarily as degraded forests in the TMF map.

This extended manual interpretation (to Africa and Southeast Asia) led us to observe that the WRI dataset has heterogeneous quality over Africa and Asia, with overall well-delineated polygons in West Africa and Kalimantan while in Central and East Africa and Sulawesi, and Papua, polygons were sometimes encroaching over other land uses such as small scale agriculture and human settlements. From the delineated selective logging polygons we use only the degraded forest pixels from the JRC-TMF dataset that appears within these polygons. This avoids specifically other types of forest cover changes in our analysis (e.g. deforestation is discarded). In addition to the fact that, as the reviewer rightly points out, the approach adopted in the Americas was not consistent with the two other continents, the WRI dataset in Africa and Asia integrates other types of land use other than selective logging. For instance,

the WRI polygons may include pixels of short term disturbances in shifting cultivation areas that can appear as degraded forests in the JRC TMF dataset.

Figure A1: Selective logging Polygons (red) in Central and West Africa, Indonesia, Malaysia and Papua New Guinea delineated on the basis of the spatial patterns of the degraded forest class (light green) of the JRC Transition Map. WRI concession polygons are shown in black. Note that the delineated polygons do not include other land use (e.g. urban, small-scale agriculture). Zooms A and B show in both cases an extract of area that is fully included in a polygon of the WRI dataset and that includes small scale agriculture (south of zoom A) and/or human settlements (North East of zoom B) Selective logging polygons that are delineated manually in our dataset appear as red lines.

The comparison of results from the two approaches (using WRI attribution scheme “Logged wri” versus the manual delineation approach “Logged manual” in Figure A2) shows that the trends in recovery after selective logging impacts for the three forest structure metrics (RH98 in panel B, Rh50 in panel C and AGBD in panel D) are consistent. However, we note that the WRI attribution scheme yields a larger immediate change in canopy structure after selective logging compared to the manual approach but the relative difference between the two approaches is similar for Asia and Africa. In other words, the impacts of logging in Asia remains higher than in Africa with the manually delineated dataset as it was with the WRI dataset. Impacts of logging in Africa using the manual approach are now similar to those in the Americas while they were larger using the WRI dataset (applied in Africa) which may be

explained by the inclusion of other types of land use as explained above (see new Figure 3). Finally, we notice that the TMF class of several logging events (named logging 2x+) is not represented anymore when using the dataset from the manual delineation approach. We interpret that this type of repeated disturbance corresponds to shifting agricultural use with short fallow cycle and is now excluded from our analysis of impacts of selective logging using the manual delineation.

b Forest degradation from selective logging (RH98)

c Forest degradation from selective logging (Rh50)

Figure A2: Comparative analysis of the impact of selective logging (with timing attribution to the JRC TMF dataset) between the use of manual delineations (“Logged manual”) and the use of WRI managed forest concessions dataset for Africa and Asia (“Logged wri”). Panels a to d refer to the number of sampled GEDI footprints, impacts of logging on RH98, RH50 and AGBD respectively for each continent and attribution scheme. Note that Americas is not shown here as not covered by the WRI dataset.

Action taken: In order to have a consistent approach on how selective logging is attributed over the tropics, as suggested by reviewer 1, we decided to replace the WRI concession dataset with the manually delineated approach for Africa and Southeast Asia. Figure 3 (see

below) was updated accordingly along with related figures for RH50, AGBD and count of GEDI footprints in extended data. Extended data Figure 4, where we quantify differences in RH98 and RH50 for forest classified as undisturbed in the JRC-TMF dataset located within and outside a buffer area (120 m radius) around logged or burned forest, was also updated (see below). We changed the paragraph in the Method section and it now reads as such (line 371):

“Regarding degradation due to selective logging, we performed an extensive visual interpretation and delineation of selective logging operations based on their specific spatial features visible on the JRC-TMF Transition Map. The selected degraded forest pixels correspond to temporary logging roads, logging felling gaps, decks, and skid trails. This dataset covers Brazil, French Guiana, Guyana, Cameroon, Central African Republic, Gabon, Congo, the Democratic Republic of Congo, Indonesia, Malaysia and Papua New Guinea. The managed forest concessions dataset from the World Resource Institute (<https://data.globalforestwatch.org/documents/gfw::managed-forest-concessions-downloadable/about>, accessed in February 2022), which displays concession areas allocated by a government for the harvesting of timber and other wood products in public forests, was used to guide the collection of polygons in Central Africa and SouthEast Asia while the delineation in the Amazon was generated from previous scientific experience^{4,7}”.

Figure 3 from the revised manuscript.

Extended Data Figure 4 from the revised manuscript.

Specific comments

Line 38-39: Little is known about structural changes in degraded forests Is this true though? The authors cite some of this work, but I think it's fair to say we do know quite a bit about this but it is usually regional rather than pan-tropical / comparative.

Longo et al 2020, water energy and c cycling, Amazon

Milodowski et al. 2021, vertical canopy structure, Borneo

Silverio et al 2019 forest. Fire, fragmentation and wind, Amazon

Stark et al 2020, structure and energy balance, Amazon

Dalagnol et al 2023, C emissions v degradation from NICFI Planet data, Amazon

Jayanthilake et al. 2021 Drivers of this (social science approach), global

Thank you for this comment. We completely agree that there are many regional studies that exist on this topic, and we have already referenced the most relevant ones in our introduction and results. Despite our agreement, due to word limitations, we decided to rephrase this sentence into (line 16):

“Tropical forest degradation from selective logging, fire, and edge effects is a major driver of carbon and biodiversity loss⁸⁻¹⁰. The rate of degradation is comparable to that of deforestation and exceeds it during extreme climatic events⁵. However, the actual extent of degradation impacts is likely underestimated, and large uncertainties remain in quantifying its persistence at global tropical scale¹¹”.

We acknowledge later on in the main text that this is an active field of research, the existence of regional rather than pantropical analysis, and we compare the results of these regional studies with ours.

Line 41 Not biomass measurements, estimates

We agree and rephrased the sentence as such (line 19):

“Here, we combine satellite remote sensing data (1990-2022) on pantropical moist forest cover changes⁵ with canopy height and biomass estimates from a spaceborne¹² Light

Detection and Ranging (LiDAR) to quantify the magnitude and persistence of multiple types of degradation on forest structure”.

Still slightly confusing degradation from ... edge effects. This isn't clear how this happens – it's not a single thing, but a complex mix of logging, development, fires etc.

We agree with the reviewer that human-driven forest degradation is driven by three main causes (selective logging, fires and edge effects) that are interrelated in space and time¹³. This is further explained in the main text (see revised introduction), and one added value of this manuscript is the attempt to disentangle these three causes using remote sensing products, and to quantify their individual and cumulative impacts on forest structure. Due to word restrictions in the abstract, we could not explicit the complexity of forest degradation and how its causes may overlap over space/time. We addressed this comment by rephrasing this sentence as follows:

“Here, we combine satellite remote sensing data (1990-2022) on pantropical moist forest cover changes⁵ with canopy height and biomass estimates from a spaceborne¹² Light Detection and Ranging (LiDAR) to quantify the magnitude and persistence of multiple types of degradation on forest structure”.

Line 44 – showing understory regrowth ... the implication is that it is anthrop disturbance that is showing this. Is this per area, or where impact on H is highest, or ? Not clear.

Given that more explanations are given in the main text, we decided to remove this part of the sentence from the abstract to avoid confusion and emphasize on the persistence of degradation impacts. It now reads as such (line 22):

“ We show that selective logging and fire decrease forest height by 15 and 50 % respectively, and show low rates of recovery even after 20 years”.

From the abstract and lower down the key results seems to be the persistence of these impacts away from the edges (but need to define persistent impacts eg > 20 years) and the underestimate of prev edge impacts. If so, the title and abstract could be better framed eg something like: Human degradation of tropical moist forests is far more extensive and persistent than previously estimated

We thank the reviewer for this comment. We agree and like the suggestion of this new title, summarizing the two main results and the way to analyze and quantify tropical forest degradation. We have modified the title of the manuscript as suggested: “*Human degradation of global tropical moist forests is far more extensive and persistent than previously estimated*”.

We also followed this recommendation to modify the abstract and emphasize on these two aspects. For instance, the need for study has been rephrased as such (line 18):

“However, the actual extent of degradation impacts is likely underestimated, and large uncertainties remain in quantifying its persistence at global tropical scale¹¹”

Note that the subtitles of result sections 2 and 3 have also been modified into “*Magnitude and scale of edge effects*” and “*Persistence and cumulative impacts of forest degradation*”, respectively.

Line 121/2 – can degradation occur at the same rate as deforestation if they are essentially quite different things?

Forest disturbances due to selective logging and fire (and to natural disturbances to a lesser extent) that do not imply a change in land use (detected over a short time period compared to long term forest conversion) are occurring at a similar rate as deforestation according to Vancutsem et al. 2021⁵. Tables 4 and 5 from Vancutsem et al. 2021 show for the humid tropics that total forest degradation (followed or not by deforestation) is slightly lower than deforestation (with or without prior degradation) in 2000-2009 (7.29 Mha vs 8.31 Mha) but is higher in 2010-2019 (6.09 Mha vs 5.51 Mha). Similar results are found in Bullock et al.⁶ in the Amazon (1995-2017) or in Shapiro et al.² for the Congo basin (2015-2020). Additionally, if we also consider the increase of forest edge area (indirect consequence of deforestation and habitat fragmentation) as reported by Fischer et al.¹⁴ for the Tropics or Matricardi et al.¹⁵ in the Brazilian Amazon, then we would assume that overall degradation (from logging, fire and edge effect) may surpass deforestation rate.

Note that we are here referring to degradation from logging/fire, we both consider degradation not followed by deforestation (i.e. that stays forest in 2022) and degradation followed (after 7.5 years on average across the pantropical belt) by deforestation. The latter represents around 45% of total new degradation proving that degradation and

deforestation can be linked. This is actually why we have done this analysis in the last result section « Fate of degraded forests ».

To make the distinction between deforestation and degradation clearer to the reader, we rephrased this sentence as such (line 35):

“ In addition, degradation from selective logging, fires, edge effects or a combination of these disturbances, is affecting forests and their capacity to provide ecosystem services at a rate comparable and in some years larger than deforestation^{5,13,16}”

The interrelation between degradation and deforestation is now more explicit and placed in the introduction (while it was presented at the beginning of result #4 in the previous version) (line 39):

“ Furthermore, degraded forests are more vulnerable to additional disturbances like climate extremes, reducing their potential resilience and threatening their long-term future^{17–20}. In 45% of cases, forest degradation represents the initial stage that eventually leads to deforestation⁵.“

Edge effects are a tricky thing to wrap up into the general discussion of degradation as they are co-varying (driven by, exacerbated by) with the other effects. This part has never quite been clear in the different versions. It is explained well in the Methods, but ideally needs really carefully laying out here in the main text ie Here we define edge effects as these will increase in response to and so are closely-dependent on

As suggested we used the edge effect definition from the Method to clarify this section. It now reads as such (line 35):

“In addition, degradation from selective logging, fires, edge effects or a combination of these disturbances, is affecting forests and their capacity to provide ecosystem services at a rate comparable and in some years larger than deforestation^{5,13,16}. Here we define edge effects as changes in forest structure and functionality occurring at forest edges, driven by habitat fragmentation²¹. “

Line 125 – moreover ... is this needed? Can you have degradation without deforestation or vice versa? If so, then prob need to state this either way ie are they inextricably causally linked and which way(s).

According to Vancutsem et al.⁵, around 45% of degraded forests end up being deforested so forest degradation and deforestation may be linked in some cases. Their conclusions actually motivated us to look at the link between the degradation state of forest canopy structure and their subsequent land use (deforested or forest recovery).

We now state clearly this potential causality between degradation and deforestation in the introduction:

“ Furthermore, degraded forests are more vulnerable to additional disturbances like climate extremes, reducing their potential resilience and threatening their long-term future^{17–20}. In 45% of cases, forest degradation represents the initial stage that eventually leads to deforestation⁵.“

Line 131 – not a clear sentence.

We meant *“About ¼ of current deforestation emissions are offset by the carbon absorption of current degraded and secondary forests”*. Due to word restriction for the introduction (~500 words) we decided to delete this sentence. This result from Heinrich et al. 2023 is introduced in the conclusion of the manuscript (line 303): *“Although the current areas of fast-regrowing forests allow offsetting ~25% of carbon loss from deforestation”*.

Line 150 – remains localized and limited in access – what does that mean? If you mean coverage, then these are just the same thing. I'd argue that thousands of ha that can be covered by ALS is not so local. It's not pan-tropical but can be regional eg see the CAO work in Peru. If you mean cost and availability (access) then that is surely changing – nearly everywhere has some ALS systems available either from state, local gov or commercial or some combination.

We completely agree with the reviewer and decided to delete this sentence to avoid any confusion and erroneous statements. This paragraph has been rephrased as such (line 47):

“ Despite the advancements in remote sensing capabilities for assessing carbon fluxes associated with each type of disturbance^{22–26}, a pantropical assessment of forest degradation on forest structure is still lacking. Furthermore, regional studies likely underestimate the extent and impact of forest degradation²⁷, particularly regarding the depth of edge effect penetration within forest interiors²⁸.”

Note that this section has been greatly reduced to fit the journal guidelines.

151 – what is small-scale?

Small-scale refers to forest disturbances of less than 0.09 ha (less than the size of a Landsat pixel, Landsat images being the main source of forest cover change products at pantropical and global scale). This sentence was removed with this revision but we added this precision in the result section #2 (Magnitude and scale of edge effects) (line 153):

“ These results are not only due to localized edge effects from degradation but also to the omission in the Landsat-based forest cover change datasets of small-scale (<0.09 ha) and low-intensity disturbances at the interface between ‘undisturbed’ and degraded forests²⁹, highlighting the added value of LiDAR-based assessment to compensate for the intrinsic limitations of optical sensors in detecting these phenomena.”

Lines 158 on - Use of pan-tropical scale, global, near-global-scale in different places and in different ways. These don't really mean anything – it's either pan-tropical or it's not. Neither of these are global or even near-global. Then it switches to global tropical moist forest. Just say tropical MF, or pantropical. There are many different uses of the word scale throughout without clear definition or context, which is confusing.

We modified it into “*pantropical scale*” as suggested and used it consistently throughout the manuscript.

Line 159 – overall? Meaning what?

We agree this term was not clear and we simplified the sentence as such (line 55):

“ Here we provide an assessment of the impacts of human-induced degradation on global tropical moist forest structure and of the forest's ability to recover to its pre-disturbance condition”

Line 162 – now it's tropical humid belt (see above). Needs to be consistent terminology throughout.

We agree and this term has been homogenized throughout the manuscript.

Line 170 – sample dataset. I know what is meant but this suggests a kind of test dataset. Better to phrase as a spatially discontinuous or point-based.

We modified this accordingly and it now reads as such (line 63):

“Within this scope, we combine a wall-to-wall dataset on forest degradation, deforestation and regrowth dynamics derived from Landsat imagery at 30 m spatial resolution⁵ with spatially discontinuous estimates of forest canopy structure from GEDI ”

Line 175 – you need to explain what RH98 and RH50 are here as these are your critical metrics and non-experts will have no idea what they actually represent. They are defined in the methods, but this is critical info and so should be summarised here. Also you then refer to RH98 as the proxy for canopy height – fine, but then don't use top height and other terms elsewhere.

Thank you for this suggestion, we now explain clearly what the three metrics represent in this section of the paper, allowing the reader to grasp the ecological meaning of these structural metrics before reaching to the result section (line 65):

“ We jointly analyze the canopy height (RH98), i.e. the top of the canopy, or the nearest tallest vegetation in the footprint; the height of median energy (RH50), describing the vertical distribution of canopy elements and gaps³⁰; and the Above Ground Biomass Density (AGBD) representing the above-ground woody biomass per unit area³¹”

We homogenized the term canopy height when referring to RH98 throughout the manuscript and deleted “top canopy height”.

Line 177 – I don't think you can argue this is comprehensive. It doesn't for eg give leaf area density, n layers, gap fraction etc. It's definitely a good set of metrics, but

We fully agree and have deleted this sentence to avoid redundancy. This set of metrics is useful to capture different aspects of forest structure. See the point above with the revised sentence.

Line 208 – I think intact needs defining given all that is to come, particularly as you are defining disturbed/degraded relative to this (eg impact distance from an edge). Are forest edges 120 m width, then how far does the other side of this extend before this becomes eg a patch? And is there some linear extent?

We agree with the reviewer on the importance of defining intact forests early in the narrative of our manuscript. We revised this sentence and reads as such (line 71):

“ The analysis of more than 23 million sample footprints of GEDI data over intact tropical moist forest, i.e. areas with no signs of human activity detected during the last three decades and located at least 3 km from forest/non-forest edge (see Methods), reveals regional and continental variability in forest structure, with the tallest canopies found in Insular Asia, Western Africa and Western and Eastern Amazonia ”.

For figure 1 and 3, forest edges are 120m width between other land cover (deforested land for agriculture and road/settlement/other uses) and forest (undisturbed or degraded) and this is a conservative distance where impacts from edge effects have been quantified in terms of biomass loss²¹⁻²³.

Note that we masked out natural edges by discarding transitions from water surfaces to forest. In this new revision, we added another methodological step to mitigate the inclusion of natural edges (e.g. transition forest/savanna) in our analysis. We computed the total area of deforestation detected from 1990 to 2022 and the area of forest cover in 1990 to yield a percentage of deforestation from the JRC TMF dataset for each grid cell (1.5 degrees). We then discarded grid cells showing a value of accumulated deforestation (1991-2022) compared to forest area 1990 less or equal than 2% (i.e. 0.7-0.1% percentage of annual deforestation over the last 20-30 years) which is a conservative value compared to FAO FRA numbers of

gross deforestation rates over the tropics. The Method section and Figure 1 have been updated accordingly.

What lies beyond this distance of 120m is not of interest for Figure 1 and Figure 3 and related analysis. Multiple slices of distance from forest core to the edge are analyzed in Figure 2 to assess the extent of edge effects. Looking at the spatial configuration of forest landscape (e.g. shape, connectivity, patch size) is not analyzed here although it is interesting and could explain the variability in the magnitude of the impact of edge effects on canopy structure. We have mentioned that in the discussion of Figure 2 (line 127):

“The variability in the magnitude of change and extent of the edge effect within each continent is likely to be linked to forest structure composition and environmental conditions^{23,28,32} but also to different landscape configurations³³ (e.g, forest patch size and connectivity) and composition (e,g, proportion and type of agricultural land) that can increase forest sensitivity to edge effects³⁴”.

Line 210 – highest canopy heights – can you just say tallest canopies? Then, overall canopies are taller

We agree and have modified the text as suggested.

Our results show that the mean canopy height and AGB densities of degraded forests are at least 10 m and 100 Mg/ha lower, respectively, than intact old-growth forests (Fig.1b, Extended Data Fig 3-4).

This is of course what we'd expect – by definition almost - but the interesting part is quantifying this.

Thank you for this comment, we modified this sentence in order to emphasize the quantification aspect rather than the description (line 84):

“In fact, we quantify that the minimum difference between intact old growth and degraded forests is 10m for mean RH98 and 122 Mg/ha for mean AGBD ”.

In figure 1, why mask out the hexagons where there is no difference between intact and other forest types? This is a different category from < 600 footprints, but both are masked out. Also, that sentence needs to be rephrased as Statistically significant differences in canopy height

We decided to mask out hexagons where there is no statistical significance between intact forest and the other forest types to simplify the visualization of the map. In a previous version of the figure (Figure A3), we used to add a dot within each grid cell to indicate the result of this statistical test:

Figure A3: First version of figure 1 from the initial submission of the manuscript

Based on several reviewer's comments made on the readability of this figure, we decided to simplify it. Note that "100%, 99% and 99% of the hexagons for degraded, edge and regrowth forests, respectively were statistically significant." meaning that masked grids are mainly driven by the condition of having a minimum of 600 GEDI footprints per grid cell for each forest type (which also depends on the forest type occurrence in the JRC TMF dataset).

Regarding the structure of the sentence, we rephrased it as recommended (line 97):

“Grid cells with less than 600 GEDI samples or with no statistically significant differences in canopy height between intact and non intact forests were masked (Welch t-test $p < 0.05$)”

Line 226 Regrowth forests 10m to 15m lower RH98 ... ? Which is it? Or if it's different based on location, state this. Same for lots of the other statements like this – the range is confusing as it implies uncertainty, rather than comparing specific regional differences.

We refer here to forest regrowth on deforested land across the pantropical belt i.e. a two-phase transition in the TMF dataset from moist forest to (i) deforested land and then (ii) vegetative regrowth. A minimum 3-year duration (2020-2022) of permanent moist forest cover presence is needed to classify a pixel as forest regrowth (to avoid confusion with agriculture). This is further explained in the Method section.

We agree that the range is confusing and we now report the average difference with intact old growth forest at the pantropical scale for RH98 and AGBD. Note that we have applied this correction throughout the manuscript modifying a range with continental or pantropical averages. We modified extended data tables 1 and 2 accordingly in order to give access to these descriptive statistics for each forest type analyzed here. The sentence has been modified and reads as such (line 88):

“Regrowing forests on deforested land have, on average, a 16 m lower RH98 than intact forests, and an average AGBD of 80.4 Mg/ha \pm 87.3 (Fig.1d).”

Line 315 Distance to edge impact in the Americas up to 7 km – this is a major result. I'd mention this in the abstract. Also, say where this is (not just the Americas, but specifically). And it IS greater isn't it (not may)? The impact on AGB is not similar scale though (line 317) – it's a lot smaller, and a different pattern (much more similar between continents).

We thank the reviewer for this comment. We performed one-way ANOVAs for each continent followed by a post hoc Tukey test to evaluate differences in RH98 (resp. RH50 and AGB) between the undisturbed forest located at various distances to the forest edge and the

obtained scale of edge effect differs from the other approach where we identify the distance at which the RH98 of the edge forest reaches 95% of the RH98 of the intact forest (more conservative approach). There are synergies in the results of the two approaches except for the Americas (Fig A4). This may be due to the higher variability of the extent of edge effect at sub-continental level.

Figure A4: Canopy height (RH98) of undisturbed forests (located at more than 120 metres from degraded forests) and all forests (including undisturbed and degraded forests) at various distances to the forest edge separating forest cover from agricultural land and other land covers. Pasting the results of the ANOVA and Tukey in figure 3 blurs the message and makes the figures unreadable, Tukey tests are provided as supplementary files due to visibility reasons.

We decided to prioritize the conservative approach when reporting the scale of the edge effects in the abstract. This approach is also coherent with the one developed by Chaplin-Karmer et al.²⁸ for the tropics where they used a threshold of 90% of the estimated asymptotic

biomass (i.e. the average biomass density has reached 90% of the asymptotic density) to assess the scale of edge effects.

To clarify the fact we use two methods to assess the scale of edge effect, we added in the text the following paragraph:

“ We used two indicators, the first, and more conservative, is the distance at which the RH98 of the edge forest reach 95% of the RH98 of the intact forest, and the second is the distance at which the differences in RH98 between edge and intact forest are not any more significant based on an ANOVA test (See Methods).”

To answer the second question, we mapped the scale of edge effects (i.e. the distance at which the RH98 of the edge forest reaches 95% of the RH98 of the intact forest) at the grid level for all cells where we have GEDI data over intact forest and the 17 classes of distance to edge, and where we estimated a minimum accumulated deforestation (1991-2022) compared to forest area 1990 of 2% to target anthropogenic edge effect (following deforestation and fragmentation). This map now constitutes panel c of Figure 2 and reinforces one of the main messages of the manuscript showing how extensive edge effects can be at a subcontinental scale:

New Figure 2c

Finally regarding your third point on the scale of edge effects for AGBD, our final results show that we detect differences between intact forest AGBD and edge forest AGBD up to 1000m in South America, 750m in Africa and 1020m in Asia, on average. We have modified this sentence and it now reads as such (line 132):

“ We detected larger scales of edge effects on AGBD in the Americas (1000m) and Africa (750m) and lower scales in Asia (1020m) compared to RH98 (Extended Data Fig.3). These results show that the edge effect on biomass is far exceeding the 120m over which have previously been measured^{22,23,35,36}”

Line 333 – small-scale again. Meaning?

We added a precision on the maximum size of Landsat-based detection of forest disturbances (line 154):

“ These results are not only due to localized edge effects from degradation but also to the omission in the Landsat-based forest cover change datasets of small-scale (<0.09 ha)... ”

And these numbers 15-25%, 30-40% RH50, 25-40% AGB – see my comment above re line 226 - are these variations those between continents? If so, you should probably say so as otherwise it doesn't make sense given you have discussed differences between continents elsewhere.

Thank you for this comment, these numbers were referring to an intercontinental range in the average value of RH98 (resp. Rh50, AGBD) at the forest edge. We see that it could bring confusion and we have now rephrased this sentence as such:

“ We detect the cumulative impacts of selective logging and forest fire up to ~1.5 km into forest interiors and we quantify an additional 10% decrease in RH98 on average compared to the distribution of forest affected by edge-desiccation effects only ”.

Fig 2: A lot of info here and not all of it clear. The colorscale of the vertical bars in 2a just aren't clear enough – lines would need to be much thicker. Solid lines – I think this is the loess fit (seeing the caption) but there is no blue line – previous version? Why is loess fit necessary? If the fig was clearer you could just use the points & don't need some arbitrary line fit, which seems to have been added just to help navigate the dots. It's not a model prediction / interpolation so don't use. Same for the inset – what do these lines represent? Also the red dotted horiz line referred to later is almost invisible, but this is important information.

Some similar issues on fig 2b. Can't see the intact line. Also these are histograms so why have lines (again)? they should be bars with errors or even box-whisker would be better. Overall this figure is important but it needs to be clearer. If the 2b histos could be overlaid on a single plot that would help compare one continent to the other also.

As suggested, we have increased the size of the standard deviation bars and the dots representing the mean. We deleted the loess fit line. We darkened the intact forest horizontal line and changed the vertical turquoise line into a red-dotted one that matches the horizontal red-dotted line (now thicker). We kept the fit line in the inset as it helps visualizing the proportion of forest fire and selective logging activities at various distances to the edge. For Figure 2b, we deleted the intact forest distribution as it was visible enough. We thickened the lines and decided to keep a line distribution as we have 17 groups (17 distances to the edge) to display and a bar plot or box-whisker would not be visible. Same reason that made us have a plot per continent.

The updated Figure 2 is shown below:

a

Edge Effect (RH98)

b

Tree height distribution of undisturbed forest (RH98)

c

Scale of edge effects

New Figure 2: **Spatial scale and magnitude of edge effects caused by deforestation.** Canopy height (RH98) of undisturbed forests (located at more than 120 meters from degraded forests) and all forests shown in grey (including undisturbed and degraded forests) at various distances to the forest edge separating forest cover from agricultural land and other land covers (a) average distribution of undisturbed forest (located at more than 120 meters from degraded forests) canopy height at various distances to the forest edge (b) scale of the edge effect represented as the distance from forest edge at which RH98 reaches 95% of intact forest RH98 (c). The colours for the undisturbed forest at various distances from the forest edge in panel a correspond to those in panels b and c. Gray cells in panel c represent cells where the accumulated deforestation (1991-2022) compared to forest area 1990 is lower than 2%. The inset on panel a represents the degradation area due to fire (red curve) and selective logging (purple curve) calculated at various distances to the edge. The red dotted vertical line is placed at a distance equal to 350, 400, and 1500 m for America, Africa, and Asia, respectively, and corresponds to the distance between the forest edge and the point at which 95% of intact forest RH98 is reached (red horizontal dotted line). Vertical bars indicate the spatial standard deviation. The number of GEDI sample footprints for each distance to the forest edge is reported in Supplementary Fig.4. F represents the F-Value in ANOVA and asterisks indicate the level of statistical significance for ANOVA: * $p \leq 0.05$, ** $p \leq 0.01$, *** $p \leq 0.001$, **** $p \leq 0.0001$, ns stands for not significant. Tukey post-hoc tests are available in supplementary files 1-3.

474: you start using top canopy height here when you haven't defined or used it before.

'Top canopy height' has been replaced by 'canopy height' throughout the manuscript.

Line 486: twice as strong as logging alone. And how long time series would you need? I'd leave this sentence out, or you need to justify what you mean here w some evidence on what 'longer' means.

We deleted both sentences on the immediate and long term impacts of logging followed by fire due to the low representativity of this class (limited number of GEDI footprints for the three continents). We updated the method section accordingly. We now stress in the conclusion of the manuscript the need to develop dense and long time series of forest recovery post-disturbance which relies on 1) the long-term monitoring of forest cover from multi-source remote sensing and 2) the densification of forest structure measurements from ongoing (e.g.

GEDI or Advanced Land Observing Satellite -ALOS) and planned (e.g. ESA BIOMASS and NASA-ISRO Synthetic Aperture Radar -NISAR) missions.

This part of the conclusion reads as such (line 300):

“ We show that the cumulative impacts of selective logging, fires and edge effects have significant long-term effects on the structure of global TMFs, but as the 30-year period of our study is too short to observe a full recovery of the forest structure for most types of forest disturbances and regions, future studies should further address this question. ”

Line 493: half that of?

We changed the sentence accordingly.

Lines 498-9: do you know they are not representative? If not, leave this out.

Agreed, we deleted this part, it reads as such (line 247):

“However, the slowdown in regrowth rates of AGBD after 10 years of regeneration may indicate that several drivers are affecting forest growth and are not captured by Poorter et al. (Supplementary Files 4-7).”

Line 499: a complete time series? Not sure what that means (annual?) – implies measurements all the time which is definitely not the case here, so leave this out.

Agreed, we delete this part.

Fig 3 – also a bit unclear. You are referring to top canopy height again but not defined. Also the panels say one thing eg Forest Edge then (RH98) – so do you mean the impact of edge on RH98? If so, just say that. Same for others. And the line fits again. Not needed, and not even consistent eg missing on Americas panel in line b, and the line fit for 1x logged in Asia has a huge jump of 8 years! The logged + burned points in these plots are impossible to distinguish, except for 2 in the Americas panel. They are either too similar in color to the logged 1+ or not there.

We modified this figure as suggested. We changed the labels and deleted the fit line and the classes where only a few points were shown (logged 2x+ and logged then burned). Here is the final version of this figure:

New Figure 3 : Impacts of forest degradation from selective logging, fire and edge effects. Long-term impacts on canopy height (RH98) from edge-desiccation effect (2a), degradation (fire or logging) of edge forest (2a), selective logging (logged 1x corresponds to logged once over the last 3 decades), fire (2b) and secondary forests regrowing on abandoned deforested lands (2c). Results are reported as the percentage of intact forest canopy height (solid line) after normalising the difference in canopy height (RH98) within each grid cell between intact forest and each forest type (degraded, edge forest, regrowth) and age. Vertical bars indicate the spatial standard deviation. GEDI samples for each

disturbance type and related time since disturbance are reported in Supplementary Fig.6. F represents the F-Value in ANOVA and asterisks indicate the level of statistical significance for ANOVA: * $p \leq 0.05$, ** $p \leq 0.01$, *** $p \leq 0.001$, **** $p \leq 0.0001$, ns stands for not significant. Tukey post-hoc tests are available in supplementary files 5-7. Underlying data is available in supplementary file 4.

Line 541 – this is a much clearer statement of the problem, which should come right at the top.

We agree and we introduced this sentence in the introduction. It now reads as such (line 39):

“ Furthermore, degraded forests are more vulnerable to additional disturbances like climate extremes, reducing their potential resilience and threatening their long-term future^{17–20}. In 45% of cases, forest degradation represents the initial stage that eventually leads to deforestation⁵.“

Line 543

Here we show using three years of GEDI data (more details in the Modelling deforestation risk section of the Method) that degradation is also a determinant of future deforestation, where the probability of full removal of forest cover and change in land use increases with the degradation stage of forests.

You can just say, here we show that degradation is (and then refer to Methods at the end).

Thank you, we changed this sentence accordingly (line 268):

“ Here we show that degradation also plays a crucial role in predicting future deforestation, where the likelihood of total deforestation and land-use change increases with the degree of forest degradation.”

This is an issue throughout. There is a lot of reference to Extended Data, Methods etc that is quite distracting from the flow and the overall message. If they are important enough that they need to be referred to throughout then they are not ED. I think some more

consideration is needed in terms of what is left in/out in terms of what the key results are and what needs to be in ED or left out altogether. See my comments on that part below also.

We thank you for pointing this out. We have greatly reshaped this whole result section, deleting unnecessary references to the extended data figures/tables or to the supplementary information section. We have improved the flow of sentences not only in this result section but throughout the manuscript and invite the reviewer to look at our track change version of the revised manuscript. Note that the extended data contains less than 10 figures (and 2 tables) compared to 17 figures and 4 tables in the previous version.

Line 547: recently degraded? Same w line 549 – this whole bit is hard to follow and needs rephrasing eg Forests that are recently (within X years) degraded and are then deforested within X years, showed reductions in mean ... This corresponds to ‘severe’ impacts from unsustainable logging and/or fire (where severe is defined as)

We have rephrased this whole section as recommended, it now reads as such (line 267):

“The stage of forest degradation is linked to the type, intensity, and recurrence of past disturbances, as well as to the time since the previous disturbance. Here we show that degradation also plays a crucial role in predicting future deforestation, where the likelihood of total deforestation and land-use change increases with the degree of forest degradation. Our results indicate that degraded forests followed by recent deforestation (2020-2022) have significantly lower canopy heights and AGBD compared to those not subjected to deforestation (Extended Data Fig.8, Supplementary Fig.7). On average, degraded forests followed by deforestation experienced severe impacts, with average reductions in RH50, RH98 and AGBD of 60%, 45% and 65%, respectively. These impacts are likely due to unsustainable logging and/or fire, as shown in Fig.3. Moreover, these structural parameters have a large spatial variability ($\pm 12.8\%$, $\pm 13.3\%$ and $\pm 14.6\%$ for RH50, RH98 and AGBD respectively), reflecting the complexity of the degradation processes and underlying factors driving deforestation in the tropics³⁷”

Line 555: see comments above. Using Generalised Linear Models (GLM) (see the section on statistical analyses in the Methods) we confirmed that ... why do you need that qualifier? Just state Parameters related to ... were shown to be strong predictors ...

We changed this sentence accordingly (line 278):

“ We found that forest relative heights (RH50, RH98) and distance to the edge were strong predictors of the probability of deforestation (Extended Data Fig.9, Supplementary Fig.8). Degraded forests in America showed, on average, a higher deforestation risk than in Africa or Asia, as 50% of deforestation probability was reached when forests lost 50% of their initial heights (60% in Africa and Asia). ”

Line 560: again, this section is not clear and needs clarifying – what confounding effects, and is this different to previous studies?

We rephrase this sentence as such (line 282):

“Furthermore, proximity to the forest edge, recognised in previous research as a key factor in assessing deforestation risk³⁸, showed complex interactions with canopy height in degraded forests. This observation highlights the interplay between different factors such as degradation, exposure to human activities and edge-desiccation effects within the first kilometer from the forest edge, contributing to an increased likelihood of subsequent deforestation.”

Fig 4 – if you were looking to leave something out, I would suggest this – it is interesting but it doesn't really need a whole figure as you've already stated the main result in the text. A more interesting figure would be the map of where this puts the most vulnerable to deforestation places & even a statement on that eg from our results, the top X most vulnerable locations are X, Y and Z.

As suggested we have moved this figure from the main to the extended data figure section as the text was self-explanatory.

We have also deleted the sentence *“The combination of the JRC-TMF for monitoring disturbance history and GEDI datasets on canopy heights⁷⁰ and AGBD provides a very promising solution to identify those vulnerable forest areas.”* which suggested that a combination of degraded forest from JRC-TMF (or other forest disturbance mapping system) and a map of continuous canopy heights (e.g. Gridded maps from Lang et al.³⁹ or Potapov et al.⁴⁰ using GEDI and Landsat or Sentinel as input) could lead to an assessment of most vulnerable forests to deforestation. This was an overly simplified message. We show that

deforestation trajectory may be explained partly by forest canopy structure and degradation history. These two aspects are the added value of our study compared to other studies predicting deforestation.

We could technically provide a map of forest vulnerability to deforestation using: 1) all degraded forests from the JRC-TMF (still standing and mapped as forest in 2022), 2) a gridded map of canopy height at medium resolution (10-30m) and 3) mean value of intact forest RH98 (derived from this study). However, this map would be prone to a lot of criticism. Besides the fact that we would need to ingest new data (i.e. gridded map of canopy height from Lang et al.) and propagate the error that comes with it, predicting deforestation would be anyway much more complex. A whole range of anthropogenic and natural drivers would need to be considered and this is to our view beyond the scope of our study.

Line 591 You mean GEDI samples not plots. Also, what is the intention of using seemingly large numbers like this (see above also) – are they meaningful? If you're using a Sentinel-2 image it has 125 M samples (or pixels as they are usually called). Nobody ever says 'we processed X trillion' samples of Sentinel or Landsat data, but it's effectively the same thing. So unless there's a specific reason for using this number, leave it out. A more meaningful number might be eg the total ha of TMF covered by GEDI samples out of the total in each continent, but that wouldn't sound so impressive.

We agree and have removed the number of GEDI footprints used. Here is the revised sentence (line 291):

"Our study demonstrates that the integration of recent and spatially sparse spaceborne LiDAR observations (i.e. GEDI), with long-term and spatially continuous spaceborne optical datasets (i.e. JRC-TMF) provides a novel approach to assess forest degradation and recovery at the pantropical scale"

Line 594 – this could be clearer. We show that the spatial extent and magnitude of degradation impacts on canopy structure are greater than aboveground biomass doesn't need all caps.

Then – the impact of edges on forest vertical structure was previously assumed to extend no more than ~ 100 m into intact forest. Our results show this is a significant underestimate, and that

Thank you for your suggestion, we revised the text accordingly (line 294):

“ We show that the magnitude of degradation impacts on canopy structure are greater than previously reported, ranging from a 20% to 80% decrease in canopy height and above ground biomass density. The impacts of edges on forest vertical structure were previously assumed to extend no more than $\sim 100\text{m}^2$. Our results show this is a significant underestimate with edge effects measured up to ~ 1.5 km into the forest interior implying that the overall spatial impact of fragmentation across the pantropical belt is severely overlooked by a factor of over 200%.”

Line 610: these terms are a bit vague. Can you either define them or just leave as ‘spatially explicit, at scale of X m, over Y ha’?

We agree and decided to simplify the sentence (line 309):

“ This type of spatially explicit information on tropical forest degradation is crucial for implementing more effective forest-based mitigation policies⁴¹ and conservation activities agreed under the UN Conventions on Climate Change and Biodiversity^{42,43} (UNFCCC and CBD). ”

Line 620: might want to revise in light of the re-deployment plan. Also, is this really all that relevant – seems more like context. Could just start from Line 624. I think this could also be shortened quite a lot – stick to the actual methods used here.

Thank you for this comment, we have revised this introductory paragraph of the method section and have greatly simplified it as recommended:

“ In this study, we use the spaceborne Global Ecosystem Dynamics Investigation (GEDI¹²) from NASA to analyze the extent of forest degradation on canopy structure at pantropical scale, but its short lifetime limits long-term monitoring. To overcome this limitation, we combine GEDI data with long-term information on forest dynamics from Landsat using a space-time substitution strategy. This allows us to also study the effects of different types of forest degradation and regrowth. While this approach has been used in previous studies⁴⁴, it assumes that differences in neighbouring land characteristics can be used as a proxy for changes over time and that climate and vegetation remain relatively constant over the 20-30-

year analysis period. For example, when studying forest recovery, we assume that different height metrics from GEDI represent different ages since the last disturbance.”

Don't really need any of the detail lines 669 to 680 as that is described elsewhere.

The details on how we build the class intact forest are presented briefly at the beginning of result 1 of the paper however not all details are provided. Here we decided to simplify the description of intact forest and keep essential methodological details. It reads as such (line 352):

“Intact forests undisturbed forest (forest without any disturbance observed over the Landsat time series) located at more than 120 m (4 Landsat pixels) from degraded forests and more than 3000 m from the forest/non-forest edge. Note that we cannot consider intact forests as primary forests, as the Landsat observation period is too short to discriminate between primary forests that have never been cut and secondary forests that are older than the observation period.”

Line 717; Do you mean complemented / augmented? And what does extensive visual interpretation mean? How accurate are these data and have they been validated and if so how? This is quite important as it suggests you are using a different scheme for S America than you are for Africa and SE Asia. If so, could that explain any of the differences you observe for SA v Afr v SEA? Is there a way to show that if you did the same for eg in Congo, Gabon etc that you get the same / similar results as in the WRI data? And that if different people did the analysis they would get the same results?

This has been fully addressed above (our detailed response to your 2nd general comment).

Lines 850-857 I think this is what needs to go into the main section to define what RH98 and RH50 actually are. This is where you also define RH98 as canopy height, but then go on to refer to top canopy height in a number of places still.

We agree and have added at the end of the introduction (which has been reshaped and pruned to fit the journal guidelines in terms of length):

“ We jointly analyze the canopy height (RH98), i.e. the top of the canopy, or the nearest tallest vegetation in the footprint; the height of median energy (RH50), describing the vertical distribution of canopy elements and gaps³⁰; and the Above Ground Biomass Density (AGBD) representing the above-ground woody biomass per unit area³¹. ”

As previously said, ‘top canopy height’ has been modified into ‘canopy height’ throughout the manuscript.

Lines 928-32: why these numbers (7, 600, 30)?

These numbers have been established after sensitivity analysis at the beginning of the analysis to ensure a balanced representation of observations for subsequent calculations. These sensitivity analyses were presented during the first round of review (to referee #2). Please find below a summary of how these numbers were set (note that the layout of the figures presented below corresponds to the old version of figures 2 and 3):

- **Why 600?**

We implemented a subsampling procedure across grid cells of approximately 1.5 degrees, covering the study area. This involved selecting a sufficiently large random sample that accurately represents all forest disturbance classes. Grid cells with an insufficient number of observations were excluded from the subsampling procedure, with a minimum threshold of 600 observations set. By applying this threshold, only a negligible fraction of forested areas across the pantropics were excluded (for Intact forest only 46 gridcell - out of a total of 658 - were excluded), while still assuring statistical robustness to the analysis.

Fig. A5 to A8 show the distribution of the number of GEDI shots per gridcell for Intact Forest, Forest Edge, Degraded Forest and Forest Regrowth. While it is generally possible to raise this threshold to 1200 samples for Intact Forest, Forest Edge and Degraded Forest, for Forest Regrowth we might encounter some issues due to the scarcity of samples especially in Africa. In other words, higher thresholds are reducing the number of acceptable grid cells. Results of this loss of information are also reported in Table A1, where we reported the percent decrease in the study area as a function of the minimum sample threshold adopted. Cell in green in Table A1 shows the high value of decrease in study area covered by selected grids when a selection criterion of a minimum of 1200 samples is adopted over Asia.

Figure A5: Histogram of the number of GEDI samples per gridcell for Intact Forest. Vertical dotted lines are at 300, 600 and 1200 samples per cell.

Figure A6: Histogram of number of GEDI samples per gridcell for Forest Edge

Figure A7: Histogram of the number of GEDI samples per gridcell for Degraded Forest

Figure A8: Histogram of the number of GEDI samples per gridcell for Forest Regrowth

Table A1: Percent decrease in study area covered by selected grids when a selection criterion of a minimum of 600 and 1200 samples is adopted

	minimum of 600 samples	minimum of 1200 samples
Intact	7%	14%
Edge	3%	13%
Degraded	4%	15%
Regrowth	20%	44%

- **Why 30?**

When we are analyzing forest height with an annual time step (Figure 3), we selected only years when we had at least 30 GEDI shots per year. In other words, for logged forest class (i.e. Fig. 3b) we selected only cells with a minimum of 600 samples (total) and 30 samples per year. We performed a sensitivity analysis to explore how changing this annual threshold to 50 or 100 samples per year is changing the results. Fig.A9 to A11 show the effect of this annual threshold. Results show that by raising the threshold to 50 samples per year, we start losing annual information while raising it to 100 samples per year we completely lose forest classes (e.g. logged forest 1x in panel b).

Regarding Fig.2, instead of annual classes, we had distance classes, and, as for Fig.3, we used the same 30 sample thresholds per distance to edge/non forest edge class. Results of the sensitivity analysis are presented in Fig. A12 to A14. Figures show that raising the threshold for distance classes to 50 or 100 has a limited impact on the results. However, to be coherent throughout the manuscript we always used the 30-sample threshold.

Figure A9: Fig. 3 with at least 30 samples per year (and total sample per category equal or greater than 600 samples).

Figure A10: Fig. 3 with at least 50 samples per year (and total sample per category equal or greater than 600 samples)

Figure A11: Fig. 3 with at least 100 samples per year (and total sample per category equal or greater than 600 samples)

Edge Effect (RH50)

Figure A12: Fig.2 with at least 30 samples per distance (and total sample per category equal or greater than 600 samples)

Edge Effect (RH50)

Figure A13: Fig.2 with at least 50 samples per distance (and total sample per category equal to or greater than 600 samples)

Edge Effect (RH50)

Figure A14: Fig.2 with at least 100 samples per distance (and total sample per category equal to or greater than 600 samples)

- **Why 7?**

We did not perform any sensitivity analysis to set the minimum number of 7 grid cells when presenting continental scale results. It was set empirically to ensure the representativity of all types of disturbed forest analyzed across space (Figures 1 and 2) and time (Figures 3 and 4 that is now extended data 8).

Line 1026 on: don't need a description of GEE really.

We removed these two sentences as suggested.

Extended data

There is a LOT of material here, some of which is interesting, but a lot of which seems almost like draft material – possibly nice to have, but not essential. I think unless the

material is vital to evidence a specific result or point in the main text, it should probably be cut. It would help if it were also organised so that all results here are organised under sub-headings that relate specifically to which result/point in the main text they are related to.

Thank you, as said previously we have greatly reduced the number of figures and tables in this ED section (moved to Supplementary Information or deleted as non-essential) from ~17 to 8 figures., keeping only the essential.

You could show here for eg comparisons of the visual assessment of the S American selective logging impacts flagged above.

Due to the restriction in the number of ED figures, we decided not to provide comparisons of the visual assessment of selective logging in South America versus the two other continents, the approach that is now the same for the whole pantropical scale as the WRI dataset on forest concessions (for Africa and SEA) is no longer used. We state in the Data availability section that the delineation of selective logging is freely and openly accessible as Google Earth Engine assets (links in the GEE codes of our public GitHub repository). The revised sentence reads as such (line 612):

“ Pre-processed data, post-processed data, drivers of forest degradation, maps, codes and final figures developed in this study are made publicly available and briefly described to facilitate reproducibility and applicability”.

ED Fig 6 – this is interesting and useful to show. Might be good to see other examples of these from eg locations where you find differences for S America, Africa and SE Asia. It's a good way for reader to visualise what you're talking about – refer to this v early on in the main text to show what it is you're actually deriving. The RH98 suggests the canopy at the deforested end is around 3m, RH50 says it's flat - & it looks like bare soil (although I know that might be misleading). If it's bare wouldn't RH98 and RH50 both be zero? Or is this because GEDI transect was at a different time to the image data when crops were present? Can you give the dates of the GEDI and image data?

Thank you for this relevant comment, this figure was requested by referee #2 at the previous round of review. We have added two new transects representing different human-modified landscapes of Central Africa (mosaic forest/small-scale agriculture) and

Indonesia (mosaic forest/oil palm plantation). Altogether, we show over different landscape compositions and spatial configurations that the scale of the edge effect can vary from 350m to 1000m (see figure below). This transect was referenced when we introduced the concept of forest/non-forest edge effect (line 111):

“We first assessed the scale and magnitude of forest-non-forest effect on forest structure metrics in the vicinity of deforested lands at varying distances to the forest edge”.

Regarding the deforested land (probably a soybean field) in the Brazilian Amazon transect, we interpret the differences between RH98 and RH50 by the fact that RH50 is not the median of RH98 values but it is an energy metric (i.e. height at which the cumulative return is at 50%). Consequently, we can find discrepancies between these two metrics over cultivated land or bare land, where RH98 will pick the nearest tallest vegetation in the footprint (soybean) while RH50 can show low values. The coarse resolution of the background image from Google prevents us from better interpreting the land use of the agriculture field. All GEDI shots were taken during the year 2020 and we also indicated in the caption the year of the background image).

Note that, we have also found discrepancies between RH98 and RH50 after selective logging or understory fires (Figure 3 or in extended data fig.7) where RH98 picks up remaining canopy or emergent trees (high values of RH98) while RH50 shows low values because it is sensitive to the vertical distribution of vegetation.

Extended Data Fig.2: Transects of GEDI relative heights (RH50 and RH98 from the year 2020) in the Brazilian Amazon, Congo basin and West Sumatra crossing deforested land, edge forest (edge width of 350 m, 800m and 1000m, respectively) and intact forest. The two black lines represent the height of RH98 and RH50. Background data: Google (08/2019, 12/2023 and 03/2022 for panels a-c, respectively), © 2024 Maxar Technologies

Referee #2 (Remarks to the Author):

Summary

Thank you for the effort in responding to my second round of comments. I applaud your patience and work in this effort.

At this point I think we have cleared up many of the major issues, including things that were not quite right, adding clarity, and most importantly using nearly the full archive of GEDI data. This last point, however, does make review difficult because each subsequent version thus far has had significant changes, especially with respect to the kinds and quantity of GEDI data used. And each review of a revision starts to find different things that were missed in previous reviews because the entire paper needs to be evaluated again. I apologize for bringing in a few new concerns, but this essentially what has happened and is unavoidable.

We thank the reviewer for the appreciation of our work.

In summary, we have improved the structure and flow of the sentences, especially those with too many references to the extended data or supplementary information. We have changed the title of the paper. We have also removed sentences that were redundant with the Methods section, in order to go more directly to the main results. We have simplified the titles of the Results section to emphasise the two new aspects of this paper, namely the extensive and persistent aspects of forest degradation. We have moved the ecological explanation of the key forest structure metrics we use to the very beginning of the paper. We have simplified the ED and moved non-essential figures and tables to supplementary information.

However, please note that we found few contrasting suggestions between reviewers 1 and 2 on how to change the structure and message of the paper. Therefore, we had to make use of the different suggestions in the narrative suggested by referee #1 and referee #2.

We have also modified, as suggested by referee #1, the scheme for attributing forest degradation to selective logging drivers; we now have a more robust and consistent way of doing this, based on visual interpretation across the pantropical belt.

My main issue still concerns the statistical methods and the meaning of some of the descriptive statistics and uncertainties that are being reported. These are described below but center on the monte carlo approach to propagating regression uncertainty and the results presented in Extended Data Tables 1-3. For example, the precisions shown for height are greater than what GEDI reports as RMSE against airborne data globally, so it's not clear why aggregates of these data and the precision of those means should be nearly as large as reported. So either there is a methodological issue or the description of what is being reported lacks clarity and needs to be refined.

We thank the reviewer for pointing to this issue. We inform that in the previous version of the manuscript in the calculation of the descriptive statistics reported in Extended Data Tables 1 to 3 we used average values of RH98 and AGBD over intact forest computed at grid cell level (1.5 degree size) and derived the continental mean, standard deviation and standard error (or precision of the mean assuming random error distribution as suggested by⁴⁵).

Following the suggestion of the reviewer, we have now updated the tables with footprint level values of RH98 or AGBD mean uncertainties over intact forest to derive continental descriptive statistics. GEDI reports an RMSE of 3.6m against airborne data globally³⁹. Assuming random distribution of the error over large spatial scale, the estimated precision of the mean intact forest Rh98 is 0.002m, 0.004m and 0.009m for Americas, Africa and Asia, respectively.

Comments on Responses

1. "AGB estimates derived from GEDI are still a preliminary product with larger uncertainties than height retrieval, mainly due to the scarcity of calibration data for the allometric models in some regions². We believe that our work opens the door towards the quantification of emission and removal factors from various forest degradation pathways at the pantropical scale, but this would require further fine-tuning of AGB modeling and validation."

=> This is not so important but what you assert here is not actually true. GEDI V2 biomass data was released, and these are no more “preliminary” than any other global biomass map; indeed, GEDI has at least as much and maybe more calibration data than any other product.

Thank you for pointing this out. We have realized that we are actually using version 2 of GEDI L4A and not version 1, as previously stated. We apologize for any confusion caused. We deleted the paragraph highlighted here and modified the text accordingly (line 438):

“ For our analysis, we used GEDI Level 2A⁴⁶ Elevation and Height Metrics (version 2) and GEDI L4A⁴⁷ Above Ground Biomass (version 2.1) which represent returned laser's energy metrics on canopy height and estimated Above Ground Biomass Density for each 25 m diameter GEDI footprint. ”

2. “We have redone all analyses involving AGB (also removing the night filter as suggested) and have now taken into account the associated prediction error (i.e. Aboveground biomass density prediction standard error) into our results.”

=> Thank you for doing these analyses. I am still not sure about how you propagated the prediction error. The process is described under "Statistical Tests" in the manuscript. In the above description this seems to suggest that all the prediction errors at 25m from GEDI are used to find the standard deviation of this distribution? The manuscript says the sigma of the distribution is set to “the standard error of the AGB product”. That product does not have a single standard error, it only has prediction standard errors for each footprint and at 1 km cells. So, it's not clear what the parameters of the distribution are being sampled from are exactly. It seems as if it's the standard deviation of the standard error distribution?

We have clarified the details of the error propagation process in the Methods section. In our Monte Carlo analysis, we used the prediction standard error (AGBD_SE) for each GEDI footprint as part of the error propagation process. We based the analysis on the assumption that the error at each GEDI footprint is spatially independent as reported by ⁴⁸. As the reviewer pointed out, the prediction standard error varies spatially; to address this issue, a random value for the biomass data was derived using different prediction standard error values. Our strategy involved drawing a series of random values, $X = [X_j]_{j=1}^n$, from a normal distribution denoted as $X_j \sim N(\mu, \sigma^2)$ with a mean of 0 ($\mu = 0$) and the standard deviation parameter (σ)

set to the AGBD prediction standard error value. Below we provide a simplified explanation of our algorithm:

For each iteration i , where $i = 1, 2, \dots, 500$

- Generate noise, $noise_j$, for i -th iteration using by drawing a random value from a normal distribution with mean 0 and standard deviation equal to the prediction error of the biomass (σ_i) for each biomass value.

The noise can be represented as:

$$noise_i \sim N(\mu, \sigma_j^2)$$

- Perturb the biomass values by adding the generated noise to the original dataset ($biomass_{original,j}$) for the i -th iteration ($biomass_{perturbed,i}$).
- Perform an ANOVA for each iteration using the perturbed dataset and record the results

We modified the text as such (line 533):

“ We performed a series of one-way Analysis of Variance (ANOVAs) to test for differences in the impacts of edge effects at different distances and times on the long-term recovery of the relative heights and biomass variables. ANOVAs were performed separately for each continent. For the height variables (RH50 and RH98), a series of standard one-way ANOVAs were used. In the analyses involving AGBD, we used a modified approach to propagate the prediction standard error associated with the AGBD dataset values which involved using a Monte Carlo approach ($n = 500$). In brief, we generated random noise that was added to the AGBD data. For each iteration i we generated $noise_j$ by drawing a random value from a normal distribution with mean μ of 0 and standard deviation equal to the prediction standard error of the AGBD (σ_i) for each GEDI footprint. The noise can be represented as: $noise_i \sim N(\mu, \sigma_j^2)$. We then perturbed the AGBD values by adding the generated noise to the original dataset ($biomass_{original,j}$) for the i -th iteration ($biomass_{perturbed,i}$). We then performed an ANOVA for each iteration using the perturbed dataset and recorded the results. We subsequently examined the distribution density of the F -values. The results showed minimal variability suggesting that observed differences are robust to uncertainty associated with the AGBD values (Supplementary Fig.10). For each ANOVA, we conducted a series of Tukey HSD post-hoc tests to assess significant differences between distance classes or time-steps. The significance level was set to $p < 0.05$ ”.

Secondly, randomly permuted the AGB data set 500 times and calculated an F statistic. I do not see why this would work as the prediction errors are highly correlated with the AGB value. They tend to get larger as the value gets larger, so as you sample from that noised distribution you should take that into account. why this would work as the prediction errors are highly correlated with the AGB value. They tend to get larger as the value gets larger, so as you sample from that noised distribution you should take that into account.

Third, even if we randomly sample from the AGB, I do not see how we overcome the issue that the residuals are spatially correlated unless we specifically draw samples that are not spatially correlated. All the above, and the actual F values make me think something is not quite right. Would you please provide more exact description of the methods here, for which data (in 1.5 deg cells, in polygons, etc) so that it can be recreated (that is step by step)?

Note that the Committee on Earth Observation Satellites (CEOS) has published a good practices protocol that includes methodology on using monte carlo methods for estimating biomass uncertainty and at least one of the manuscript authors (Avitabile) is a coauthor on this report.

https://lpvs.gsfc.nasa.gov/PDF/CEOS_WGCV_LPV_Biomass_Protocol_2021_V1.0.pdf

Thank you for pointing this out. We recognise the importance of this issue and agree with the reviewer on the importance of the spatial correlation of the error and of heteroscedasticity in the estimation of uncertainties at small spatial scale (e.g. the 1km scale of the GEDI biomass data product). However, we maintain confidence in the robustness of our results. ANOVA can still be relatively robust to the violations of certain assumptions, including heteroskedasticity. This is particularly the case when dealing with large sample sizes, as in our study for which the statistical tests were performed for spatial grids of 1.5 degree and therefore 4 orders of magnitude larger than the standard 1km GEDI data product.

Large samples ensure that our computed means closely approximate the true population means, effectively mitigating the influence of heteroskedasticity. Furthermore, the Central Limit Theorem implies that, when the sample size is large enough, the distribution of the sample means will approach a normal distribution, regardless of the original distribution of the data. This helps to ensure that the overall distribution of the test statistic i.e. the F statistic remains approximately as expected under the null hypothesis, thus preserving the test's validity (Devore et al.⁴⁹, Kutner et al.⁵⁰)

To support this point, we have conducted a simulation with a modified version of our Monte Carlo analysis, incorporating a subset of GEDI data (all footprints from 2019 to the end of 2022 passing the same quality filters as the ones described in our Methodology) extracted over three randomly chosen study sites of ~200x200km in the Brazilian Amazon, Congo basin and South East Asia (Fig. A15 below). This simulation involved propagating error into the biomass values (Fig. A16) in such a way that the error would be both heteroskedastic (i.e. increasing with the value of the biomass) and spatially autocorrelated, therefore addressing the concern raised by the reviewer. The results from this experiment are in concordance with those from the original Monte Carlo analysis, which involved introducing noise by simply drawing from a normal distribution, as explained in our previous response. Below we explain the simulation in detail:

For each iteration i where $i = 1, 2, \dots, 300$

- Generate noise $Z_i(X)$ by drawing values from a Gaussian random field with a mean μ of 0 and sill equal to the prediction error of the biomass values σ_i^2 . The spatially correlated noise follows a Gaussian distribution: $Z_i(X) \sim N(\mu, \sigma_i^2)$ where X represents spatial coordinates (Fig. A17)
- Rescale the noise generated in the previous step to ensure that the error would be heteroskedastic (i.e., proportional to the biomass values): *Rescaled noise* $e_i = Z_i(X) \cdot \theta$ where θ is the ratio between biomass values and the *RSME* of the prediction error (Fig. A18)
- Perturb the biomass values by adding the generated noise (Fig. A19) to the original dataset ($biomass_{original,j}$) (Fig. A16) for the i -th iteration ($biomass_{perturbed,i}$) (Fig. A20)
- Perform an ANOVA for each iteration using the perturbed dataset (Fig. A20) and record the results

For completeness, we applied the above mentioned framework assuming three different scales of spatial autocorrelation (0.02 degrees: ~2km, 0.1 degree: ~10km and 0.5 degree: ~50km). This was achieved by adjusting the range parameter of the simulated Gaussian random field. The range parameter allowed us to set the maximum distance over which the spatial autocorrelation diminishes to negligible levels, essentially dictating the scale at which spatial patterns exhibit independence. For each repetition, we extracted the perturbed AGBD values from the approach defined in the manuscript, the perturbed AGBD values using combined heteroskedasticity and spatial autocorrelation noises along with JRC-TMF class values (only undisturbed forest class was selected), and distance class from the forest/non-forest edge (i.e. 0 - 60, 60 - 120, 120 - 240, 240 - 420, 420 - 720, 720 - 1020, 1020 - 1500, 1500 - 2040, 2040 - 2580, 2580 - 3120, 3120 - 4020, 4020 - 5100, 5100 - 6000, 6000 - 7200, 7200 - 8100, 8100 - 9000, 9000 - 10200 m). We considered intact forest any footprints of GEDI falling within the class undisturbed forest and located at a distance higher than 3 km from the edge. We then computed the difference between intact forest AGBD values and the perturbed AGBD values at each distance class to the edge, performed the ANOVA test and saved as output the p-value and F-test value.

Figure A15: Location of the three study sites where GEDI footprints from 2019-2022 (in black) were extracted to conduct simulations of error propagation on our results concerning the scale of AGBD edge effects. Background image: JRC-TMF Transition Map-Sub types 2022.

Figure A16: original AGBD values for site B (see Fig. A15)

Figure A17: Spatial Gaussian noise (Exponential, scale = 10km)

Figure A18: Rescaled noise (heteroskedasticity)

Spatial Gaussian Error Scaled

Figure A19: Combined heteroskedastic and spatially autocorrelated noise

Figure A20: perturbed AGBD values

The results from our numerical experiment (Fig A21) reveal that the introduction of heteroscedasticity and spatial correlation in the error generate a broader distribution of ANOVA F-values compared to the original approach based on the assumption of random error distribution. Despite these differences, all the differences between forest categories remain statistically significant, with p-values less than the chosen significance level of 0.05 ($\alpha=0.05$). Importantly, these findings are also insensitive to the scale of the spatial correlation of the error, indicating that our approach maintains its validity across varying spatial scales in the error structure. This outcome underscores the overall robustness of our approach based on the original error propagation procedure, demonstrating that our approach provides reliable statistical inferences even with the introduction of complex error structures (i.e. heteroscedastic and spatially correlated error). Moreover, the results are coherent between different error propagation approaches, indicating the robustness of our findings.

Note that we have shared the code implemented for this analysis in the dedicated GitHub repository. You can also find the input files and R code in this shared Google Drive folder (sub-folder 'Error_propagation'):

https://drive.google.com/drive/folders/1Vtm-0KEfFFuuWCFkmjGoiGQI55p_FmZ2?usp=sharing

Figure A21: Distribution of ANOVA F-test after Monte Carlo simulation to propagate AGBD prediction standard error (“Manuscript” shown as red curves) and after Monte Carlo simulation to propagate heteroskedastic and spatially autocorrelated noise (“New simulation” shown as blue curves) onto the analysis of AGBD edge effects at three spatial autocorrelation scales (impacting the “New simulation” method only) and across the three study sites presented in Fig. A15.

3. Figure S8.

=> I cannot say I understand Figure S8. Looks like for each continent there is no distribution of the F-stat or the way its plotted is masking it. Can we explain this figure more? Why does it look like a single vertical line?

The density plot would be a smooth curve that peaks where Anova Fs for MonteCarlo simulations are most common. However, in this case, we have only one ANOVA F value constant for each Monte Carlo simulation and the result is a vertical line instead of a curve. In other words, the results showed no variability in F. This is due to the large number of observations used in the computation that leads to very stable estimates. Table A2 below shows the results of the ANOVA F statistics for the first 10 MonteCarlo simulations over Africa (column "statistic" refers to the F value).

Table A2: Anova results for the first 10 MonteCarlo simulations over Africa

```
# A tibble: 1,500 × 8
  term      df      sumsq    meansq statistic p.value Continent
  <chr> <dbl>    <dbl>    <dbl>    <dbl>    <dbl> <chr>
1 Class    16 28550275. 1784392.    853.      0 Africa
2 Class    16 28550275. 1784392.    853.      0 Africa
3 Class    16 28550275. 1784392.    853.      0 Africa
4 Class    16 28550275. 1784392.    853.      0 Africa
5 Class    16 28550275. 1784392.    853.      0 Africa
6 Class    16 28550275. 1784392.    853.      0 Africa
7 Class    16 28550275. 1784392.    853.      0 Africa
8 Class    16 28550275. 1784392.    853.      0 Africa
9 Class    16 28550275. 1784392.    853.      0 Africa
10 Class   16 28550275. 1784392.    853.      0 Africa
# i 1,490 more rows
```

5. Extended Table 1 and 2

=> If we look at Extended Table 1, we report the spatial S.D. and the precision of the mean. I understand these column headings. But how did we find that precision of the mean? If the samples are random and uncorrelated, that precision should be the $\sqrt{(\text{Merr}^2 + \text{var})/n}$, where Merr is the measurement error for GEDI heights and var is the sample height variance of the GEDI samples in the area – this is not what you did (nor should you for GEDI). So, how was this found for height? Unless there is a bias, can

explain how you can get such a large error about the mean when the S.D. are (a) on the order of 5-6 m, (b) you have millions of shots, and (c) you are ignoring measurement error?

Alternately, do we mean this is the precision of grid cell means? If so, how did we propagate the uncertainty about each grid cell mean to get a precision for the entire continent? If we could see the formula and the inputs used to recreate one of the rows, that would help greatly.

As previously reported in the response to your general comment, we acknowledge that we have made a mistake in the original reporting of the summary statistics and we are sorry for the confusion it created. Extended data tables 1-2 and Supplementary Table 1 presenting summary statistics of RH98, AGBD and AGBD_SE at continental scale have now been corrected (please see below).

You can find the input files and R code in this shared google drive folder (sub-folder 'Extended_data_tables1_2') to go through our process of handling footprint level data of GEDI (RH98, AGBD and AGBD_SE) using Google Earth Engine and deriving summary statistics:

https://drive.google.com/drive/folders/1Vtm-0KEfFFuuWCFkmjGojGQl55p_FmZ2?usp=sharing

We agree with the reviewer on the importance of taking into account the spatial autocorrelation and the non-randomness of GEDI footprints, particularly when the objective is to yield robust and spatially explicit predictions of forest structure with a limited number of GEDI footprint (as in the case of the 1km data product). In that case, the overall uncertainty can be underestimated if spatial autocorrelation of the error is not taken into account.

The scope of our analysis was to compute summary statistics (mean, standard deviation and precision of the mean) of intact forests for each continent and given the large number of observations, we can assume that the height variance of the GEDI samples in the area and the spatial autocorrelation of errors has very marginal effects on the final estimates of the mean⁵¹.

We would like to bring back here some tests that we have done regarding the potential uncertainty in the reported canopy height summary statistics due to the height variance of the GEDI samples in the area. In order to reduce sampling bias in the structural variable dataset,

we randomly resampled GEDI observations 500 times within each $1.5^\circ \times 1.5^\circ$ degrees gridcell. We then summarized the random samples by calculating the mean and standard deviation of RH98, for each grid cell. Using this random sampling procedure based on the iteration (500 times) of random sampling 300 GEDI footprints for each grid cell, we found that with this sample size the intra-grid variability of canopy heights was not leading to any significant uncertainty in the estimation of the grid mean. In fact, the results of the random sampling procedure show the low variability of the intact forest class of RH98 distribution (Fig.A22). This analysis consolidates the robustness of our summary statistics of RH98 and shows that the sample height variance of the GEDI samples in the area do not fundamentally affect these statistics if the sample size is sufficiently large.

Figure A22: Random sampling procedure result showing the variability of canopy height distribution (RH98) for intact forest class. The random sampling procedure consists of sampling 300 GEDI observations for each grid cell, iterating 500 times this sampling and computing the average values. Panel a shows the mean RH98 of intact forest after random sampling procedure along with its associated 5m binned distribution of RH98 for the whole tropical moist forest domain where vertical bars show the standard deviation (supplementary

fig.9). Panel b shows the variations of the average RH98 of intact forest after 500 iterations (expressed as a standard deviation).

Action taken: We acknowledge in the caption of Extended Data Table 1 that the summary statistics of RH98 at continental level have potential limitations for not tackling the non-randomness and spatial autocorrelation issues from GEDI samples. Please find below the new caption and corrected summary of RH98 values:

Extended Data Table.1 Summary statistics of canopy heights for intact, degraded, edge forests and regrowths. Number of GEDI shots and statistics (mean, standard deviation and precision) of grid cells by continent of RH98 for intact forest, degraded forest, forest edge and forest regrowth. The precision of the RH98 was obtained by computing the standard error of the RH98 values at footprint levels for each continent. Note that issues linked to the non-randomness and spatial autocorrelation of GEDI samples are not integrated in this computation of summary statistics.

RH98	Continent	Number of GEDI shots	Mean Height [m]	Standard Deviation [m]	Precision of the Mean [m]
Intact forest	Americas	16087545	28.563	7.440	0.002
	Africa	5880126	29.259	8.572	0.004
	Asia	1523450	34.397	10.750	0.009
Degraded forest	Americas	1728729	18.885	8.661	0.007
	Africa	704074	17.965	9.821	0.012
	Asia	1749092	20.173	10.718	0.008
Forest edge (120m width)	Americas	4234908	19.253	9.037	0.004
	Africa	2654101	19.799	10.052	0.006
	Asia	1585174	21.139	10.971	0.008
Forest regrowth	Americas	594291	13.944	7.765	0.010
	Africa	67313	14.299	9.234	0.036
	Asia	177179	15.721	9.348	0.022

4. Extended Data Table 3.

=> I apologize but I still do not understand the SE's in this table. I'm also confused about what is being presented. Each GEDI footprint has a prediction error from its associated regression calibration. The table reports something called "Mean Standard Error", say for Africa, with 17.0 Mg/ha. What is that number and how do you find it? We then see "precision of the AGB" (37.9), but again how do we find it? That is just the standard deviation of all the GEDI shots in the area? Is it the precision of mean biomass which we can use to form a confidence interval?

=> The usual metric associated with uncertainty where sampling is involved is the "precision of the mean", or the or "standard error the mean". If GEDI shots were actual observations of AGB and if they were random, the precision of the mean would be the common formula of $\sqrt{\text{variance}/n}$. But this is entirely incorrect for GEDI. The observations have a regression error and they are not random, hence the montecarlo approach. It does not seem to me that we have calculated a "precision" as normally thought of in this table, where the precision is a function of both the regression error and the sampling error and would tell us the limits of the mean for some confidence level. Or am I missing something here? I also point that even if we take the sampled data and make a grid, we still must propagate the grid level sample uncertainty when reporting values that combine grid cells, say over a continent.

=> It might help if we note everywhere where we mean standard deviation and where we mean "standard error". I could not always tell.

We agree with the reviewer and acknowledge that the hybrid inference methodology^{45,48} is the most robust approach to deal with both regression error from AGBD estimation and sampling error in the production of the 1km gridded product of GEDI (L4B).

We argue that our analysis has a very different purpose, that is the statistical inference of differences between intermixed sub-population of GEDI retrievals (e.g. intact versus degraded) over a very large domain (1.5 degree, about 22000 km²). This task is ideally addressed with a different statistical approach (e.g. ANOVA). Given that these sub-population are spatially intermixed at the scale of the analysis, the effect of the spatial dependence of the error tends to cancel out when computing differences between classes, leading to a higher robustness of the statistical tests. Our analysis and synthetic experiment about the effect of heteroscedasticity and spatial error correlation demonstrate that biases from the spatial autocorrelation and from the associated regression error do not fundamentally affect our results and conclusions about the impact of degradation of structural canopy parameters, particularly on the analysis of the magnitude and spatial scale of the edge effect. The summary statistics of AGBD at continental level are a side analysis to complement the first section of our results where we characterize the general patterns of intact forest and the disturbed types in terms of canopy structure and AGBD, which to our point of view is a useful section for the reader to step into the subject.

Action taken: We acknowledge in the caption of Extended Data Table 2 that the summary statistics of AGBD at continental level have potential limitations for not tackling the non-randomness, spatial autocorrelation and regression error issues coming from GEDI samples and estimates. Please find below the new caption and corrected summary of AGBD values:

Extended Data Table.2 Summary statistics of AGBD for intact, degraded, edge forests and regrowths. Number of GEDI shots and statistics (mean, standard deviation and precision) of grid cells by continent of AGBD and its associated prediction standard error for intact forest, degraded forest, forest edge and forest regrowth. The precision of the AGBD was obtained by computing the standard error of the AGBD values at footprint levels for each continent. The mean, standard deviation and precision of the AGBD predictive standard error (AGBD_SE) are shown in Supplementary Table 1. Note that issues linked to the non-randomness and spatial autocorrelation of GEDI samples and the propagation of the regression error associated to each AGBD estimate are not integrated in this computation of summary statistics^{45,48}.

AGBD	Continent	Number of GEDI shots	Mean AGBD [Mg/ha]	Standard Deviation [Mg/ha]	Precision of the Mean [Mg/ha]
Intact forest	Americas	25303980	239.512	129.876	0.026
	Africa	8986956	225.661	110.906	0.037
	Asia	2519005	370.812	205.194	0.129
Degraded forest	Americas	2791301	111.762	115.661	0.069
	Africa	1099960	103.272	87.832	0.084
	Asia	2766798	142.991	128.692	0.077
Forest edge (120m width)	Americas	6903997	116.365	117.970	0.045
	Africa	4494123	121.980	96.790	0.046
	Asia	2809352	147.523	133.289	0.080
Forest regrowth	Americas	936495	68.972	94.260	0.097
	Africa	120529	73.730	73.817	0.213
	Asia	306946	98.413	93.695	0.169

Corrected supplementary Table.1 Number of GEDI shots and statistics (mean, standard deviation and precision) of grid cells by continent of AGBD prediction standard error (AGBD_SE) for intact forest. The precision of the AGBD_SE i was obtained by computing the standard error of the AGBD_SE values at footprint level for each continent.

Continent	Number of GEDI shots	Mean AGBD_SE [Mg/ha]	Standard Deviation [Mg/ha]	Precision of the Mean AGBD_SE [Mg/ha]
Africa	8986956	17	0.0043875	1.464E-06
Americas	25303980	12.99978	0.0265894	5.286E-06
Asia	2519005	12.49522	4.476483	0.0028205

6. “We assessed that the large majority of GEDI footprints (>75%) with canopy heights from 2 and 5m fall within 0-5 degree slope, meaning that the potential risk of pulse spreading on high slopes and thus yield unreliable heights is minimal (Figure A3).”

=> This is not true, even on a flat surface. GEDI cannot reliably get down to 2 m. It's not just pulse spreading. The GEDI pulse width is 15 ns which is 4.5 m (i.e. the full-width half max is that). You can see from this value that it will be rare that you would believe you can consistently capture heights < 3m with a pulse-width that large.

Thank you for this information. Following on this suggestion, we decided to modify this threshold to 5m and we re-extracted all GEDI data (RH98, RH50 and AGBD) accordingly. We updated the text in the Method section as such (line 470):

“Finally, we excluded GEDI footprints with RH98 values below 5 m to be compliant with the Food, and Agriculture Organization (FAO) forest definition”.

Action taken: We updated all figures concerned by this change, i.e. Figure 3 and its equivalent for AGBD and RH50, Figure 4 that has been moved to Extended Data Figures following the suggestion of Reviewer#1 and Extended Data Fig 8. Here are some examples of the new figures to show that the observed trends were not affected by this change:

New Fig 3. Note that Fig3 was modified following suggestions from reviewer #1. For instance, classes showing a low number of GEDI footprints throughout the time series were deleted. The attribution scheme of forest degradation to selective logging was also modified, now relying on visual interpretation of logging patterns for the whole pantropical belt (discarding WRI data on forest concessions that was bringing inconsistencies and other issues into the results). This is fully analyzed and documented in our response to reviewer #1.

New extended data figure 4.

Vulnerability of degraded and edge forest to deforestation

New extended data figure 8 (former fig 4 but was moved to the extended data section following reviewer #1 suggestion).

Notes on the Revised Manuscript

60. As with last revision, “It” is ambiguous. Suggest you substitute “such estimates” for it. We have replaced “it” with “such estimates” accordingly.

74. its Global Dynamics Ecosystem Investigation, no LiDAR at the end of that which appears in multiple places

We have corrected the text accordingly and throughout the manuscript.

94. I find there is ambiguity throughout the manuscript with “AGB”. AGB usually refers to a stock and AGBD refers to a density. You seem to use these interchangeably in the text and figures. Perhaps just define up front and let the reader know what is meant where?

Thank you for pointing to this issue, we have corrected the term AGB into AGBD and homogenized this all throughout the text, figures and tables.

104. I don’t think it’s clear what the number after +/- refers to. When we mean the standard deviation of a sampled distribution, we should specify somewhere S.D. so there is no ambiguity. When we mean the standard error of the sample mean of that distribution, say S.E. (i.e. the precision of the mean).

Thank you for this comment, these numbers refer to the mean and to the standard deviation.

We added this precision to the sentence and now reads like this (line 76):

“(34.4 m \pm 10.7, mean +/- sd)”.

115. Can we make this more precise? To say “at least 10 m” means that the minimum value your observed was 10 m. Is that correct?

We rephrase this sentence into (line 84):

“ In fact, we quantify that the minimum difference between intact old growth and degraded forests is 10m for mean RH98 and 122 Mg/ha for mean AGBD ”

119. Same thing when we say “on average 10 m to 15 m lower” -> this range covers some set of averages?

Thank you, we deleted the range as it was bringing too much confusion and now report in the text the average difference of RH98 between intact old growth and forest regrowth, across the whole pantropical belt. The revised sentence reads as such (line 89):

“ Regrowing forests on deforested land have, on average, a 16 m lower RH98 than intact forests, and an average AGBD of 80.4 Mg/ha \pm 87.3”

265. We defined RH50 early in the paper as “hereafter median height”, however RH50 continues to be used throughout the manuscript. Secondly, I still think there is confusion over what RH50 is. It is most definitely not the “median height of forests” if what you are referencing here is RH50. The median height of forests that were burned would be the median of all the RH98 values. Is this what is meant or do we mean RH50 recovered faster in this sentence? Remember that RH50 is an energy metric, it's the height at which the cumulative return is at 50%.

We thank the reviewer for the relevant comment and decided to remove the term ‘median height of forests’ throughout the text to avoid confusion between the median of RH98 values and what RH50 actually represents. We rephrase this paragraph accordingly (line 232):

“ No recovery trend in RH98 or AGBD was detected even 10 years after the last disturbance, confirming the long-lasting impacts of fire on tree mortality and losses of AGBD^{52,53}. Manipulative studies of post-fire degradation in the Amazon showed strong understory vegetation regrowth under the remaining dominant and taller trees within five years after the disturbance, resulting in partial canopy closure (70-80%⁵⁴). This vegetation dynamic is better captured by changes in RH50 than by changes in RH98. The high variability in recovery rates is likely due to different fire frequencies, intensity, climate and forest-type specific responses⁵⁵”

Before this paragraph, the differences between post-logging dynamics between RH50 and RH98 were interpreted as such (line 220):

“ The absence of recovery trends in RH98 can be explained by the slow regrowth of late successional, large and emergent trees²⁴, whereas forest understory dynamics, including tree removals, collateral damages from selective logging (e.g. dead fallen trees) and the fast

regrowth of pioneer and understory species, affect the average vertical distribution of plant material, captured by RH50, making this metric a robust indicator of the long-term impacts from degradation and subsequent recovery⁵⁶ ”.

520. GEDI is **not** an instrument. GEDI refers to an investigation that uses an instrument. So, this sentence should really be something like “NASA’s GEDI mission uses a lidar deployed on the ISS.” I know this seems somewhat trivial but is more accurate.

Thank you, we have rephrased the sentence accordingly (line 433): “NASA’s *GEDI mission uses a LiDAR deployed on the International Space Station from April 2019 until March 2023.*”

521: The website date is wrong (so I understand why you report it!). GEDI took its last observations on March 16, 2023.

Thank you, we have modified this date accordingly.

528: The current GEDI geolocation is 11 m (1 sigma) and has been for some time now.

Thank you, we have modified this accordingly.

539: Just delete “was returned by objects 20m above the ground” as this is not true. There may be nothing at all at 20m, it’s just the cumulative return energy as you say to start the sentence.

Thank you, we have deleted this sentence accordingly.

550. Suggest we substitute for “(now version 1)” with “(we used version 1 in this paper)”.

This whole paragraph has been deleted (see comment above). The versioning of GEDI data was added as such (line 438):

“ For our analysis, we used GEDI Level 2A⁴⁶ Elevation and Height Metrics (version 2) and GEDI L4A⁴⁷ Above Ground Biomass (version 2.1) which represent returned laser’s energy metrics on canopy height and estimated Above Ground Biomass Density for each 25 m diameter GEDI footprint. ”

Referee #3 (Remarks to the Author):

I thank the authors for their revisions.

References

1. Asner, G. P., Keller, M., Pereira, R. & Zweede, J. C. Remote sensing of selective logging in Amazonia: Assessing limitations based on detailed field observations, Landsat ETM+, and textural analysis. *Remote Sensing of Environment* **80**, 483–496 (2002).
2. Shapiro, A. *et al.* Small scale agriculture continues to drive deforestation and degradation in fragmented forests in the Congo Basin (2015–2020). *Land Use Policy* **134**, 106922 (2023).
3. Souza, C. Mapping forest degradation in the Eastern Amazon from SPOT 4 through spectral mixture models. *Remote Sensing of Environment* **87**, 494–506 (2003).
4. Lima, T. A. *et al.* Comparing Sentinel-2 MSI and Landsat 8 OLI Imagery for Monitoring Selective Logging in the Brazilian Amazon. *Remote Sensing* **11**, 961 (2019).
5. Vancutsem, C. *et al.* Long-term (1990–2019) monitoring of forest cover changes in the humid tropics. *Sci. Adv.* **7**, eabe1603 (2021).
6. Bullock, E. L., Woodcock, C. E., Souza Jr., C. & Olofsson, P. Satellite-based estimates reveal widespread forest degradation in the Amazon. *Global Change Biology* **26**, 2956–2969 (2020).
7. Shimabukuro, Y. E., Beuchle, R., Grecchi, R. C. & Achard, F. Assessment of forest degradation in Brazilian Amazon due to selective logging and fires using time series of fraction images derived from Landsat ETM+ images. *Remote Sensing Letters* **5**, 773–782 (2014).

8. Pearson, T. R. H., Brown, S., Murray, L. & Sidman, G. Greenhouse gas emissions from tropical forest degradation: an underestimated source. *Carbon Balance Manage* **12**, 3 (2017).
9. Baccini, A. *et al.* Tropical forests are a net carbon source based on aboveground measurements of gain and loss. *Science* **358**, 230–234 (2017).
10. Barlow, J. *et al.* Anthropogenic disturbance in tropical forests can double biodiversity loss from deforestation. *Nature* **535**, 144–147 (2016).
11. Gao, Y., Skutsch, M., Paneque-Gálvez, J. & Ghilardi, A. Remote sensing of forest degradation: a review. *Environ. Res. Lett.* **15**, 103001 (2020).
12. Dubayah, R. *et al.* The Global Ecosystem Dynamics Investigation: High-resolution laser ranging of the Earth's forests and topography. *Science of Remote Sensing* **1**, 100002 (2020).
13. Lapola, D. M. *et al.* The drivers and impacts of Amazon forest degradation. *Science* **379**, eabp8622 (2023).
14. Fischer, R. *et al.* Accelerated forest fragmentation leads to critical increase in tropical forest edge area. *Sci. Adv.* **7**, eabg7012 (2021).
15. Matricardi, E. A. T. *et al.* Long-term forest degradation surpasses deforestation in the Brazilian Amazon. *Science* **369**, 1378–1382 (2020).
16. Fischer, R. *et al.* Accelerated forest fragmentation leads to critical increase in tropical forest edge area. *Sci. Adv.* **7**, eabg7012 (2021).
17. Boulton, C. A., Lenton, T. M. & Boers, N. Pronounced loss of Amazon rainforest resilience since the early 2000s. *Nat. Clim. Chang.* **12**, 271–278 (2022).
18. Réjou-Méchain, M. *et al.* Unveiling African rainforest composition and vulnerability to global change. *Nature* **593**, 90–94 (2021).
19. Flores, B. M. *et al.* Critical transitions in the Amazon forest system. *Nature* **626**, 555–564 (2024).
20. Qie, L. *et al.* Long-term carbon sink in Borneo's forests halted by drought and vulnerable to edge effects. *Nat Commun* **8**, 1966 (2017).

21. Brinck, K. *et al.* High resolution analysis of tropical forest fragmentation and its impact on the global carbon cycle. *Nat Commun* **8**, 14855 (2017).
22. Silva Junior, C. H. L. *et al.* Persistent collapse of biomass in Amazonian forest edges following deforestation leads to unaccounted carbon losses. *Sci. Adv.* **6**, eaaz8360 (2020).
23. Ordway, E. M. & Asner, G. P. Carbon declines along tropical forest edges correspond to heterogeneous effects on canopy structure and function. *Proc Natl Acad Sci USA* **117**, 7863–7870 (2020).
24. Milodowski, D. T. *et al.* The impact of logging on vertical canopy structure across a gradient of tropical forest degradation intensity in Borneo. *Journal of Applied Ecology* **58**, 1764–1775 (2021).
25. Rangel Pinagé, E. *et al.* Long-Term Impacts of Selective Logging on Amazon Forest Dynamics from Multi-Temporal Airborne LiDAR. *Remote Sensing* **11**, 709 (2019).
26. Rappaport, D. I. *et al.* Quantifying long-term changes in carbon stocks and forest structure from Amazon forest degradation. *Environmental Research Letters* (2018) doi:10.1088/1748-9326/aac331.
27. de Andrade, R. B. *et al.* Scenarios in tropical forest degradation: carbon stock trajectories for REDD+. *Carbon Balance and Management* **12**, (2017).
28. Chaplin-Kramer, R. *et al.* Degradation in carbon stocks near tropical forest edges. *Nat Commun* **6**, 10158 (2015).
29. Dalagnol, R. *et al.* Mapping tropical forest degradation with deep learning and Planet NICFI data. *Remote Sensing of Environment* **298**, 113798 (2023).
30. Drake, J. B. *et al.* Estimation of tropical forest structural characteristics using large-footprint lidar. *Remote Sensing of Environment* **79**, 305–319 (2002).
31. Duncanson, L. *et al.* Aboveground biomass density models for NASA's Global Ecosystem Dynamics Investigation (GEDI) lidar mission. *Remote Sensing of Environment* **270**, 112845 (2022).

32. Blanchard, G. *et al.* UAV-Lidar reveals that canopy structure mediates the influence of edge effects on forest diversity, function and microclimate. *Journal of Ecology* **n/a**, (2023).
33. Briant, G., Gond, V. & Laurance, S. G. W. Habitat fragmentation and the desiccation of forest canopies: A case study from eastern Amazonia. *Biological Conservation* **143**, 2763–2769 (2010).
34. Almeida, D. R. A. *et al.* Persistent effects of fragmentation on tropical rainforest canopy structure after 20 yr of isolation. *Ecol Appl* **29**, (2019).
35. Shapiro, A. C., Aguilar-Amuchastegui, N., Hostert, P. & Bastin, J.-F. Using fragmentation to assess degradation of forest edges in Democratic Republic of Congo. *Carbon Balance and Management* **11**, (2016).
36. Zhao, Z. *et al.* Fire enhances forest degradation within forest edge zones in Africa. *Nat. Geosci.* **14**, 479–483 (2021).
37. Pendrill, F. *et al.* Disentangling the numbers behind agriculture-driven tropical deforestation. *Science* **377**, eabm9267 (2022).
38. Hansen, M. C. *et al.* The fate of tropical forest fragments. *Sci. Adv.* **6**, eaax8574 (2020).
39. Lang, N. *et al.* Global canopy height regression and uncertainty estimation from GEDI LIDAR waveforms with deep ensembles. *Remote Sensing of Environment* **268**, 112760 (2022).
40. Potapov, P. *et al.* Mapping global forest canopy height through integration of GEDI and Landsat data. *Remote Sensing of Environment* **253**, 112165 (2021).
41. Silva Junior, C. H. L. *et al.* Amazonian forest degradation must be incorporated into the COP26 agenda. *Nat. Geosci.* **14**, 634–635 (2021).
42. Gasser, T., Ciais, P. & Lewis, S. L. How the Glasgow Declaration on Forests can help keep alive the 1.5 °C target. *Proc. Natl. Acad. Sci. U.S.A.* **119**, e2200519119 (2022).
43. Cop15 - Meeting Documents. <https://www.cbd.int/meetings/COP-15>.

44. Heinrich, V. H. A. *et al.* The carbon sink of secondary and degraded humid tropical forests. *Nature* **615**, 436–442 (2023).
45. Dubayah, R. *et al.* GEDI launches a new era of biomass inference from space. *Environ. Res. Lett.* **17**, 095001 (2022).
46. Dubayah, Ralph *et al.* GEDI L2A Elevation and Height Metrics Data Global Footprint Level V002. NASA EOSDIS Land Processes DAAC
https://doi.org/10.5067/GEDI/GEDI02_A.002 (2021).
47. Dubayah, R. O. *et al.* GEDI L4A Footprint Level Aboveground Biomass Density, Version 1. *ORNL DAAC* (2021) doi:10.3334/ORNLDAAC/1907.
48. Patterson, P. L. *et al.* Statistical properties of hybrid estimators proposed for GEDI—NASA’s global ecosystem dynamics investigation. *Environ. Res. Lett.* **14**, 065007 (2019).
49. Devore, J. L. & Berk, K. N. *Modern Mathematical Statistics with Applications*. (Springer, New York, NY, 2012). doi:10.1007/978-1-4614-0391-3.
50. *Applied Linear Statistical Models*. (McGraw-Hill Irwin, Boston, Mass., 2005).
51. F. Dormann, C. *et al.* Methods to account for spatial autocorrelation in the analysis of species distributional data: a review. *Ecography* **30**, 609–628 (2007).
52. Barni, P. E. *et al.* Logging Amazon forest increased the severity and spread of fires during the 2015–2016 El Niño. *Forest Ecology and Management* **500**, 119652 (2021).
53. Silva, C. V. J. *et al.* Estimating the multi-decadal carbon deficit of burned Amazonian forests. *Environ. Res. Lett.* **15**, 114023 (2020).
54. Brando, P. M. *et al.* Prolonged tropical forest degradation due to compounding disturbances: Implications for CO₂ and H₂O fluxes. *Glob Change Biol* **25**, 2855–2868 (2019).
55. Pontes-Lopes, A. *et al.* Drought-driven wildfire impacts on structure and dynamics in a wet Central Amazonian forest. *Proc. R. Soc. B.* **288**, 20210094 (2021).
56. Ghizoni Santos, E., Henrique Nunes, M., Jackson, T. & Eiji Maeda, E. Quantifying tropical forest disturbances using canopy structural traits derived from terrestrial laser scanning. *Forest Ecology and Management* **524**, 120546 (2022).

Reviewer Reports on the Third Revision:

Referees' comments:

Referee #1 (Remarks to the Author):

The manuscript is definitely improved and I thank the authors for having put in significant work to address many of the previous comments. On the whole I think a number of the issues I raised have been addressed, but not all. One of the main issues I raised previously was the difference between the WRI and the manually delimited data. The authors have now moved to a manually delimited dataset "Regarding degradation due to selective logging, we performed an extensive visual interpretation and delineation of selective logging operations based on their specific spatial features visible on the JRC-TMF Transition Map Asia while the delineation in the Amazon was generated from previous scientific experience^{4,7}".

While this is certainly a more consistent approach across continents, the problem now is how dependent this is on who has done it and how. This is critical as it underpins so much of what comes after. Extensive doesn't mean anything quantitative, nor does an appeal to previous scientific experience. The authors have provided a comparison of the manual v WRI impact on canopy height, w time in fig A2. This show that the two are different. But now they need to somehow provide a more quantitative assessment of the manual delineation approach ie how replicable / robust is this? One approach would be to have multiple people do sample delineations and comparing the outputs. I appreciate this is tricky but as I say, their analysis really rests on how robust this is.

More generally, while there is a lot of improvement in the clarity overall, there are still some areas where they're a qualitative or subjective statements. I have included some of those in the annotated docx document.

Overall, I think this is definitely an improvement over the previous version, but there are still questions over the robustness of the analysis due to the issue raised above. I would be happy to see the paper published if they were able to address that finally, as I still think the results are interesting.

Referee #2 (Remarks to the Author):

I thank the authors for responding to the comments, and again, for their patience in working through the issues that were raised. I am very sensitive to holding up what I feel is an important paper based on continued disagreements over statistical analyses. I do not think there is a way to resolve these, in an anonymous fashion, without a more closely coupling of the methods employed here relative to those that have been published previously. I do not think the simulations performed by the authors close these issues out. That said, the case can be made that in a relative sense, the statistical inferences of significant differences between degraded and undegraded areas may not hinge on whether we get the errors correct in an absolute sense. From my perspective, we know how to do this in an absolute sense. I also recognize this is mostly an issue for biomass, which has a model-based error, and that I advocated for a more central inclusion of biomass. I also underscore the authors have made an honest effort to provide evidence that the approach they have followed is defensible and would lead to results that are significant. Based on all the above, I believe the paper should proceed. I continue to advocate that this is exactly the kind of discovery science we expect space observations of forest structure to facilitate, and I am quite happy to see such a comprehensive effort.

I have only a few comments for the authors, again on the statistics and based on their responses, and only for their consideration.

“...distribution of the error over large spatial scale, the estimated precision of the mean intact forest Rh98 is 0.002m, 0.004m and 0.009m for Americas, Africa and Asia, respectively.”

But obviously it cannot be this small, either, in reality. Only under the assumption of random measurements, random error and no bias in measurements. It would take downstream work to incorporate any biases (say from looking at cal/val data) and applying these, which is beyond the scope of this work.

“My main issue still concerns the statistical methods and the meaning of some of the descriptive statistics and uncertainties that are being reported. These are described below but center on the monte carlo approach to propagating regression uncertainty and the results presented in Extended Data Tables 1-3. For example, the precisions shown for height are greater than what GEDI reports as RMSE against airborne data globally, so its

not clear why aggregates of these data and the precision of those means should be nearly as large as reported. So, either there is a methodological issue or the description of what is being reported lacks clarity and needs to be refined.

We thank the reviewer for pointing to this issue. We inform that in the previous version of the manuscript in the calculation of the descriptive statistics reported in Extended Data Tables 1 to 3 we used average values of RH98 and AGBD over intact forest computed at grid cell level (1.5 degree size) and derived the continental mean, standard deviation and standard error (or precision of the mean assuming random error distribution as suggested by⁴⁵)

Ref 45 does not assert random error distribution in biomass in general. There is some scale at which orbital tracks may be considered random, but the shots along the track (cluster samples) are never considered random.

“We based the analysis on the assumption that the error at each GEDI footprint is spatially independent as reported by 48.”

Please see above. Not sure that is reported by 48.

“ANOVA can still be relatively robust to the violations of certain assumptions, including heteroskedasticity.

This is particularly the case when dealing with large sample sizes, as in our study for which, the statistical tests were performed for spatial grids of 1.5 degree and therefore 4 orders of magnitude larger than the standard 1km GEDI data product.” And the following paragraphs.

The paper you cite (48) provides SE for biomass far beyond 1.5 deg. They are provided for entire *countries*, vastly larger than 1.5 degree cells. Yes, of course, the distribution of the sample mean is gaussian upon repeat samplings and the sampling error gets smaller, but the model-based regression errors do not deflate in the same manner.

It would have been good to compare your error you get from bootstrapping with the country-wide estimates of error published by GEDI (in Ref 48). This would have been preferable to the approach taken here. If the parameter of interest is height, then all that is missing is to add measurement error, which I think your method should capture correctly, again knowing that bias may be present, but could not be accounted for.

Fig A22 and comments. Thank you for this. I found it quite helpful.

“Our analysis and synthetic experiment about the effect of heteroscedasticity and spatial error correlation demonstrate that biases from the spatial autocorrelation and from the associated regression error do not fundamentally affect our results and conclusions about the impact of degradation of structural canopy parameters, particularly on the analysis of the magnitude and spatial scale of the edge effect.”

Thank you for this. I think this is OK. I don't think the associated regression error is handled correctly but I agree that this paper is attacking this from a different perspective and hopefully the conclusions are robust.

Table 1 for AGBD.

Your precisions seem too small, consider that you are asserting that the AGBD of intact forest, e.g. is 239 ± 0.026 . This seems too small by maybe one or two orders of magnitude. If you look at Ref 48, say, for the whole of the US, the SE reported using hybrid inference is around 1.2 Mg/ha (the US being a big area so maybe comparable in size). Maybe add some comment such as “Our method of obtaining the precision of the mean does not account for several factors that may depress our estimates of the precision of the mean relative to Ref (48)”. Or something like that? Your explanation did not seem to match the headings in that table as well.

Author Rebuttals to Third Revision:

Referee #1 (Remarks to the Author):

The manuscript is definitely improved and I thank the authors for having put in significant work to address many of the previous comments. On the whole I think a number of the issues I raised have been addressed, but not all. One of the main issues I raised previously was the difference between the WRI and the manually delimited data. The authors have now moved to a manually delimited dataset “Regarding degradation due to selective logging, we performed an extensive visual interpretation and delineation of selective logging operations based on their specific spatial features visible on the JRC-TMF Transition Map Asia while the delineation in the Amazon was generated from previous scientific experience^{4,7}”.

While this is certainly a more consistent approach across continents, the problem now is how dependent this is on who has done it and how. This is critical as it underpins so much of what comes after. Extensive doesn't mean anything quantitative, nor does an appeal to previous scientific experience. The authors have provided a comparison of the manual v WRI impact on canopy height, w time in fig A2. This show that the two are different. But now they need to somehow provide a more quantitative assessment of the manual delineation approach ie how replicable / robust is this? One approach would be to have multiple people do sample delineations and comparing the outputs. I appreciate this is tricky but as I say, their analysis really rests on how robust this is.

We thank the reviewer for the appreciation of our work. We recognize the importance of the issue raised by the reviewer on the robustness and quality of the delineation of selective logging in quantifying its impacts on forest structure. In our manuscript, the manual delineation of selective logging across the pantropical scale, specifically covering Brazil, French Guiana, Guyana, Cameroon, Central, African Republic, Gabon, Congo, the Democratic Republic of Congo, Indonesia, Malaysia and Papua New Guinea, is used to disentangle the causes of degradation in the assessment of long-term impacts and recovery on canopy height and other structural metrics. We also use this delineation to look at the status of so-called undisturbed forest from the JRC TMF dataset surrounding selectively logged forest (i.e. degraded forest pixels from the JRC TMF dataset overlaying with selective logging delineation) and quantify small-scale edge effects and structural damage induced by selective logging activity.

This manual delineation of selective logging activity was based on the identification of regular and irregular spatial patterns of degraded forest pixels, i.e. temporary canopy openings detected by the JRC TMF dataset that are visible for less than 900 days because they are followed by fast regrowth. These disturbances correspond in the field to secondary logging roads, logging gaps, logging decks where the forest recovers after a few years (Figure A1) and have already been described in the literature across the pantropical region (Souza 2003; Asner et al. 2002; Dupuis et al. 2023; Melendy et al. 2018).

Figure A1: *Examples of regular (a) and irregular (b) spatial patterns of selective logging mapped as degraded forest (light green) in the JRC TMF dataset.*

Additionally, the WRI-managed data on forest concessions (<https://data.globalforestwatch.org/documents/gfw::managed-forest-concessions-downloadable/about>) was used by the interpreter as a guide to delineate selective logging areas in Africa and Asia. In the absence of public data, the interpreter relied on past experience to target the main hotspots of selective logging (Bourgoin et al. 2020; Blanc et al. 2009; Miettinen, Stibig, and Achard 2014; Grecchi et al. 2017; Lima et al. 2019; Langner et al. 2018; Vancutsem et al. 2021; Shimabukuro et al. 2014; Rozak et al. 2018; Shapiro et al. 2023).

In the previous review, we performed a sensitivity analysis between the WRI-managed forest concessions and the manual delineation. Here, following the suggestion of the reviewer we perform an additional test on the robustness and quality of the manual delineation. Following the reviewer's suggestion, a new independent delineation of selective logging was carried out by another interpreter. The same countries of interest (see list above) were analyzed and the interpreter followed the same guidelines to identify selective logging as presented in the Methods section of the manuscript (i.e. spatial patterns of degraded forest pixels from JRC TMF dataset and, when available, the extent of WRI managed forest concessions). Based on the new delineation of selective logging (total of 1293 polygons), we re-extracted GEDI data as explained in the method section of the manuscript, we ran part of the analysis and compared the results with the ones obtained from the delineation presented in the manuscript (performed by interpreter 1).

Regarding the analysis on the persistence and cumulative impacts of forest degradation (cf. figure 3 of the manuscript), we could observe higher GEDI samples in Americas and Africa (similar to Asia) with the initial manual delineation (hereafter referred to as interpreter 1) compared to the new manual delineation (hereafter referred to as interpreter 2). However, GEDI samples with the new delineation approach were still higher than 1000 samples for each time step and continent, ensuring a consistent and robust comparison of results between the two delineations of selective logging (Figure A2 panel a). Only GEDI samples after more than 25 years since the last logging event in Americas and Asia are missing compared to the initial

delineation. This is not affecting the comparison of the immediate changes in canopy structure and the general recovery trends post-logging.

The comparison of results from the two approaches shows that the immediate changes in canopy structure and trends in recovery after selective logging impact on RH98 (panel B) and RH50 (panel C) are very similar (Figure A2 panels b and c). For instance, The magnitude of immediate decrease of RH50 after selective logging is 31% compared to intact forest RH50 vs 35% between delineation 1 and 2 in Americas, 32% vs 35% in Africa and 48% vs 49% in Asia (Figure A2 panel b). We also observe and quantify no recovery trends in RH98 between the two approaches, compared to a clear and similar recovery trajectory for RH50: 18% of RH50 is recovered on average in Americas after 20 years since logging with the initial delineation vs 20% with the new delineation (interpreter 2); 7% of RH50 is recovered in Africa after 5 years since logging with the initial delineation vs 5% with the new delineation; 19% of RH50 is recovered in Asia after 20 years since logging with the initial delineation vs 21% with the new delineation.

a

Forest degradation from selective logging (samples)

b Forest degradation from selective logging (RH98)

Figure A2: Comparative analysis of the impact of selective logging (with timing attribution to the JRC TMF dataset) between the use of manual delineation performed by interpreter 1 (corresponding to the data collection of the manuscript) vs the delineation performed by interpreter 2. Panels a to c refer to the number of sampled GEDI footprints, impacts of logging on RH98 and RH50 respectively for each continent and attribution scheme.

Regarding the analysis of changes in forest height in the vicinity of degraded forest from selective logging (i.e. forest-logged forest edge effects), our comparison shows similar results in canopy heights and RH50 of undisturbed forests near logged forest identified via delineation by interpreter 1 or interpreter 2 (Figure A3). Statistical tests between forests located at a distance less than 120m from logging and forests located beyond 120m from logging show similar results in terms of statistical significance comparing delineation by interpreter 1 and interpreter 2. The magnitude of forest-logged forest edge effects in RH98 and RH50 are also similar. For instance, RH50 of undisturbed forest near (<120m) logged forest is 9%, 15% and 21% lower on average than intact forest RH50 in Americas, Africa and Asia respectively using delineation by interpreter 2 while it is 10%, 16% and 21% lower than intact forest RH50 in Americas, Africa and Asia respectively using delineation by interpreter 1.

a

Undisturbed forest Surrounding Degradation (RH98)

b

Undisturbed forest Surrounding Degradation (RH50)

Figure A3: *Difference in RH98 and RH50 for forest classified as undisturbed in the JRC-TMF dataset located within and outside a buffer area (120 m radius) around logged forest delineated by interpreter 1 vs interpreter 2. Asterisks indicate the level of statistical significance of these comparisons: * $p \leq 0.05$, ** $p \leq 0.01$, *** $p \leq 0.001$*

Action taken: we have demonstrated that our results on the impacts of selective logging on forest structure across time and space dimensions are robust to manual delineation and not biased by the visual interpretation and delineation of selective logging areas across the pantropical belt. This sensitivity analysis proves that the manually delineated data is reliable and it also consolidates our methodology and relevance of input data in capturing this type of forest disturbance. Consequently, we added to the method section (line 382) this following sentence along with Figures A2 and A3 as new supplementary figures.

“An independent visual interpretation of selective logging was performed and the sensitivity analysis of how the delineation of logging influenced our results proved to be robust and unbiased (Supplementary Figs 8-9).”

Note that both sets of manual delineation of selective logging will be made publically available as google earth engine assets. This is already stated in the data availability section of the manuscript.

More generally, while there is a lot of improvement in the clarity overall, there are still some areas where they're a qualitative or subjective statements. I have included some of those in the annotated docx document.

We thank the reviewer for these specific comments that have now been addressed below.

Overall, I think this is definitely an improvement over the previous version, but there are still questions over the robustness of the analysis due to the issue raised above. I would be happy to see the paper published if they were able to address that finally, as I still think the results are interesting.

Annotated comments:

- 1) In 45% of cases[MOU1], forest degradation represents the initial stage that eventually leads to deforestation⁴

[MOU1]Instances? Areas? This needs explaining as is not clear.

Here, we are referring to areas of forest degradation followed by deforestation (after on average 7 years since the last degradation event) compared to the area of total degradation (followed or not by deforestation), one of the main results from Vancutsem et al. 2021. We have rephrased this sentence as such:

“For instance, Vancutsem et al.⁴ showed that nearly half of TMFs are ultimately deforested”.

- 2) Furthermore, regional studies likely underestimate [MOU1] the extent and impact of forest degradation²², particularly regarding the depth of edge effect penetration within forest interiors²³. [MOU1]Why though - can you explain why regional studies underestimate compared to something pan tropical? If it's not to do with it being regional per se, you should prob say - current studies are likely to be an underestimate because of X.

Thank you for pointing out this issue. Here we wanted to stress that despite the increasing collection of ALS data over tropical forest, it still remains difficult to access, limiting pan-tropical assessment of forest degradation. We rephrased this sentence as such:

“Furthermore, the depth of edge effect penetration within forest interiors is likely to be underestimated, mainly due to the scarcity of forest structure data across the tropics”.

- 3) The combination of a recent pantropical wall-to-wall dataset on disturbances of moist forests [MOU1] with the deployment of the Global Dynamics Ecosystem Investigation (GEDI⁶) [MOU1]Add ref [4] for that here? This is kind of duplicated in the intro to the next para.

Thank you for this comment, to avoid redundancy with the intro of the next paragraph we rephrased this sentence as such: **“The deployment of the Global Dynamics Ecosystem Investigation (GEDI⁶) instrument on the International Space Station in late 2018, which specifically targets forest structure, offers a unique opportunity to shed light on forest degradation at pantropical scale”**

- 4) the height of median energy (RH50), describing the [MOU1] vertical distribution of canopy elements and gaps²⁵; [MOU1]No it's related to that, but it's an energy measure primarily

We completely agree with the reviewer and RH50 is described more technically in the methodology section (line 542): “RH50 (also known as “Height Of Median Energy”-HOME) is the median height at which the 50th percentile of the cumulative waveform energy returned relative to the ground and has been identified as one of the LiDAR metrics with the greatest potential for estimating structural characteristics in tropical forests”.

- 5) We found comparable [MOU1] among-continent intercontinental variations for AGBD (370.8 Mg/ha ± 205.2 in Asia, 225.5 Mg/ha ± 110.9 in Africa and 239.5 Mg/ha ± 129.9

in Americas), emphasizing the consistency [MOU2] of these forest structural metrics across continents (Extended Data Fig.1, Supplementary Fig.1). [MOU1]In what sense? The ordering is different as Am > Africa for AGBD. [MOU2]That doesn't follow here - the ordering is different, and this has nothing to do with consistency of metrics unless you make clear what you mean and why that is.

Thank you, we rephrased this section as such:

“Overall, canopy heights are higher in Asia (34.4 m ± 4.76 ± 10.7, mean +/- sd) than in Africa (29.3 m ± 8.6) and in Americas (28.6 m ± 7.4). Similar results are found for AGBD (370.8 Mg/ha ± 205.2 in Asia, 225.5 Mg/ha ± 110.9 in Africa and 239.5 Mg/ha ± 129.9 in Americas).”

Note that Supplementary Fig.1 has been deleted as it is not used in the text anymore.

- 6) These results support previous observations²⁷, showing that old-growth tropical forests in Asia, which are typically dominated by hardwood wind-dispersed species, show a higher frequency of large trees [MOU1] compared to Africa and South America. [MOU1]How do those figures above support this? You need to at least provide some evidence for that here, or highlight what the key result in the ED or SI fig 1. And what is your threshold for 'large' trees?

Thank you for pointing this out, it seems that we omitted to add the reference to Supplementary Figure 1 (former Supplementary Fig. 2 in the last round) at the end of this sentence. The revised sentence reads as such:

“These results support previous observations²⁷, showing that intact tropical forests in Asia, which are typically dominated by hardwood wind-dispersed species, show a higher frequency of large and tall trees (>30m in RH98) compared to Africa and South America (Supplementary Fig.1).”

- 7) In fact, we quantify that the minimum difference between intact old growth [MOU1] and degraded forests [MOU1]Defined as?

Because old-growth is not defined in our manuscript, we decided to rephrase the sentence into:

“In fact, we quantify that the minimum difference between intact and degraded forests...”

- 8) Our results show this is a significant underestimate with edge effects measured up to ~1.5 km into the forest interior implying that the overall spatial impact of fragmentation across the pantropical belt is severely underestimated by a factor of over 200[MOU1]%. [MOU1]This is not a factor, it's a %

We agree and rephrased the sentence as such:

“...spatial impact of fragmentation across the pantropical belt is severely underestimated by at least 200%”

- 9) Regarding degradation due to selective logging, we used the performed an extensive [MOU1] visual interpretation and delineation of selective logging operations based on their specific spatial features visible on the JRC-TMF Transition Map [MOU1] Meaning what? How are you quantifying this?

We have fully addressed this comment, see our response to the general comment above.

- Asner, Gregory P., Michael Keller, Rodrigo Pereira, and Johan C. Zweede. 2002. "Remote Sensing of Selective Logging in Amazonia: Assessing Limitations Based on Detailed Field Observations, Landsat ETM+, and Textural Analysis." *Remote Sensing of Environment* 80 (3): 483–96.
- Blanc, Lilian, Marion Echard, Bruno Herault, Damien Bonal, Eric Marcon, Jerome Chave, and Christopher Baraloto. 2009. "Dynamics of Aboveground Carbon Stocks in a Selectively Logged Tropical Forest." *Ecological Applications* 19 (6): 1397–1404.
- Bourgoin, Clément, Julie Betbeder, Pierre Couteron, Lilian Blanc, Hélène Dessard, Johan Oszwald, Renan Le Roux, et al. 2020. "UAV-Based Canopy Textures Assess Changes in Forest Structure from Long-Term Degradation." *Ecological Indicators* 115 (August): 106386. <https://doi.org/10.1016/j.ecolind.2020.106386>.
- Dupuis, Chloé, Adeline Fayolle, Jean-François Bastin, Nicolas Latte, and Philippe Lejeune. 2023. "Monitoring Selective Logging Intensities in Central Africa with Sentinel-1: A Canopy Disturbance Experiment." *Remote Sensing of Environment* 298 (December): 113828. <https://doi.org/10.1016/j.rse.2023.113828>.
- Grecchi, Rosana Cristina, René Beuchle, Yosio Edemir Shimabukuro, Luiz E.O.C. Aragão, Egidio Arai, Dario Simonetti, and Frédéric Achard. 2017. "An Integrated Remote Sensing and GIS Approach for Monitoring Areas Affected by Selective Logging: A Case Study in Northern Mato Grosso, Brazilian Amazon." *International Journal of Applied Earth Observation and Geoinformation* 61 (September): 70–80. <https://doi.org/10.1016/j.jag.2017.05.001>.
- Langner, Andreas, Jukka Miettinen, Markus Kukkonen, Christelle Vancutsem, Dario Simonetti, Ghislain Vieilledent, Astrid Verhegghen, Javier Gallego, and Hans-Jürgen Stibig. 2018. "Towards Operational Monitoring of Forest Canopy Disturbance in Evergreen Rain Forests: A Test Case in Continental Southeast Asia." *Remote Sensing* 10 (4): 544. <https://doi.org/10.3390/rs10040544>.
- Lima, Thaís Almeida, René Beuchle, Andreas Langner, Rosana Cristina Grecchi, Verena C. Griess, and Frédéric Achard. 2019. "Comparing Sentinel-2 MSI and Landsat 8 OLI Imagery for Monitoring Selective Logging in the Brazilian Amazon." *Remote Sensing* 11 (8): 961. <https://doi.org/10.3390/rs11080961>.
- Melendy, L., S. C. Hagen, F. B. Sullivan, T. R. H. Pearson, S. M. Walker, P. Ellis, Kustiyo, et al. 2018. "Automated Method for Measuring the Extent of Selective Logging Damage with Airborne LiDAR Data." *ISPRS Journal of Photogrammetry and Remote Sensing* 139 (May): 228–40. <https://doi.org/10.1016/j.isprsjprs.2018.02.022>.
- Miettinen, Jukka, Hans-Jürgen Stibig, and Frédéric Achard. 2014. "Remote Sensing of Forest Degradation in Southeast Asia—Aiming for a Regional View through 5–30 m Satellite Data." *Global Ecology and Conservation* 2 (December): 24–36. <https://doi.org/10.1016/j.gecco.2014.07.007>.
- Rozak, Andes Hamuraby, Ervan Rutishauser, Karsten Raulund-Rasmussen, and Plinio Sist. 2018. "The Imprint of Logging on Tropical Forest Carbon Stocks: A Bornean Case-Study." *Forest Ecology and Management* 417 (May): 154–66. <https://doi.org/10.1016/j.foreco.2018.03.007>.

- Shapiro, Aurélie, Rémi d'Annunzio, Baudouin Desclée, Quentin Jungers, Héritier Koy Kondjo, Josefina Mbulito Iyanga, Francis Inicko Gangyo, et al. 2023. "Small Scale Agriculture Continues to Drive Deforestation and Degradation in Fragmented Forests in the Congo Basin (2015–2020)." *Land Use Policy* 134 (November): 106922. <https://doi.org/10.1016/j.landusepol.2023.106922>.
- Shimabukuro, Yosio Edemir, René Beuchle, Rosana Cristina Grecchi, and Frédéric Achard. 2014. "Assessment of Forest Degradation in Brazilian Amazon Due to Selective Logging and Fires Using Time Series of Fraction Images Derived from Landsat ETM+ Images." *Remote Sensing Letters* 5 (9): 773–82. <https://doi.org/10.1080/2150704X.2014.967880>.
- Souza, C. 2003. "Mapping Forest Degradation in the Eastern Amazon from SPOT 4 through Spectral Mixture Models." *Remote Sensing of Environment* 87 (4): 494–506. <https://doi.org/10.1016/j.rse.2002.08.002>.
- Vancutsem, C., F. Achard, J.-F. Pekel, G. Vieilledent, S. Carboni, D. Simonetti, J. Gallego, L. E. O. C. Aragão, and R. Nasi. 2021. "Long-Term (1990–2019) Monitoring of Forest Cover Changes in the Humid Tropics." *Science Advances* 7 (10): eabe1603. <https://doi.org/10.1126/sciadv.abe1603>.

Referee #2 (Remarks to the Author):

I thank the authors for responding to the comments, and again, for their patience in working through the issues that were raised. I am very sensitive to holding up what I feel is an important paper based on continued disagreements over statistical analyses. I do not think there is a way to resolve these, in an anonymous fashion, without a more closely coupling of the methods employed here relative to those that have been published previously. I do not think the simulations performed by the authors close these issues out. That said, the case can be made that in a relative sense, the statistical inferences of significant differences between degraded and undegraded areas may not hinge on whether we get the errors correct in an absolute sense. From my perspective, we know how to do this in an absolute sense. I also recognize this is mostly an issue for biomass, which has a model-based error, and that I advocated for a more central inclusion of biomass. I also underscore the authors have made an honest effort to provide evidence that the approach they have followed is defensible and would lead to results that are significant. Based on all the above, I believe the paper should proceed. I continue to advocate that this is exactly the kind of discovery science we expect space observations of forest structure to facilitate, and I am quite happy to see such a comprehensive effort.

We thank the reviewer for the comprehensive and thoughtful comments, which have significantly contributed to the improvement of the manuscript throughout the review process. We acknowledge the inherent challenges in achieving perfect alignment between different methodologies. Through the various rounds of review, our aim has been to transparently present our methodology, underlining its robustness and its specific relevance to the research context at hand.

While we believe our simulation exercises substantiate the robustness of our findings, we also recognize the complexities of applying analytical methods at a pantropical scale, particularly when addressing heteroskedasticity and spatial autocorrelation, which present persistent

challenges. The reviewer's recognition of our efforts to construct a robust methodological framework, and their acknowledgment of the significance of our results, serve to reaffirm the scientific merit and potential impact of our work.

We share the reviewer's enthusiasm for the transformative potential of discovery science enabled by space observations of forest structure, looking forward to the insights that GEDI may provide.

I have only a few comments for the authors, again on the statistics and based on their responses, and only for their consideration.

- 1) "...distribution of the error over large spatial scale, the estimated precision of the mean intact forest Rh98 is 0.002m, 0.004m and 0.009m for Americas, Africa and Asia, respectively."

But obviously it cannot be this small, either, in reality. Only under the assumption of random measurements, random error and no bias in measurements. It would take downstream work to incorporate any biases (say from looking at cal/val data) and applying these, which is beyond the scope of this work.

We thank the reviewer for this comment. In agreement with the reviewer's assessment, we recognize that the presence of non-random samples and bias in measurements indeed poses challenges to the estimation of the precision parameter. We also agree with the reviewer's feedback that the problem should be more thoroughly addressed downstream. However, we would like to stress that every care was taken to emphasize this important point in the table caption. In particular, we would like to note that the following text was added to the caption: "Note that issues related to the non-randomness and spatial autocorrelation of GEDI samples are not incorporated into this computation of summary statistics."

- 2) "My main issue still concerns the statistical methods and the meaning of some of the descriptive statistics and uncertainties that are being reported. These are described below but center on the monte carlo approach to propagating regression uncertainty and the results presented in Extended Data Tables 1-3. For example, the precisions shown for height are greater than what GEDI reports as RMSE against airborne data globally, so its not clear why aggregates of these data and the precision of those means should be nearly as large as reported. So, either there is a methodological issue or the description of what is being reported lacks clarity and needs to be refined.

We thank the reviewer for pointing to this issue. We inform that in the previous version of the manuscript in the calculation of the descriptive statistics reported in Extended Data Tables 1 to 3 we used average values of RH98 and AGBD over intact forest computed at grid cell level (1.5 degree size) and derived the continental mean, standard deviation and standard error (or precision of the mean assuming random error distribution as suggested by 45"

Ref 45 does not assert random error distribution in biomass in general. There is some scale at which orbital tracks may be considered random, but the shots along the track (cluster samples) are never considered random.

- 3) “We based the analysis on the assumption that the error at each GEDI footprint is spatially independent as reported by 48.”

Please see above. Not sure that is reported by 48.

- 4) “ANOVA can still be relatively robust to the violations of certain assumptions, including heteroskedasticity. This is particularly the case when dealing with large sample sizes, as in our study for which, the statistical tests were performed for spatial grids of 1.5 degree and therefore 4 orders of magnitude larger than the standard 1km GEDI data product.” And the following paragraphs.

The paper you cite (48) provides SE for biomass far beyond 1.5 deg. They are provided for entire *countries*, vastly larger than 1.5 degree cells. Yes, of course, the distribution of the sample mean is gaussian upon repeat samplings and the sampling error gets smaller, but the model-based regression errors do not deflate in the same manner.

We appreciate the reviewer's valid point regarding the distinct behaviour of model-based regression errors, which indeed do not necessarily deflate with increased sample size in the same manner as sampling error. While a large sample size enhances the stability of mean estimations via the Central Limit Theorem, it does not automatically correct for model-based errors. Acknowledging this, our synthetic experiment was designed to test the robustness of our results against these types of errors. This analysis demonstrated that our conclusions remain reliable, despite the complexities introduced by model-based errors. We remain confident in the robustness of our findings, supported by these additional efforts.

It would have been good to compare your error you get from bootstrapping with the country-wide estimates of error published by GEDI (in Ref 48). This would have been preferable to the approach taken here. If the parameter of interest is height, then all that is missing is to add measurement error, which I think your method should capture correctly, again knowing that bias may be present, but could not be accounted for.

- 5) Fig A22 and comments. Thank you for this. I found it quite helpful.

“Our analysis and synthetic experiment about the effect of heteroscedasticity and spatial error correlation demonstrate that biases from the spatial autocorrelation and from the associated regression error do not fundamentally affect our results and conclusions about the impact of degradation of structural canopy parameters, particularly on the analysis of the magnitude and spatial scale of the edge effect.”

Thank you for this. I think this is OK. I don't think the associated regression error is handled correctly but I agree that this paper is attacking this from a different perspective and hopefully the conclusions are robust.

6) Table 1 for AGBD.

Your precisions seem too small, consider that you are asserting that the AGBD of intact forest, e.g. is 239 ± 0.026 . This seems too small by maybe one or two orders of magnitude. If you look at Ref 48, say, for the whole of the US, the SE reported using hybrid inference is around 1.2 Mg/ha (the US being a big area so maybe comparable in size). Maybe add some comment such as “Our method of obtaining the precision of the mean does not account for several factors that may depress our estimates of the precision of the mean relative to Ref (48)”. Or something like that? Your explanation did not seem to match the headings in that table as well.

Thank you for this comment, we agree with the reviewer and we have already included in the caption of this table the following text pointing to references of Dubayah et al. 2022 and Patterson et al. 2019: “Note that issues linked to the non-randomness and spatial autocorrelation of GEDI samples and the propagation of the regression error associated to each AGBD estimate are not integrated in this computation of summary statistics^{68,75}”.

Reviewer Reports on the Fourth Revision:

Referees' comments:

Referee #1 (Remarks to the Author):

I think the main issue I raised in the last round ie the issue of how reliable the manual delineation aspect is has mainly been addressed by the inclusion of delineation by a second interpreter. But I still think this needs a bit more work. Fig A2 a, b, c - why not plot both sets of points on the same scale? As it is, it is unnecessarily difficult to compare - for eg even the y scale on Fig A2a is different. This could be analysed as a scatter plot which would indicate whether there was actually bias. There are some differences here but they are broadly dismissed as being not significant. They may be but that isn't really tested for the time series.

"18% of RH50 is recovered on average in Americas after 20 years since logging with the initial delineation vs 20% with the new delineation (interpreter 2); 7% of RH50 is recovered in Africa after 5 years since logging with the initial delineation vs 5% with the new delineation; 19% of RH50 is recovered in Asia after 20 years since logging with the initial delineation vs 21% with the new delineation."

18:20 is a 10% difference. 7:5 is a 40% difference; 19:21 also 10% difference. These may or may not be significant. As I say, I think this mostly addresses the issue of how robust the resulting delineation is, but this could be more definitive & the authors need to be v clear that different interpreters arrive at results that may have small differences but do not alter the subsequent analysis and conclusions.

Author Rebuttals to Fourth Revision:

Referee #1 (Remarks to the Author):

I think the main issue I raised in the last round ie the issue of how reliable the manual delineation aspect is has mainly been addressed by the inclusion of delineation by a second interpreter. But I still think this needs a bit more work. Fig A2 a, b, c - why not plot both sets of points on the same scale? As it is, it is unnecessarily difficult to compare - for eg even the y scale on Fig A2a is different. This could be analysed as a scatter plot which would indicate whether there was actually bias. There are some differences here but they are broadly dismissed as being not significant. They may be but that isn't really tested for the time series.

We thank the reviewer for raising this last comment on the readability of FigA2 (also found as Supplementary Fig 8). As recommended, we modified this figure into scatter plots which allow a direct comparison of the results of the long-term impacts of selective logging on RH50 and RH98 (scaled as a percentage of intact forest values) between interpretation 1 and 2 of logging. Fig A2 b and c from the previous version become Fig A2 b and c in this revised version and Supplementary Figure 8b and 8c in the manuscript.

a

Forest degradation from selective logging (samples)

Updated figure A2 (and updated Supplementary Fig 8): *Comparative analysis of the impact of selective logging (with timing attribution to the JRC TMF dataset) between the use of manual delineation performed by interpreter 1 (corresponding to the data collection of the manuscript) vs the delineation performed by interpreter 2 (sensitivity analysis). Panels a to c refer to the number of sampled GEDI footprints, impacts of logging on RH50 and RH98 respectively for each continent and attribution scheme. In panels b and c, horizontal and vertical bars indicate the spatial standard deviation following the delineation of selective logging made by interpretation #1 and #2 respectively. The dashed line in panel b and c is the 1:1 line.*

"18% of RH50 is recovered on average in Americas after 20 years since logging with the initial delineation vs 20% with the new delineation (interpreter 2); 7% of RH50 is recovered in Africa after 5 years since logging with the initial delineation vs 5% with the new delineation; 19% of RH50 is recovered in Asia after 20 years since logging with the initial delineation vs 21% with the new delineation."

18:20 is a 10% difference. 7:5 is a 40% difference; 19:21 also 10% difference. These may or may not be significant. As I say, I think this mostly addresses the issue of how robust the resulting delineation is, but this could be more definitive & the authors need to be v clear that different interpreters arrive at results that may have small differences but do not alter the subsequent analysis and conclusions.

We agree with the referee that we need to be more nuanced in the conclusion of this sensitivity analysis. In the Methods section, we now acknowledge that different interpretations of selective logging may result in small differences in magnitude and trends of logging impacts on forest structure without altering the subsequent analysis and conclusions. This was also clearly visible on supplementary fig 9 (or figure A3 from the last round of review) where despite small differences in RH98 and RH50 for forest classified as undisturbed in the JRC-TMF dataset located within and outside a buffer area (120 m radius) around logged forest delineated by interpreter 1 vs interpreter 2, all the differences between forest categories remain statistically significant, with p-values less than the chosen significance level of 0.05 ($\alpha=0.05$).

The updated sentence (line 382) reads as such: **“An independent visual interpretation of selective logging was performed in order to analyze how the delineation influenced our results. This sensitivity analysis showed small differences in the magnitude and trends of logging impacts on forest structure without altering the subsequent analysis and conclusions (Supplementary Fig 8). It also proved to be unbiased and robust when comparing changes in forest height in the vicinity of forest degraded by selective logging (Supplementary Fig 9)”**.